# Stochastic Regret Guarantees for Online Zeroth- and First-Order Bilevel Optimization

**Parvin Nazari**
Amirkabir University of Technology
p.nazari17@gmail.com

**Bojian Hou**
University of Pennsylvania
bojianh@upenn.edu

**Davoud Ataee Tarzanagh** *
Samsung SDS Research America
d.tarzanagh@samsung.com

**Li Shen**
University of Pennsylvania
li.shen@pennmedicine.upenn.edu

**George Michailidis**
University of California, Los Angeles
gmichail@ucla.edu

## Abstract

Online bilevel optimization (OBO) is a powerful framework for machine learning problems where both outer and inner objectives evolve over time, requiring dynamic updates. Current OBO approaches rely on deterministic *window-smoothed* regret minimization, which may not accurately reflect system performance when functions change rapidly. In this work, we introduce a novel search direction and show that both first- and zeroth-order (ZO) stochastic OBO algorithms leveraging this direction achieve sublinear stochastic bilevel regret without window smoothing. Beyond these guarantees, our framework enhances efficiency by: (i) reducing oracle dependence in hypergradient estimation, (ii) updating inner and outer variables alongside the linear system solution, and (iii) employing ZO-based estimation of Hessians, Jacobians, and gradients. Experiments on online parametric loss tuning and black-box adversarial attacks validate our approach.

## 1 Introduction

Bilevel optimization (BO) minimizes an outer objective dependent on an inner problem's solution. Originating in game theory [64] and formalized in mathematical optimization [9], BO finds applications in operations research, engineering, economics [16], and image processing [14]. Recently, BO has gained traction in machine learning, including hyperparameter optimization [22], meta-learning [18], reinforcement learning [65], and neural architecture search [51].

In the *offline* setting, BO solves the following problem:

$$\mathbf{x}^* \in \operatorname{argmin}_{\mathbf{x} \in \mathbb{R}^{d_1}} f(\mathbf{x}, \mathbf{y}^*(\mathbf{x})) \quad \text{subj. to} \quad \mathbf{y}^*(\mathbf{x}) = \operatorname{argmin}_{\mathbf{y} \in \mathbb{R}^{d_2}} g(\mathbf{x}, \mathbf{y}), \tag{BO}$$

where $f$ and $g$ are the outer and inner objectives, with $\mathbf{x}$ and $\mathbf{y}$ as their respective variables.

OBO [67] addresses dynamic scenarios where objectives evolve over time, requiring the agent to update the outer decision in response to the optimal inner decision. Similar to online single-level optimization (OSO) [72], OBO involves iterative decision-making without prior knowledge of outcomes [67, 50, 8]. Let $T$ be the total number of rounds. Define $\mathbf{x}_t \in \mathcal{X} \subset \mathbb{R}^{d_1}$ as the

---

*Corresponding author

| OBO Method | Window Size in Regret ($w$) | System Iters. | Stochastic Regret | Const. Regret Min. | Only Func. Feedback | Local Regret Bound |
|---|---|---|---|---|---|---|
| OAGD [67] | $o(T)$ | N.A. (Exact) | ✗ | ✗ | ✗ | $\frac{T}{w} + H_{1,T} + H_{2,T}$ |
| SOBOW [50] | $o(T)$ | $\mathcal{O}(\kappa_g \log \kappa_g)$ | ✗ | ✗ | ✗ | $\frac{T}{w} + V_T + H_{2,T}$ |
| SOBBO [8] | $o(T)$ | $\mathcal{O}(\kappa_g \log \kappa_g)$ | ✓ | ✓ | ✗ | $\frac{T}{w}\sigma^2 + V_T + H_{2,T}$ |
| SOGD | 1 | 1 | ✓ | ✓ | ✗ | $T^{\frac{1}{3}}(\sigma^2 + \Delta_T) + T^{\frac{2}{3}}\Psi_T$ |
| ZO-SOGD | 1 | 1 | ✓ | ✓ | ✓ | $(d_1 + d_2)^{\frac{3}{4}}T^{\frac{1}{3}}(\hat{\sigma}^2 + \hat{\Delta}_T)$ $+(d_1 + d_2)^{\frac{3}{2}}T^{\frac{2}{3}}\hat{\Psi}_T$ |

Table 1: Comparison of OBO algorithms based on regret window $w$, solver iterations, stochastic/constrained regrets, feedback type, and local bounds. $\kappa_g$ denotes the condition number of the inner objective $g_t$. $V_T$, $H_{p,T}$, $\Delta_T$, $\Psi_T$, $\hat{\Delta}_T$, $\hat{\Psi}_T$, $\sigma$, and $\hat{\sigma}$ are defined in (11), (14), (30), (10), and (28), respectively.

decision variable and $f_t : \mathcal{X} \times \mathbb{R}^{d_2} \to \mathbb{R}$ as the outer function. Similarly, define $\mathbf{y}_t \in \mathbb{R}^{d_2}$ and $g_t : \mathcal{X} \times \mathbb{R}^{d_2} \to \mathbb{R}$ for the inner problem, where $\mathbf{y}_t^*(\mathbf{x}) = \mathrm{argmin}_{\mathbf{y} \in \mathbb{R}^{d_2}} g_t(\mathbf{x}, \mathbf{y})$. OBO can be seen as a *single-player* problem, where the player selects $\mathbf{x}_t$ without knowing $\mathbf{y}_t^*(\mathbf{x})$, using $\mathbf{y}_t$ as an estimate based on $g_t$. Alternatively, it can be framed as a *two-player* game [64], where the leader ($\mathbf{x}_t$) competes with the follower ($\mathbf{y}_t$), who selects $\mathbf{y}_t^*(\mathbf{x})$ based on limited knowledge of $g_t$. This framework includes online and adversarial variants of (BO), such as online actor-critic algorithms [71], online meta-learning [19], and online hyperparameter optimization [50]. The inner and outer functions may be time-varying, adversarial, unavailable *a priori*, and require *nonstationary* optimization.

**Our Contributions.** This paper addresses stochastic OBO, introducing novel first - and zeroth-order methods to minimize stochastic bilevel regret. Key contributions are summarized below.

• **Stochastic regret minimization without window-smoothing.** Existing OBO methods [67, 50, 40, 8] rely on deterministic *window-smoothed* regret minimization, which may not accurately reflect system performance when functions change rapidly. We address these limitations by introducing a novel search direction (Section 2) and proving that both first-order and ZO methods achieve sublinear *stochastic bilevel regret without window-smoothing ($w = 1$)*; see Theorems 2.6 and 3.2 and Table 1.

• **OBO with function value oracle feedback.** In large-scale and black-box settings [11, 58], first- and second-order information is often unavailable or costly. Constructing accurate (hyper)-gradient estimators using only function value oracles is particularly challenging due to BO's nested structure. Existing methods rely on gradient, Hessian, and Jacobian oracles, limiting scalability [21, 27]. We propose Algorithm 2, which estimates Hessians, Jacobians, and gradients using function value oracles, achieving sublinear local regret (Theorem 3.2).

• **OBO with one subproblem solver iteration.** A major challenge in BO is solving implicit systems to approximate the hypergradient [43, 12]. While efficient offline BO methods exist [43, 15], extending them to OBO is difficult due to time-varying objectives. SOBOW [50] partially addresses this using a conjugate gradient (CG) algorithm with increasing iterations (Table 1). We improve upon SOBOW by introducing Algorithms 1 and 2, which require only a *single* subproblem solver iteration.

## 2 Stochastic OBO with Access to First- and Inner Second-Order Oracles

**Notation.** $\mathbb{R}^d$ is the $d$-dimensional real space; $\mathbb{R}^d_+$ and $\mathbb{R}^d_{++}$ denote its nonnegative and positive orthants. Bold lowercase letters (e.g., $\mathbf{x}, \mathbf{y}$) represent vectors, $\langle \mathbf{x}, \mathbf{y} \rangle$ is the inner product, and $\|\cdot\|$ is the Euclidean norm. $\nabla_{\mathbf{x}}$ denotes the gradient, and $\nabla^2_{\mathbf{xy}} = \nabla_{\mathbf{x}}\nabla_{\mathbf{y}}$. A function is $L$-smooth if its gradient is $L$-Lipschitz. The projection onto a convex set $\mathcal{X}$ is $\Pi_{\mathcal{X}}(\mathbf{z}) = \mathrm{argmin}_{\mathbf{x} \in \mathcal{X}} \frac{1}{2}\|\mathbf{x} - \mathbf{z}\|^2$. We use $[T]$ for $\{1, \ldots, T\}$, $\mathbb{E}[\cdot]$ for expectation, and $\mathcal{O}(\cdot)$ to hide problem-independent constants.

**Stochastic OBO Setting**. Let $T$ be the total rounds [67]. Define $\mathbf{x}_t \in \mathcal{X} \subset \mathbb{R}^{d_1}$ as the decision variable and $f_t : \mathcal{X} \times \mathbb{R}^{d_2}$ as the outer objective. The inner decision variable and objective are $\mathbf{y}_t \in \mathbb{R}^{d_2}$ and $g_t : \mathcal{X} \times \mathbb{R}^{d_2}$, where the optimal inner decision is:

$$\mathbf{y}_t^*(\mathbf{x}) \in \mathrm{argmin}_{\mathbf{y} \in \mathbb{R}^{d_2}} \left\{ g_t(\mathbf{x}, \mathbf{y}) := \mathbb{E}_{\zeta_t \sim \mathcal{D}_{g,t}} [g_t(\mathbf{x}, \mathbf{y}; \zeta_t)] \right\}. \tag{1}$$

Further, we have

$$f_t(\mathbf{x}, \mathbf{y}_t^*(\mathbf{x})) := \mathbb{E}_{\xi_t \sim \mathcal{D}_{f,t}} \left[ f_t(\mathbf{x}, \mathbf{y}_t^*(\mathbf{x}); \xi_t) \right].$$

Here, $(\mathcal{D}_{f,t}, \mathcal{D}_{g,t})$ denote data distributions at time $t$. Our setting is stochastic, with only noisy evaluations of functions, gradients, and Hessians. Unlike OSO [72], where true losses are revealed, in OBO the outer function $f_t(\mathbf{x}, \mathbf{y}_t^*(\mathbf{x}))$ is inaccessible for updating $\mathbf{x}_t$ and is generally non-convex in $\mathbf{x}$, making standard regret notions from online convex optimization [33] unsuitable.

Given a sequence $\{\alpha_t \in \mathbb{R}_{++}\}_{t=1}^T$, we define the following notion of *bilevel local regret*:

$$\text{BL-Reg}_T := \sum_{t=1}^T \mathbb{E}\left[\left\| \mathcal{P}_{\mathcal{X}, \alpha_t}\left(\mathbf{x}_t; \nabla f_t(\mathbf{x}_t, \mathbf{y}_t^*(\mathbf{x}_t))\right) \right\|^2\right], \quad \text{with} \tag{2a}$$

$$\mathcal{P}_{\mathcal{X}, \alpha_t}\left(\mathbf{x}_t; \nabla f_t(\mathbf{x}_t, \mathbf{y}_t^*(\mathbf{x}_t))\right) = \frac{1}{\alpha_t}\left(\mathbf{x}_t - \Pi_{\mathcal{X}}\left[\mathbf{x}_t - \alpha_t \nabla f_t(\mathbf{x}_t, \mathbf{y}_t^*(\mathbf{x}_t))\right]\right). \tag{2b}$$

The local regret (2) compares the leader's decision $\mathbf{x}_t$ to the stationary points $\mathbf{x}_t^*$ satisfying $\mathcal{P}_{\mathcal{X}, \alpha_t}\left(\mathbf{x}_t^*; \nabla f_t(\mathbf{x}_t^*, \mathbf{y}_t^*(\mathbf{x}_t^*))\right) = 0$. This can also be viewed as dynamic local regret, as the baseline corresponds to a stationary point of the leader's objective $f_t$.

Previous work on (nonconvex) OBO examined unconstrained local regret using window-smoothed objectives: $F_{t,w}(\mathbf{x}, \mathbf{y}) = (1/w)\sum_{i=0}^{w-1} f_{t-i}(\mathbf{x}, \mathbf{y})$. For $w = 1$ and $\mathcal{X} = \mathbb{R}^{d_1}$, this reduces to (2). [67, 50] showed that $w = o(T)$ ensures sublinear regret under slow variations in $\{F_{t,w}\}_{t=1}^T$, while rapid changes can lead to deviations. However, smoothing may misrepresent regret (Figure 1). This paper introduces a new projection-based local regret notion (2) without smoothing, and establishes sublinear regret for constrained OBO.

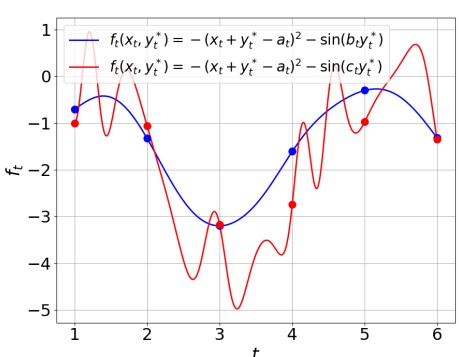

Figure 1: Smoothly and rapidly changing $f_t$ in OBO with $g_t(x_t, y_t) = (y_t - \cos(x_t))^2$, $a_t = 1 + 0.5\sin(t)$, $b_t = 1 + \sin(0.5t)$, and $c_t = 10b_t$.

**Online Gradient Descent (OGD).** One of the most widely used algorithms for online (single-level) optimization is OGD [72]. The procedure for OGD is as follows: For each $t \in [T]$, the algorithm selects $\mathbf{x}_t \in \mathcal{X}$, observes the function $f_t : \mathcal{X} \subset \mathbb{R}^d \to \mathbb{R}$, and updates according to

$$\mathbf{x}_{t+1} = \Pi_{\mathcal{X}}\left(\mathbf{x}_t - \alpha_t \nabla f_t(\mathbf{x}_t)\right), \qquad \alpha_t > 0. \tag{OGD}$$

In the following, we adapt OGD to OBO and introduce a novel framework that requires limited feedback and can utilize ZO updates within a single-loop structure.

To adapt OGD to OBO, [67, 50, 8] developed a variant alternating between inner and outer OGD, achieving sublinear bilevel regret bounds. We introduce a new search direction that enables sublinear bilevel regret without window smoothing. To compute the hypergradient $\nabla f_t(\mathbf{x}, \mathbf{y}_t^*(\mathbf{x}))$ where $\mathbf{y}_t^*(\mathbf{x})$ is defined in (1), since $\nabla_{\mathbf{y}} g_t(\mathbf{x}, \mathbf{y}_t^*(\mathbf{x})) = 0$, using the implicit function theorem, yields

$$\nabla f_t(\mathbf{x}, \mathbf{y}_t^*(\mathbf{x})) = \nabla_{\mathbf{x}} f_t\left(\mathbf{x}, \mathbf{y}_t^*(\mathbf{x})\right) + \nabla_{\mathbf{xy}}^2 g_t\left(\mathbf{x}, \mathbf{y}_t^*(\mathbf{x})\right) \mathbf{v}_t^*(\mathbf{x}), \tag{3}$$

where $\mathbf{v}_t^*(\mathbf{x}) \in \mathbb{R}^{d_2}$ is the solution to the following linear system:

$$\nabla_{\mathbf{y}}^2 g_t\left(\mathbf{x}, \mathbf{y}_t^*(\mathbf{x})\right) \mathbf{v}_t^*(\mathbf{x}) + \nabla_{\mathbf{y}} f_t\left(\mathbf{x}, \mathbf{y}_t^*(\mathbf{x})\right) = 0. \tag{4}$$

As the exact $\mathbf{y}_t^*(\mathbf{x})$ is not available, we estimate the hypergradient of $f_t$ at $(\mathbf{x}, \mathbf{y})$ and introduce an auxiliary variable $\mathbf{v} := \mathbf{v}(\mathbf{x}, \mathbf{y})$ to effectively decouple the nonlinear structure in $\nabla f_t(\mathbf{x}, \mathbf{y}_t^*(\mathbf{x}))$, i.e.

$$\tilde{\nabla} f_t(\mathbf{x}, \mathbf{y}) := \nabla_{\mathbf{x}} f_t(\mathbf{x}, \mathbf{y}) + \nabla_{\mathbf{xy}}^2 g_t\left(\mathbf{x}, \mathbf{y}\right) \mathbf{v}_t, \tag{5a}$$

where $\mathbf{v}_t$ serves as an inexact solution to the linear system

$$\nabla_{\mathbf{y}}^2 g_t\left(\mathbf{x}, \mathbf{y}\right) \mathbf{v}_t + \nabla_{\mathbf{y}} f_t(\mathbf{x}, \mathbf{y}) = 0. \tag{5b}$$

**Algorithm 1** SOGD

**Require:** $(\mathbf{x}_1, \mathbf{y}_1, \mathbf{v}_1) \in \mathcal{X} \times \mathbb{R}^{d_2} \times \mathcal{Z}_p$; $p \in \mathbb{R}_{++}$; $T \in \mathbb{N}$; stepsizes $\{(\alpha_t, \beta_t, \delta_t) \in \mathbb{R}^3_{++}\}_{t=1}^T$;
  parameters $\{(\gamma_t, \lambda_t, \eta_t)\}_{t=1}^T \in (0,1)$; $\mathbf{z}_t := (\mathbf{x}_t, \mathbf{y}_t)$.
  **For** $t = 1$ **to** $T$ **do**:
  **S1.** Draw samples $\mathcal{B}_t$ and $\bar{\mathcal{B}}_t$ with batch sizes $b$ and $\bar{b}$. Get search directions $\mathbf{d}_t^{\mathbf{y}}, \mathbf{d}_t^{\mathbf{v}}$, and $\mathbf{d}_t^{\mathbf{x}}$:

$$\mathbf{d}_t^{\mathbf{yy}}(\mathbf{z}_t; \bar{\mathcal{B}}_t) = \nabla_{\mathbf{y}} g_t(\mathbf{z}_t; \bar{\mathcal{B}}_t), \tag{9a}$$

$$\mathbf{d}_t^{\mathbf{y}} = \mathbf{d}_t^{\mathbf{yy}}(\mathbf{z}_t; \bar{\mathcal{B}}_t) + (1 - \gamma_t)(\mathbf{d}_{t-1}^{\mathbf{y}} - \mathbf{d}_t^{\mathbf{yy}}(\mathbf{z}_{t-1}; \bar{\mathcal{B}}_t)),$$

$$\mathbf{d}_t^{\mathbf{vv}}(\mathbf{z}_t; \mathcal{B}_t) = \nabla_{\mathbf{y}} f_t(\mathbf{z}_t; \mathcal{B}_t) + \nabla_{\mathbf{y}}^2 g_t(\mathbf{z}_t; \bar{\mathcal{B}}_t) \mathbf{v}_t, \tag{9b}$$

$$\mathbf{d}_t^{\mathbf{v}} = \mathbf{d}_t^{\mathbf{vv}}(\mathbf{z}_t; \mathcal{B}_t) + (1 - \lambda_t)(\mathbf{d}_{t-1}^{\mathbf{v}} - \mathbf{d}_t^{\mathbf{vv}}(\mathbf{z}_{t-1}; \mathcal{B}_t)),$$

$$\mathbf{d}_t^{\mathbf{xx}}(\mathbf{z}_t; \mathcal{B}_t) = \nabla_{\mathbf{x}} f_t(\mathbf{z}_t; \mathcal{B}_t) + \nabla_{\mathbf{xy}}^2 g_t(\mathbf{z}_t; \bar{\mathcal{B}}_t) \mathbf{v}_t, \tag{9c}$$

$$\mathbf{d}_t^{\mathbf{x}} = \mathbf{d}_t^{\mathbf{xx}}(\mathbf{z}_t; \mathcal{B}_t) + (1 - \eta_t)(\mathbf{d}_{t-1}^{\mathbf{x}} - \mathbf{d}_t^{\mathbf{xx}}(\mathbf{z}_{t-1}; \mathcal{B}_t)).$$

  **S2.** Update inner, system, and outer solutions:

$$\mathbf{y}_{t+1} = \mathbf{y}_t - \beta_t \mathbf{d}_t^{\mathbf{y}}, \quad \mathbf{v}_{t+1} = \Pi_{\mathcal{Z}_p}\big[\mathbf{v}_t - \delta_t \mathbf{d}_t^{\mathbf{v}}\big], \quad \mathbf{x}_{t+1} = \Pi_{\mathcal{X}}[\mathbf{x}_t - \alpha_t \mathbf{d}_t^{\mathbf{x}}].$$

An accurate solution of (5b) is crucial for tight regret bounds. [67] assumes an exact solution, which is restrictive in large-scale settings. To address this, [50] proposed an efficient OBO algorithm with window averaging, using CG methods to solve (5b), which is equivalent to:

$$\min_{\mathbf{v}_t \in \mathbb{R}^{d_2}} \frac{1}{2} \left\| \nabla_{\mathbf{y}}^2 g_t(\mathbf{x}, \mathbf{y}) \mathbf{v}_t + \nabla_{\mathbf{y}} f_t(\mathbf{x}, \mathbf{y}) \right\|^2. \tag{6}$$

Next, we introduce a novel search direction that enables both first- and ZO stochastic OBO algorithms to achieve sublinear bilevel regret without smoothing. We first state the following lemma:

**Lemma 2.1.** *Let* $w = t$, $W = 1/\eta$ *and* $\nu = 1 - \eta$ *for* $\eta \in (0,1)$ *in the window-smoothed gradient* $\nabla F_{t,\nu}(\mathbf{x}_t, \mathbf{y}_t; \mathcal{B}_t) = \frac{1}{W} \sum_{i=0}^{w-1} \nu^i \nabla f_{t-i}(\mathbf{x}_{t-i}, \mathbf{y}_{t-i}; \mathcal{B}_{t-i})$, *where* $\mathcal{B}_t := \{\xi_{t,1}, \ldots, \xi_{t,b}\}$ *is drawn i.i.d. from* $\mathcal{D}_{f,t}$. *Then,* $\nabla F_{t,\nu}(\mathbf{x}_t, \mathbf{y}_t; \mathcal{B}_t) = \sum_{j=1}^t \eta(1-\eta)^{t-j} \nabla f_j(\mathbf{x}_j, \mathbf{y}_j; \mathcal{B}_j)$, *and we have* $\nabla F_{t,\nu}(\mathbf{x}_t, \mathbf{y}_t; \mathcal{B}_t) = \mathbf{d}_t^{\mathbf{x}}$ *with* $\mathbf{d}_t^{\mathbf{x}} = \eta \nabla f_t(\mathbf{x}_t, \mathbf{y}_t; \mathcal{B}_t) + (1-\eta)\mathbf{d}_{t-1}^{\mathbf{x}}$, *and* $\mathbf{d}_1^{\mathbf{x}} = \frac{1}{W}\nabla f_1(\mathbf{x}_1, \mathbf{y}_1; \mathcal{B}_1)$ *for all* $t \geq 2$.

Proof is given in Appendix C.1. As shown in Lemma 2.1, for a specific choice of $w$ and $W$, the time-smoothed gradient forms a recursive momentum-type search direction. However, achieving sublinear regret in stochastic OBO requires large-window smoothing ($w = o(T)$) [67, 50, 8]. To address this, we propose the following search direction:

$$\mathbf{d}_t^{\mathbf{x}} = \eta \nabla f_t(\mathbf{x}_t, \mathbf{y}_t; \mathcal{B}_t) + (1-\eta)\mathbf{d}_{t-1}^{\mathbf{x}} + (1-\eta)(\nabla f_t(\mathbf{x}_t, \mathbf{y}_t; \mathcal{B}_t) - \nabla f_t(\mathbf{x}_{t-1}, \mathbf{y}_{t-1}; \mathcal{B}_t)). \tag{7}$$

This direction is used for updating $\mathbf{x}$, with similar updates for $\mathbf{y}$ and $\mathbf{v}$, as discussed below and detailed in Algorithm 1. The quadratic formulation of (5b) in (6) motivates single-loop methods such as [15]. Building on this, we propose Simultaneous Online Gradient Descent (SOGD) for constrained OBO, presented in Algorithm 1. At each step, SOGD jointly updates the follower variable $\mathbf{y}_t$, auxiliary variable $\mathbf{v}_t$, and leader variable $\mathbf{x}_t$ using batches $\mathcal{B}_t = \{\xi_{t,1}, \ldots, \xi_{t,b}\}$ and $\bar{\mathcal{B}}_t := \{\zeta_{t,1}, \ldots, \zeta_{t,\bar{b}}\}$ sampled i.i.d. from $\mathcal{D}_{f,t}$ and $\mathcal{D}_{g,t}$. Step **S1.** only requires computing Hessian-vector products, avoiding explicit computation of $\nabla_{\mathbf{y}}^2 g_t$ or $\nabla_{\mathbf{xy}}^2 g_t$. Step **S2.** uses the projection:

$$\Pi_{\mathcal{Z}_p}(\mathbf{v}) = \operatorname{argmin}_{\mathbf{z} \in \mathcal{Z}_p} \frac{1}{2}\|\mathbf{v} - \mathbf{z}\|^2 = \min\left\{1, \frac{p}{\|\mathbf{v}\|}\right\}\mathbf{v}, \quad \text{where}$$

$$\mathcal{Z}_p := \left\{\mathbf{v} \in \mathbb{R}^{d_2} \mid \|\mathbf{v}\| \leq p\right\}. \tag{8}$$

Unlike OAGD [67] with alternating loops, and SOBOW [50] using CG, SOGD performs a single OGD step for all variables.

**Assumption 2.2.** $g_t(\mathbf{x}, \mathbf{y})$ is twice continuously differentiable and $\mu_g$-strongly convex in $\mathbf{y}$ for all $\mathbf{x} \in \mathcal{X}, t \in [T]$.

**Assumption 2.3.** Let $\mathbf{z} = [\mathbf{x}; \mathbf{y}]$ and $\mathbf{z}' = [\mathbf{x}'; \mathbf{y}']$, where $\mathbf{x}, \mathbf{x}' \in \mathcal{X}$ and $\mathbf{y}, \mathbf{y}' \in \mathbb{R}^{d_2}$. For any $\mathbf{z}, \mathbf{z}'$, and $t \in [T]$:

B1. $\exists\, \ell_{f,0} \in \mathbb{R}_+$ s.t. $\|f_t(\mathbf{z}; \xi_t) - f_t(\mathbf{z}'; \xi_t)\| \leq \ell_{f,0} \|\mathbf{z} - \mathbf{z}'\|$;
B2. $\exists\, \ell_{f,1} \in \mathbb{R}_+$ s.t. $\|\nabla f_t(\mathbf{z}; \xi_t) - \nabla f_t(\mathbf{z}'; \xi_t)\| \leq \ell_{f,1} \|\mathbf{z} - \mathbf{z}'\|$;
B3. $\exists\, \ell_{g,1} \in \mathbb{R}_+$ s.t. $\|\nabla g_t(\mathbf{z}; \zeta_t) - \nabla g_t(\mathbf{z}'; \zeta_t)\| \leq \ell_{g,1} \|\mathbf{z} - \mathbf{z}'\|$;
B4. $\exists\, \ell_{g,2} \in \mathbb{R}_+$ s.t. $\|\nabla^2 g_t(\mathbf{z}; \zeta_t) - \nabla^2 g_t(\mathbf{z}'; \zeta_t)\| \leq \ell_{g,2} \|\mathbf{z} - \mathbf{z}'\|$.

**Assumption 2.4.** For any $t \in [T]$, $|f_t(\mathbf{x}, \mathbf{y}_t^*(\mathbf{x}))| \leq M$ for some $M \in \mathbb{R}_{++}$ and any $\mathbf{x} \in \mathcal{X}$.

**Assumption 2.5.** There exist constants $\sigma_{g_{\mathbf{y}}}, \sigma_{g_{\mathbf{yy}}}, \sigma_{g_{\mathbf{xy}}}, \sigma_{f_{\mathbf{y}}}, \sigma_{f_{\mathbf{x}}}$ such that, for all $\mathbf{z} = [\mathbf{x}, \mathbf{y}]$:

C1. $\mathbb{E}\|\nabla_{\mathbf{y}} g_t(\mathbf{z}; \zeta_t) - \nabla_{\mathbf{y}} g_t(\mathbf{z})\|^2 \leq \sigma_{g_{\mathbf{y}}}^2$;   C4. $\mathbb{E}\|\nabla_{\mathbf{y}} f_t(\mathbf{z}; \xi_t) - \nabla_{\mathbf{y}} f_t(\mathbf{z})\|^2 \leq \sigma_{f_{\mathbf{y}}}^2$;
C2. $\mathbb{E}\|\nabla_{\mathbf{y}}^2 g_t(\mathbf{z}; \zeta_t) - \nabla_{\mathbf{y}}^2 g_t(\mathbf{z})\|^2 \leq \sigma_{g_{\mathbf{yy}}}^2$;   C5. $\mathbb{E}\|\nabla_{\mathbf{x}} f_t(\mathbf{z}; \xi_t) - \nabla_{\mathbf{x}} f_t(\mathbf{z})\|^2 \leq \sigma_{f_{\mathbf{x}}}^2$.
C3. $\mathbb{E}\|\nabla_{\mathbf{xy}}^2 g_t(\mathbf{z}; \zeta_t) - \nabla_{\mathbf{xy}}^2 g_t(\mathbf{z})\|^2 \leq \sigma_{g_{\mathbf{xy}}}^2$;

Throughout this paper, we define

$$\sigma^2 := \sigma_{g_{\mathbf{y}}}^2 + \sigma_{g_{\mathbf{yy}}}^2 + \sigma_{f_{\mathbf{y}}}^2 + \sigma_{g_{\mathbf{xy}}}^2 + \sigma_{f_{\mathbf{x}}}^2. \tag{10}$$

Assumptions 2.2 and 2.3 are standard in BO [12, 43] and OBO [67], and hold for many bilevel ML problems [22]. Assumption 2.4 is typical in non-convex OSO [36, 50], while Assumption 2.5 assumes unbiased stochastic gradient, Hessian, and Jacobian estimators with bounded variance [12].

Achieving sublinear dynamic regret is generally infeasible under arbitrary time variations [7]. Prior analyses [67, 50] bound regret by enforcing regularity on the comparator sequence. To attain sublinear regret, [67] introduces the following regularity metrics for bilevel sequences:

$$H_{p,T} := \sum_{t=2}^{T} \sup_{\mathbf{x} \in \mathcal{X}} \|\mathbf{y}_{t-1}^*(\mathbf{x}) - \mathbf{y}_t^*(\mathbf{x})\|^p, \qquad V_T := \sum_{t=2}^{T} \sup_{\mathbf{x} \in \mathcal{X}} |f_{t-1}(\mathbf{x}, \mathbf{y}_{t-1}^*(\mathbf{x})) - f_t(\mathbf{x}, \mathbf{y}_t^*(\mathbf{x}))|. \tag{11}$$

Path-length $H_{p,T}$ measures changes in the follower's costs, while $V_T$ captures the leader's objective smoothness. We use path-length for the follower and function variation for the leader due to the follower's strong convexity (Assumption 2.2) versus the leader's nonconvexity. Another regularity is the sequential gradient difference of the outer objective:

$$D_{\mathbf{x},T} := \sum_{t=2}^{T} \sup_{\mathbf{x}, \mathbf{y}} \|\nabla_{\mathbf{x}} f_{t-1}(\mathbf{x}, \mathbf{y}) - \nabla_{\mathbf{x}} f_t(\mathbf{x}, \mathbf{y})\|^2, \tag{12a}$$

$$D_{\mathbf{y},T} := \sum_{t=2}^{T} \sup_{\mathbf{x}, \mathbf{y}} \|\nabla_{\mathbf{y}} f_{t-1}(\mathbf{x}, \mathbf{y}) - \nabla_{\mathbf{y}} f_t(\mathbf{x}, \mathbf{y})\|^2. \tag{12b}$$

As in [41, 31], $D_{\mathbf{x},T}$ and $D_{\mathbf{y},T}$ measure the gradient drift of $f_t$ relative to $f_{t-1}$ for $\mathbf{x}$ and $\mathbf{y}$, respectively. We define deviations in the gradient, Hessian, and Jacobian of the inner objective as:

$$G_{\mathbf{y},T} := \sum_{t=2}^{T} \|\nabla_{\mathbf{y}} g_{t-1}(\mathbf{x}_t, \mathbf{y}_t) - \nabla_{\mathbf{y}} g_t(\mathbf{x}_t, \mathbf{y}_t)\|^2, \quad G_{\mathbf{yy},T} := \sum_{t=2}^{T} \|\nabla_{\mathbf{y}}^2 g_{t-1}(\mathbf{x}_t, \mathbf{y}_t) - \nabla_{\mathbf{y}}^2 g_t(\mathbf{x}_t, \mathbf{y}_t)\|^2,$$

$$G_{\mathbf{xy},T} := \sum_{t=2}^{T} \|\nabla_{\mathbf{xy}}^2 g_{t-1}(\mathbf{x}_t, \mathbf{y}_t) - \nabla_{\mathbf{xy}}^2 g_t(\mathbf{x}_t, \mathbf{y}_t)\|^2. \tag{13}$$

We introduce the following notations for simplicity:

$$\Delta_T := E_1 + V_T, \qquad \Psi_T := H_{2,T} + G_T + D_T, \tag{14}$$

where $(V_T, H_{p,T})$ are defined in (11), and

$$E_1 := \|\mathbf{y}_1 - \mathbf{y}_1^*(\mathbf{x}_1)\|^2 + \|\mathbf{v}_1 - \mathbf{v}_1^*(\mathbf{x}_1)\|^2, \quad G_T := G_{\mathbf{y},T} + G_{\mathbf{yy},T} + G_{\mathbf{xy},T},$$
$$D_T := D_{\mathbf{x},T} + D_{\mathbf{y},T}. \tag{15}$$

By accounting for both $D_T$ and $G_T$, we can represent the variations in the environments of OBO.

**Theorem 2.6.** *Let $\{(f_t, g_t)\}_{t=1}^{T}$ be the sequence of functions presented to Algorithm 1, satisfying Assumptions 2.2-2.5. For all $t \in [T]$, let*

$$\alpha_t = \frac{1}{(c+t)^{1/3}}, \quad \beta_t = c_\beta \alpha_t, \quad \delta_t = c_\delta \alpha_t, \quad b = \bar{b} = 1,$$
$$\gamma_{t+1} = c_\gamma \alpha_t^2, \quad \eta_{t+1} = c_\eta \alpha_t^2, \quad \lambda_{t+1} = c_\lambda \alpha_t^2. \tag{16}$$

Here, $c$, $c_\beta$, $c_\delta$, $c_\gamma$, $c_\eta$, and $c_\lambda$ are specified in (107). Algorithm 1 guarantees:

$$\text{BL-Reg}_T \leq \mathcal{O}\left(T^{\frac{1}{3}}(\sigma^2 + \Delta_T) + T^{\frac{2}{3}}\Psi_T\right), \tag{17}$$

where $\sigma$ and $(\Delta_T, \Psi_T)$ are defined in (10) and (14).

*Remark* 2.7 (**Stochastic Regret Guarantee for OBO and OSO with** $w = 1$). Theorem 2.6 bounds the regret of Algorithm 1 without window-smoothing, based on the regularities in (14). We note that the average dynamic regret $\text{BL-Reg}_T/T \leq \mathcal{O}(T^{-2/3}(\sigma^2 + \Delta_T) + T^{-1/3}\Psi_T)$ remains sublinear under suitable conditions on $\Delta_T$, $\Psi_T$, and $\sigma$. Specifically, if $\Delta_T = o(T^{2/3})$, $\Psi_T = o(T^{1/3})$, and $\sigma = o(T^{1/3})$, then the dynamic regret grows sublinearly, i.e., $\text{BL-Reg}_T = o(T)$; see Appendix B.2 for further examples and discussion. This result also yields a sharper $T^{-2/3}\sigma^2$ regret—improving over the $T^{-1/2}\sigma^2$ bound for stochastic OBO [8]—and removes the need for window-smoothing [8, 67, 50, 40]. For OSO, this result surpasses the $T^{-1/2}\sigma^2$ rate in [31].

## 3 Stochastic OBO with Zeroth-Order Oracles

Black-box optimization arises when gradients are unavailable [11]. We study ZO-OBO methods with limited access to leader and follower objectives. Let $\mathbf{s} \in \mathbb{R}^{d_1}$ and $\mathbf{r} \in \mathbb{R}^{d_2}$ be vectors uniformly sampled from unit balls $B_1$ and $B_2$. Given smoothing parameters $\boldsymbol{\rho} = (\rho_\mathbf{s}, \rho_\mathbf{r})$, we define Gaussian-smoothed objectives using [59]:

$$f_{t,\boldsymbol{\rho}}\left(\mathbf{x}, \hat{\mathbf{y}}_t^*(\mathbf{x})\right) = \underset{(\mathbf{s},\mathbf{r},\xi_t)}{\mathbb{E}}\left[f_t(\mathbf{x} + \rho_\mathbf{s}\mathbf{s}, \hat{\mathbf{y}}_t^*(\mathbf{x}) + \rho_\mathbf{r}\mathbf{r}; \xi_t)\right], \quad \text{where} \tag{18}$$

$$\hat{\mathbf{y}}_t^*(\mathbf{x}) \in \underset{\mathbf{y} \in \mathbb{R}^{d_2}}{\arg\min}\left\{g_{t,\boldsymbol{\rho}}(\mathbf{x}, \mathbf{y}) := \underset{(\mathbf{s},\mathbf{r},\zeta_t)}{\mathbb{E}}\left[g_t(\mathbf{x} + \rho_\mathbf{s}\mathbf{s}, \mathbf{y} + \rho_\mathbf{r}\mathbf{r}; \zeta_t)\right]\right\}. \tag{19}$$

To solve stochastic OBO with (18), we need to obtain the hyper-gradient of $f_{t,\boldsymbol{\rho}}$ in (18) at $(\mathbf{x}, \mathbf{y})$ as

$$\nabla f_{t,\boldsymbol{\rho}}(\mathbf{x}, \hat{\mathbf{y}}_t^*(\mathbf{x})) := \nabla_\mathbf{x} f_{t,\boldsymbol{\rho}}(\mathbf{x}, \hat{\mathbf{y}}_t^*(\mathbf{x})) + \nabla_{\mathbf{xy}}^2 g_{t,\boldsymbol{\rho}}\left(\mathbf{x}, \hat{\mathbf{y}}_t^*(\mathbf{x})\right)\hat{\mathbf{v}}_t^*(\mathbf{x}), \quad \text{where}$$

$$\hat{\mathbf{v}}_t^*(\mathbf{x}) \text{ is the solution to } \nabla_\mathbf{y}^2 g_{t,\boldsymbol{\rho}}\left(\mathbf{x}, \hat{\mathbf{y}}_t^*(\mathbf{x})\right)\hat{\mathbf{v}}_t^*(\mathbf{x}) + \nabla_\mathbf{y} f_{t,\boldsymbol{\rho}}(\mathbf{x}, \hat{\mathbf{y}}_t^*(\mathbf{x})) = 0. \tag{20}$$

Obtaining $\hat{\mathbf{y}}_t^*(\mathbf{x})$ in closed-form is usually a challenging task, so it is natural to use the following gradient surrogate. At any $(\mathbf{x}, \mathbf{y})$, we introduce an auxiliary variable $\mathbf{v} = \mathbf{v}(\mathbf{x}, \mathbf{y})$ and define:

$$\tilde{\nabla} f_{t,\boldsymbol{\rho}}(\mathbf{x}, \mathbf{y}) := \nabla_\mathbf{x} f_{t,\boldsymbol{\rho}}(\mathbf{x}, \mathbf{y}) + \nabla_{\mathbf{xy}}^2 g_{t,\boldsymbol{\rho}}(\mathbf{x}, \mathbf{y})\mathbf{v}, \quad \text{where} \tag{21a}$$

$$\mathbf{v} \text{ is the solution to } \nabla_\mathbf{y}^2 g_{t,\boldsymbol{\rho}}(\mathbf{x}, \mathbf{y})\mathbf{v} + \nabla_\mathbf{y} f_{t,\boldsymbol{\rho}}(\mathbf{x}, \mathbf{y}) = 0. \tag{21b}$$

To do so, we also introduce $\mathbf{d}_{t,\boldsymbol{\rho}}^\mathbf{y}$, $\mathbf{d}_{t,\boldsymbol{\rho}}^\mathbf{v}$ and $\mathbf{d}_{t,\boldsymbol{\rho}}^\mathbf{x}$ as follows:

$$\mathbf{d}_{t,\boldsymbol{\rho}}^\mathbf{y}(\mathbf{x}, \mathbf{y}) = \nabla_\mathbf{y} g_{t,\boldsymbol{\rho}}(\mathbf{x}, \mathbf{y}), \tag{22a}$$

$$\mathbf{d}_{t,\boldsymbol{\rho}}^\mathbf{v}(\mathbf{x}, \mathbf{y}, \mathbf{v}) = \nabla_\mathbf{y} f_{t,\boldsymbol{\rho}}(\mathbf{x}, \mathbf{y}) + \nabla_\mathbf{y}^2 g_{t,\boldsymbol{\rho}}(\mathbf{x}, \mathbf{y})\mathbf{v}, \tag{22b}$$

$$\mathbf{d}_{t,\boldsymbol{\rho}}^\mathbf{x}(\mathbf{x}, \mathbf{y}, \mathbf{v}) = \nabla_\mathbf{x} f_{t,\boldsymbol{\rho}}(\mathbf{x}, \mathbf{y}) + \nabla_{\mathbf{xy}}^2 g_{t,\boldsymbol{\rho}}(\mathbf{x}, \mathbf{y})\mathbf{v}. \tag{22c}$$

Next, we approximate these directions using stochastic zeroth-order oracles (*SZO*), which produce the quantities $\hat{\nabla}_\mathbf{y} f_t(\mathbf{x}, \mathbf{y}; \xi_t)$, $\hat{\nabla}_\mathbf{y} g_t(\mathbf{x}, \mathbf{y}; \zeta_t)$, $\hat{\nabla}_\mathbf{x} f_t(\mathbf{x}, \mathbf{y}; \xi_t)$, and $\hat{\nabla}_\mathbf{x} g_t(\mathbf{x}, \mathbf{y}; \zeta_t)$. These are unbiased estimators of the true gradients $\nabla_\mathbf{y} f_{t,\boldsymbol{\rho}}(\mathbf{x}, \mathbf{y})$, $\nabla_\mathbf{y} g_{t,\boldsymbol{\rho}}(\mathbf{x}, \mathbf{y})$, $\nabla_\mathbf{x} f_{t,\boldsymbol{\rho}}(\mathbf{x}, \mathbf{y})$, and $\nabla_\mathbf{x} g_{t,\boldsymbol{\rho}}(\mathbf{x}, \mathbf{y})$, respectively, as shown in [20], such that the following assumption holds:

$$\underset{(\mathbf{r},\xi_t)}{\mathbb{E}}\left[\hat{\nabla}_\mathbf{y} f_t(\mathbf{x}, \mathbf{y}; \xi_t)\right] = \nabla_\mathbf{y} f_{t,\boldsymbol{\rho}}(\mathbf{x}, \mathbf{y}), \quad \underset{(\mathbf{s},\xi_t)}{\mathbb{E}}\left[\hat{\nabla}_\mathbf{x} f_t(\mathbf{x}, \mathbf{y}; \xi_t)\right] = \nabla_\mathbf{x} f_{t,\boldsymbol{\rho}}(\mathbf{x}, \mathbf{y}),$$

$$\underset{(\mathbf{r},\zeta_t)}{\mathbb{E}}\left[\hat{\nabla}_\mathbf{y} g_t(\mathbf{x}, \mathbf{y}; \zeta_t)\right] = \nabla_\mathbf{y} g_{t,\boldsymbol{\rho}}(\mathbf{x}, \mathbf{y}), \quad \underset{(\mathbf{s},\zeta_t)}{\mathbb{E}}\left[\hat{\nabla}_\mathbf{x} g_t(\mathbf{x}, \mathbf{y}; \zeta_t)\right] = \nabla_\mathbf{x} g_{t,\boldsymbol{\rho}}(\mathbf{x}, \mathbf{y}). \tag{23}$$

Specifically, following [62], we estimate the gradient of a function $h : \mathbb{R}^d \to \mathbb{R}$, querying at $\mathbf{x} - \lambda\mathbf{s}$ and $\mathbf{x} + \lambda\mathbf{s}$, yielding an estimator $(d/2\lambda)\left(h(\mathbf{x} + \lambda\mathbf{s}) - h(\mathbf{x} - \lambda\mathbf{s})\right)\mathbf{s}$. Using this strategy, the finite-difference estimation of $\nabla g_{t,\boldsymbol{\rho}}(\mathbf{x}, \mathbf{y})$, denoted by $\hat{\nabla} g_t(\mathbf{x}, \mathbf{y})$, is constructed for given smoothing

**Algorithm 2** ZO-SOGD

**Require:** In addition to parameters in SOGD, choose $\rho_{\mathbf{v}}, \rho_{\mathbf{r}}, \rho_{\mathbf{s}} \in \mathbb{R}_{++}$.
**For** $t = 1$ **to** $T$ **do**:

    **S1.** Draw samples $\mathcal{B}_t$ and $\bar{\mathcal{B}}_t$ with batch sizes $b$ and $\bar{b}$. Using (24)–(26), get:

$$\mathbf{d}_t^{\mathbf{y}}\left(\mathbf{z}_t; \bar{\mathcal{B}}_t\right) = \hat{\nabla}_{\mathbf{y}} g_t(\mathbf{z}_t; \bar{\mathcal{B}}_t), \tag{27a}$$

$$\hat{\mathbf{d}}_t^{\mathbf{y}} = \mathbf{d}_t^{\mathbf{y}}(\mathbf{z}_t; \bar{\mathcal{B}}_t) + (1 - \gamma_t)(\hat{\mathbf{d}}_{t-1}^{\mathbf{y}} - \mathbf{d}_t^{\mathbf{y}}(\mathbf{z}_{t-1}; \bar{\mathcal{B}}_t)),$$

$$\mathbf{d}_t^{\mathbf{vv}}\left(\mathbf{z}_t; \mathcal{B}_t\right) = \hat{\nabla}_{\mathbf{y}} f_t\left(\mathbf{z}_t; \mathcal{B}_t\right) + \hat{\nabla}_{\mathbf{y}}^2 g_t\left(\mathbf{z}_t; \bar{\mathcal{B}}_t\right), \tag{27b}$$

$$\hat{\mathbf{d}}_t^{\mathbf{v}} = \mathbf{d}_t^{\mathbf{vv}}(\mathbf{z}_t; \mathcal{B}_t) + (1 - \lambda_t)(\hat{\mathbf{d}}_{t-1}^{\mathbf{v}} - \mathbf{d}_t^{\mathbf{vv}}(\mathbf{z}_{t-1}; \mathcal{B}_t)),$$

$$\mathbf{d}_t^{\mathbf{xy}}\left(\mathbf{z}_t; \mathcal{B}_t\right) = \hat{\nabla}_{\mathbf{x}} f_t\left(\mathbf{z}_t; \mathcal{B}_t\right) + \hat{\nabla}_{\mathbf{xy}}^2 g_t\left(\mathbf{z}_t; \bar{\mathcal{B}}_t\right), \tag{27c}$$

$$\hat{\mathbf{d}}_t^{\mathbf{x}} = \mathbf{d}_t^{\mathbf{xy}}(\mathbf{z}_t; \mathcal{B}_t) + (1 - \eta_t)(\hat{\mathbf{d}}_{t-1}^{\mathbf{x}} - \mathbf{d}_t^{\mathbf{xy}}(\mathbf{z}_{t-1}; \mathcal{B}_t)),$$

    **S2.** Update inner, system, and outer solutions:

$$\mathbf{y}_{t+1} = \mathbf{y}_t - \beta_t \hat{\mathbf{d}}_t^{\mathbf{y}}, \quad \mathbf{v}_{t+1} = \Pi_{\mathcal{Z}_p}\left[\mathbf{v}_t - \delta_t \hat{\mathbf{d}}_t^{\mathbf{v}}\right], \quad \mathbf{x}_{t+1} = \Pi_{\mathcal{X}}\left[\mathbf{x}_t - \alpha_t \hat{\mathbf{d}}_t^{\mathbf{x}}\right].$$

parameters $\boldsymbol{\rho} = (\rho_{\mathbf{s}}, \rho_{\mathbf{r}})$, and a set $\bar{\mathcal{B}}_t = \{\zeta_{t,1}, \ldots, \zeta_{t,\bar{b}}\}$ drawn i.i.d. from $\mathcal{D}_{g,t}$, as follows:

$$\hat{\nabla}_{\mathbf{y}} g_t(\mathbf{x}, \mathbf{y}; \bar{\mathcal{B}}_t) := \frac{d_2}{2\bar{b}\rho_{\mathbf{r}}} \sum_{i=1}^{\bar{b}} \left(g_t(\mathbf{x}, \mathbf{y} + \rho_{\mathbf{r}}\mathbf{r}_i; \zeta_{t,i}) - g_t(\mathbf{x}, \mathbf{y} - \rho_{\mathbf{r}}\mathbf{r}_i; \zeta_{t,i})\right)\mathbf{r}_i, \tag{24a}$$

$$\hat{\nabla}_{\mathbf{x}} g_t(\mathbf{x}, \mathbf{y}; \bar{\mathcal{B}}_t) := \frac{d_1}{2\bar{b}\rho_{\mathbf{s}}} \sum_{i=1}^{\bar{b}} \left(g_t(\mathbf{x} + \rho_{\mathbf{s}}\mathbf{s}_i, \mathbf{y}; \zeta_{t,i}) - g_t(\mathbf{x} - \rho_{\mathbf{s}}\mathbf{s}_i, \mathbf{y}; \zeta_{t,i})\right)\mathbf{s}_i. \tag{24b}$$

Similarly, we estimate $\nabla_{\mathbf{y}} f_{t,\boldsymbol{\rho}}(\mathbf{x}, \mathbf{y}; \mathcal{B}_t)$ and $\nabla_{\mathbf{x}} f_{t,\boldsymbol{\rho}}(\mathbf{x}, \mathbf{y}; \mathcal{B}_t)$, respectively, using a batch $\mathcal{B}_t = \{\xi_{t,1}, \ldots, \xi_{t,b}\}$ drawn i.i.d. from $\mathcal{D}_{f,t}$, by

$$\hat{\nabla}_{\mathbf{y}} f_t(\mathbf{x}, \mathbf{y}; \mathcal{B}_t) := \frac{d_2}{2b\rho_{\mathbf{r}}} \sum_{i=1}^{b} (f_t(\mathbf{x}, \mathbf{y} + \rho_{\mathbf{r}}\mathbf{r}_i; \xi_{t,i}) - f_t(\mathbf{x}, \mathbf{y} - \rho_{\mathbf{r}}\mathbf{r}_i; \xi_{t,i}))\mathbf{r}_i, \tag{25a}$$

$$\hat{\nabla}_{\mathbf{x}} f_t(\mathbf{x}, \mathbf{y}; \mathcal{B}_t) := \frac{d_1}{2b\rho_{\mathbf{s}}} \sum_{i=1}^{b} (f_t(\mathbf{x} + \rho_{\mathbf{s}}\mathbf{s}_i, \mathbf{y}; \xi_{t,i}) - f_t(\mathbf{x} - \rho_{\mathbf{s}}\mathbf{s}_i, \mathbf{y}; \xi_{t,i}))\mathbf{s}_i. \tag{25b}$$

Furthermore, given a smoothing parameter $\rho_{\mathbf{v}} > 0$, we approximate the Hessian-vector product $\nabla_{\mathbf{y}}^2 g_{t,\boldsymbol{\rho}}(\mathbf{x}, \mathbf{y})\mathbf{v}$ and the Jacobian-vector product $\nabla_{\mathbf{xy}}^2 g_{t,\boldsymbol{\rho}}(\mathbf{x}, \mathbf{y})\mathbf{v}$ as the finite difference between two gradients, respectively, as

$$\hat{\nabla}_{\mathbf{y}}^2 g_t(\mathbf{x}, \mathbf{y}; \bar{\mathcal{B}}_t) := \frac{1}{2\bar{b}\rho_{\mathbf{v}}} \sum_{i=1}^{\bar{b}} (\hat{\nabla}_{\mathbf{y}} g_t(\mathbf{x}, \mathbf{y} + \rho_{\mathbf{v}}\mathbf{v}; \zeta_{t,i}) - \hat{\nabla}_{\mathbf{y}} g_t(\mathbf{x}, \mathbf{y} - \rho_{\mathbf{v}}\mathbf{v}; \zeta_{t,i})), \tag{26a}$$

$$\hat{\nabla}_{\mathbf{xy}}^2 g_t(\mathbf{x}, \mathbf{y}; \bar{\mathcal{B}}_t) := \frac{1}{2\bar{b}\rho_{\mathbf{v}}} \sum_{i=1}^{\bar{b}} (\hat{\nabla}_{\mathbf{x}} g_t(\mathbf{x}, \mathbf{y} + \rho_{\mathbf{v}}\mathbf{v}; \zeta_{t,i}) - \hat{\nabla}_{\mathbf{x}} g_t(\mathbf{x}, \mathbf{y} - \rho_{\mathbf{v}}\mathbf{v}; \zeta_{t,i})). \tag{26b}$$

Using (24)–(26), the first-order terms in (9) are approximated by $\hat{\mathbf{d}}_t^{\mathbf{y}}$, $\hat{\mathbf{d}}_t^{\mathbf{v}}$, and $\hat{\mathbf{d}}_t^{\mathbf{x}}$ in (27). The approximations in (26a) and (26b) introduce errors in the hypergradient, which must be controlled. (26) depends on the dimension of $\mathbf{y}$, as in ZO optimization [59, 62]. The projection $\Pi_{\mathcal{Z}_p}$ in (8) bounds $\mathbf{v}$, controlling variance in $\mathbf{v}$ and $\mathbf{x}$ updates for convergence.

**Assumption 3.1.** There exist constants $\hat{\sigma}_{g_{\mathbf{y}}}, \hat{\sigma}_{g_{\mathbf{x}}}, \hat{\sigma}_{f_{\mathbf{y}}}, \hat{\sigma}_{f_{\mathbf{x}}}$ such that, for all $\mathbf{z} = [\mathbf{x}, \mathbf{y}]$:

D1. $\mathbb{E}\|\hat{\nabla}_{\mathbf{y}} g_t(\mathbf{z}; \zeta_t) - \nabla_{\mathbf{y}} g_{t,\boldsymbol{\rho}}(\mathbf{z})\|^2 \leq \hat{\sigma}_{g_{\mathbf{y}}}^2$,     D3. $\mathbb{E}\|\hat{\nabla}_{\mathbf{y}} f_t(\mathbf{z}; \xi_t) - \nabla_{\mathbf{y}} f_{t,\boldsymbol{\rho}}(\mathbf{z})\|^2 \leq \hat{\sigma}_{f_{\mathbf{y}}}^2$,

D2. $\mathbb{E}\|\hat{\nabla}_{\mathbf{x}} g_t(\mathbf{z}; \zeta_t) - \nabla_{\mathbf{x}} g_{t,\boldsymbol{\rho}}(\mathbf{z})\|^2 \leq \hat{\sigma}_{g_{\mathbf{x}}}^2$,     D4. $\mathbb{E}\|\hat{\nabla}_{\mathbf{x}} f_t(\mathbf{z}; \xi_t) - \nabla_{\mathbf{x}} f_{t,\boldsymbol{\rho}}(\mathbf{z})\|^2 \leq \hat{\sigma}_{f_{\mathbf{x}}}^2$.

Assumption 3.1 is analogous to the upper bound on the variance of stochastic partial gradients discussed in [54, 68]. We simplify the notation by introducing the following shorthand.

$$\hat{\sigma}^2 := \hat{\sigma}_{g_{\mathbf{y}}}^2 + \hat{\sigma}_{g_{\mathbf{x}}}^2 + \hat{\sigma}_{f_{\mathbf{y}}}^2 + \hat{\sigma}_{f_{\mathbf{x}}}^2. \tag{28}$$

Next, we establish a regret bound for ZO-SOGD. Similar to the previous results, we introduce regularity conditions for the smoothed functions defined in (18) and (19).

**Inner Gradient Variations**: In ZO setting, we use a set of gradient variations at the perturbed point as follows:

$$G_{\mathbf{v},T} := \sum_{t=2}^{T} (\chi_{1t} + \chi_{2t}), \qquad G_{\mathbf{x},T} := \sum_{t=2}^{T} (\chi_{3t} + \chi_{4t}), \tag{29}$$

where $\mathbf{z}_t^+ := (\mathbf{x}_{t-1}, \mathbf{y}_{t-1} + \rho_{\mathbf{v}} \mathbf{v}_{t-1})$, $\mathbf{z}_t^- := (\mathbf{x}_{t-1}, \mathbf{y}_{t-1} - \rho_{\mathbf{v}} \mathbf{v}_{t-1})$, and

$$\chi_{1t} := \|\nabla_{\mathbf{y}} g_t(\mathbf{z}_t^+) - \nabla_{\mathbf{y}} g_{t-1}(\mathbf{z}_t^+)\|^2, \quad \chi_{2t} := \|\nabla_{\mathbf{y}} g_t(\mathbf{z}_t^-) - \nabla_{\mathbf{y}} g_{t-1}(\mathbf{z}_t^-)\|^2,$$

$$\chi_{3t} := \|\nabla_{\mathbf{x}} g_t(\mathbf{z}_t^+) - \nabla_{\mathbf{x}} g_{t-1}(\mathbf{z}_t^+)\|^2, \quad \chi_{4t} := \|\nabla_{\mathbf{x}} g_t(\mathbf{z}_t^-) - \nabla_{\mathbf{x}} g_{t-1}(\mathbf{z}_t^-)\|^2.$$

Further, for simplicity of notation, we define

$$\hat{\Delta}_T := E_1 + V_T + D_T + G_{\mathbf{y},T}, \qquad \hat{\Psi}_T := H_{2,T} + G_{\mathbf{v},T} + G_{\mathbf{x},T}, \tag{30}$$

where $(V_T, H_{p,T})$ and $(E_1, D_T)$ are defined in (11), and (15), respectively. Moreover, $G_{\mathbf{y},T}$ and $(G_{\mathbf{v},T}, G_{\mathbf{x},T})$ are defined in (13) and (29), respectively.

**Theorem 3.2.** *Let $\{(f_t, g_t)\}_{t=1}^{T}$ be the sequence of functions presented to Algorithm 2, satisfying Assumptions 2.2-2.4 and 3.1. For all $t \in [T]$, let*

$$\alpha_t = \frac{1}{(d_1 + d_2)^{3/4}(c+t)^{1/3}}, \quad \beta_t = c_\beta \alpha_t, \quad \delta_t = c_\delta \alpha_t, \quad \gamma_{t+1} = c_\gamma \alpha_t,$$

$$\eta_{t+1} = c_\eta \alpha_t, \quad \lambda_{t+1} = c_\lambda \alpha_t, \quad \rho_{\mathbf{v}}^2 = c_{\mathbf{v}} \alpha_t, \quad \rho_{\mathbf{r}}^2 = \frac{1}{d_2^2 T}, \quad \rho_{\mathbf{s}}^2 = \frac{1}{d_1^2 T},$$

$$b = \frac{T^{1/3}}{(d_1 + d_2)^{3/2}}, \quad \bar{b} = \frac{T^{2/3}}{(d_1 + d_2)^{3/4}}, \tag{31}$$

*where $c$, $c_\beta$, $c_\delta$, $c_\gamma$, $c_\eta$, $c_{\mathbf{v}}$, and $c_\lambda$ are specified in (226). Let $p = \ell_{f,0}/\mu_g$ for the set $\mathcal{Z}_p$ defined in (8). Then, Algorithm 2 guarantees:*

$$\text{BL-Reg}_T \leq \mathcal{O}\left((d_1 + d_2)^{\frac{3}{4}} T^{\frac{1}{3}} \left(\hat{\sigma}^2 + \hat{\Delta}_T\right) + (d_1 + d_2)^{\frac{3}{2}} T^{\frac{2}{3}} \hat{\Psi}_T\right).$$

*where $\hat{\sigma}^2$ and $(\hat{\Delta}_T, \hat{\Psi}_T)$ are defined in (28) and (30).*

Theorem 3.2 bounds the regret of Algorithm 2 without window-smoothing, based on the regularities in (30). We note that the average dynamic regret $\text{BL-Reg}_T/T \leq \mathcal{O}((d_1 + d_2)^{3/4} T^{-2/3} \left(\hat{\sigma}^2 + \hat{\Delta}_T\right) + (d_1 + d_2)^{3/2} T^{-1/3} \hat{\Psi}_T)$ remains sublinear under suitable conditions on $\hat{\Delta}_T$, $\hat{\Psi}_T$, and $\hat{\sigma}$.

*Remark* 3.3 (**Regret Guarantee for Zeroth Order OBO**). Theorem 3.2 provides the first regret guarantee for OBO with access only to noisy function evaluations of the leader and follower. The dimensional dependence $\mathcal{O}(d_1 + d_2)$ in Theorem 3.2 aligns with optimal results for simpler offline min-max problems [39]. The bound also depends on the sample sizes $b, \bar{b}$ and smoothing parameters $\rho_{\mathbf{v}}, \rho_{\mathbf{r}}, \rho_{\mathbf{s}}$ at each iteration.

*Remark* 3.4 (**Improved Regret for OSO**). Our dynamic regret for single-level non-stationary optimization is $\mathcal{O}((d_1 + d_2)^{3/4} T^{-2/3} (\hat{\sigma}^2 + E_1 + V_T + D_T))$, improving the result in [60], which is $\mathcal{O}(T^{-1/2} \sigma^2 \sqrt{d})$. [60] proposed a zeroth-order stochastic gradient descent algorithm for unconstrained, non-convex, time-varying objective functions, achieving a regret bound of $\mathcal{O}(T^{-1/2} \sigma^2 \sqrt{d W_T})$ using a two-point gradient estimator, where $W_T$ bounds the nonstationarity. Additionally, [29] showed that the local regret for standard online stochastic gradient descent with the standard two-point gradient estimator [1] is $\mathcal{O}(T^{-1/2} d \sqrt{V_T})$.

# 4 Experimental Results

In this section, we present experimental results for two applications: online black-box attacks on deep neural networks and parametric loss tuning for imbalanced data. Code is available at ⭕. Additional experiments and details on hyperparameter tuning are provided in Appendix E.

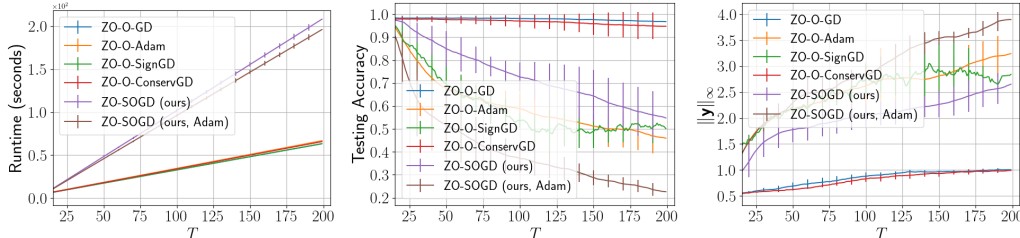

Figure 2: Performance comparison (mean±std) of optimizers including ZO-O-GD, ZO-O-Adam, ZO-O-SignGD, ZO-O-ConservSGD, ZO-SOGD, and ZO-SOGD (Adam) on **online adversarial attack** for MNIST data across five runs.

**Bilevel Optimization for Black-Box Adversarial Attacks (BBAA)**   Deep neural networks are vulnerable to adversarial examples—inputs subtly perturbed to mislead classifiers. These examples can fool models without access to their internals, as in [11, 52, 13]. We first review the ZO single-level formulation for BBAA [11]. Let $(\mathbf{a}, b)$ be a clean image $\mathbf{a} \in \mathbb{R}^d$ with label $b \in \{1, \ldots, J\}$, and define $\mathbf{a}' = \mathbf{a} + \mathbf{y}$, where $\mathbf{y}$ is the adversarial perturbation. Let $\mathcal{Y} := [-5, 5]^d$, and $\ell : \mathbb{R}^d \to \mathbb{R}$ be the black-box attack loss. For a given hyperparameter $\lambda > 0$, the BBAA problem is:

$$\min_{\mathbf{y} \in \mathcal{Y}} \frac{1}{m} \sum_{i=1}^{m} \ell(\mathbf{a}_i + \mathbf{y}) + \lambda \|\mathbf{y}\|^2. \tag{32}$$

To adapt (32) to our OBO, consider OBO for supervised learning: at each timestep $t$, new samples $(\mathbf{a}_t, b_t) \in \mathcal{D}_t := \{\mathcal{D}_t^{\text{val}}, \mathcal{D}_t^{\text{tr}}\}$ are received, where $\mathbf{a}_t \in \mathbb{R}^{d_2}$ is the feature vector (image) and $b_t \in \mathbb{R}$ is the corresponding target. Note that the correct decision can change abruptly. We consider an $S$-stage scenario where $(\mathbf{x}_s^*, \mathbf{y}_s^*(\mathbf{x}_s^*))$ represents the best decisions for the $s$-th stage, for all $s \in [S]$:

$$\mathbf{x}_s^* \in \operatorname*{argmin}_{\mathbf{x} \in \mathcal{X}} \sum_{t=1}^{T_s} f\left(\mathbf{y}_s^*(\mathbf{x}); \mathcal{D}_t^{\text{val}}\right) \quad \text{s.t.} \quad \mathbf{y}_s^*(\mathbf{x}) \in \operatorname*{argmin}_{\mathbf{y} \in \mathcal{Y}} \sum_{t=1}^{T_s} g\left(\mathbf{x}, \mathbf{y}; \mathcal{D}_t^{\text{tr}}\right) \tag{33}$$

$$g(\mathbf{x}_t, \mathbf{y}_t; \mathcal{D}_t^{\text{tr}}) = \frac{1}{|\mathcal{D}_t^{\text{tr}}|} \sum_{i \in \mathcal{D}_t^{\text{tr}}} \ell(\mathbf{a}_t^{(i)} + \mathbf{y}_t) + \frac{1}{2} \sum_{\iota=1}^{p} e^{[\mathbf{x}_t]_\iota} [\mathbf{y}_t]_\iota^2,$$

$$f(\mathbf{y}_t(\mathbf{x}_t); \mathcal{D}_t^{\text{val}}) = \frac{1}{|\mathcal{D}_t^{\text{val}}|} \sum_{i \in \mathcal{D}_t^{\text{val}}} \ell(\mathbf{a}_t^{(i)} + \mathbf{y}_t). \tag{34a}$$

Here, $\{\mathbf{a}_t^{(i)}\}_{i \in \mathcal{D}_t^{\text{tr}}}$ and $\{\mathbf{a}_t^{(i)}\}_{i \in \mathcal{D}_t^{\text{val}}}$ are batches of training and validation samples at timestep $t$; $\mathbf{a}_t^{(i)}$ is the $i$th sample in that batch; and $[\mathbf{x}_t]_\iota$ and $[\mathbf{y}_t]_\iota$ denote the $\iota$th component of $\mathbf{x}_t$ and $\mathbf{y}_t$, respectively.

We normalize the pixel values to $\mathcal{Y}$. For an untargeted attack, the loss in (34) is $\ell(\mathbf{a}_t') = \max\{Z(\mathbf{a}_t')_{b_t} - \max_{j \neq b_t} Z(\mathbf{a}_t')_j, -\kappa\}$, where $Z(\mathbf{a}_t')_j$ is the prediction score for class $j$ given input $\mathbf{a}_t' = \mathbf{a}_t + \mathbf{y}_t$, and $\kappa > 0$ controls the confidence gap. In our experiments, we set $\kappa = 0$. Eq. (33) introduces the first OBO formulation of BBAA. Using a vector $\mathbf{x} \in \mathbb{R}_+^d$ for hyperparameters instead of $\lambda \in \mathbb{R}_{++}$ in (32) enables finer control over model components, enhancing performance for complex models and heterogeneous data [53]. For a fair comparison with single-level BBAA, we replace $\lambda$ with a fixed vector multiplied by each component of $\mathbf{y}$ in (32). We compare our ZO-SOGD and ZO-SOGD (Adam) with the following competing methods in the online setting: **ZO-O-GD**, a single-level method that updates $\mathbf{y}_t$ with a fixed $\mathbf{x}$ at each timestep using ZO gradient descent [59]; **ZO-O-Adam**, a single-level method that updates $\mathbf{y}_t$ with a fixed $\mathbf{x}$ at each timestep using ZO Adam [45, 13]; **ZO-O-SignSGD**, a single-level method that updates $\mathbf{y}_t$ with a fixed $\mathbf{x}$ at each timestep using ZO SignSGD [6]; and **ZO-O-ConservSGD**, a single-level method that updates $\mathbf{y}_t$ with a fixed $\mathbf{x}$ at each timestep using ZO Conservative SGD [44]. Note that ZO-SOGD (ours, Adam) is a variant of our algorithm with an adaptive stepsize, similar to that of [45].

We evaluated the proposed algorithms based on runtime, test accuracy on perturbed samples, and the infinity norm of $\mathbf{y}_t$. Figure 2 compares the methods. The left panel shows that ZO-SOGD has a slower runtime than single-level baselines due to outer-level optimization on $\mathbf{x}$. The middle panel illustrates that accuracy decreases as the adversarial attack $\mathbf{y}$ strengthens, with ZO-SOGD outperforming ZO-O-GD and ZO-O-ConservGD, while ZO-SOGD (Adam) surpasses ZO-O-Adam and all baselines. The right panel indicates that the infinity norm of $\mathbf{y}_t$ increases over time for all methods, reducing accuracy. However, perturbations remain minor, with $\max \mathbf{y}_t$ not exceeding 4, demonstrating that ZO-SOGD achieves effective attacks with superior performance.

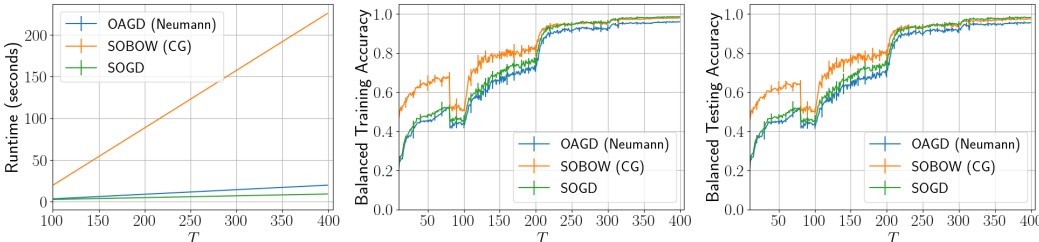

Figure 3: Performance (mean±std) on **online parametric loss tuning** with distribution shift on MNIST across five runs, comparing OGD [72], OAGD [67], SOBOW [50], and our SOGD.

**Parametric Loss Tuning for Imbalanced Data** Imbalanced datasets are common in modern machine learning, causing challenges in generalization and fairness due to underrepresented classes and sensitive attributes. Deep NNs often overfit, seeming accurate and fair during training but performing poorly during testing. A common solution is designing a parametric training loss that balances accuracy and fairness while preventing overfitting [49]. We consider an optimization problem similar to that in (33). For a new sample $(\mathbf{a}_t, b_t)$, the follower and leader incur a parametric and balanced cross-entropy loss, respectively:

$$g(\mathbf{x}_t, \mathbf{y}_t; \mathcal{D}_t^{\mathrm{tr}}) = -\log \frac{e^{\gamma_{b_t}[\mathbf{y}_t(\mathbf{a}_t)]_{b_t} + \Delta_{b_t}}}{\sum_{j=1}^{J} e^{\gamma_j [\mathbf{y}_t(\mathbf{a}_t)]_j + \Delta_j}}, \quad f(\mathbf{y}_t(\mathbf{x}_t); \mathcal{D}_t^{\mathrm{val}}) = -u_{b_t} \log \frac{e^{[\mathbf{y}_t(\mathbf{a}_t)]_{b_t}}}{\sum_{j=1}^{J} e^{[\mathbf{y}_t(\mathbf{a}_t)]_j}}. \quad (35)$$

Here, $\mathbf{x}_t := (\Delta_j, \gamma_j)_{j=1}^{J}$ represents the logit adjustments, with $j$ indexing the $J$ classes, and $u_j$ is the reciprocal of the proportion of samples from the $j$-th class to the total number of samples [49].

In (35), $\mathbf{y}_t(\mathbf{x}_t)$ is the follower conditioned on the leader, and $[\mathbf{y}_t(\mathbf{a}_t)]_{b_t}$ is the logit for class $b_t$ on sample $\mathbf{a}_t$. The follower $\mathbf{y}_t$ uses a 4-layer CNN, inducing a nonconvex bilevel objective. We compare SOGD with **OAGD** [67], a static method using the Neumann series, and **SOBOW** [50], a dynamic method using conjugate gradients (CG). Experiments were conducted on MNIST [48] with batch size 64. We evaluated cumulative runtime, test accuracy, and balanced accuracy, defined as $\frac{1}{J} \sum_{j=1}^{J} \mathbb{P}_{\mathbf{a}_t \sim \mathcal{D}_j} [\mathrm{argmax}_i([\mathbf{y}_t(\mathbf{a}_t)]_i) = j]$, where $\mathcal{D}_j$ is the class-$j$ sample distribution [49]. Learning rates were tuned as $\beta_t = \delta_t = \beta \in \{0.001, 0.005, 0.01, 0.05, 0.1\}$, $\alpha_t = \alpha \in \{0.0001, 0.0005, 0.001, 0.005, 0.01\}$, and $\gamma_t = \lambda_t = \eta_t = \gamma \in \{0.9, 0.99, 0.999\}$. Both OAGD and SOBOW used 5 iterations for their respective system solvers.

We evaluated performance over 400 timesteps in four 100-timestep phases, transitioning from an imbalanced $(0.4^i)$ to a balanced $(0.8^i)$ distribution for each class $(i = 0, 1, \ldots, 9)$. Figure 3 (left) shows SOBOW's longer runtime due to CG complexity, while SOGD is the fastest with simultaneous updates. Figures 3 (middle, right) show accuracy gains as balance increases, with SOGD achieving competitive accuracy.

## 5 Conclusion

This work introduced a novel online bilevel optimization framework that overcomes the limitations of existing algorithms, which often depend on extensive oracle information and incur high computational costs. Our method leverages limited feedback and zeroth-order updates for efficient hypergradient estimation and simultaneous updates of decision variables, achieving *sublinear* bilevel regret without window smoothing. Experiments on online parametric loss tuning and black-box adversarial attacks validate its effectiveness. A limitation of this study is that the results focus on nonconvex regret bounds, without extending guarantees to convex settings.

## Acknowledgments and Disclosure of Funding

We thank the reviewers for their valuable comments. The work of DAT was supported by Samsung SDS Research America, Mountain View. The work of GM was supported in part by NSF grants DMS–2348640 and DMS–2319552.

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

# Contents

# A   Related Work

**BO** was introduced in game theory by [64] and modeled mathematically in [9]. Initial works [32, 55] reduced it to single-level optimization. Recently, gradient-based approaches have gained popularity for their simplicity and efficacy [21, 27, 43, 12, 56], though they assume offline objectives.

**OBO** was initiated by [67], proposing the OAGD method with regret bounds. [40] developed algorithms for online minimax optimization, special cases of OBO with local regret guarantees. [50] introduced SOBOW, a single-loop optimizer using window-smoothed functions and multiple CGs for nonconvex-strongly-convex cases. Unlike these works, we propose using *projected gradient* as a more general performance measure for constrained objectives, focusing on the original functions and their regret; See Table 1 for a comparison.

**Single-Level Regret Minimization.** Single-level online optimization predominantly focuses on convex problems, either with static or dynamic convex regret minimization [72, 34, 61]. Non-convex online optimization [36, 30, 29] poses greater challenges than its convex counterparts [61, 72, 35, 7]. Notable contributions in this field include adversarial multi-armed bandit algorithms [10, 37, 38, 47] and the Follow-the-Perturbed-Leader approach [2, 46, 66]. Hazan et al. [36] introduced window-smoothed local regret for gradient averaging in non-convex models, which Hallak et al. [31] extended to non-smooth, non-convex problems. Inspired by their work, we employ local regret for Online Bilevel Optimization (OBO) without window-smoothing.

**Zeroth-Order Optimization.** Single-Level ZO Optimization has been widely studied in both offline [25, 17, 1, 59, 57] and online settings [52, 29, 30, 70, 5]. We next review closely related work. Liu et al. [52] proposed ZOO-ADMM, a gradient-free online optimization algorithm utilizing ADMM. Guan et al. [30] studied online non-convex optimization with limited oracle feedback. Research on online non-convex optimization with bandit feedback includes work by Heliou et al. [37], which established bounds on global static and dynamic regret using dual averaging, further refined in [38]. Gao et al. [23] extended these ideas to ZO algorithms. Flaxman et al. [20] provided algorithms for bandit online optimization of convex functions using ZO gradient approximation. Our work closely relates to [63], which proposes a Hessian-free method approximating the Jacobian matrix using a ZO method based on finite differences of gradients. In contrast, our method uses function oracles to approximate both the Hessian and gradients and is derivative-free. We also point out the recent work [3] on ZO stochastic algorithms for solving bilevel problems when neither the upper/lower objective values nor their unbiased gradient estimates are available. Their approach, limited to the *offline* setting, does not include numerical results, thus leaving its practical efficiency unclear.

# B   Additional Preliminaries and Notations

## B.1   Preliminary Lemmas

We first provide several useful lemmas for the main proofs.

**Definition B.1 (Projected gradient [26]).** Let $\mathcal{X} \subset \mathbb{R}^{d_1}$ be a closed convex set. Then, the projected gradient for any $\alpha_t > 0$ and $\mathbf{p} \in \mathbb{R}^{d_1}$ is defined as

$$\mathcal{P}_{\mathcal{X}, \alpha_t}(\mathbf{x}; \mathbf{p}) := \frac{1}{\alpha_t}\left(\mathbf{x} - \mathbf{x}^+\right),$$

where

$$\mathbf{x}^+ = \Pi_{\mathcal{X}}\left(\mathbf{x} - \alpha_t \mathbf{p}\right), \tag{36}$$

and $\Pi_{\mathcal{X}}[\cdot]$ denotes the orthogonal projection operator onto set $\mathcal{X}$.

**Lemma B.2.** [28, Lemma 13] *If $f : \mathcal{X} \to \mathbb{R}$ is a $\mu_f$-strongly convex function with respect to some norm $\|\cdot\|$, and $\mathbf{x}^*$ is the minimizer of $f$ (i.e. $\mathbf{x}^* = \arg\min_{\mathbf{x} \in \mathcal{X}} f(\mathbf{x})$), then we have $\forall \mathbf{x} \in \mathcal{X}$,*

$$\frac{\mu_f}{2}\|\mathbf{x} - \mathbf{x}^*\|^2 \le f(\mathbf{x}) - f(\mathbf{x}^*) \le \frac{1}{2\mu_f}\|\nabla f(\mathbf{x})\|^2.$$

**Lemma B.3.** *Suppose $f(\mathbf{x})$ is $L$-smooth, and $\mathbf{x}^* \in argmin_{\mathbf{x} \in \mathcal{X}} f(\mathbf{x})$. Then, we can upper bound the magnitude of the gradient at any given point $\mathbf{x} \in \mathbb{R}^d$ in terms of the objective sub optimality at $\mathbf{x}$, as follows:*

$$\frac{1}{2L}\|\nabla f(\mathbf{x})\|^2 \le f(\mathbf{x}) - f(\mathbf{x}^*) \le \frac{L}{2}\|\mathbf{x} - \mathbf{x}^*\|^2. \tag{37}$$

**Lemma B.4.** *For any* $\mathbf{x}, \mathbf{y} \in \mathbb{R}^d$, *the following holds for any* $c > 0$:

$$\|\mathbf{x} + \mathbf{y}\|^2 \leq (1 + c)\|\mathbf{x}\|^2 + \left(1 + \frac{1}{c}\right)\|\mathbf{y}\|^2.$$

We also utilize a basic yet important property of the projected-gradient mapping.

**Lemma B.5.** [26, Proposition 1] *Let* $\mathcal{P}_{\mathcal{X},\alpha_t}(\mathbf{x};\mathbf{p})$ *denote the projected gradient as defined in Definition B.1. For any* $\mathbf{x}, \mathbf{p}_1, \mathbf{p}_2 \in \mathbb{R}^d$ *and* $\alpha_t > 0$, *it holds that*

$$\|\mathcal{P}_{\mathcal{X},\alpha_t}(\mathbf{x};\mathbf{p}_1) - \mathcal{P}_{\mathcal{X},\alpha_t}(\mathbf{x};\mathbf{p}_2)\| \leq \|\mathbf{p}_1 - \mathbf{p}_2\|.$$

**Lemma B.6.** [36, Proposition 2.4] *Let* $\mathcal{P}_{\mathcal{X},\alpha_t}(\mathbf{x};\mathbf{p})$ *denote the projected gradient as defined in Definition B.1. For any* $\mathbf{x}, \mathbf{p}_1, \mathbf{p}_2 \in \mathbb{R}^d$ *and* $\alpha_t > 0$, *it holds that*

$$\|\mathcal{P}_{\mathcal{X},\alpha_t}(\mathbf{x};\mathbf{p}_1 + \mathbf{p}_2)\| \leq \|\mathcal{P}_{\mathcal{X},\alpha_t}(\mathbf{x};\mathbf{p}_1)\| + \|\mathbf{p}_2\|.$$

**Lemma B.7.** *Let* $\mathcal{P}_{\mathcal{X},\alpha_t}(\mathbf{x};\mathbf{p})$ *be as given in Definition B.1. Then, for any* $\mathbf{p} \in \mathbb{R}^d$ *and* $\alpha_t > 0$, *we have*

$$\langle \mathbf{p}, \mathcal{P}_{\mathcal{X},\alpha_t}(\mathbf{x};\mathbf{p})\rangle \geq \|\mathcal{P}_{\mathcal{X},\alpha_t}(\mathbf{x};\mathbf{p})\|^2.$$

*Proof.* By the definition of $\mathbf{x}^+$, the optimality condition of (36) is

$$\left\langle \mathbf{p} + \frac{1}{\alpha_t}(\mathbf{x}^+ - \mathbf{x}), \mathbf{z} - \mathbf{x}^+ \right\rangle \geq 0, \quad \forall \mathbf{z} \in \mathcal{X}.$$

Letting $\mathbf{z} = \mathbf{x}$, we obtain

$$\langle \mathbf{p}, \mathbf{x} - \mathbf{x}^+ \rangle \geq \frac{1}{\alpha_t}\langle \mathbf{x} - \mathbf{x}^+, \mathbf{x} - \mathbf{x}^+ \rangle,$$

which can be rearranged to

$$\langle \mathbf{p}, \mathcal{P}_{\mathcal{X},\alpha_t}(\mathbf{x};\mathbf{p})\rangle = \frac{1}{\alpha_t}\langle \mathbf{p}, \mathbf{x} - \mathbf{x}^+ \rangle \geq \frac{1}{\alpha_t^2}\langle \mathbf{x} - \mathbf{x}^+, \mathbf{x} - \mathbf{x}^+ \rangle$$

$$= \|\mathcal{P}_{\mathcal{X},\alpha_t}(\mathbf{x};\mathbf{p})\|^2.$$

$\square$

## B.2 Examples Illustrating Regularity Conditions

Theorem 2.6 achieves sublinear bilevel regret when the variations $V_T$ and $H_{2,T}$ are $o(T^{2/3})$ and $o(T^{1/3})$, respectively. Below, we provide some examples of online optimization in both single-level and bilevel settings to illustrate when this occurs.

*Example* B.8. Consider function $f_t(\mathbf{x}) = \|\mathbf{A}_t\mathbf{x} - \mathbf{b}_t\|^2$, where $\mathbf{A}_t = [1, 0; 0, 1 + \frac{1}{t}]$, $\mathbf{b}_t = (1, 1)$. It follows from (11) that $V_T = \sum_{t=2}^{T} \max_{\mathbf{x}} |f_t(\mathbf{x}) - f_{t-1}(\mathbf{x})| = \sum_{t=2}^{T} |\left(\frac{1}{t}\right)^2 - \left(\frac{1}{t-1}\right)^2|$, and

$$V_T = \sum_{t=2}^{T} |\left(\frac{1}{t} - \frac{1}{t-1}\right) - \left(\frac{1}{t} + \frac{1}{t-1}\right)|$$

$$= \sum_{t=2}^{T} |\left(\frac{t-1-t}{t(t-1)}\right) - \left(\frac{1}{t} + \frac{1}{t-1}\right)|$$

$$= \sum_{t=2}^{T} |\left(-\frac{1}{t(t-1)}\right) - \left(\frac{1}{t} + \frac{1}{t-1}\right)|$$

$$= \sum_{t=2}^{T} |\frac{1}{t(t-1)}||\frac{t-1+t}{t(t-1)}|$$

$$= \sum_{t=2}^{T} |\frac{2}{t(t-1)^2}|.$$

Then, $V_T \leq \sum_{t=2}^{T} \frac{2}{t^3} \approx \int_2^T \frac{2}{t^3} dt = \frac{1}{4} - \frac{1}{T^2}$. As $T \to \infty$, $V_T$ becomes bounded and approaches a constant value, indicating that $V_T$ grows slower than $T$ itself.

*Example* B.9. Let

$$f_t(\mathbf{x}) = \begin{cases} \left(-\frac{1}{T}, 0\right) & \text{if } t \text{ is even;} \\ \left(0, -\frac{1}{T}\right) & \text{if } t \text{ is odd.} \end{cases}$$

Then, $V_T = \sum_{t=2}^{T} \max_{\mathbf{x}} |f_t(\mathbf{x}) - f_{t-1}(\mathbf{x})| = \mathcal{O}(1)$.

*Example* B.10. Let $x \in \mathcal{X} = [-1, 1] \subset \mathbb{R}$, $y \in \mathbb{R}$, and consider a sequence of quadratic cost functions

$$f_t(x, y) = \frac{1}{2}\left(x + 2a_t^{(1)}\right)^2 + \frac{1}{2}\left(y - a_t^{(2)}\right)^2,$$

$$g_t(x, y) = \frac{1}{2}y^2 - \left(x - a_t^{(2)}\right)y,$$

where $a_t^{(1)} = 1/t$ and $a_t^{(2)} = 1/\sqrt{t}$ for all $t \in [T]$.

We have

$$y_t^*(x) = x - a_t^{(2)},$$

and

$$f_t(x, y_t^*(x)) - f_{t-1}(x, y_{t-1}^*(x))$$

$$= \frac{1}{2}\left[\left(x + 2a_t^{(1)}\right)^2 - \left(x + 2a_{t-1}^{(1)}\right)^2\right] + \frac{1}{2}\left[\left(y_t^*(x) - a_t^{(2)}\right)^2 - \left(y_{t-1}^*(x) - a_{t-1}^{(2)}\right)^2\right]$$

$$= \frac{1}{2}\left[\left(x^2 + 4xa_t^{(1)} + 4(a_t^{(1)})^2\right) - \left(x^2 + 4xa_{t-1}^{(1)} + 4(a_{t-1}^{(1)})^2\right)\right]$$

$$+ \frac{1}{2}\left[\left((x - a_t^{(2)})^2 - 2(x - a_t^{(2)})a_t^{(2)} + (a_t^{(2)})^2\right) - \left((x - a_{t-1}^{(2)})^2 - 2(x - a_{t-1}^{(2)})a_{t-1}^{(2)} + (a_{t-1}^{(2)})^2\right)\right]$$

$$= 2x\left(a_t^{(1)} - a_{t-1}^{(1)} - a_t^{(2)} + a_{t-1}^{(2)}\right) + 2\left((a_t^{(1)})^2 - (a_{t-1}^{(1)})^2 + (a_t^{(2)})^2 - (a_{t-1}^{(2)})^2\right).$$

Taking the maximum over $x$ and using $x \in [-1, 1]$ :

$$\sup_x |f_t(x, y_t^*(x)) - f_{t-1}(x, y_{t-1}^*(x))| = 2\left|a_t^{(1)} - a_{t-1}^{(1)}\right| + 2\left|-a_t^{(2)} + a_{t-1}^{(2)}\right|$$

$$+ 2\left|(a_t^{(1)})^2 - (a_{t-1}^{(1)})^2\right| + 2\left|(a_t^{(2)})^2 - (a_{t-1}^{(2)})^2\right|.$$

Since $a_t^{(1)} = 1/t$ and $a_t^{(2)} = 1/\sqrt{t}$ for all $t \in [T]$, then we have

$$|a_t^{(1)} - a_{t-1}^{(1)}| \approx \frac{1}{t^2}, \quad |a_t^{(2)} - a_{t-1}^{(2)}| \approx \frac{1}{2t^{3/2}},$$

$$|(a_t^{(1)})^2 - (a_{t-1}^{(1)})^2| \approx \frac{1}{t^3}, \quad |(a_t^{(2)})^2 - (a_{t-1}^{(2)})^2| \approx \frac{1}{t^2}.$$

Then, we get

$$V_T := \sum_{t=2}^{T} \sup_x |f_t(x, y_t^*(x)) - f_{t-1}(x, y_{t-1}^*(x))| = \sum_{t=2}^{T}\left(\frac{2}{t^2} + \frac{1}{2t^{3/2}} + \frac{1}{t^3}\right).$$

The series $\sum_{t=2}^{T}\left(\frac{2}{t^2} + \frac{1}{2t^{3/2}} + \frac{1}{t^3}\right)$ converges, implying $V_T = \mathcal{O}(1)$. Moreover, we have

$$H_{2,T} = \sum_{t=2}^{T} \sup_x \|y_t^*(x) - y_{t-1}^*(x)\|^2 = \sum_{t=2}^{T} \sup_x \|x - a_t^{(2)} - x + a_{t-1}^{(2)}\|^2$$

$$= \sum_{t=2}^{T} |-a_t^{(2)} + a_{t-1}^{(2)}|^2 = \sum_{t=2}^{T} |a_t^{(2)} - a_{t-1}^{(2)}|^2 \approx \sum_{t=2}^{T} \frac{1}{4t^3},$$

which implies $H_{2,T} = \mathcal{O}(1)$.

To achieve $V_T = o(T^{2/3})$ and $H_{2,T} = o(T^{1/3})$, the changes in the cost functions $f_t(\mathbf{x}, y_t^*(\mathbf{x}))$ and $y_t^*(\mathbf{x})$ should decay to zero faster than $\mathcal{O}(1/t^a)$ with $a > 1/3$. For example, if the coefficients in the functions change as $\mathcal{O}(1/t^a)$ with $a > 1/3$, then the cumulative sum over $T$ will be $o(T^{2/3})$. When $f_t(\mathbf{x}, y_t^*(\mathbf{x}))$ and $y_t^*(\mathbf{x})$ decay as $\mathcal{O}(1/\sqrt{t})$, then the total variation grows at most as $\mathcal{O}(\sqrt{T})$.

# C  Proof of Regret Bounds for Simultaneous Online Gradient Descent (SOGD)

**Proof Roadmap.** We introduce Lemma C.2, which quantifies the error between the approximated direction of the momentum-based gradient estimator, $\mathbf{d}_t^{\mathbf{y}}$, and the true direction, $\nabla_{\mathbf{y}} g_t(\mathbf{x}_t, \mathbf{y}_t)$, at each iteration. To bound the error of the lower-level variable, we provide Lemma C.4, which captures the gap $\|\mathbf{y}_{t+1} - \mathbf{y}_t^*(\mathbf{x}_t)\|^2$ and incorporates the error introduced in Lemma C.2. Moreover, we provide Lemma C.6, which quantifies the error between the approximated direction of the momentum-based gradient estimator, $\mathbf{d}_t^{\mathbf{y}}$, and the true direction, $\nabla_{\mathbf{y}}^2 g_t(\mathbf{z}_t) \mathbf{v}_t + \nabla_{\mathbf{y}} f_t(\mathbf{z}_t)$, at each iteration. To bound the error of the system solution, we provide Lemma C.8, which captures the gap $\|\mathbf{v}_{t+1} - \mathbf{v}_t^*(\mathbf{x}_t)\|^2$ and incorporates the error introduced in Lemma C.6. Moreover, we provide Lemma C.9, which quantifies the error between the approximated direction of the momentum-based hypergradient estimator, $\mathbf{d}_t^{\mathbf{x}}$, and the true direction, $\nabla_{\mathbf{x}} f_t(\mathbf{z}_t) + \nabla_{\mathbf{xy}}^2 g_t(\mathbf{z}_t) \mathbf{v}_t$, at each iteration. We also present Lemma C.11, which provides an upper bound for the projection mapping and relates to the three errors discussed in Lemmas C.4, C.8, and C.9. Finally, by combining these lemmas and appropriately setting the parameters, we achieve the desired result.

## C.1  Proof of Lemma 2.1

*Proof.* By letting $\nu = 1 - \eta$ for $\eta \in (0, 1)$, the window-smoothed gradient

$$\nabla F_{t,\nu}(\mathbf{x}_t, \mathbf{y}_t; \mathcal{B}_t) = \frac{1}{W} \sum_{i=0}^{w-1} \nu^i \nabla f_{t-i}(\mathbf{x}_{t-i}, \mathbf{y}_{t-i}; \mathcal{B}_{t-i}),$$

is equivalent to

$$\nabla F_{t,\nu}(\mathbf{x}_t, \mathbf{y}_t; \mathcal{B}_t) = \frac{1}{W} \sum_{j=t-w+1}^{t} (1-\eta)^{t-j} \nabla f_j(\mathbf{x}_j, \mathbf{y}_j; \mathcal{B}_j). \tag{38}$$

Let $\mathbf{d}_t^{\mathbf{x}} = \nabla F_{t,\nu}(\mathbf{x}_t, \mathbf{y}_t; \mathcal{B}_t)$. Then (38) is equivalent to

$$\mathbf{d}_t^{\mathbf{x}} = \frac{1}{W} \nabla f_t(\mathbf{x}_t, \mathbf{y}_t; \mathcal{B}_t) + \frac{1}{W} \sum_{j=t-w+1}^{t-1} (1-\eta)^{t-j} \nabla f_j(\mathbf{x}_j, \mathbf{y}_j; \mathcal{B}_j).$$

Since

$$(1-\eta)\mathbf{d}_{t-1}^{\mathbf{x}} = \frac{(1-\eta)}{W} \sum_{j=t-w}^{t-1} (1-\eta)^{t-1-j} \nabla f_j(\mathbf{x}_j, \mathbf{y}_j; \mathcal{B}_j),$$

we have

$$\mathbf{d}_t^{\mathbf{x}} = \frac{1}{W} \nabla f_t(\mathbf{x}_t, \mathbf{y}_t; \mathcal{B}_t) + (1-\eta)\mathbf{d}_{t-1}^{\mathbf{x}} - \frac{(1-\eta)^w}{W} \nabla f_{t-w}(\mathbf{x}_{t-w}, \mathbf{y}_{t-w}; \mathcal{B}_{t-w}),$$

with $f_i(\cdot) = 0$ for all $i \leq 0$.

If $w = t$ and $W = \frac{1}{\eta}$ then, we have

$$\mathbf{d}_t^{\mathbf{x}} = \eta \nabla f_t(\mathbf{x}_t, \mathbf{y}_t; \mathcal{B}_t) + (1-\eta)\mathbf{d}_{t-1}^{\mathbf{x}}.$$

$\square$

## C.2  Bounds on the Inner Decision Variable

In the following, inspired by offline BO [69, 15] and OBO [67, 50], we provide a set of lemmas for the analysis of SOGD. We first present a lemma that characterizes the Lipschitz continuity of the approximate gradients, as well as the inner and system solutions.

**Lemma C.1.** *Under Assumptions 2.2 and 2.3, for all* $\mathbf{x}, \mathbf{x}' \in \mathcal{X}$, *and the search directions* $\{\mathbf{d}_t^{\mathbf{x}}\}_{t=1}^{T}$ *and* $\{\mathbf{d}_t^{\mathbf{v}}\}_{t=1}^{T}$ *generated by Algorithm 1, we have*

$$\|\mathbf{d}_t^{\mathbf{x}} - \nabla f_t(\mathbf{x}_t, \mathbf{y}_t^*(\mathbf{x}_t))\|^2 \leq M_f^2 \left( \|\mathbf{y}_t - \mathbf{y}_t^*(\mathbf{x}_t)\|^2 + \|\mathbf{v}_t - \mathbf{v}_t^*(\mathbf{x}_t)\|^2 \right), \qquad (39a)$$

$$\|\mathbf{d}_t^{\mathbf{v}}\|^2 \leq M_{\mathbf{v}}^2 \left( \|\mathbf{y}_t - \mathbf{y}_t^*(\mathbf{x}_t)\|^2 + \|\mathbf{v}_t - \mathbf{v}_t^*(\mathbf{x}_t)\|^2 \right), \qquad (39b)$$

$$\|\nabla f_t(\mathbf{x}, \mathbf{y}_t^*(\mathbf{x})) - \nabla f_t(\mathbf{x}', \mathbf{y}_t^*(\mathbf{x}'))\| \leq L_f \|\mathbf{x} - \mathbf{x}'\|, \qquad (39c)$$

$$\|\mathbf{y}_t^*(\mathbf{x}) - \mathbf{y}_t^*(\mathbf{x}')\| \leq L_{\mathbf{y}} \|\mathbf{x} - \mathbf{x}'\|, \qquad (39d)$$

$$\|\mathbf{v}_t^*(\mathbf{x}) - \mathbf{v}_t^*(\mathbf{x}')\| \leq L_{\mathbf{v}} \|\mathbf{x} - \mathbf{x}'\|, \qquad (39e)$$

*where* $M_f$, $M_{\mathbf{v}}$, *and* $(L_{\mathbf{y}}, L_{\mathbf{v}}, L_f)$ *are defined in* (42), (43), *and* (44), *respectively.*

*Proof.* We first show (39a).

Using Assumptions 2.2 and 2.3, we have $\nabla_{\mathbf{y}}^2 g_t(\mathbf{x}_t, \mathbf{y}_t^*(\mathbf{x}_t)) \succeq \mu_g$, and

$$\|\mathbf{v}_t^*(\mathbf{x}_t)\| = \| \left( \nabla_{\mathbf{y}}^2 g_t(\mathbf{x}_t, \mathbf{y}_t^*(\mathbf{x}_t)) \right)^{-1} \nabla_{\mathbf{y}} f_t(\mathbf{x}_t, \mathbf{y}_t^*(\mathbf{x}_t)) \| \leq \frac{\ell_{f,0}}{\mu_g}. \qquad (40)$$

Observe that

$$\begin{aligned}
\|\mathbf{d}_t^{\mathbf{x}} - \nabla f_t(\mathbf{x}_t, \mathbf{y}_t^*(\mathbf{x}_t))\| &\leq \|\nabla_{\mathbf{x}} f_t(\mathbf{x}_t, \mathbf{y}_t) - \nabla_{\mathbf{x}} f_t(\mathbf{x}_t, \mathbf{y}_t^*(\mathbf{x}_t))\| \\
&\quad + \|\mathbf{v}_t \nabla_{\mathbf{x}\mathbf{y}}^2 g_t(\mathbf{x}_t, \mathbf{y}_t) - \mathbf{v}_t^*(\mathbf{x}_t) \nabla_{\mathbf{x}\mathbf{y}}^2 g_t(\mathbf{x}_t, \mathbf{y}_t^*(\mathbf{x}_t)) \| \\
&\leq \|\nabla_{\mathbf{x}} f_t(\mathbf{x}_t, \mathbf{y}_t) - \nabla_{\mathbf{x}} f_t(\mathbf{x}_t, \mathbf{y}_t^*(\mathbf{x}_t))\| \\
&\quad + \|\nabla_{\mathbf{x}\mathbf{y}}^2 g_t(\mathbf{x}_t, \mathbf{y}_t)\| \|\mathbf{v}_t - \mathbf{v}_t^*(\mathbf{x}_t)\| \\
&\quad + \|\mathbf{v}_t^*(\mathbf{x}_t)\| \|\nabla_{\mathbf{x}\mathbf{y}}^2 g_t(\mathbf{x}_t, \mathbf{y}_t) - \nabla_{\mathbf{x}\mathbf{y}}^2 g_t(\mathbf{x}_t, \mathbf{y}_t^*(\mathbf{x}_t))\| \\
&\leq \left( \ell_{f,1} + \frac{\ell_{g,2} \ell_{f,0}}{\mu_g} \right) \|\mathbf{y}_t - \mathbf{y}_t^*(\mathbf{x}_t)\| + \ell_{g,1} \|\mathbf{v}_t - \mathbf{v}_t^*(\mathbf{x}_t)\| \\
&\leq M_f^2 \left( \|\mathbf{y}_t - \mathbf{y}_t^*(\mathbf{x}_t)\| + \|\mathbf{v}_t - \mathbf{v}_t^*(\mathbf{x}_t)\| \right), \qquad (41)
\end{aligned}$$

where

$$M_f := \sqrt{2} \max \left\{ \ell_{f,1} + \frac{\ell_{g,2} \ell_{f,0}}{\mu_g}, \ell_{g,1} \right\}, \qquad (42)$$

the third inequality is by Assumption 2.3, and the last inequality follows from (40).

Next, we establish (39b).
Since $\mathbf{d}_t^{\mathbf{v}*} := \nabla_{\mathbf{y}} f_t(\mathbf{x}_t, \mathbf{y}_t^*(\mathbf{x}_t)) + \nabla_{\mathbf{y}}^2 g_t(\mathbf{x}_t, \mathbf{y}_t^*(\mathbf{x}_t)) \mathbf{v}_t^*(\mathbf{x}_t) = 0$, we have

$$\begin{aligned}
\|\mathbf{d}_t^{\mathbf{v}}\| &= \|\mathbf{d}_t^{\mathbf{v}} - \mathbf{d}_t^{\mathbf{v}*}\| \\
&= \|\mathbf{v}_t \nabla_{\mathbf{y}}^2 g_t(\mathbf{x}_t, \mathbf{y}_t) + \nabla_{\mathbf{y}} f_t(\mathbf{x}_t, \mathbf{y}_t) \\
&\quad - \left( \mathbf{v}_t^*(\mathbf{x}_t) \nabla_{\mathbf{y}}^2 g_t(\mathbf{x}_t, \mathbf{y}_t^*(\mathbf{x}_t)) + \nabla_{\mathbf{y}} f_t(\mathbf{x}_t, \mathbf{y}_t^*(\mathbf{x}_t)) \right) \| \\
&\leq \| \left( \nabla_{\mathbf{y}}^2 g_t(\mathbf{x}_t, \mathbf{y}_t) - \nabla_{\mathbf{y}}^2 g_t(\mathbf{x}_t, \mathbf{y}_t^*(\mathbf{x}_t)) \right) \mathbf{v}_t^*(\mathbf{x}_t)\| \\
&\quad + \|\nabla_{\mathbf{y}}^2 g_t(\mathbf{x}_t, \mathbf{y}_t) \left( \mathbf{v}_t - \mathbf{v}_t^*(\mathbf{x}_t) \right) \| \\
&\quad + \|\nabla_{\mathbf{y}} f_t(\mathbf{x}_t, \mathbf{y}_t) - \nabla_{\mathbf{y}} f_t(\mathbf{x}_t, \mathbf{y}_t^*(\mathbf{x}_t))\|.
\end{aligned}$$

Then, from Assumption 2.3 and (40), we have

$$\begin{aligned}
\|\mathbf{d}_t^{\mathbf{v}}\| &\leq \ell_{g,2} \|\mathbf{y}_t - \mathbf{y}_t^*(\mathbf{x}_t)\| \|\mathbf{v}_t^*(\mathbf{x}_t)\| + \ell_{g,1} \|\mathbf{v}_t - \mathbf{v}_t^*(\mathbf{x}_t)\| + \ell_{f,1} \|\mathbf{y}_t - \mathbf{y}_t^*(\mathbf{x}_t)\| \\
&\leq \left( \frac{\ell_{g,2} \ell_{f,0}}{\mu_g} + \ell_{f,1} \right) \|\mathbf{y}_t - \mathbf{y}_t^*(\mathbf{x}_t)\| + \ell_{g,1} \|\mathbf{v}_t - \mathbf{v}_t^*(\mathbf{x}_t)\| \\
&\leq M_{\mathbf{v}} \left( \|\mathbf{y}_t - \mathbf{y}_t^*(\mathbf{x}_t)\| + \|\mathbf{v}_t - \mathbf{v}_t^*(\mathbf{x}_t)\| \right),
\end{aligned}$$

where

$$M_{\mathbf{v}} := \sqrt{2} \max \left\{ \frac{\ell_{g,2} \ell_{f,0}}{\mu_g} + \ell_{f,1}, \ell_{g,1} \right\}. \qquad (43)$$

The proofs of Eqs. (39c)-(39e) follow from [67, Lemma 17] by setting

$$L_{\mathbf{y}} := \frac{\ell_{g,1}}{\mu_g},$$

$$L_{\mathbf{v}} := \ell_{f,1} + \frac{\ell_{g,1}\ell_{f,1}}{\mu_g} + \frac{\ell_{f,0}}{\mu_g}\left(\ell_{g,2} + \frac{\ell_{g,1}\ell_{g,2}}{\mu_g}\right),$$

$$L_f := \ell_{f,1} + \frac{\ell_{g,1}(\ell_{f,1} + M_f)}{\mu_g} + \frac{\ell_{f,0}}{\mu_g}\left(\ell_{g,2} + \frac{\ell_{g,1}\ell_{g,2}}{\mu_g}\right),$$

(44)

where the other constants are defined in Assumption 2.3. □

The following lemma is inspired by [69] and can be viewed as an extension of [69] to the online setting.

**Lemma C.2.** *Suppose Assumptions* B3. *and* C1. *hold. Let* $\{(\mathbf{x}_t, \mathbf{y}_t, \mathbf{v}_t)\}_{t=1}^T$ *be generated according to Algorithm* 1. *For* $e_t^g$ *defined as*

$$e_t^g := \mathbf{d}_t^{\mathbf{y}} - \nabla_{\mathbf{y}} g_t(\mathbf{x}_t, \mathbf{y}_t),$$

(45)

*we have:*

$$
\mathbb{E}\|e_{t+1}^g\|^2 \le (1-\gamma_{t+1})^2(1+48\ell_{g,1}^2\beta_t^2)\mathbb{E}\|e_t^g\|^2 + 2\gamma_{t+1}^2\frac{\sigma_{g_{\mathbf{y}}}^2}{\bar{b}} + 24(1-\gamma_{t+1})^2\ell_{g,1}^2\mathbb{E}\|\mathbf{x}_{t+1} - \mathbf{x}_t\|^2
$$
$$
+ 6(1-\gamma_{t+1})^2\mathbb{E}\|\nabla_{\mathbf{y}} g_t(\mathbf{z}_{t+1}) - \nabla_{\mathbf{y}} g_{t+1}(\mathbf{z}_{t+1})\|^2
$$
$$
+ 48(1-\gamma_{t+1})^2\ell_{g,1}^2\beta_t^2\mathbb{E}\|\nabla_{\mathbf{y}} g_t(\mathbf{x}_t, \mathbf{y}_t)\|^2.
$$

(46)

*Proof.* From Algorithm 1, we have

$$\mathbf{d}_{t+1}^{\mathbf{y}} = \nabla_{\mathbf{y}} g_{t+1}(\mathbf{z}_{t+1}; \bar{\mathcal{B}}_{t+1}) + (1-\gamma_{t+1})(\mathbf{d}_t^{\mathbf{y}} - \nabla_{\mathbf{y}} g_{t+1}(\mathbf{z}_t; \bar{\mathcal{B}}_{t+1})).$$

Then, we have

$$
\mathbb{E}\|e_{t+1}^g\|^2 = \mathbb{E}\|\mathbf{d}_{t+1}^{\mathbf{y}} - \nabla_{\mathbf{y}} g_{t+1}(\mathbf{z}_{t+1})\|^2
$$
$$
= \mathbb{E}\|\nabla_{\mathbf{y}} g_{t+1}(\mathbf{z}_{t+1}; \bar{\mathcal{B}}_{t+1}) + (1-\gamma_{t+1})(\mathbf{d}_t^{\mathbf{y}} - \nabla_{\mathbf{y}} g_{t+1}(\mathbf{z}_t; \bar{\mathcal{B}}_{t+1})) - \nabla_{\mathbf{y}} g_{t+1}(\mathbf{z}_{t+1})\|^2
$$
$$
= \mathbb{E}\|(1-\gamma_{t+1})e_t^g + (\nabla_{\mathbf{y}} g_{t+1}(\mathbf{z}_{t+1}; \bar{\mathcal{B}}_{t+1}) - \nabla_{\mathbf{y}} g_{t+1}(\mathbf{z}_{t+1}))
$$
$$
- (1-\gamma_{t+1})\left(\nabla_{\mathbf{y}} g_{t+1}(\mathbf{z}_t; \bar{\mathcal{B}}_{t+1})\right) - \nabla_{\mathbf{y}} g_t(\mathbf{z}_t)\|^2,
$$

which implies that

$$
\mathbb{E}\|e_{t+1}^g\|^2 = (1-\gamma_{t+1})^2\mathbb{E}\|e_t^g\|^2 + \mathbb{E}\|(\nabla_{\mathbf{y}} g_{t+1}(\mathbf{z}_{t+1}; \bar{\mathcal{B}}_{t+1}) - \nabla_{\mathbf{y}} g_{t+1}(\mathbf{z}_{t+1}))
$$
$$
- (1-\gamma_{t+1})\left(\nabla_{\mathbf{y}} g_{t+1}(\mathbf{z}_t; \bar{\mathcal{B}}_{t+1})\right) - \nabla_{\mathbf{y}} g_t(\mathbf{z}_t)\|^2
$$
$$
\le (1-\gamma_{t+1})^2\mathbb{E}\|e_t^g\|^2 + 2\gamma_{t+1}^2\mathbb{E}\|\nabla_{\mathbf{y}} g_{t+1}(\mathbf{z}_{t+1}; \bar{\mathcal{B}}_{t+1}) - \nabla_{\mathbf{y}} g_{t+1}(\mathbf{z}_{t+1})\|^2
$$
$$
+ 2(1-\gamma_{t+1})^2\mathbb{E}\|\nabla_{\mathbf{y}} g_{t+1}(\mathbf{z}_{t+1}; \bar{\mathcal{B}}_{t+1})
$$
$$
- \nabla_{\mathbf{y}} g_{t+1}(\mathbf{z}_{t+1}) - \nabla_{\mathbf{y}} g_{t+1}(\mathbf{z}_t; \bar{\mathcal{B}}_{t+1}) + \nabla_{\mathbf{y}} g_t(\mathbf{z}_t)\|^2
$$
$$
\le (1-\gamma_{t+1})^2\mathbb{E}\|e_t^g\|^2 + 2\gamma_{t+1}^2\frac{\sigma_{g_{\mathbf{y}}}^2}{\bar{b}}
$$
$$
+ 2(1-\gamma_{t+1})^2\mathbb{E}\|\nabla_{\mathbf{y}} g_{t+1}(\mathbf{z}_{t+1}; \bar{\mathcal{B}}_{t+1})
$$
$$
- \nabla_{\mathbf{y}} g_{t+1}(\mathbf{z}_{t+1}) - \nabla_{\mathbf{y}} g_{t+1}(\mathbf{z}_t; \bar{\mathcal{B}}_{t+1}) + \nabla_{\mathbf{y}} g_t(\mathbf{z}_t)\|^2,
$$

where the second inequality follows from Cauchy–Schwartz inequality and Assumption C1.. Moreover, from Cauchy–Schwartz inequality, we have

$$
\mathbb{E}\|e_{t+1}^g\|^2 \le (1-\gamma_{t+1})^2\mathbb{E}\|e_t^g\|^2 + 2\gamma_{t+1}^2\frac{\sigma_{g_{\mathbf{y}}}^2}{\bar{b}}
$$
$$
+ 6(1-\gamma_{t+1})^2\mathbb{E}\|\nabla_{\mathbf{y}} g_t(\mathbf{z}_t) - \nabla_{\mathbf{y}} g_t(\mathbf{z}_{t+1})\|^2
$$
$$
+ 6(1-\gamma_{t+1})^2\mathbb{E}\|\nabla_{\mathbf{y}} g_t(\mathbf{z}_{t+1}) - \nabla_{\mathbf{y}} g_{t+1}(\mathbf{z}_{t+1})\|^2
$$
$$
+ 6(1-\gamma_{t+1})^2\mathbb{E}\|\nabla_{\mathbf{y}} g_{t+1}(\mathbf{z}_{t+1}; \bar{\mathcal{B}}_{t+1}) - \nabla_{\mathbf{y}} g_{t+1}(\mathbf{z}_t; \bar{\mathcal{B}}_{t+1})\|^2.
$$

From Assumption B3., we have

$$\mathbb{E}\|\nabla_{\mathbf{y}} g_t(\mathbf{z}_{t+1}) - \nabla_{\mathbf{y}} g_t(\mathbf{z}_t)\|^2$$
$$\leq 2\mathbb{E}\|\nabla_{\mathbf{y}} g_t(\mathbf{x}_{t+1}, \mathbf{y}_{t+1}) - \nabla_{\mathbf{y}} g_t(\mathbf{x}_{t+1}, \mathbf{y}_t)\|^2 + 2\mathbb{E}\|\nabla_{\mathbf{y}} g_t(\mathbf{x}_{t+1}, \mathbf{y}_t) - \nabla_{\mathbf{y}} g_t(\mathbf{x}_t, \mathbf{y}_t)\|^2$$
$$\leq 2\ell_{g,1}^2 \mathbb{E}\|\mathbf{x}_{t+1} - \mathbf{x}_t\|^2 + 2\ell_{g,1}^2 \mathbb{E}\|\mathbf{y}_{t+1} - \mathbf{y}_t\|^2$$
$$= 2\ell_{g,1}^2 \mathbb{E}\|\mathbf{x}_{t+1} - \mathbf{x}_t\|^2 + 2\ell_{g,1}^2 \beta_t^2 \mathbb{E}\|\mathbf{d}_t^{\mathbf{y}}\|^2,$$

and

$$\mathbb{E}\|\nabla_{\mathbf{y}} g_{t+1}(\mathbf{z}_{t+1}; \bar{\mathcal{B}}_{t+1}) - \nabla_{\mathbf{y}} g_{t+1}(\mathbf{z}_t; \bar{\mathcal{B}}_{t+1})\|^2$$
$$\leq 2\mathbb{E}\|\nabla_{\mathbf{y}} g_{t+1}(\mathbf{x}_{t+1}, \mathbf{y}_{t+1}; \bar{\mathcal{B}}_{t+1}) - \nabla_{\mathbf{y}} g_{t+1}(\mathbf{x}_{t+1}, \mathbf{y}_t; \bar{\mathcal{B}}_{t+1})\|^2$$
$$+ 2\mathbb{E}\|\nabla_{\mathbf{y}} g_{t+1}(\mathbf{x}_{t+1}, \mathbf{y}_t; \bar{\mathcal{B}}_{t+1}) - \nabla_{\mathbf{y}} g_{t+1}(\mathbf{x}_t, \mathbf{y}_t; \bar{\mathcal{B}}_{t+1})\|^2$$
$$\leq 2\ell_{g,1}^2 \mathbb{E}\|\mathbf{x}_{t+1} - \mathbf{x}_t\|^2 + 2\ell_{g,1}^2 \mathbb{E}\|\mathbf{y}_{t+1} - \mathbf{y}_t\|^2$$
$$= 2\ell_{g,1}^2 \mathbb{E}\|\mathbf{x}_{t+1} - \mathbf{x}_t\|^2 + 2\ell_{g,1}^2 \beta_t^2 \mathbb{E}\|\mathbf{d}_t^{\mathbf{y}}\|^2.$$

From the two inequalities above, we have

$$\mathbb{E}\|e_{t+1}^g\|^2 \leq (1 - \gamma_{t+1})^2 \mathbb{E}\|e_t^g\|^2 + 2\gamma_{t+1}^2 \frac{\sigma_{g_{\mathbf{y}}}^2}{\bar{b}}$$
$$+ 6(1 - \gamma_{t+1})^2 \mathbb{E}\|\nabla_{\mathbf{y}} g_t(\mathbf{z}_{t+1}) - \nabla_{\mathbf{y}} g_{t+1}(\mathbf{z}_{t+1})\|^2$$
$$+ 24(1 - \gamma_{t+1})^2 \ell_{g,1}^2 \left( \mathbb{E}\|\mathbf{x}_{t+1} - \mathbf{x}_t\|^2 + \beta_t^2 \mathbb{E}\|\mathbf{d}_t^{\mathbf{y}}\|^2 \right).$$

Since $e_t^g := \mathbf{d}_t^{\mathbf{y}} - \nabla_{\mathbf{y}} g_t(\mathbf{x}_t, \mathbf{y}_t)$, we have

$$\mathbb{E}\|e_{t+1}^g\|^2 \leq (1 - \gamma_{t+1})^2 \mathbb{E}\|e_t^g\|^2 + 2\gamma_{t+1}^2 \frac{\sigma_{g_{\mathbf{y}}}^2}{\bar{b}} + 24(1 - \gamma_{t+1})^2 \ell_{g,1}^2 \mathbb{E}\|\mathbf{x}_{t+1} - \mathbf{x}_t\|^2$$
$$+ 6(1 - \gamma_{t+1})^2 \mathbb{E}\|\nabla_{\mathbf{y}} g_t(\mathbf{z}_{t+1}) - \nabla_{\mathbf{y}} g_{t+1}(\mathbf{z}_{t+1})\|^2$$
$$+ 48(1 - \gamma_{t+1})^2 \ell_{g,1}^2 \beta_t^2 \mathbb{E}\|e_t^g\|^2 + 48(1 - \gamma_{t+1})^2 \ell_{g,1}^2 \beta_t^2 \mathbb{E}\|\nabla_{\mathbf{y}} g_t(\mathbf{x}_t, \mathbf{y}_t)\|^2$$
$$\leq (1 - \gamma_{t+1})^2 (1 + 48\ell_{g,1}^2 \beta_t^2) \mathbb{E}\|e_t^g\|^2 + 2\gamma_{t+1}^2 \frac{\sigma_{g_{\mathbf{y}}}^2}{\bar{b}} + 24(1 - \gamma_{t+1})^2 \ell_{g,1}^2 \mathbb{E}\|\mathbf{x}_{t+1} - \mathbf{x}_t\|^2$$
$$+ 6(1 - \gamma_{t+1})^2 \mathbb{E}\|\nabla_{\mathbf{y}} g_t(\mathbf{z}_{t+1}) - \nabla_{\mathbf{y}} g_{t+1}(\mathbf{z}_{t+1})\|^2$$
$$+ 48(1 - \gamma_{t+1})^2 \ell_{g,1}^2 \beta_t^2 \mathbb{E}\|\nabla_{\mathbf{y}} g_t(\mathbf{x}_t, \mathbf{y}_t)\|^2.$$

$\square$

**Lemma C.3.** *Suppose Assumptions 2.2, and B3. hold. Then, for the sequence $\{(\mathbf{x}_t, \mathbf{y}_t)\}_{t=1}^T$ generated by Algorithm 1, we have*

$$\mathbb{E}\left[\|\mathbf{y}_{t+1} - \mathbf{y}_t^*(\mathbf{x}_t)\|^2\right] \leq (1+a)\left(1 - 2\beta_t \frac{\mu_g \ell_{g,1}}{\mu_g + \ell_{g,1}}\right) \mathbb{E}\left[\|\mathbf{y}_t - \mathbf{y}_t^*(\mathbf{x}_t)\|^2\right]$$
$$+ \left(-(1+a)\left(\frac{2\beta_t}{\mu_g + \ell_{g,1}} - \beta_t^2\right)\right) \mathbb{E}\left[\|\nabla_{\mathbf{y}} g_t(\mathbf{x}_t, \mathbf{y}_t)\|^2\right]$$
$$+ (1 + \frac{1}{a})\beta_t^2 \mathbb{E}\left[\|e_t^g\|^2\right],$$

*where $e_t^g$ defined in (45), $\mathbf{y}_t^*(\mathbf{x}_t)$ is defined in (1) and $a > 0$ is a constant.*

*Proof.* From Lemma B.4, we have

$$\mathbb{E}\left[\|\mathbf{y}_{t+1} - \mathbf{y}_t^*(\mathbf{x}_t)\|^2\right] = \mathbb{E}\left[\|\mathbf{y}_t - \beta_t \mathbf{d}_t^{\mathbf{y}} - \mathbf{y}_t^*(\mathbf{x}_t)\|^2\right]$$
$$\leq (1+a)\mathbb{E}\left[\|\mathbf{y}_t - \beta_t \nabla_{\mathbf{y}} g_t(\mathbf{x}_t, \mathbf{y}_t) - \mathbf{y}_t^*(\mathbf{x}_t)\|^2\right]$$
$$+ (1 + \frac{1}{a})\beta_t^2 \mathbb{E}\left[\|\mathbf{d}_t^{\mathbf{y}} - \nabla_{\mathbf{y}} g_t(\mathbf{x}_t, \mathbf{y}_t)\|^2\right]. \tag{47}$$

Next, we will bound the first term on the RHS of (47).
We have

$$\mathbb{E}\left[\|\mathbf{y}_t - \beta_t \nabla_\mathbf{y} g_t(\mathbf{x}_t, \mathbf{y}_t) - \mathbf{y}_t^*(\mathbf{x}_t)\|^2\right] = \mathbb{E}\left[\|\mathbf{y}_t - \mathbf{y}_t^*(\mathbf{x}_t)\|^2\right] + \beta_t^2 \mathbb{E}\left[\|\nabla_\mathbf{y} g_t(\mathbf{x}_t, \mathbf{y}_t)\|^2\right]$$
$$- 2\beta_t \mathbb{E}\left[\langle \nabla_\mathbf{y} g_t(\mathbf{x}_t, \mathbf{y}_t), \mathbf{y}_t - \mathbf{y}_t^*(\mathbf{x}_t)\rangle\right]$$
$$\leq \left(1 - 2\beta_t \frac{\mu_g \ell_{g,1}}{\mu_g + \ell_{g,1}}\right) \mathbb{E}\left[\|\mathbf{y}_t - \mathbf{y}_t^*(\mathbf{x}_t)\|^2\right]$$
$$- \left(\frac{2\beta_t}{\mu_g + \ell_{g,1}} - \beta_t^2\right) \mathbb{E}\left[\|\nabla_\mathbf{y} g_t(\mathbf{x}_t, \mathbf{y}_t)\|^2\right], \qquad (48)$$

where the inequality results from the strong convexity of $g_t$ by Assumption 2.2, which implies

$$\langle \nabla_\mathbf{y} g_t(\mathbf{x}_t, \mathbf{y}_t), \mathbf{y}_t - \mathbf{y}_t^*(\mathbf{x}_t)\rangle \geq \frac{\mu_g \ell_{g,1}}{\mu_g + \ell_{g,1}}\|\mathbf{y}_t - \mathbf{y}_t^*(\mathbf{x}_t)\|^2 + \frac{1}{\mu_g + \ell_{g,1}}\|\nabla_\mathbf{y} g_t(\mathbf{x}_t, \mathbf{y}_t)\|^2.$$

Substituting (48) into (47), gives the desired result.

$\square$

To simplify the notation in the analysis, we introduce the definitions

$$\theta_t^\mathbf{y} := \|\mathbf{y}_t - \mathbf{y}_t^*(\mathbf{x}_t)\|^2, \quad \text{and} \quad \theta_t^\mathbf{v} := \|\mathbf{v}_t - \mathbf{v}_t^*(\mathbf{x}_t)\|^2. \qquad (49)$$

The following lemma, inspired by the offline bilevel optimization framework in [69], characterizes the descent behavior of the iterates in the inner problem.

**Lemma C.4.** *Suppose Assumptions 2.2, and B3. hold. Let $\theta_t^\mathbf{y}$ be defined as in (49). Then, for the sequence $\{(\mathbf{x}_t, \mathbf{y}_t)\}_{t=1}^T$ generated by Algorithm 1, the following bound is guaranteed:*

$$\sum_{t=1}^T \left(\mathbb{E}[\theta_{t+1}^\mathbf{y}] - \mathbb{E}[\theta_t^\mathbf{y}]\right) \qquad (50)$$

$$\leq -\frac{L_{\mu_g}}{2} \sum_{t=1}^T \beta_t \mathbb{E}[\theta_t^\mathbf{y}] + \frac{2}{L_{\mu_g}} \sum_{t=1}^T \beta_t \mathbb{E}\left[\|e_t^g\|^2\right] + \frac{4L_\mathbf{y}^2}{L_{\mu_g}} \sum_{t=1}^T \frac{1}{\beta_t} \mathbb{E}\|\mathbf{x}_t - \mathbf{x}_{t+1}\|^2$$

$$+ \frac{4}{L_{\mu_g}} \sum_{t=2}^T \frac{1}{\beta_t} \sup_{\mathbf{x}\in\mathcal{X}} \mathbb{E}\|\mathbf{y}_{t-1}^*(\mathbf{x}) - \mathbf{y}_t^*(\mathbf{x})\|^2 + \sum_{t=1}^T \left(-\frac{2\beta_t}{\mu_g + \ell_{g,1}} + \beta_t^2\right) \mathbb{E}\left[\|\nabla_\mathbf{y} g_t(\mathbf{x}_t, \mathbf{y}_t)\|^2\right],$$

*where $L_{\mu_g} = \frac{\mu_g \ell_{g,1}}{\mu_g + \ell_{g,1}}$, $L_\mathbf{y} = \frac{\ell_{g,1}}{\mu_g}$ is defined as in (44); $H_{2,T}$ is defined in (11). Moreover, $e_t^g$ is defined in (45).*

*Proof.* From Lemma B.4, we have for any $\acute{c} > 0$

$$\mathbb{E}\left[\|\mathbf{y}_{t+1} - \mathbf{y}_{t+1}^*(\mathbf{x}_{t+1})\|^2\right] = \mathbb{E}\left[\|\mathbf{y}_{t+1} - \mathbf{y}_t^*(\mathbf{x}_t) + \mathbf{y}_t^*(\mathbf{x}_t) - \mathbf{y}_{t+1}^*(\mathbf{x}_{t+1})\|^2\right]$$
$$\leq (1 + \acute{c}) \mathbb{E}\left[\|\mathbf{y}_{t+1} - \mathbf{y}_t^*(\mathbf{x}_t)\|^2\right]$$
$$+ \left(1 + \frac{1}{\acute{c}}\right) \mathbb{E}\left[\|\mathbf{y}_{t+1}^*(\mathbf{x}_{t+1}) - \mathbf{y}_t^*(\mathbf{x}_t)\|^2\right]. \qquad (51)$$

From Lemma C.3, we have for any $a > 0$

$$\mathbb{E}\left[\|\mathbf{y}_{t+1} - \mathbf{y}_t^*(\mathbf{x}_t)\|^2\right] \leq (1 + a)\left(1 - 2\beta_t \frac{\mu_g \ell_{g,1}}{\mu_g + \ell_{g,1}}\right) \mathbb{E}\left[\|\mathbf{y}_t - \mathbf{y}_t^*(\mathbf{x}_t)\|^2\right]$$
$$+ \left(-(1 + a)\left(\frac{2\beta_t}{\mu_g + \ell_{g,1}} - \beta_t^2\right)\right) \mathbb{E}\left[\|\nabla_\mathbf{y} g_t(\mathbf{x}_t, \mathbf{y}_t)\|^2\right]$$
$$+ \left(1 + \frac{1}{a}\right) \beta_t^2 \mathbb{E}\left[\|e_t^g\|^2\right]. \qquad (52)$$

Substituting (52) into (51), we get

$$
\mathbb{E}\left[\|\mathbf{y}_{t+1} - \mathbf{y}_{t+1}^*(\mathbf{x}_{t+1})\|^2\right]
$$
$$
\leq (1+\acute{c})(1+a)\left(1 - 2\beta_t\frac{\mu_g\ell_{g,1}}{\mu_g+\ell_{g,1}}\right)\mathbb{E}\left[\|\mathbf{y}_t - \mathbf{y}_t^*(\mathbf{x}_t)\|^2\right]
$$
$$
+ \left(-(1+\acute{c})(1+a)\left(\frac{2\beta_t}{\mu_g+\ell_{g,1}} - \beta_t^2\right)\right)\mathbb{E}\left[\|\nabla_\mathbf{y} g_t(\mathbf{x}_t,\mathbf{y}_t)\|^2\right]
$$
$$
+ (1+\acute{c})(1+\frac{1}{a})\beta_t^2\mathbb{E}\left[\|e_t^g\|^2\right]
$$
$$
+ \left(1+\frac{1}{\acute{c}}\right)\mathbb{E}\left[\|\mathbf{y}_{t+1}^*(\mathbf{x}_{t+1}) - \mathbf{y}_t^*(\mathbf{x}_t)\|^2\right]. \tag{53}
$$

Choose $\acute{c} = \frac{\beta_t L_{\mu_g}/2}{1-\beta_t L_{\mu_g}}$ and $a = \frac{\beta_t L_{\mu_g}}{1-2\beta_t L_{\mu_g}}$. Let $L_{\mu_g} := \frac{\mu_g\ell_{g,1}}{\mu_g+\ell_{g,1}}$. Then, the following equations and inequalities are satisfied.

$$
(1+\acute{c})(1+a)\left(1 - 2\beta_t L_{\mu_g}\right) = 1 - \frac{\beta_t L_{\mu_g}}{2},
$$
$$
(1+a)\left(1 - 2\beta_t L_{\mu_g}\right) = 1 - \beta_t L_{\mu_g},
$$
$$
(1+\acute{c})\left(1 - \beta_t L_{\mu_g}\right) = 1 - \frac{\beta_t L_{\mu_g}}{2}, \tag{54}
$$
$$
1 + \frac{1}{a} \leq \frac{1}{\beta_t L_{\mu_g}}, \quad 1 + \frac{1}{\acute{c}} \leq \frac{2}{\beta_t L_{\mu_g}}.
$$

Based on (53) and (54), we get

$$
\mathbb{E}\left[\|\mathbf{y}_{t+1} - \mathbf{y}_{t+1}^*(\mathbf{x}_{t+1})\|^2\right] - \mathbb{E}\left[\|\mathbf{y}_t - \mathbf{y}_t^*(\mathbf{x}_t)\|^2\right]
$$
$$
\leq -\frac{\beta_t L_{\mu_g}}{2}\mathbb{E}\left[\|\mathbf{y}_t - \mathbf{y}_t^*(\mathbf{x}_t)\|^2\right] + \left(-\left(\frac{2\beta_t}{\mu_g+\ell_{g,1}} - \beta_t^2\right)\right)\mathbb{E}\left[\|\nabla_\mathbf{y} g_t(\mathbf{x}_t,\mathbf{y}_t)\|^2\right]
$$
$$
+ \frac{2}{\beta_t L_{\mu_g}}\beta_t^2\mathbb{E}\left[\|e_t^g\|^2\right] + \frac{2}{\beta_t L_{\mu_g}}\mathbb{E}\left[\|\mathbf{y}_{t+1}^*(\mathbf{x}_{t+1}) - \mathbf{y}_t^*(\mathbf{x}_t)\|^2\right]. \tag{55}
$$

Next, we upper-bound the last term of the above inequality.

$$
\mathbb{E}\left[\|\mathbf{y}_{t+1}^*(\mathbf{x}_{t+1}) - \mathbf{y}_t^*(\mathbf{x}_t)\|^2\right]
$$
$$
\leq 2\left(\mathbb{E}\left[\|\mathbf{y}_{t+1}^*(\mathbf{x}_{t+1}) - \mathbf{y}_{t+1}^*(\mathbf{x}_t)\|^2\right] + \mathbb{E}\left[\|\mathbf{y}_{t+1}^*(\mathbf{x}_t) - \mathbf{y}_t^*(\mathbf{x}_t)\|^2\right]\right)
$$
$$
\leq 2\left(L_\mathbf{y}^2\mathbb{E}\left[\|\mathbf{x}_t - \mathbf{x}_{t+1}\|^2 + \|\mathbf{y}_{t+1}^*(\mathbf{x}_t) - \mathbf{y}_t^*(\mathbf{x}_t)\|^2\right]\right), \tag{56}
$$

where the second inequality is by Eq. (39d) in Lemma C.1.

Substituting (56) into (55) and summing over $t \in [T]$, give the desired result.

$\square$

## C.3 Bounds on the Linear System Solution

**Lemma C.5.** *Suppose Assumptions 2.2 and B3. hold. Then, for the sequence $\{(\mathbf{x}_t, \mathbf{y}_t, \mathbf{v}_t)\}_{t=1}^T$ generated by Algorithm 1, we have*

$$
\mathbb{E}\|\mathbf{v}_{t+1} - \mathbf{v}_t^*(\mathbf{x}_t)\|^2 \leq (1+\acute{c})\left(1 - 2\delta_t\frac{(\ell_{g,1}+\ell_{g,1}^3)\mu_g}{\mu_g+\ell_{g,1}} + \delta_t^2\ell_{g,1}^2\right)\mathbb{E}\|\mathbf{v}_t - \mathbf{v}_t^*(\mathbf{x}_t)\|^2
$$
$$
+ (1+\frac{1}{\acute{c}})\delta_t^2\mathbb{E}\|e_t^\mathbf{v}\|^2,
$$

*for any $\acute{c} > 0$, where $\mathbf{v}_t^*(\mathbf{x}_t)$ is the solution of the system in Eq. (4), and $e_t^\mathbf{v}$ is defined in (60).*

*Proof.* From the update rules in Algorithm 1, we have the following:

$$\mathbb{E}\|\mathbf{v}_{t+1} - \mathbf{v}_t^*(\mathbf{x}_t)\|^2 = \mathbb{E}\|\Pi_{\mathcal{Z}_p}\left[\mathbf{v}_t - \delta_t\mathbf{d}_t^{\mathbf{v}}\right] - \Pi_{\mathcal{Z}_p}\left[\mathbf{v}_t^*(\mathbf{x}_t)\right]\|^2$$
$$\leq \mathbb{E}\|\mathbf{v}_t - \delta_t\mathbf{d}_t^{\mathbf{v}} - \mathbf{v}_t^*(\mathbf{x}_t)\|^2$$
$$\leq (1+\acute{c})\mathbb{E}\|\mathbf{v}_t - \delta_t\nabla P_t(\mathbf{x}_t, \mathbf{y}_t^*(\mathbf{x}_t), \mathbf{v}_t) - \mathbf{v}_t^*(\mathbf{x}_t)\|^2$$
$$+ (1+\frac{1}{\acute{c}})\delta_t^2\mathbb{E}\|\mathbf{d}_t^{\mathbf{v}} - \nabla P_t(\mathbf{x}_t, \mathbf{y}_t^*(\mathbf{x}_t), \mathbf{v}_t)\|^2, \tag{57}$$

where $\nabla P_t(\mathbf{x}_t, \mathbf{y}_t^*(\mathbf{x}_t), \mathbf{v}_t) := \nabla_{\mathbf{y}}^2 g_t\left(\mathbf{x}_t, \mathbf{y}_t^*(\mathbf{x}_t)\right)\mathbf{v}_t + \nabla_{\mathbf{y}}f_t(\mathbf{x}_t, \mathbf{y}_t^*(\mathbf{x}_t))$.

For the first term of Eq. (57) above, we have

$$\mathbb{E}\|\mathbf{v}_t - \delta_t\nabla P_t(\mathbf{x}_t, \mathbf{y}_t^*(\mathbf{x}_t), \mathbf{v}_t) - \mathbf{v}_t^*(\mathbf{x}_t)\|^2$$
$$= \mathbb{E}\|\mathbf{v}_t - \mathbf{v}_t^*(\mathbf{x}_t)\|^2 - 2\delta_t\mathbb{E}\langle\mathbf{v}_t - \mathbf{v}_t^*(\mathbf{x}_t), \nabla P_t(\mathbf{x}_t, \mathbf{y}_t^*(\mathbf{x}_t), \mathbf{v}_t)\rangle + \delta_t^2\mathbb{E}\|\nabla P_t(\mathbf{x}_t, \mathbf{y}_t^*(\mathbf{x}_t), \mathbf{v}_t)\|^2$$
$$\leq \left(1 - 2\delta_t\frac{\mu_g\ell_{g,1}}{\mu_g + \ell_{g,1}}\right)\mathbb{E}\|\mathbf{v}_t - \mathbf{v}_t^*(\mathbf{x}_t)\|^2 - (2\delta_t\frac{\mu_g\ell_{g,1}}{\mu_g + \ell_{g,1}} - \delta_t^2)\mathbb{E}\|\nabla P_t(\mathbf{x}_t, \mathbf{y}_t^*(\mathbf{x}_t), \mathbf{v}_t)\|^2$$
$$\leq \left(1 - 2\delta_t\frac{(\ell_{g,1} + \ell_{g,1}^3)\mu_g}{\mu_g + \ell_{g,1}} + \delta_t^2\ell_{g,1}^2\right)\mathbb{E}\|\mathbf{v}_t - \mathbf{v}_t^*(\mathbf{x}_t)\|^2, \tag{58}$$

where the first inequality follows from the strong convexity of the function $P_t$, which is the gradient of the strongly convex quadratic program $\frac{1}{2}\mathbf{v}^\top\nabla_{\mathbf{y}}^2 g_t\left(\mathbf{x}, \mathbf{y}_t^*(\mathbf{x})\right)\mathbf{v} + \mathbf{v}^\top\nabla_{\mathbf{y}}f_t(\mathbf{x}, \mathbf{y}_t^*(\mathbf{x}))$. Then, we have

$$\mathbb{E}\langle\mathbf{v}_t - \mathbf{v}_t^*(\mathbf{x}_t), \nabla P_t(\mathbf{x}_t, \mathbf{y}_t^*(\mathbf{x}_t), \mathbf{v}_t)\rangle \geq \frac{\mu_g\ell_{g,1}}{\mu_g + \ell_{g,1}}\mathbb{E}\|\mathbf{v}_t - \mathbf{v}_t^*(\mathbf{x}_t)\|^2$$
$$+ \frac{1}{\mu_g + \ell_{g,1}}\mathbb{E}\|\nabla P_t(\mathbf{x}_t, \mathbf{y}_t^*(\mathbf{x}_t), \mathbf{v}_t)\|^2.$$

The second inequality is derived from the following inequality.

$$\mathbb{E}\|\nabla P_t(\mathbf{x}_t, \mathbf{y}_t^*(\mathbf{x}_t), \mathbf{v}_t)\|^2 = \mathbb{E}\|\nabla_{\mathbf{y}}^2 g_t\left(\mathbf{x}_t, \mathbf{y}_t^*(\mathbf{x}_t)\right)\mathbf{v}_t + \nabla_{\mathbf{y}}f_t(\mathbf{x}_t, \mathbf{y}_t^*(\mathbf{x}_t))\|^2$$
$$= \mathbb{E}\|\nabla_{\mathbf{y}}^2 g_t\left(\mathbf{x}_t, \mathbf{y}_t^*(\mathbf{x}_t)\right)\left(\mathbf{v}_t - \mathbf{v}_t^*(\mathbf{x}_t)\right)\|^2$$
$$\leq \ell_{g,1}^2\mathbb{E}\|\mathbf{v}_t - \mathbf{v}_t^*(\mathbf{x}_t)\|^2, \tag{59}$$

where the second equality follows from (4).
Combining (57) and (58), we get the desired result. $\qquad\square$

**Lemma C.6.** *Suppose Assumptions B2., B3., B4., C2. and C4. hold. Let $\{(\mathbf{x}_t, \mathbf{y}_t, \mathbf{v}_t)\}_{t=1}^T$ be generated according to Algorithm 1. For $e_{t+1}^{\mathbf{v}}$ defined as*

$$e_t^{\mathbf{v}} := \mathbf{d}_t^{\mathbf{v}} - \nabla P_t(\mathbf{x}_t, \mathbf{y}_t, \mathbf{v}_t), \qquad \text{where} \tag{60a}$$
$$\nabla P_t(\mathbf{x}_t, \mathbf{y}_t, \mathbf{v}_t) := \nabla_{\mathbf{y}}^2 g_t\left(\mathbf{x}_t, \mathbf{y}_t\right)\mathbf{v}_t + \nabla_{\mathbf{y}}f_t(\mathbf{x}_t, \mathbf{y}_t), \tag{60b}$$

*we have:*

$$\mathbb{E}\|e_{t+1}^{\mathbf{v}}\|^2 \leq (1 - \lambda_{t+1})^2(1 + 72\ell_{g,1}^2\delta_t^2)\mathbb{E}\|e_t^{\mathbf{v}}\|^2 + 4\lambda_{t+1}^2(\frac{\sigma_{g_{\mathbf{yy}}}^2}{\bar{b}}p^2 + \frac{\sigma_{f_{\mathbf{y}}}^2}{b})$$
$$+ 12p^2(1 - \lambda_{t+1})^2\mathbb{E}\|\nabla_{\mathbf{y}}^2 g_t\left(\mathbf{x}_{t+1}, \mathbf{y}_{t+1}\right) - \nabla_{\mathbf{y}}^2 g_{t+1}\left(\mathbf{x}_{t+1}, \mathbf{y}_{t+1}\right)\|^2$$
$$+ 12(1 - \lambda_{t+1})^2\mathbb{E}\|\nabla_{\mathbf{y}}f_t(\mathbf{x}_{t+1}, \mathbf{y}_{t+1}) - \nabla_{\mathbf{y}}f_{t+1}(\mathbf{x}_{t+1}, \mathbf{y}_{t+1})\|^2$$
$$+ 72(1 - \lambda_{t+1})^2(\ell_{g,2}^2p^2 + \ell_{f,1}^2)\left(\mathbb{E}\|\mathbf{x}_{t+1} - \mathbf{x}_t\|^2 + 2\beta_t^2\mathbb{E}\|e_t^g\|^2 + 2\beta_t^2\mathbb{E}\|\nabla_{\mathbf{y}}g_t(\mathbf{x}_t, \mathbf{y}_t)\|^2\right)$$
$$+ 144(1 - \lambda_{t+1})^2\ell_{g,1}^4\delta_t^2\mathbb{E}[\theta_t^{\mathbf{v}}] + 288\ell_{g,1}^2(p^2\ell_{g,2}^2 + \ell_{f,1}^2)\delta_t^2\mathbb{E}[\theta_t^{\mathbf{y}}], \tag{61}$$

*for all $t \in [T]$ and $(\theta_t^{\mathbf{v}}, \theta_t^{\mathbf{y}})$ and $e_t^g$ are defined in (49) and (45), respectively.*

*Proof.* Note that

$$e_{t+1}^{\mathbf{v}} := \mathbf{d}_{t+1}^{\mathbf{v}} - \nabla P_{t+1}(\mathbf{x}_{t+1}, \mathbf{y}_{t+1}, \mathbf{v}_{t+1}),$$

where

$$\nabla P_{t+1}(\mathbf{x}_{t+1}, \mathbf{y}_{t+1}, \mathbf{v}_{t+1}) := \nabla_{\mathbf{y}}^2 g_{t+1}\left(\mathbf{x}_{t+1}, \mathbf{y}_{t+1}\right) \mathbf{v}_{t+1} + \nabla_{\mathbf{y}} f_{t+1}(\mathbf{x}_{t+1}, \mathbf{y}_{t+1}).$$

From Algorithm 1, we have

$$\mathbf{d}_{t+1}^{\mathbf{v}} = \mathbf{d}_{t+1}^{\mathbf{vv}}\left(\mathbf{x}_{t+1}, \mathbf{y}_{t+1}; \mathcal{B}_{t+1}\right) + (1 - \lambda_{t+1})(\mathbf{d}_t^{\mathbf{v}} - \mathbf{d}_{t+1}^{\mathbf{vv}}(\mathbf{x}_t, \mathbf{y}_t; \mathcal{B}_{t+1})).$$

Let $\mathbf{u} = [\mathbf{x}; \mathbf{y}; \mathbf{v}]$. Then, we have

$$
\begin{aligned}
\mathbb{E}\|e_{t+1}^{\mathbf{v}}\|^2 &= \mathbb{E}\|\mathbf{d}_{t+1}^{\mathbf{v}} - \nabla P_{t+1}(\mathbf{u}_{t+1})\|^2 \\
&= \mathbb{E}\|\nabla P_{t+1}(\mathbf{u}_{t+1}; \mathcal{B}_{t+1}) + (1 - \lambda_{t+1})(\mathbf{d}_t^{\mathbf{v}} - \nabla P_{t+1}(\mathbf{u}_t; \mathcal{B}_{t+1})) - \nabla P_{t+1}(\mathbf{u}_{t+1})\|^2 \\
&= \mathbb{E}\|(1 - \lambda_{t+1})e_t^{\mathbf{v}} + \nabla P_{t+1}(\mathbf{u}_{t+1}; \mathcal{B}_{t+1}) - \nabla P_{t+1}(\mathbf{u}_{t+1}) \\
&\quad - (1 - \lambda_{t+1})\left(\nabla P_{t+1}(\mathbf{u}_t; \mathcal{B}_{t+1}) - \nabla P_t(\mathbf{u}_t)\right)\|^2,
\end{aligned}
$$

which implies that

$$
\begin{aligned}
&\mathbb{E}\|e_{t+1}^{\mathbf{v}}\|^2 \\
&= (1 - \lambda_{t+1})^2 \mathbb{E}\|e_t^{\mathbf{v}}\|^2 + \mathbb{E}\|\lambda_{t+1}\left(\nabla P_{t+1}(\mathbf{u}_{t+1}; \mathcal{B}_{t+1}) - \nabla P_{t+1}(\mathbf{u}_{t+1})\right) \\
&\quad - (1 - \lambda_{t+1})\left(\nabla P_{t+1}(\mathbf{u}_t; \mathcal{B}_{t+1}) - \nabla P_{t+1}(\mathbf{u}_{t+1}; \mathcal{B}_{t+1}) + \nabla P_{t+1}(\mathbf{u}_{t+1}) - \nabla P_t(\mathbf{u}_t)\right)\|^2 \\
&\leq (1 - \lambda_{t+1})^2 \mathbb{E}\|e_t^{\mathbf{v}}\|^2 + 2\lambda_{t+1}^2 \mathbb{E}\|\nabla P_{t+1}(\mathbf{u}_{t+1}; \mathcal{B}_{t+1}) - \nabla P_{t+1}(\mathbf{u}_{t+1})\|^2 \\
&\quad + 2(1 - \lambda_{t+1})^2 \mathbb{E}\|\nabla P_{t+1}(\mathbf{u}_{t+1}; \mathcal{B}_{t+1}) - \nabla P_{t+1}(\mathbf{u}_{t+1}) - \nabla P_{t+1}(\mathbf{u}_t; \mathcal{B}_{t+1}) + \nabla P_t(\mathbf{u}_t)\|^2,
\end{aligned}
$$

where the inequality follows from Cauchy–Schwartz inequality.
For the first term, from Assumptions C2. and C4., we have

$$
\begin{aligned}
&\mathbb{E}\|\nabla P_{t+1}(\mathbf{u}_{t+1}; \mathcal{B}_{t+1}) - \nabla P_{t+1}(\mathbf{u}_{t+1})\|^2 \\
&= \mathbb{E}\|\left(\nabla_{\mathbf{y}}^2 g_{t+1}\left(\mathbf{x}_{t+1}, \mathbf{y}_{t+1}; \bar{\mathcal{B}}_{t+1}\right) - \nabla_{\mathbf{y}}^2 g_{t+1}\left(\mathbf{x}_{t+1}, \mathbf{y}_{t+1}\right)\right) \mathbf{v}_{t+1} \\
&\quad + \nabla_{\mathbf{y}} f_{t+1}(\mathbf{x}_{t+1}, \mathbf{y}_{t+1}; \mathcal{B}_{t+1}) - \nabla_{\mathbf{y}} f_{t+1}(\mathbf{x}_{t+1}, \mathbf{y}_{t+1})\|^2 \\
&\leq 2\mathbb{E}\|\left(\nabla_{\mathbf{y}}^2 g_{t+1}\left(\mathbf{x}_{t+1}, \mathbf{y}_{t+1}; \bar{\mathcal{B}}_{t+1}\right) - \nabla_{\mathbf{y}}^2 g_{t+1}\left(\mathbf{x}_{t+1}, \mathbf{y}_{t+1}\right)\right) \mathbf{v}_{t+1}\|^2 \\
&\quad + 2\mathbb{E}\|\nabla_{\mathbf{y}} f_{t+1}(\mathbf{x}_{t+1}, \mathbf{y}_{t+1}; \mathcal{B}_{t+1}) - \nabla_{\mathbf{y}} f_{t+1}(\mathbf{x}_{t+1}, \mathbf{y}_{t+1})\|^2 \\
&\leq 2(\frac{\sigma_{g_{\mathbf{yy}}}^2}{\bar{b}}p^2 + \frac{\sigma_{f_{\mathbf{y}}}^2}{b}),
\end{aligned}
$$

where the last inequality follows from (8).
Then, from the above inequality and $\|a + b + c\|^2 \leq 3(\|a\|^2 + \|b\|^2 + \|c\|^2)$, we have

$$
\begin{aligned}
\mathbb{E}\|e_{t+1}^{\mathbf{v}}\|^2 &\leq (1 - \lambda_{t+1})^2 \mathbb{E}\|e_t^{\mathbf{v}}\|^2 + 4\lambda_{t+1}^2(\frac{\sigma_{g_{\mathbf{yy}}}^2}{\bar{b}}p^2 + \frac{\sigma_{f_{\mathbf{y}}}^2}{b}) \\
&\quad + 6(1 - \lambda_{t+1})^2 \mathbb{E}\|\nabla P_t(\mathbf{u}_t) - \nabla P_t(\mathbf{u}_{t+1})\|^2 \\
&\quad + 6(1 - \lambda_{t+1})^2 \mathbb{E}\|\nabla P_t(\mathbf{u}_{t+1}) - \nabla P_{t+1}(\mathbf{u}_{t+1})\|^2 \\
&\quad + 6(1 - \lambda_{t+1})^2 \mathbb{E}\|\nabla P_{t+1}(\mathbf{u}_{t+1}; \mathcal{B}_{t+1}) - \nabla P_{t+1}(\mathbf{u}_t; \mathcal{B}_{t+1})\|^2. \quad (62)
\end{aligned}
$$

Moreover, from $\|a + b + c\|^2 \leq 3(\|a\|^2 + \|b\|^2 + \|c\|^2)$, we have

$$
\begin{aligned}
&\mathbb{E}\|\nabla P_t(\mathbf{u}_{t+1}) - \nabla P_t(\mathbf{u}_t)\|^2 \\
&\leq 3\mathbb{E}\|\nabla P_t(\mathbf{x}_{t+1}, \mathbf{y}_{t+1}, \mathbf{v}_{t+1}) - \nabla P_t(\mathbf{x}_t, \mathbf{y}_{t+1}, \mathbf{v}_{t+1})\|^2 \\
&\quad + 3\mathbb{E}\|\nabla P_t(\mathbf{x}_t, \mathbf{y}_{t+1}, \mathbf{v}_{t+1}) - \nabla P_t(\mathbf{x}_t, \mathbf{y}_t, \mathbf{v}_{t+1})\|^2 \\
&\quad + 3\mathbb{E}\|\nabla P_t(\mathbf{x}_t, \mathbf{y}_t, \mathbf{v}_{t+1}) - \nabla P_t(\mathbf{x}_t, \mathbf{y}_t, \mathbf{v}_t)\|^2 \\
&\leq 3\mathbb{E}\|(\nabla_{\mathbf{y}}^2 g_t\left(\mathbf{x}_{t+1}, \mathbf{y}_{t+1}\right) - \nabla_{\mathbf{y}}^2 g_t\left(\mathbf{x}_t, \mathbf{y}_{t+1}\right))\mathbf{v}_{t+1} + \nabla_{\mathbf{y}} f_t(\mathbf{x}_{t+1}, \mathbf{y}_{t+1}) - \nabla_{\mathbf{y}} f_t(\mathbf{x}_t, \mathbf{y}_{t+1})\|^2 \\
&\quad + 3\mathbb{E}\|(\nabla_{\mathbf{y}}^2 g_t\left(\mathbf{x}_t, \mathbf{y}_{t+1}\right) - \nabla_{\mathbf{y}}^2 g_t\left(\mathbf{x}_t, \mathbf{y}_t\right))\mathbf{v}_{t+1} + \nabla_{\mathbf{y}} f_t(\mathbf{x}_t, \mathbf{y}_{t+1}) - \nabla_{\mathbf{y}} f_t(\mathbf{x}_t, \mathbf{y}_t)\|^2 \\
&\quad + 3\mathbb{E}\|\nabla P_t(\mathbf{x}_t, \mathbf{y}_t, \mathbf{v}_{t+1}) - \nabla P_t(\mathbf{x}_t, \mathbf{y}_t, \mathbf{v}_t)\|^2 \\
&\leq 6(\ell_{g,2}^2 \mathbb{E}\|\mathbf{v}_{t+1}\|^2 + \ell_{f,1}^2)\left(\mathbb{E}\|\mathbf{x}_{t+1} - \mathbf{x}_t\|^2 + \mathbb{E}\|\mathbf{y}_{t+1} - \mathbf{y}_t\|^2\right) + 3\ell_{g,1}^2 \mathbb{E}\|\mathbf{v}_{t+1} - \mathbf{v}_t\|^2, \quad (63)
\end{aligned}
$$

where the last inequality follows from Assumptions B2., B3. and B4.;

From Eq. (63) and the inequality $\|a + b\|^2 \leq 2(\|a\|^2 + \|b\|^2)$, we obtain

$$
\begin{aligned}
&\mathbb{E}\|\nabla P_t(\mathbf{u}_{t+1}) - \nabla P_t(\mathbf{u}_t)\|^2 \\
&\leq 6(\ell_{g,2}^2 p^2 + \ell_{f,1}^2)\left(\mathbb{E}\|\mathbf{x}_{t+1} - \mathbf{x}_t\|^2 + \beta_t^2 \mathbb{E}\|\mathbf{d}_t^{\mathbf{y}}\|^2\right) + 3\ell_{g,1}^2 \delta_t^2 \mathbb{E}\|\mathbf{d}_t^{\mathbf{v}}\|^2 \\
&\leq 6(\ell_{g,2}^2 p^2 + \ell_{f,1}^2)\left(\mathbb{E}\|\mathbf{x}_{t+1} - \mathbf{x}_t\|^2 + 2\beta_t^2 \mathbb{E}\|e_t^g\|^2 + 2\beta_t^2 \mathbb{E}\|\nabla_{\mathbf{y}} g_t(\mathbf{x}_t, \mathbf{y}_t)\|^2\right) \\
&\quad + 6\ell_{g,1}^2 \delta_t^2 (\mathbb{E}\|e_t^{\mathbf{v}}\|^2 + \mathbb{E}\|\nabla P_t(\mathbf{x}_t, \mathbf{y}_t, \mathbf{v}_t)\|^2) \\
&\leq 6(\ell_{g,2}^2 p^2 + \ell_{f,1}^2)\left(\mathbb{E}\|\mathbf{x}_{t+1} - \mathbf{x}_t\|^2 + 2\beta_t^2 \mathbb{E}\|e_t^g\|^2 + 2\beta_t^2 \mathbb{E}\|\nabla_{\mathbf{y}} g_t(\mathbf{x}_t, \mathbf{y}_t)\|^2\right) \\
&\quad + 6\ell_{g,1}^2 \delta_t^2 (\mathbb{E}\|e_t^{\mathbf{v}}\|^2 + 2\mathbb{E}\|\nabla P_t(\mathbf{x}_t, \mathbf{y}_t^*(\mathbf{x}_t), \mathbf{v}_t)\|^2 + 2\mathbb{E}\|\nabla P_t(\mathbf{x}_t, \mathbf{y}_t, \mathbf{v}_t) - \nabla P_t(\mathbf{x}_t, \mathbf{y}_t^*(\mathbf{x}_t), \mathbf{v}_t)\|^2) \\
&\leq 6(\ell_{g,2}^2 p^2 + \ell_{f,1}^2)\left(\mathbb{E}\|\mathbf{x}_{t+1} - \mathbf{x}_t\|^2 + 2\beta_t^2 \mathbb{E}\|e_t^g\|^2 + 2\beta_t^2 \mathbb{E}\|\nabla_{\mathbf{y}} g_t(\mathbf{x}_t, \mathbf{y}_t)\|^2\right) \\
&\quad + 6\ell_{g,1}^2 \delta_t^2 \left(\mathbb{E}\|e_t^{\mathbf{v}}\|^2 + 2\ell_{g,1}^2 \mathbb{E}\|\mathbf{v}_t - \mathbf{v}_t^*(\mathbf{x}_t)\|^2 + 4(p^2 \ell_{g,2}^2 + \ell_{f,1}^2)\mathbb{E}\|\mathbf{y}_t - \mathbf{y}_t^*(\mathbf{x}_t)\|^2\right),
\end{aligned}
\tag{64}
$$

where the last inequality follows from (59).
Similarly, we have

$$
\begin{aligned}
&\mathbb{E}\|\nabla P_{t+1}(\mathbf{u}_{t+1}; \mathcal{B}_{t+1}) - \nabla P_{t+1}(\mathbf{u}_t; \mathcal{B}_{t+1})\|^2 \\
&\leq 6(\ell_{g,2}^2 p^2 + \ell_{f,1}^2)\left(\mathbb{E}\|\mathbf{x}_{t+1} - \mathbf{x}_t\|^2 + 2\beta_t^2 \mathbb{E}\|e_t^g\|^2 + 2\beta_t^2 \mathbb{E}\|\nabla_{\mathbf{y}} g_t(\mathbf{x}_t, \mathbf{y}_t)\|^2\right) \\
&\quad + 6\ell_{g,1}^2 \delta_t^2 \left(\mathbb{E}\|e_t^{\mathbf{v}}\|^2 + 2\ell_{g,1}^2 \mathbb{E}\|\mathbf{v}_t - \mathbf{v}_t^*(\mathbf{x}_t)\|^2 + 4(p^2 \ell_{g,2}^2 + \ell_{f,1}^2)\mathbb{E}\|\mathbf{y}_t - \mathbf{y}_t^*(\mathbf{x}_t)\|^2\right).
\end{aligned}
\tag{65}
$$

Substituting (65) and (64) into (62), we have

$$
\begin{aligned}
\mathbb{E}\|e_{t+1}^{\mathbf{v}}\|^2 &\leq (1 - \lambda_{t+1})^2(1 + 72\ell_{g,1}^2 \delta_t^2)\mathbb{E}\|e_t^{\mathbf{v}}\|^2 + 4\lambda_{t+1}^2\left(\frac{\sigma_{g_{\mathbf{y}\mathbf{y}}}^2}{\bar{b}}p^2 + \frac{\sigma_{f_{\mathbf{y}}}^2}{b}\right) \\
&\quad + 6(1 - \lambda_{t+1})^2 \mathbb{E}\|\nabla P_t(\mathbf{u}_{t+1}) - \nabla P_{t+1}(\mathbf{u}_{t+1})\|^2 \\
&\quad + 72(1 - \lambda_{t+1})^2(\ell_{g,2}^2 p^2 + \ell_{f,1}^2)\left(\mathbb{E}\|\mathbf{x}_{t+1} - \mathbf{x}_t\|^2 + 2\beta_t^2 \mathbb{E}\|e_t^g\|^2 + 2\beta_t^2 \mathbb{E}\|\nabla_{\mathbf{y}} g_t(\mathbf{x}_t, \mathbf{y}_t)\|^2\right) \\
&\quad + 144(1 - \lambda_{t+1})^2 \ell_{g,1}^4 \delta_t^2 \mathbb{E}\|\mathbf{v}_t - \mathbf{v}_t^*(\mathbf{x}_t)\|^2 + 288\ell_{g,1}^2(p^2 \ell_{g,2}^2 + \ell_{f,1}^2)\delta_t^2 \mathbb{E}\|\mathbf{y}_t - \mathbf{y}_t^*(\mathbf{x}_t)\|^2.
\end{aligned}
$$

From $\|a + b\|^2 \leq 2\|a\|^2 + 2\|b\|^2$ and (8), we have

$$
\begin{aligned}
\mathbb{E}\|\nabla P_t(\mathbf{u}_{t+1}) - \nabla P_{t+1}(\mathbf{u}_{t+1})\|^2 &= \mathbb{E}\|\nabla_{\mathbf{y}}^2 g_t(\mathbf{x}_{t+1}, \mathbf{y}_{t+1})\mathbf{v}_{t+1} - \nabla_{\mathbf{y}}^2 g_{t+1}(\mathbf{x}_{t+1}, \mathbf{y}_{t+1})\mathbf{v}_{t+1} \\
&\quad + \nabla_{\mathbf{y}} f_t(\mathbf{x}_{t+1}, \mathbf{y}_{t+1}) - \nabla_{\mathbf{y}} f_{t+1}(\mathbf{x}_{t+1}, \mathbf{y}_{t+1})\|^2 \\
&\leq 2\mathbb{E}\|\left(\nabla_{\mathbf{y}}^2 g_t(\mathbf{x}_{t+1}, \mathbf{y}_{t+1}) - \nabla_{\mathbf{y}}^2 g_{t+1}(\mathbf{x}_{t+1}, \mathbf{y}_{t+1})\right)\mathbf{v}_{t+1}\|^2 \\
&\quad + 2\mathbb{E}\|\nabla_{\mathbf{y}} f_t(\mathbf{x}_{t+1}, \mathbf{y}_{t+1}) - \nabla_{\mathbf{y}} f_{t+1}(\mathbf{x}_{t+1}, \mathbf{y}_{t+1})\|^2 \\
&\leq 2\mathbb{E}\|\nabla_{\mathbf{y}}^2 g_t(\mathbf{x}_{t+1}, \mathbf{y}_{t+1}) - \nabla_{\mathbf{y}}^2 g_{t+1}(\mathbf{x}_{t+1}, \mathbf{y}_{t+1})\|^2 p^2 \\
&\quad + 2\mathbb{E}\|\nabla_{\mathbf{y}} f_t(\mathbf{x}_{t+1}, \mathbf{y}_{t+1}) - \nabla_{\mathbf{y}} f_{t+1}(\mathbf{x}_{t+1}, \mathbf{y}_{t+1})\|^2.
\end{aligned}
$$

This completes the proof. □

As demonstrated in Lemma C.6, the gradient estimation error $e_{t+1}^{\mathbf{v}}$ for the linear system consists of four key components: (1) an iteratively refined error term $(1 - \lambda_{t+1})^2(1 + 72\ell_{g,1}^2 \delta_t^2)\mathbb{E}\|e_t^{\mathbf{v}}\|^2$, which depends on the stepsize $\delta_t$; (2) the error arising from the variation in the Hessian of the lower-level objectiv; (3) the error resulting from the variation in the gradient of the upper-level objective, and (4) approximation error terms of order $\mathcal{O}(\delta_t^2 \mathbb{E}[\theta_t^{\mathbf{v}}])$ and $\mathcal{O}(\delta_t^2 \mathbb{E}[\theta_t^{\mathbf{y}}])$ associated with solving the linear system and the iterates in the inner problem, respectively.

**Lemma C.7.** *Suppose Assumptions 2.2, B1., B2. and B4. hold. Let $\mathbf{v}_t^*(\mathbf{x})$ is a solution of Subproblem (4). Then, we have*

$$
\left\|\mathbf{v}_t^*(\mathbf{x}_t) - \mathbf{v}_{t+1}^*(\mathbf{x}_{t+1})\right\|^2 \leq 2\frac{\nu^2}{\mu_g^2}\left(\left\|\mathbf{y}_{t+1}^*(\mathbf{x}_{t+1}) - \mathbf{y}_t^*(\mathbf{x}_t)\right\|^2 + \|\mathbf{x}_{t+1} - \mathbf{x}_t\|^2\right),
$$

*where $\nu := \ell_{f,1} + \frac{\ell_{g,2}\ell_{f,0}}{\mu_g}$.*

*Proof.* Based on (4), we have that

$$\left\| \mathbf{v}_t^*(\mathbf{x}_t) - \mathbf{v}_{t+1}^*(\mathbf{x}_{t+1}) \right\|^2$$

$$= \| \left( \nabla_{\mathbf{y}}^2 g_t(\mathbf{x}_t, \mathbf{y}_t^*(\mathbf{x}_t)) \right)^{-1} \nabla_{\mathbf{y}} f_t(\mathbf{x}_t, \mathbf{y}_t^*(\mathbf{x}_t))$$

$$- \left( \nabla_{\mathbf{y}}^2 g_{t+1}(\mathbf{x}_{t+1}, \mathbf{y}_{t+1}^*(\mathbf{x}_{t+1})) \right)^{-1} \nabla_{\mathbf{y}} f_{t+1}(\mathbf{x}_{t+1}, \mathbf{y}_{t+1}^*(\mathbf{x}_{t+1})) \|^2$$

$$\leq 2 \left\| \left( \left( \nabla_{\mathbf{y}}^2 g_t(\mathbf{x}_t, \mathbf{y}_t^*(\mathbf{x}_t)) \right)^{-1} - \left( \nabla_{\mathbf{y}}^2 g_{t+1}(\mathbf{x}_{t+1}, \mathbf{y}_{t+1}^*(\mathbf{x}_{t+1})) \right)^{-1} \right) \nabla_{\mathbf{y}} f_t(\mathbf{x}_t, \mathbf{y}_t^*(\mathbf{x}_t)) \right\|^2 \quad \text{(66a)}$$

$$+ 2 \left\| \left( \nabla_{\mathbf{y}}^2 g_{t+1}(\mathbf{x}_{t+1}, \mathbf{y}_{t+1}^*(\mathbf{x}_{t+1})) \right)^{-1} \left( \nabla_{\mathbf{y}} f_t(\mathbf{x}_t, \mathbf{y}_t^*(\mathbf{x}_t)) - \nabla_{\mathbf{y}} f_{t+1}(\mathbf{x}_{t+1}, \mathbf{y}_{t+1}^*(\mathbf{x}_{t+1})) \right) \right\|^2. \quad \text{(66b)}$$

In the following steps, we bound the terms (66a) and (66b), respectively.

For (66a), we have:

$$\left\| \left( \nabla_{\mathbf{y}}^2 g_t(\mathbf{x}_t, \mathbf{y}_t^*(\mathbf{x}_t)) \right)^{-1} - \left( \nabla_{\mathbf{y}}^2 g_{t+1}(\mathbf{x}_{t+1}, \mathbf{y}_{t+1}^*(\mathbf{x}_{t+1})) \right)^{-1} \right\|^2$$

$$= \| \left( \nabla_{\mathbf{y}}^2 g_t(\mathbf{x}_t, \mathbf{y}_t^*(\mathbf{x}_t)) \right)^{-1} \left( \nabla_{\mathbf{y}}^2 g_{t+1}(\mathbf{x}_{t+1}, \mathbf{y}_{t+1}^*(\mathbf{x}_{t+1})) \right)$$

$$- \nabla_{\mathbf{y}}^2 g_t(\mathbf{x}_t, \mathbf{y}_t^*(\mathbf{x}_t)) \right) \left( \nabla_{\mathbf{y}}^2 g_{t+1}(\mathbf{x}_{t+1}, \mathbf{y}_{t+1}^*(\mathbf{x}_{t+1})) \right)^{-1} \|^2$$

$$\leq \frac{1}{\mu_g^2} \left\| \nabla_{\mathbf{y}}^2 g_t(\mathbf{x}_t, \mathbf{y}_t^*(\mathbf{x}_t)) - \nabla_{\mathbf{y}}^2 g_{t+1}(\mathbf{x}_{t+1}, \mathbf{y}_{t+1}^*(\mathbf{x}_{t+1})) \right\|^2$$

$$\leq \frac{\ell_{g,2}}{\mu_g^2} \left\| (\mathbf{x}_t, \mathbf{y}_t^*(\mathbf{x}_t)) - (\mathbf{x}_{t+1}, \mathbf{y}_{t+1}^*(\mathbf{x}_{t+1})) \right\|^2$$

$$\leq \frac{\ell_{g,2}}{\mu_g^2} \left( \left\| \mathbf{y}_t^*(\mathbf{x}_t) - \mathbf{y}_{t+1}^*(\mathbf{x}_{t+1}) \right\|^2 + \left\| \mathbf{x}_t - \mathbf{x}_{t+1} \right\|^2 \right), \quad \text{(67)}$$

where the equality holds since for any invertible matrix $\mathbf{A}$ and $\mathbf{B}$ we have $\|\mathbf{A}^{-1} - \mathbf{B}^{-1}\| = \|\mathbf{A}^{-1}(\mathbf{B} - \mathbf{A})\mathbf{B}^{-1}\|$, and inequalities are obtained from Assumptions 2.2 and B4..

Thus, from (67) and Assumption B1., we get

$$(66a) \leq \frac{\ell_{f,0}\ell_{g,2}}{\mu_g^2} \left( \left\| \mathbf{y}_t^*(\mathbf{x}_t) - \mathbf{y}_{t+1}^*(\mathbf{x}_{t+1}) \right\|^2 + \left\| \mathbf{x}_t - \mathbf{x}_{t+1} \right\|^2 \right). \quad \text{(68)}$$

For (66b), we have

$$(66b) \leq \frac{1}{\mu_g} \| \nabla_{\mathbf{y}} f_t(\mathbf{x}_t, \mathbf{y}_t^*(\mathbf{x}_t)) - \nabla_{\mathbf{y}} f_{t+1}(\mathbf{x}_{t+1}, \mathbf{y}_{t+1}^*(\mathbf{x}_{t+1})) \|^2$$

$$\leq \frac{\ell_{f,1}}{\mu_g} \| (\mathbf{x}_t, \mathbf{y}_t^*(\mathbf{x}_t)) - (\mathbf{x}_{t+1}, \mathbf{y}_{t+1}^*(\mathbf{x}_{t+1})) \|^2$$

$$\leq \frac{\ell_{f,1}}{\mu_g} \left( \left\| \mathbf{y}_{t+1}^*(\mathbf{x}_{t+1}) - \mathbf{y}_t^*(\mathbf{x}_t) \right\|^2 + \left\| \mathbf{x}_{t+1} - \mathbf{x}_t \right\|^2 \right). \quad \text{(69)}$$

Combining (68) and (69), we have

$$\left\| \mathbf{v}_t^*(\mathbf{x}_t) - \mathbf{v}_{t+1}^*(\mathbf{x}_{t+1}) \right\|^2 \leq \frac{1}{\mu_g} \left( \frac{\ell_{f,0}\ell_{g,2}}{\mu_g} + \ell_{f,1} \right) \left( \left\| \mathbf{y}_{t+1}^*(\mathbf{x}_{t+1}) - \mathbf{y}_t^*(\mathbf{x}_t) \right\|^2 + \left\| \mathbf{x}_{t+1} - \mathbf{x}_t \right\|^2 \right).$$

By raising both sides of the above inequality to the power 2 and using $(a + b)^2 \leq 2a^2 + 2b^2$, we complete the proof. □

The following lemma characterizes the decrease in $\theta_t^{\mathbf{y}}$ defined in (49) and can be viewed as an extension of the offline BO result in [69] to the OBO setting.

**Lemma C.8.** *Suppose Assumptions 2.2 and 2.3 hold. Let $\theta_t^{\mathbf{y}}$ be defined in (49). Then, for any positive choice of step size $\delta_t$ as*

$$\delta_t \leq \frac{\acute{L}_{\mu_g}}{\ell_{g,1}^2}, \quad \text{where} \quad \acute{L}_{\mu_g} := \frac{(\ell_{g,1} + \ell_{g,1}^3)\mu_g}{(\mu_g + \ell_{g,1})},$$

*for all $t \in [T]$, the sequence $\{(\mathbf{x}_t, \mathbf{y}_t, \mathbf{v}_t)\}_{t=1}^T$ generated by Algorithm 1 satisfy*

$$\sum_{t=1}^T \left( \mathbb{E}[\theta_{t+1}^{\mathbf{v}}] - \mathbb{E}[\theta_t^{\mathbf{v}}] \right) \leq -\frac{\delta_t \acute{L}_{\mu_g}}{4} \sum_{t=1}^T \mathbb{E}[\theta_t^{\mathbf{v}}] + \frac{4}{\acute{L}_{\mu_g}} \delta_t \sum_{t=1}^T \mathbb{E}\|e_t^{\mathbf{v}}\|^2$$

$$+ \frac{16\nu^2}{\acute{L}_{\mu_g} \mu_g^2 \delta_t} \sum_{t=1}^T \mathbb{E}\left\|\mathbf{y}_{t+1}^*(\mathbf{x}_t) - \mathbf{y}_t^*(\mathbf{x}_t)\right\|^2$$

$$+ \frac{8\nu^2}{\acute{L}_{\mu_g} \mu_g^2 \delta_t} (1 + 2L_{\mathbf{y}}^2) \sum_{t=1}^T \mathbb{E}\left\|\mathbf{x}_{t+1} - \mathbf{x}_t\right\|^2, \tag{70}$$

*where $e_t^{\mathbf{v}}$ is defined in (60), $\nu$ and $L_{\mathbf{y}}$, are defined in Lemmas C.7 and C.4, respectively.*

*Proof.* By Lemma B.4, for any $a > 0$, we have

$$\mathbb{E}\left\|\mathbf{v}_{t+1} - \mathbf{v}_{t+1}^*(\mathbf{x}_{t+1})\right\|^2 = \mathbb{E}\left\|\mathbf{v}_{t+1} - \mathbf{v}_t^*(\mathbf{x}_t) + \mathbf{v}_t^*(\mathbf{x}_t) - \mathbf{v}_{t+1}^*(\mathbf{x}_{t+1})\right\|^2$$

$$\leq (1 + a)\, \mathbb{E}\|\mathbf{v}_{t+1} - \mathbf{v}_t^*(\mathbf{x}_t)\|^2$$

$$+ \left(1 + \frac{1}{a}\right) \mathbb{E}\left\|\mathbf{v}_{t+1}^*(\mathbf{x}_{t+1}) - \mathbf{v}_t^*(\mathbf{x}_t)\right\|^2. \tag{71}$$

From Lemma C.5, we have for any $\acute{c} > 0$:

$$\mathbb{E}\|\mathbf{v}_{t+1} - \mathbf{v}_t^*(\mathbf{x}_t)\|^2 \leq (1 + \acute{c}) \left( 1 - 2\delta_t \frac{(\ell_{g,1} + \ell_{g,1}^3)\mu_g}{\mu_g + \ell_{g,1}} + \delta_t^2 \ell_{g,1}^2 \right) \mathbb{E}\|\mathbf{v}_t - \mathbf{v}_t^*(\mathbf{x}_t)\|^2$$

$$+ (1 + \frac{1}{\acute{c}})\delta_t^2 \mathbb{E}\|e_t^{\mathbf{v}}\|^2. \tag{72}$$

Substituting (72) into (71), we get

$$\mathbb{E}\left\|\mathbf{v}_{t+1} - \mathbf{v}_{t+1}^*(\mathbf{x}_{t+1})\right\|^2 \leq (1 + a)\,(1 + \acute{c}) \left( 1 - 2\delta_t \frac{(\ell_{g,1} + \ell_{g,1}^3)\mu_g}{\mu_g + \ell_{g,1}} + \delta_t^2 \ell_{g,1}^2 \right) \mathbb{E}\|\mathbf{v}_t - \mathbf{v}_t^*(\mathbf{x}_t)\|^2$$

$$+ (1 + a)\,(1 + \frac{1}{\acute{c}})\delta_t^2 \mathbb{E}\|e_t^{\mathbf{v}}\|^2$$

$$+ \left(1 + \frac{1}{a}\right) \mathbb{E}\left\|\mathbf{v}_{t+1}^*(\mathbf{x}_{t+1}) - \mathbf{v}_t^*(\mathbf{x}_t)\right\|^2. \tag{73}$$

In the following, we provide a bound for the third term on the right-hand side of (73). To this end, we have from Lemma C.7:

$$\mathbb{E}\left\|\mathbf{v}_{t+1}^*(\mathbf{x}_{t+1}) - \mathbf{v}_t^*(\mathbf{x}_t)\right\|^2 \leq 2\frac{\nu^2}{\mu_g^2} \left( \mathbb{E}\left\|\mathbf{y}_{t+1}^*(\mathbf{x}_{t+1}) - \mathbf{y}_t^*(\mathbf{x}_t)\right\|^2 + \mathbb{E}\|\mathbf{x}_{t+1} - \mathbf{x}_t\|^2 \right)$$

$$\leq 2\frac{\nu^2}{\mu_g^2} \left( 2\mathbb{E}\left\|\mathbf{y}_{t+1}^*(\mathbf{x}_{t+1}) - \mathbf{y}_{t+1}^*(\mathbf{x}_t)\right\|^2 \right.$$

$$+ 2\mathbb{E}\left\|\mathbf{y}_{t+1}^*(\mathbf{x}_t) - \mathbf{y}_t^*(\mathbf{x}_t)\right\|^2 + \mathbb{E}\left\|\mathbf{x}_{t+1} - \mathbf{x}_t\right\|^2 \right)$$

$$\leq 2\frac{\nu^2}{\mu_g^2} \left( (1 + 2L_{\mathbf{y}}^2)\mathbb{E}\left\|\mathbf{x}_{t+1} - \mathbf{x}_t\right\|^2 + 2\mathbb{E}\left\|\mathbf{y}_{t+1}^*(\mathbf{x}_t) - \mathbf{y}_t^*(\mathbf{x}_t)\right\|^2 \right),$$

where the last inequality follows from Lemma C.1.

Combining this result with (73) gives

$$\mathbb{E}\left\|\mathbf{v}_{t+1} - \mathbf{v}_{t+1}^*(\mathbf{x}_{t+1})\right\|^2 \leq (1 + a)\,(1 + \acute{c}) \left( 1 - 2\delta_t \frac{(\ell_{g,1} + \ell_{g,1}^3)\mu_g}{\mu_g + \ell_{g,1}} + \delta_t^2 \ell_{g,1}^2 \right) \mathbb{E}\|\mathbf{v}_t - \mathbf{v}_t^*(\mathbf{x}_t)\|^2$$

$$+ (1 + a)\,(1 + \frac{1}{\acute{c}})\delta_t^2 \mathbb{E}\|e_t^{\mathbf{v}}\|^2 + 4 \left( 1 + \frac{1}{a} \right) \frac{\nu^2}{\mu_g^2} \mathbb{E}\left\|\mathbf{y}_{t+1}^*(\mathbf{x}_t) - \mathbf{y}_t^*(\mathbf{x}_t)\right\|^2$$

$$+ 2 \left( 1 + \frac{1}{a} \right) \frac{\nu^2}{\mu_g^2} (1 + 2L_{\mathbf{y}}^2) \mathbb{E}\left\|\mathbf{x}_{t+1} - \mathbf{x}_t\right\|^2. \tag{74}$$

Let $\acute{L}_{\mu_g} := \frac{(\ell_{g,1} + \ell_{g,1}^3)\mu_g}{\mu_g + \ell_{g,1}}$, then we have

$$1 - 2\delta_t \frac{(\ell_{g,1} + \ell_{g,1}^3)\mu_g}{\mu_g + \ell_{g,1}} + \delta_t^2 \ell_{g,1}^2 = 1 - 2\delta_t \acute{L}_{\mu_g} + \delta_t^2 \ell_{g,1}^2$$
$$\leq 1 - \delta_t \acute{L}_{\mu_g}, \tag{75}$$

where the last inequality follows from $\delta_t \leq \frac{\acute{L}_{\mu_g}}{\ell_{g,1}^2}$.

Choose $a = \frac{\delta_t \acute{L}_{\mu_g}/4}{1 - \frac{\delta_t \acute{L}_{\mu_g}}{2}}$ and $\acute{c} = \frac{\delta_t \acute{L}_{\mu_g}/2}{1 - \delta_t \acute{L}_{\mu_g}}$. Then, from (75), we have

$$(1 + a)(1 + \acute{c}) \left( 1 - 2\delta_t \frac{(\ell_{g,1} + \ell_{g,1}^3)\mu_g}{\mu_g + \ell_{g,1}} + \delta_t^2 \ell_{g,1}^2 \right)$$
$$\leq (1 + a)(1 + \acute{c}) \left( 1 - \delta_t \acute{L}_{\mu_g} \right) = 1 - \frac{\delta_t \acute{L}_{\mu_g}}{4},$$
$$(1 + a) \left( 1 + \frac{1}{\acute{c}} \right) \leq \frac{4}{\delta_t \acute{L}_{\mu_g}}, \tag{76}$$
$$1 + \frac{1}{\acute{c}} \leq \frac{2}{\delta_t \acute{L}_{\mu_g}}, \quad 1 + \frac{1}{a} \leq \frac{4}{\delta_t \acute{L}_{\mu_g}}.$$

Thus, from (74) and (76) we have

$$\mathbb{E}\left\| \mathbf{v}_{t+1} - \mathbf{v}_{t+1}^*(\mathbf{x}_{t+1}) \right\|^2 \leq \left( 1 - \frac{\delta_t \acute{L}_{\mu_g}}{4} \right) \mathbb{E}\|\mathbf{v}_t - \mathbf{v}_t^*(\mathbf{x}_t)\|^2$$
$$+ \frac{4}{\acute{L}_{\mu_g}} \delta_t \mathbb{E}\|e_t^{\mathbf{v}}\|^2 + \frac{16\nu^2}{\acute{L}_{\mu_g}\mu_g^2 \delta_t} \mathbb{E}\left\| \mathbf{y}_{t+1}^*(\mathbf{x}_t) - \mathbf{y}_t^*(\mathbf{x}_t) \right\|^2$$
$$+ \frac{8\nu^2}{\acute{L}_{\mu_g}\mu_g^2 \delta_t}(1 + 2L_{\mathbf{y}}^2)\mathbb{E}\left\| \mathbf{x}_{t+1} - \mathbf{x}_t \right\|^2.$$

Rearranging the terms and summing from $t = 1$ to $T$, gives the desired result. $\square$

## C.4 Bounds on the Gradient Estimation Error of Outer Objective

The following lemma, inspired by [69], provides a characterization of the descent of the gradient estimation error for the outer-level function.

**Lemma C.9.** *Suppose Assumptions* B2., B3., B4., C3. *and* C5. *hold. Let* $\{(\mathbf{x}_t, \mathbf{y}_t, \mathbf{v}_t)\}_{t=1}^T$ *be generated according to Algorithm* 1. *For* $e_t^f$ *defined as*

$$e_t^f := \mathbf{d}_t^{\mathbf{x}} - \tilde{\mathbf{d}}_t\left(\mathbf{z}_t, \mathbf{v}_t\right), \quad \text{where} \quad \tilde{\mathbf{d}}_t\left(\mathbf{z}_t, \mathbf{v}_t\right) = \nabla_{\mathbf{x}} f_t(\mathbf{z}_t) + \nabla_{\mathbf{xy}}^2 g_t\left(\mathbf{z}_t\right)\mathbf{v}_t, \tag{77}$$

*we have:*

$$\mathbb{E}\|e_{t+1}^f\|^2 \leq (1 - \eta_{t+1})^2 \mathbb{E}\|e_t^f\|^2 + 4\eta_{t+1}^2 \left( \frac{\sigma_{g_{\mathbf{xy}}}^2}{\tilde{b}} p^2 + \frac{\sigma_{f_{\mathbf{x}}}^2}{b} \right)$$
$$+ 12p^2(1 - \eta_{t+1})^2 \mathbb{E}\|\nabla_{\mathbf{xy}}^2 g_t\left(\mathbf{x}_{t+1}, \mathbf{y}_{t+1}\right) - \nabla_{\mathbf{xy}}^2 g_{t+1}\left(\mathbf{x}_{t+1}, \mathbf{y}_{t+1}\right)\|^2$$
$$+ 12(1 - \eta_{t+1})^2 \mathbb{E}\|\nabla_{\mathbf{x}} f_t(\mathbf{x}_{t+1}, \mathbf{y}_{t+1}) - \nabla_{\mathbf{x}} f_{t+1}(\mathbf{x}_{t+1}, \mathbf{y}_{t+1})\|^2$$
$$+ 72(1 - \eta_{t+1})^2 (\ell_{g,2}^2 p^2 + \ell_{f,1}^2)\left( \mathbb{E}\|\mathbf{x}_{t+1} - \mathbf{x}_t\|^2 + 2\beta_t^2 \mathbb{E}\|e_t^g\|^2 + 2\beta_t^2 \mathbb{E}\|\nabla_{\mathbf{y}} g_t(\mathbf{x}_t, \mathbf{y}_t)\|^2 \right)$$
$$+ 72\ell_{g,1}^2(1 - \eta_{t+1})^2 \delta_t^2 \mathbb{E}\|e_t^{\mathbf{v}}\|^2 + 72(1 - \eta_{t+1})^2 \ell_{g,1}^4 \delta_t^2 \mathbb{E}[\theta_t^{\mathbf{v}}], \tag{78}$$

*for all* $t \in [T]$, $\theta_t^{\mathbf{v}}$, $e_t^{\mathbf{v}}$ *and* $e_t^g$ *are defined in* (49), (60) *and* (45), *respectively.*

*Proof.* Note that

$$e_{t+1}^f = \mathbf{d}_{t+1}^{\mathbf{x}} - \tilde{\mathbf{d}}_{t+1}\left(\mathbf{x}_{t+1}, \mathbf{y}_{t+1}, \mathbf{v}_{t+1}\right),$$

where

$$\tilde{\mathbf{d}}_{t+1}\left(\mathbf{x}_{t+1}, \mathbf{y}_{t+1}, \mathbf{v}_{t+1}\right) = \nabla_{\mathbf{x}} f_{t+1}(\mathbf{x}_{t+1}, \mathbf{y}_{t+1}) + \nabla_{\mathbf{xy}}^2 g_{t+1}\left(\mathbf{x}_{t+1}, \mathbf{y}_{t+1}\right) \mathbf{v}_{t+1}. \qquad (79)$$

From Algorithm 1, we have

$$\mathbf{d}_{t+1}^{\mathbf{x}} = \mathbf{d}_{t+1}^{\mathbf{xx}}\left(\mathbf{x}_{t+1}, \mathbf{y}_{t+1}; \mathcal{B}_{t+1}\right) + (1 - \eta_{t+1})(\mathbf{d}_{t+1}^{\mathbf{x}} - \mathbf{d}_{t+1}^{\mathbf{xx}}(\mathbf{x}_{t+1}, \mathbf{y}_{t+1}; \mathcal{B}_{t+1})),$$

where $\mathbf{d}_{t+1}^{\mathbf{xx}}\left(\mathbf{x}_{t+1}, \mathbf{y}_{t+1}; \mathcal{B}_{t+1}\right) = \nabla_{\mathbf{x}} f_{t+1}(\mathbf{x}_{t+1}, \mathbf{y}_{t+1}; \mathcal{B}_{t+1}) + \nabla_{\mathbf{xy}}^2 g_{t+1}\left(\mathbf{x}_{t+1}, \mathbf{y}_{t+1}; \mathcal{B}_{t+1}\right) \mathbf{v}_{t+1}$.
Let $\mathbf{u} = [\mathbf{x}; \mathbf{y}; \mathbf{v}]$. Then, we have

$$\begin{aligned}
\mathbb{E}\|e_{t+1}^f\|^2 &= \mathbb{E}\|\mathbf{d}_{t+1}^{\mathbf{x}} - \tilde{\mathbf{d}}_{t+1}\left(\mathbf{u}_{t+1}\right)\|^2 \\
&= \mathbb{E}\|\tilde{\mathbf{d}}_{t+1}(\mathbf{u}_{t+1}; \mathcal{B}_{t+1}) + (1 - \eta_{t+1})(\mathbf{d}_t^{\mathbf{x}} - \tilde{\mathbf{d}}_{t+1}(\mathbf{u}_t; \mathcal{B}_{t+1})) - \tilde{\mathbf{d}}_{t+1}(\mathbf{u}_{t+1})\|^2 \\
&= \mathbb{E}\|(1 - \eta_{t+1})e_t^f + \tilde{\mathbf{d}}_{t+1}(\mathbf{u}_{t+1}; \mathcal{B}_{t+1}) - \tilde{\mathbf{d}}_{t+1}(\mathbf{u}_{t+1}) \\
&\quad - (1 - \eta_{t+1})(\tilde{\mathbf{d}}_{t+1}(\mathbf{u}_t; \mathcal{B}_{t+1}) - \tilde{\mathbf{d}}_t(\mathbf{u}_t))\|^2,
\end{aligned}$$

which implies that

$$\begin{aligned}
&\mathbb{E}\|e_{t+1}^f\|^2 \\
&= (1 - \eta_{t+1})^2 \mathbb{E}\|e_t^f\|^2 + \mathbb{E}\|\eta_{t+1}(\tilde{\mathbf{d}}_{t+1}(\mathbf{u}_{t+1}; \mathcal{B}_{t+1}) - \tilde{\mathbf{d}}_{t+1}(\mathbf{u}_{t+1})) \\
&\quad - (1 - \eta_{t+1})(\tilde{\mathbf{d}}_{t+1}(\mathbf{u}_t; \mathcal{B}_{t+1}) - \tilde{\mathbf{d}}_{t+1}(\mathbf{u}_{t+1}; \mathcal{B}_{t+1}) + \tilde{\mathbf{d}}_{t+1}(\mathbf{u}_{t+1}) - \tilde{\mathbf{d}}_t(\mathbf{u}_t))\|^2 \\
&\leq (1 - \eta_{t+1})^2 \mathbb{E}\|e_t^f\|^2 + 2\eta_{t+1}^2 \mathbb{E}\|\tilde{\mathbf{d}}_{t+1}(\mathbf{u}_{t+1}; \mathcal{B}_{t+1}) - \tilde{\mathbf{d}}_{t+1}(\mathbf{u}_{t+1})\|^2 \\
&\quad + 2(1 - \eta_{t+1})^2 \mathbb{E}\|\tilde{\mathbf{d}}_{t+1}(\mathbf{u}_{t+1}; \mathcal{B}_{t+1}) - \tilde{\mathbf{d}}_{t+1}(\mathbf{u}_{t+1}) - \tilde{\mathbf{d}}_{t+1}(\mathbf{u}_t; \mathcal{B}_{t+1}) + \tilde{\mathbf{d}}_t(\mathbf{u}_t)\|^2, \qquad (80)
\end{aligned}$$

where the inequality follows from $\|a + b\|^2 \leq 2\|a\|^2 + 2\|b\|^2$.
Let us bound the second term in the right-hand side of (80). Based on (79), we have

$$\begin{aligned}
&\mathbb{E}\|\tilde{\mathbf{d}}_{t+1}(\mathbf{u}_{t+1}; \mathcal{B}_{t+1}) - \tilde{\mathbf{d}}_{t+1}(\mathbf{u}_{t+1})\|^2 \\
&= \mathbb{E}\|\left(\nabla_{\mathbf{xy}}^2 g_{t+1}\left(\mathbf{x}_{t+1}, \mathbf{y}_{t+1}; \bar{\mathcal{B}}_{t+1}\right) - \nabla_{\mathbf{xy}}^2 g_{t+1}\left(\mathbf{x}_{t+1}, \mathbf{y}_{t+1}\right)\right) \mathbf{v}_{t+1} \\
&\quad + \nabla_{\mathbf{x}} f_{t+1}(\mathbf{x}_{t+1}, \mathbf{y}_{t+1}; \mathcal{B}_{t+1}) - \nabla_{\mathbf{x}} f_{t+1}(\mathbf{x}_{t+1}, \mathbf{y}_{t+1})\|^2 \\
&\leq 2\mathbb{E}\|\left(\nabla_{\mathbf{xy}}^2 g_{t+1}\left(\mathbf{x}_{t+1}, \mathbf{y}_{t+1}; \bar{\mathcal{B}}_{t+1}\right) - \nabla_{\mathbf{xy}}^2 g_{t+1}\left(\mathbf{x}_{t+1}, \mathbf{y}_{t+1}\right)\right) \mathbf{v}_{t+1}\|^2 \\
&\quad + 2\mathbb{E}\|\nabla_{\mathbf{x}} f_{t+1}(\mathbf{x}_{t+1}, \mathbf{y}_{t+1}; \mathcal{B}_{t+1}) - \nabla_{\mathbf{x}} f_{t+1}(\mathbf{x}_{t+1}, \mathbf{y}_{t+1})\|^2 \\
&\leq 2\left(\frac{\sigma_{g_{\mathbf{xy}}}^2}{\bar{b}} p^2 + \frac{\sigma_{f_{\mathbf{x}}}^2}{b}\right),
\end{aligned}$$

where the first inequality is by and $\|a + b\|^2 \leq 2\|a\|^2 + 2\|b\|^2$; the second inequality follows from Assumptions C3., C5. and (8).
Substituting the above inequality into (80) and using $\|a + b + c\|^2 \leq 3(\|a\|^2 + \|b\|^2 + \|c\|^2)$, we obtain

$$\begin{aligned}
\mathbb{E}\|e_{t+1}^f\|^2 &\leq (1 - \eta_{t+1})^2 \mathbb{E}\|e_t^f\|^2 + 4\lambda_{t+1}^2 \left(\frac{\sigma_{g_{\mathbf{xy}}}^2}{\bar{b}} p^2 + \frac{\sigma_{f_{\mathbf{x}}}^2}{b}\right) \\
&\quad + 6(1 - \eta_{t+1})^2 \mathbb{E}\|\tilde{\mathbf{d}}_t(\mathbf{u}_t) - \tilde{\mathbf{d}}_t(\mathbf{u}_{t+1})\|^2 \\
&\quad + 6(1 - \eta_{t+1})^2 \mathbb{E}\|\tilde{\mathbf{d}}_t(\mathbf{u}_{t+1}) - \tilde{\mathbf{d}}_{t+1}(\mathbf{u}_{t+1})\|^2 \\
&\quad + 6(1 - \eta_{t+1})^2 \mathbb{E}\|\tilde{\mathbf{d}}_{t+1}(\mathbf{u}_{t+1}; \mathcal{B}_{t+1}) - \tilde{\mathbf{d}}_{t+1}(\mathbf{u}_t; \mathcal{B}_{t+1})\|^2. \qquad (81)
\end{aligned}$$

Moreover, from $\|a + b + c\|^2 \leq 3(\|a\|^2 + \|b\|^2 + \|c\|^2)$, we have

$$\mathbb{E}\|\tilde{\mathbf{d}}_t(\mathbf{u}_{t+1}) - \tilde{\mathbf{d}}_t(\mathbf{u}_t)\|^2$$

$$\leq 3\mathbb{E}\|\tilde{\mathbf{d}}_t(\mathbf{x}_{t+1}, \mathbf{y}_{t+1}, \mathbf{v}_{t+1}) - \tilde{\mathbf{d}}_t(\mathbf{x}_t, \mathbf{y}_{t+1}, \mathbf{v}_{t+1})\|^2$$

$$+ 3\mathbb{E}\|\tilde{\mathbf{d}}_t(\mathbf{x}_t, \mathbf{y}_{t+1}, \mathbf{v}_{t+1}) - \tilde{\mathbf{d}}_t(\mathbf{x}_t, \mathbf{y}_t, \mathbf{v}_{t+1})\|^2$$

$$+ 3\mathbb{E}\|\tilde{\mathbf{d}}_t(\mathbf{x}_t, \mathbf{y}_t, \mathbf{v}_{t+1}) - \tilde{\mathbf{d}}_t(\mathbf{x}_t, \mathbf{y}_t, \mathbf{v}_t)\|^2$$

$$\overset{(i)}{\leq} 3\mathbb{E}\|(\nabla^2_{\mathbf{xy}} g_t(\mathbf{x}_{t+1}, \mathbf{y}_{t+1}) - \nabla^2_{\mathbf{xy}} g_t(\mathbf{x}_t, \mathbf{y}_{t+1}))\mathbf{v}_{t+1} + \nabla_{\mathbf{x}} f_t(\mathbf{x}_{t+1}, \mathbf{y}_{t+1}) - \nabla_{\mathbf{x}} f_t(\mathbf{x}_t, \mathbf{y}_{t+1})\|^2$$

$$+ 3\mathbb{E}\|(\nabla^2_{\mathbf{xy}} g_t(\mathbf{x}_t, \mathbf{y}_{t+1}) - \nabla^2_{\mathbf{xy}} g_t(\mathbf{x}_t, \mathbf{y}_t))\mathbf{v}_{t+1} + \nabla_{\mathbf{x}} f_t(\mathbf{x}_t, \mathbf{y}_{t+1}) - \nabla_{\mathbf{x}} f_t(\mathbf{x}_t, \mathbf{y}_t)\|^2$$

$$+ 3\mathbb{E}\|\tilde{\mathbf{d}}_t(\mathbf{x}_t, \mathbf{y}_t, \mathbf{v}_{t+1}) - \tilde{\mathbf{d}}_t(\mathbf{x}_t, \mathbf{y}_t, \mathbf{v}_t)\|^2$$

$$\overset{(ii)}{\leq} 6(\ell^2_{g,2}\mathbb{E}\|\mathbf{v}_{t+1}\|^2 + \ell^2_{f,1})\left(\mathbb{E}\|\mathbf{x}_{t+1} - \mathbf{x}_t\|^2 + \mathbb{E}\|\mathbf{y}_{t+1} - \mathbf{y}_t\|^2\right) + 3\ell^2_{g,1}\mathbb{E}\|\mathbf{v}_{t+1} - \mathbf{v}_t\|^2$$

$$\overset{(iii)}{\leq} 6(\ell^2_{g,2}p^2 + \ell^2_{f,1})\left(\mathbb{E}\|\mathbf{x}_{t+1} - \mathbf{x}_t\|^2 + \beta_t^2\mathbb{E}\|\mathbf{d}^{\mathbf{y}}_t\|^2\right) + 3\ell^2_{g,1}\delta_t^2\mathbb{E}\|\mathbf{d}^{\mathbf{v}}_t\|^2$$

$$\overset{(iv)}{\leq} 6(\ell^2_{g,2}p^2 + \ell^2_{f,1})\left(\mathbb{E}\|\mathbf{x}_{t+1} - \mathbf{x}_t\|^2 + 2\beta_t^2\mathbb{E}\|e^g_t\|^2 + 2\beta_t^2\mathbb{E}\|\nabla_{\mathbf{y}} g_t(\mathbf{x}_t, \mathbf{y}_t)\|^2\right)$$

$$+ 6\ell^2_{g,1}\delta_t^2(\mathbb{E}\|e^{\mathbf{v}}_t\|^2 + \mathbb{E}\|\nabla P_t(\mathbf{x}_t, \mathbf{y}_t, \mathbf{v}_t)\|^2)$$

$$\overset{(vi)}{\leq} 6(\ell^2_{g,2}p^2 + \ell^2_{f,1})\left(\mathbb{E}\|\mathbf{x}_{t+1} - \mathbf{x}_t\|^2 + 2\beta_t^2\mathbb{E}\|e^g_t\|^2 + 2\beta_t^2\mathbb{E}\|\nabla_{\mathbf{y}} g_t(\mathbf{x}_t, \mathbf{y}_t)\|^2\right)$$

$$+ 6\ell^2_{g,1}\delta_t^2\left(\mathbb{E}\|e^{\mathbf{v}}_t\|^2 + \ell^2_{g,1}\mathbb{E}\|\mathbf{v}_t - \mathbf{v}^*_t(\mathbf{x}_t)\|^2\right), \tag{82}$$

where the (i) follows from (79); (ii) follows from Assumptions B2., B3. and B4.; (iii) follows from (8); (iv) follows from (45) and (60); (vi) follows from (59).
Similarly, we have

$$\mathbb{E}\|\tilde{\mathbf{d}}_{t+1}(\mathbf{u}_{t+1}; \mathcal{B}_{t+1}) - \tilde{\mathbf{d}}_{t+1}(\mathbf{u}_t; \mathcal{B}_{t+1})\|^2$$

$$\leq 6(\ell^2_{g,2}p^2 + \ell^2_{f,1})\left(\mathbb{E}\|\mathbf{x}_{t+1} - \mathbf{x}_t\|^2 + 2\beta_t^2\mathbb{E}\|e^g_t\|^2 + 2\beta_t^2\mathbb{E}\|\nabla_{\mathbf{y}} g_t(\mathbf{x}_t, \mathbf{y}_t)\|^2\right)$$

$$+ 6\ell^2_{g,1}\delta_t^2\left(\mathbb{E}\|e^{\mathbf{v}}_t\|^2 + \ell^2_{g,1}\mathbb{E}\|\mathbf{v}_t - \mathbf{v}^*_t(\mathbf{x}_t)\|^2\right). \tag{83}$$

Substituting (83) and (82) into (81), we have

$$\mathbb{E}\|e^f_{t+1}\|^2 \leq (1 - \eta_{t+1})^2\mathbb{E}\|e^f_t\|^2 + 4\eta_{t+1}^2\left(\frac{\sigma^2_{g_{\mathbf{yy}}}}{\bar{b}}p^2 + \frac{\sigma^2_{f_{\mathbf{y}}}}{b}\right)$$

$$+ 6(1 - \eta_{t+1})^2\mathbb{E}\|\tilde{\mathbf{d}}_t(\mathbf{u}_{t+1}) - \tilde{\mathbf{d}}_{t+1}(\mathbf{u}_{t+1})\|^2$$

$$+ 72(1 - \eta_{t+1})^2(\ell^2_{g,2}p^2 + \ell^2_{f,1})\left(\mathbb{E}\|\mathbf{x}_{t+1} - \mathbf{x}_t\|^2 + 2\beta_t^2\mathbb{E}\|e^g_t\|^2 + 2\beta_t^2\mathbb{E}\|\nabla_{\mathbf{y}} g_t(\mathbf{x}_t, \mathbf{y}_t)\|^2\right)$$

$$+ 72\ell^2_{g,1}(1 - \eta_{t+1})^2\delta_t^2\mathbb{E}\|e^{\mathbf{v}}_t\|^2 + 72(1 - \eta_{t+1})^2\ell^4_{g,1}\delta_t^2\mathbb{E}\|\mathbf{v}_t - \mathbf{v}^*_t(\mathbf{x}_t)\|^2.$$

From $\|a + b\|^2 \leq 2\|a\|^2 + 2\|b\|^2$ and (8), we have

$$\mathbb{E}\|\tilde{\mathbf{d}}_t(\mathbf{u}_{t+1}) - \tilde{\mathbf{d}}_{t+1}(\mathbf{u}_{t+1})\|^2 = \mathbb{E}\|\nabla^2_{\mathbf{xy}} g_t(\mathbf{x}_{t+1}, \mathbf{y}_{t+1})\mathbf{v}_{t+1} - \nabla^2_{\mathbf{xy}} g_{t+1}(\mathbf{x}_{t+1}, \mathbf{y}_{t+1})\mathbf{v}_{t+1}$$

$$+ \nabla_{\mathbf{x}} f_t(\mathbf{x}_{t+1}, \mathbf{y}_{t+1}) - \nabla_{\mathbf{x}} f_{t+1}(\mathbf{x}_{t+1}, \mathbf{y}_{t+1})\|^2$$

$$\leq 2\mathbb{E}\|\left(\nabla^2_{\mathbf{xy}} g_t(\mathbf{x}_{t+1}, \mathbf{y}_{t+1}) - \nabla^2_{\mathbf{xy}} g_{t+1}(\mathbf{x}_{t+1}, \mathbf{y}_{t+1})\right)\mathbf{v}_{t+1}\|^2$$

$$+ 2\mathbb{E}\|\nabla_{\mathbf{x}} f_t(\mathbf{x}_{t+1}, \mathbf{y}_{t+1}) - \nabla_{\mathbf{x}} f_{t+1}(\mathbf{x}_{t+1}, \mathbf{y}_{t+1})\|^2$$

$$\leq 2\mathbb{E}\|\nabla^2_{\mathbf{xy}} g_t(\mathbf{x}_{t+1}, \mathbf{y}_{t+1}) - \nabla^2_{\mathbf{xy}} g_{t+1}(\mathbf{x}_{t+1}, \mathbf{y}_{t+1})\|^2 p^2$$

$$+ 2\mathbb{E}\|\nabla_{\mathbf{x}} f_t(\mathbf{x}_{t+1}, \mathbf{y}_{t+1}) - \nabla_{\mathbf{x}} f_{t+1}(\mathbf{x}_{t+1}, \mathbf{y}_{t+1})\|^2.$$

This completes the proof. $\qquad\square$

As demonstrated in Lemma C.9, the hypergradient estimator error $e^f_{t+1}$ comprises five key components: (1) the term $(1 - \eta_{t+1})^2\mathbb{E}\|e^f_t\|^2$, representing the per-iteration improvement achieved by the momentum-based update; (2) the error arising from the variation in the Jacobian of the lower-level objectiv; (3) the error caused by the variation in the gradient of the upper-level objective ; (4) the error term $\mathcal{O}(2\beta_t^2\mathbb{E}\|e^g_t\|^2 + 2\beta_t^2\mathbb{E}\|\nabla_{\mathbf{y}} g_t(\mathbf{x}_t, \mathbf{y}_t)\|^2)$, which is due to solving the lower-level problem; and (5) the error term $\mathcal{O}(\delta_t^2\mathbb{E}\|e^{\mathbf{v}}_t\|^2 + 72(1 - \eta_{t+1})^2\ell^4_{g,1}\delta_t^2\mathbb{E}[\theta^{\mathbf{v}}_t])$, which is introduced by the one-step momentum update in solving the linear system problem.

## C.5 Bounds on the Outer Objective and its Projected Gradient

**Lemma C.10.** *Let Assumption 2.4 holds. Then, for the sequence of functions $\{f_t\}_{t=1}^T$, we have*

$$\sum_{t=1}^T \left(f_t(\mathbf{x}_t, \mathbf{y}_t^*(\mathbf{x}_t)) - f_t(\mathbf{x}_{t+1}, \mathbf{y}_t^*(\mathbf{x}_{t+1}))\right) \leq 2M + V_T,$$

*where $M$ is defined in Assumption 2.4; $V_T$ is defined in (11).*

*Proof.* Note that, we have

$$\begin{aligned}
&\sum_{t=1}^T \left(f_t(\mathbf{x}_t, \mathbf{y}_t^*(\mathbf{x}_t)) - f_t(\mathbf{x}_{t+1}, \mathbf{y}_t^*(\mathbf{x}_{t+1}))\right) \\
&= f_1(\mathbf{x}_1, \mathbf{y}_1^*(\mathbf{x}_1)) - f_T(\mathbf{x}_{T+1}, \mathbf{y}_T^*(\mathbf{x}_{T+1})) \\
&\quad + \sum_{t=2}^T \left(f_t(\mathbf{x}_t, \mathbf{y}_t^*(\mathbf{x}_t)) - f_{t-1}(\mathbf{x}_t, \mathbf{y}_{t-1}^*(\mathbf{x}_t))\right) \\
&\leq 2M + V_T,
\end{aligned}$$

where the inequality follows from Assumption 2.4. $\qquad\square$

**Lemma C.11.** *Let $\{f_t\}_{t=1}^T$ denote the sequence of functions presented to Algorithm 1, satisfying Assumptions 2.2, 2.3 and 2.4. Let $\mathcal{P}_{\mathcal{X},\alpha_t}$ be defined as in Definition B.1. For any positive step size $\alpha_t$ such that $\alpha_t \leq 1/4L_f$ for all $t \in [T]$, Algorithm 1 ensures the following bound:*

$$\begin{aligned}
&\sum_{t=1}^T \left(\alpha_t - L_f \alpha_t^2\right) \mathbb{E} \left\|\mathcal{P}_{\mathcal{X},\alpha_t}\left(\mathbf{x}_t; \nabla f_t(\mathbf{x}_t, \mathbf{y}_t^*(\mathbf{x}_t))\right)\right\|^2 \\
&\leq 8M + 4V_T + 2M_f^2 \sum_{t=1}^T \left(2\alpha_t - L_f \alpha_t^2\right) \left(\mathbb{E}[\theta_t^{\mathbf{y}}] + \mathbb{E}[\theta_t^{\mathbf{v}}]\right) \\
&\quad + 2 \sum_{t=1}^T \left(2\alpha_t - L_f \alpha_t^2\right) \mathbb{E} \left\|e_t^f\right\|^2.
\end{aligned} \tag{84}$$

*Here, $\theta_t^{\mathbf{y}}$ and $\theta_t^{\mathbf{v}}$ are defined in (49); $V_T$, $M$, $M_f$ and $e_t^f$ are defined in (11), Assumption 2.4, Eq. (42), and (77).*

*Proof.* It follows from Lemma C.1 that

$$\begin{aligned}
&f_t(\mathbf{x}_{t+1}, \mathbf{y}_t^*(\mathbf{x}_{t+1})) - f_t(\mathbf{x}_t, \mathbf{y}_t^*(\mathbf{x}_t)) \\
&\leq \langle \nabla f_t(\mathbf{x}_t, \mathbf{y}_t^*(\mathbf{x}_t)), \mathbf{x}_{t+1} - \mathbf{x}_t \rangle + \frac{L_f}{2} \|\mathbf{x}_{t+1} - \mathbf{x}_t\|^2 \\
&= -\alpha_t \langle \nabla f_t(\mathbf{x}_t, \mathbf{y}_t^*(\mathbf{x}_t)), \mathcal{P}_{\mathcal{X},\alpha_t}\left(\mathbf{x}_t; \mathbf{d}_t^{\mathbf{x}}\right)\rangle + \frac{L_f \alpha_t^2}{2} \left\|\mathcal{P}_{\mathcal{X},\alpha_t}\left(\mathbf{x}_t; \mathbf{d}_t^{\mathbf{x}}\right)\right\|^2.
\end{aligned} \tag{85}$$

For the first term on the right hand side of (85), we have that

$$\begin{aligned}
&-\langle \nabla f_t(\mathbf{x}_t, \mathbf{y}_t^*(\mathbf{x}_t)), \mathcal{P}_{\mathcal{X},\alpha_t}\left(\mathbf{x}_t; \mathbf{d}_t^{\mathbf{x}}\right)\rangle \\
&= -\langle \mathbf{d}_t^{\mathbf{x}}, \mathcal{P}_{\mathcal{X},\alpha_t}\left(\mathbf{x}_t; \mathbf{d}_t^{\mathbf{x}}\right)\rangle - \langle \nabla f_t(\mathbf{x}_t, \mathbf{y}_t^*(\mathbf{x}_t)) - \mathbf{d}_t^{\mathbf{x}}, \mathcal{P}_{\mathcal{X},\alpha_t}\left(\mathbf{x}_t; \mathbf{d}_t^{\mathbf{x}}\right)\rangle \\
&\leq -\frac{1}{2} \left\|\mathcal{P}_{\mathcal{X},\alpha_t}\left(\mathbf{x}_t; \mathbf{d}_t^{\mathbf{x}}\right)\right\|^2 + \frac{1}{2} \left\|\mathbf{d}_t^{\mathbf{x}} - \nabla f_t(\mathbf{x}_t, \mathbf{y}_t^*(\mathbf{x}_t))\right\|^2,
\end{aligned}$$

where the inequality follows from Lemma B.7.

Let $\tilde{\mathbf{d}}_t\left(\mathbf{z}_t, \mathbf{v}_t\right) = \nabla_{\mathbf{x}} f_t(\mathbf{z}_t) + \nabla^2_{\mathbf{xy}} g_t\left(\mathbf{z}_t\right) \mathbf{v}_t$. Then, from Lemma C.1, we have

$$
\begin{aligned}
\left\|\mathbf{d}_t^{\mathbf{x}} - \nabla f_t(\mathbf{x}_t, \mathbf{y}_t^*(\mathbf{x}_t))\right\|^2 &= \left\|\mathbf{d}_t^{\mathbf{x}} - \tilde{\mathbf{d}}_t\left(\mathbf{z}_t, \mathbf{v}_t\right) + \tilde{\mathbf{d}}_t\left(\mathbf{z}_t, \mathbf{v}_t\right) - \nabla f_t(\mathbf{x}_t, \mathbf{y}_t^*(\mathbf{x}_t))\right\|^2 \\
&\leq 2\left\|\mathbf{d}_t^{\mathbf{x}} - \tilde{\mathbf{d}}_t\left(\mathbf{z}_t, \mathbf{v}_t\right)\right\|^2 + 2\left\|\tilde{\mathbf{d}}_t\left(\mathbf{z}_t, \mathbf{v}_t\right) - \nabla f_t(\mathbf{x}_t, \mathbf{y}_t^*(\mathbf{x}_t))\right\|^2 \\
&\leq 2\left\|e_t^f\right\|^2 + 2\left\|\tilde{\mathbf{d}}_t\left(\mathbf{z}_t, \mathbf{v}_t\right) - \nabla f_t(\mathbf{x}_t, \mathbf{y}_t^*(\mathbf{x}_t))\right\|^2 \\
&\leq 2\left\|e_t^f\right\|^2 + M_f^2\left(\theta_t^{\mathbf{y}} + \theta_t^{\mathbf{v}}\right),
\end{aligned}
\tag{86}
$$

where $e_t^f = \mathbf{d}_t^{\mathbf{x}} - \tilde{\mathbf{d}}_t\left(\mathbf{z}_t, \mathbf{v}_t\right)$. This implies that

$$
\begin{aligned}
&-\left\langle \nabla f_t(\mathbf{x}_t, \mathbf{y}_t^*(\mathbf{x}_t)), \mathcal{P}_{\mathcal{X}, \alpha_t}\left(\mathbf{x}_t; \mathbf{d}_t^{\mathbf{x}}\right)\right\rangle \\
&\leq -\frac{1}{2}\left\|\mathcal{P}_{\mathcal{X}, \alpha_t}\left(\mathbf{x}_t; \mathbf{d}_t^{\mathbf{x}}\right)\right\|^2 + 2\left\|e_t^f\right\|^2 + M_f^2\left(\theta_t^{\mathbf{y}} + \theta_t^{\mathbf{v}}\right).
\end{aligned}
\tag{87}
$$

Plugging the bound (87) into (85), we have that

$$
\begin{aligned}
&f_t(\mathbf{x}_{t+1}, \mathbf{y}_t^*(\mathbf{x}_{t+1})) - f_t(\mathbf{x}_t, \mathbf{y}_t^*(\mathbf{x}_t)) \\
&\leq \frac{\left(L_f \alpha_t^2 - \alpha_t\right)}{2}\left\|\mathcal{P}_{\mathcal{X}, \alpha_t}\left(\mathbf{x}_t; \mathbf{d}_t^{\mathbf{x}}\right)\right\|^2 + 2\alpha_t\left\|e_t^f\right\|^2 + M_f^2\left(\theta_t^{\mathbf{y}} + \theta_t^{\mathbf{v}}\right)\alpha_t,
\end{aligned}
$$

which can be rearranged into

$$
\begin{aligned}
&\left(\alpha_t - L_f \alpha_t^2\right)\left\|\mathcal{P}_{\mathcal{X}, \alpha_t}\left(\mathbf{x}_t; \mathbf{d}_t^{\mathbf{x}}\right)\right\|^2 \\
&\leq 2 f_t(\mathbf{x}_t, \mathbf{y}_t^*(\mathbf{x}_t)) - f_t(\mathbf{x}_{t+1}, \mathbf{y}_t^*(\mathbf{x}_{t+1})) + 4\alpha_t\left\|e_t^f\right\|^2 + 2 M_f^2\left(\theta_t^{\mathbf{y}} + \theta_t^{\mathbf{v}}\right)\alpha_t.
\end{aligned}
\tag{88}
$$

In addition, we have

$$
\begin{aligned}
&\left\|\mathcal{P}_{\mathcal{X}, \alpha_t}\left(\mathbf{x}_t; \nabla f_t(\mathbf{x}_t, \mathbf{y}_t^*(\mathbf{x}_t))\right)\right\|^2 \\
&\leq 2\left\|\mathcal{P}_{\mathcal{X}, \alpha_t}\left(\mathbf{x}_t; \mathbf{d}_t^{\mathbf{x}}\right) - \mathcal{P}_{\mathcal{X}, \alpha_t}\left(\mathbf{x}_t; \nabla f_t(\mathbf{x}_t, \mathbf{y}_t^*(\mathbf{x}_t))\right)\right\|^2 + 2\left\|\mathcal{P}_{\mathcal{X}, \alpha_t}\left(\mathbf{x}_t; \mathbf{d}_t^{\mathbf{x}}\right)\right\|^2 \\
&\leq 2\left\|\mathbf{d}_t^{\mathbf{x}} - \nabla f_t(\mathbf{x}_t, \mathbf{y}_t^*(\mathbf{x}_t))\right\|^2 + 2\left\|\mathcal{P}_{\mathcal{X}, \alpha_t}\left(\mathbf{x}_t; \mathbf{d}_t^{\mathbf{x}}\right)\right\|^2 \\
&\leq 4\left\|e_t^f\right\|^2 + 4 M_f^2\left(\theta_t^{\mathbf{y}} + \theta_t^{\mathbf{v}}\right) + 4\left\|\mathcal{P}_{\mathcal{X}, \alpha_t}\left(\mathbf{x}_t; \mathbf{d}_t^{\mathbf{x}}\right)\right\|^2,
\end{aligned}
\tag{89}
$$

where the second inequaliy follows from non-expansiveness of the projection operator and the last inequality follows from (86).

Combining (88) and (89), we have

$$
\begin{aligned}
&\sum_{t=1}^T\left(\alpha_t - L_f \alpha_t^2\right)\left\|\mathcal{P}_{\mathcal{X}, \alpha_t}\left(\mathbf{x}_t; \nabla f_t(\mathbf{x}_t, \mathbf{y}_t^*(\mathbf{x}_t))\right)\right\|^2 \\
&\leq 4 \sum_{t=1}^T\left(f_t(\mathbf{x}_t, \mathbf{y}_t^*(\mathbf{x}_t)) - f_t(\mathbf{x}_{t+1}, \mathbf{y}_t^*(\mathbf{x}_{t+1}))\right) \\
&\quad + 2 M_f^2 \sum_{t=1}^T\left(2\alpha_t - L_f \alpha_t^2\right)\left(\theta_t^{\mathbf{y}} + \theta_t^{\mathbf{v}}\right) + 2 \sum_{t=1}^T\left(2\alpha_t - L_f \alpha_t^2\right)\left\|e_t^f\right\|^2 \\
&\leq 8M + 4V_T \\
&\quad + 2 M_f^2 \sum_{t=1}^T\left(2\alpha_t - L_f \alpha_t^2\right)\left(\theta_t^{\mathbf{y}} + \theta_t^{\mathbf{v}}\right) + 2 \sum_{t=1}^T\left(2\alpha_t - L_f \alpha_t^2\right)\left\|e_t^f\right\|^2,
\end{aligned}
$$

where the second inequality is due to Lemma C.10. $\qquad \square$

**Lemma C.12.** *Let Assumptions 2.2, and 2.3 hold. Let $\{\mathbf{x}_t\}_{t=1}^T$ be generated according to Algorithm 1. Then, we have*

$$
\left\|\mathbf{x}_t - \mathbf{x}_{t+1}\right\|^2 \leq 2\alpha_t^2\left(\left\|\mathcal{P}_{\mathcal{X}, \alpha_t}\left(\mathbf{x}_t; \nabla f_t(\mathbf{x}_t, \mathbf{y}_t^*(\mathbf{x}_t))\right)\right\|^2 + M_f^2\left(\theta_t^{\mathbf{y}} + \theta_t^{\mathbf{v}}\right)\right),
$$

*where $\theta_t^{\mathbf{y}}$ and $\theta_t^{\mathbf{v}}$ are defined in (49), $M_f$ is defined in (42).*

*Proof.* From the update rule of Algorithm 1, we have

$$
\begin{aligned}
\|\mathbf{x}_t - \mathbf{x}_{t+1}\|^2 &= \alpha_t^2 \|\mathcal{P}_{\mathcal{X},\alpha_t}(\mathbf{x}_t; \mathbf{d}_t^\mathbf{x})\|^2 \\
&\leq 2\alpha_t^2 \Big( \|\mathcal{P}_{\mathcal{X},\alpha_t}(\mathbf{x}_t; \nabla f_t(\mathbf{x}_t, \mathbf{y}_t^*(\mathbf{x}_t)))\|^2 \\
&\quad + \|\mathcal{P}_{\mathcal{X},\alpha_t}(\mathbf{x}_t; \mathbf{d}_t^\mathbf{x}) - \mathcal{P}_{\mathcal{X},\alpha_t}(\mathbf{x}_t; \nabla f_t(\mathbf{x}_t, \mathbf{y}_t^*(\mathbf{x}_t)))\|^2 \Big) \\
&\leq 2\alpha_t^2 \Big( \|\mathcal{P}_{\mathcal{X},\alpha_t}(\mathbf{x}_t; \nabla f_t(\mathbf{x}_t, \mathbf{y}_t^*(\mathbf{x}_t)))\|^2 \\
&\quad + \|\mathbf{d}_t^\mathbf{x} - \nabla f_t(\mathbf{x}_t, \mathbf{y}_t^*(\mathbf{x}_t))\|^2 \Big) \\
&\leq 2\alpha_t^2 \Big( \|\mathcal{P}_{\mathcal{X},\alpha_t}(\mathbf{x}_t; \nabla f_t(\mathbf{x}_t, \mathbf{y}_t^*(\mathbf{x}_t)))\|^2 + M_f^2 (\theta_t^\mathbf{y} + \theta_t^\mathbf{v}) \Big),
\end{aligned} \tag{90}
$$

where the first inequality is by $(a+b)^2 \leq 2a^2 + 2b^2$; the second inequality follows from non-expansiveness of the projection operator; and the last inequality follows from Eq. (39a) in Lemma C.1. $\qquad \square$

## C.6 Proof of Theorem 2.6

*Proof.* **Bounding** $\mathbb{E}\|e_t^f\|^2$ **in** (78) . From (78), we have

$$
\begin{aligned}
\frac{\mathbb{E}\|e_{t+1}^f\|^2}{\alpha_t} - \frac{\mathbb{E}\|e_t^f\|^2}{\alpha_{t-1}} &\leq \left( \frac{(1-\eta_{t+1})^2}{\alpha_t} - \frac{1}{\alpha_{t-1}} \right) \mathbb{E}\|e_t^f\|^2 + \frac{4\eta_{t+1}^2}{\alpha_t} \left( \frac{\sigma_{g_{\mathbf{xy}}}^2}{\overline{b}} p^2 + \frac{\sigma_{f_\mathbf{x}}^2}{b} \right) \\
&\quad + \frac{12p^2}{\alpha_t}(1-\eta_{t+1})^2 \mathbb{E}\|\nabla_{\mathbf{xy}}^2 g_t(\mathbf{x}_{t+1}, \mathbf{y}_{t+1}) - \nabla_{\mathbf{xy}}^2 g_{t+1}(\mathbf{x}_{t+1}, \mathbf{y}_{t+1})\|^2 \\
&\quad + \frac{12}{\alpha_t}(1-\eta_{t+1})^2 \mathbb{E}\|\nabla_\mathbf{x} f_t(\mathbf{x}_{t+1}, \mathbf{y}_{t+1}) - \nabla_\mathbf{x} f_{t+1}(\mathbf{x}_{t+1}, \mathbf{y}_{t+1})\|^2 \\
&\quad + \frac{72}{\alpha_t}(1-\eta_{t+1})^2 (\ell_{g,2}^2 p^2 + \ell_{f,1}^2) \left( \mathbb{E}\|\mathbf{x}_{t+1} - \mathbf{x}_t\|^2 + 2\beta_t^2 \mathbb{E}\|e_t^g\|^2 + 2\beta_t^2 \mathbb{E}\|\nabla_\mathbf{y} g_t(\mathbf{x}_t, \mathbf{y}_t)\|^2 \right) \\
&\quad + \frac{72}{\alpha_t} \ell_{g,1}^2 (1-\eta_{t+1})^2 \delta_t^2 \mathbb{E}\|e_t^\mathbf{v}\|^2 + \frac{72}{\alpha_t}(1-\eta_{t+1})^2 \ell_{g,1}^4 \delta_t^2 \mathbb{E}[\theta_t^\mathbf{v}].
\end{aligned} \tag{91}
$$

With respect to the coefficient of the first term on the right-hand side of Eq. (91), it is important to note that we have:

$$
\frac{(1-\eta_{t+1})^2}{\alpha_t} - \frac{1}{\alpha_{t-1}} \leq \frac{1}{\alpha_t} - \frac{\eta_{t+1}}{\alpha_t} - \frac{1}{\alpha_{t-1}}. \tag{92}
$$

Using the definition of $\alpha_t$ in (16), we have

$$
\begin{aligned}
\frac{1}{\alpha_t} - \frac{1}{\alpha_{t-1}} &= (c+t)^{1/3} - (c+t-1)^{1/3} \overset{(i)}{\leq} \frac{1}{3(c+t-1)^{2/3}} \overset{(ii)}{\leq} \frac{1}{3(\frac{c}{2}+t)^{2/3}} \\
&= \frac{2^{2/3}}{3(c+2t)^{2/3}} \overset{(iii)}{\leq} \frac{2^{2/3}}{3(c+t)^{2/3}} \overset{(iv)}{\leq} \frac{2^{2/3}}{3} \alpha_t^2 \overset{(vi)}{\leq} \frac{\alpha_t}{6L_f},
\end{aligned} \tag{93}
$$

where the (i) follows from $(a+b)^{1/3} - a^{1/3} \leq b/(3a^{2/3})$; (ii) follows from $c \geq 2$ in (107); (iii) follows from (16); (iv) follows from $\alpha_t \leq 1/4L_f$ in (107).

Substituting (93) into (92) and using $\delta_t = c_\delta \alpha_t$ and $\eta_{t+1} = c_\eta \alpha_t^2$, we have

$$
\frac{(1-\eta_{t+1})^2}{\alpha_t} - \frac{1}{\alpha_{t-1}} \leq \frac{\alpha_t}{6L_f} - \frac{\eta_{t+1}}{\alpha_t} = \frac{\alpha_t}{6L_f} - c_\eta \alpha_t \leq -5\Omega\alpha_t, \tag{94}
$$

where the inequalities follow from $c_\eta = \frac{1}{6L_f} + 5\Omega$ in (106).

Then, substituting (94) into (91) yields

$$\frac{1}{\Omega}\mathbb{E}\left(\frac{\|e_{t+1}^f\|^2}{\alpha_t} - \frac{\|e_t^f\|^2}{\alpha_{t-1}}\right) \leq -5\alpha_t\mathbb{E}\|e_t^f\|^2 + \frac{4\eta_{t+1}^2}{\Omega\alpha_t}(\frac{\sigma_{g_{\mathbf{xy}}}^2}{b}p^2 + \frac{\sigma_{f_{\mathbf{x}}}^2}{b})$$

$$+ \frac{12p^2}{\Omega\alpha_t}(1-\eta_{t+1})^2\mathbb{E}\|\nabla_{\mathbf{xy}}^2 g_t(\mathbf{x}_{t+1}, \mathbf{y}_{t+1}) - \nabla_{\mathbf{xy}}^2 g_{t+1}(\mathbf{x}_{t+1}, \mathbf{y}_{t+1})\|^2$$

$$+ \frac{12}{\Omega\alpha_t}(1-\eta_{t+1})^2\mathbb{E}\|\nabla_{\mathbf{x}} f_t(\mathbf{x}_{t+1}, \mathbf{y}_{t+1}) - \nabla_{\mathbf{x}} f_{t+1}(\mathbf{x}_{t+1}, \mathbf{y}_{t+1})\|^2$$

$$+ \frac{72}{\Omega\alpha_t}(1-\eta_{t+1})^2(\ell_{g,2}^2 p^2 + \ell_{f,1}^2)\left(\mathbb{E}\|\mathbf{x}_{t+1}-\mathbf{x}_t\|^2 + 2\beta_t^2\mathbb{E}\|e_t^g\|^2 + 2\beta_t^2\mathbb{E}\|\nabla_{\mathbf{y}} g_t(\mathbf{x}_t, \mathbf{y}_t)\|^2\right)$$

$$+ \frac{72}{\Omega\alpha_t}\ell_{g,1}^2(1-\eta_{t+1})^2\delta_t^2\mathbb{E}\|e_t^{\mathbf{y}}\|^2 + \frac{72}{\Omega\alpha_t}(1-\eta_{t+1})^2\ell_{g,1}^4\delta_t^2\mathbb{E}[\theta_t^{\mathbf{y}}]. \tag{95}$$

**Bounding $\mathbb{E}\|e_t^g\|^2$ in** (46).
From (46), we have

$$\frac{\mathbb{E}\|e_{t+1}^g\|^2}{\alpha_t} - \frac{\mathbb{E}\|e_t^g\|^2}{\alpha_{t-1}} \leq \left(\frac{1}{\alpha_t}(1-\gamma_{t+1})^2(1+48\ell_{g,1}^2\beta_t^2) - \frac{1}{\alpha_{t-1}}\right)\mathbb{E}\|e_t^g\|^2$$

$$+ 2\frac{\gamma_{t+1}^2}{\alpha_t}\frac{\sigma_{g_{\mathbf{y}}}^2}{b} + \frac{24}{\alpha_t}(1-\gamma_{t+1})^2\ell_{g,1}^2\mathbb{E}\|\mathbf{x}_{t+1}-\mathbf{x}_t\|^2$$

$$+ \frac{6}{\alpha_t}(1-\gamma_{t+1})^2\mathbb{E}\|\nabla_{\mathbf{y}} g_t(\mathbf{x}_{t+1}, \mathbf{y}_{t+1}) - \nabla_{\mathbf{y}} g_{t+1}(\mathbf{x}_{t+1}, \mathbf{y}_{t+1})\|^2$$

$$+ 48(1-\gamma_{t+1})^2\ell_{g,1}^2\frac{\beta_t^2}{\alpha_t}\mathbb{E}\|\nabla_{\mathbf{y}} g_t(\mathbf{x}_t, \mathbf{y}_t)\|^2. \tag{96}$$

Let us examine the coefficient of the first term on the right-hand side of Eq. (96). Specifically, for $\gamma_{t+1} = c_\gamma\alpha_t^2$ and $\beta_t = c_\beta\alpha_t$, we have:

$$\frac{1}{\alpha_t}(1-\gamma_{t+1})^2(1+48\ell_{g,1}^2\beta_t^2) - \frac{1}{\alpha_{t-1}} \leq \frac{1}{\alpha_t}(1-\gamma_{t+1})(1+48\ell_{g,1}^2\beta_t^2) - \frac{1}{\alpha_{t-1}}$$

$$= \frac{1}{\alpha_t} - \frac{1}{\alpha_{t-1}} - \frac{\gamma_{t+1}}{\alpha_t} + \frac{1-\gamma_{t+1}}{\alpha_t}48\ell_{g,1}^2\beta_t^2$$

$$= \frac{1}{\alpha_t} - \frac{1}{\alpha_{t-1}} - c_\gamma\alpha_t + (\frac{1}{\alpha_t} - c_\gamma\alpha_t)48\ell_{g,1}^2c_\beta^2\alpha_t^2$$

$$\leq \frac{\alpha_t}{6L_f} + 48\ell_{g,1}^2c_\beta^2\alpha_t - c_\gamma\alpha_t, \tag{97}$$

where the last inequality follows from (93).

From the selected $c_\gamma$ in (107) and the definition of $\Phi$ in (106), we have

$$c_\gamma = \frac{1}{6L_f} + 48\ell_{g,1}^2c_\beta^2 + \hbar\Phi, \quad \text{where} \quad \hbar = 25\frac{M_f^2}{L_{\mu_g}^2}.$$

Combined this with Eq. (97) yields

$$\frac{1}{\alpha_t}(1-\gamma_{t+1})^2(1+48\ell_{g,1}^2\beta_t^2) - \frac{1}{\alpha_{t-1}} \leq -\hbar\Phi\alpha_t. \tag{98}$$

Substituting Eq. (98) into Eq. (96) yields

$$\frac{1}{\Phi}\left(\frac{\mathbb{E}\|e_{t+1}^g\|^2}{\alpha_t} - \frac{\mathbb{E}\|e_t^g\|^2}{\alpha_{t-1}}\right) \leq -\hbar\alpha_t\mathbb{E}\|e_t^g\|^2$$

$$+ 2\frac{\gamma_{t+1}^2}{\Phi\alpha_t}\frac{\sigma_{g_{\mathbf{y}}}^2}{b} + \frac{24}{\Phi\alpha_t}(1-\gamma_{t+1})^2\ell_{g,1}^2\mathbb{E}\|\mathbf{x}_{t+1}-\mathbf{x}_t\|^2$$

$$+ \frac{6}{\Phi\alpha_t}(1-\gamma_{t+1})^2\mathbb{E}\|\nabla_{\mathbf{y}} g_t(\mathbf{x}_{t+1}, \mathbf{y}_{t+1}) - \nabla_{\mathbf{y}} g_{t+1}(\mathbf{x}_{t+1}, \mathbf{y}_{t+1})\|^2$$

$$+ 48(1-\gamma_{t+1})^2\ell_{g,1}^2\frac{\beta_t^2}{\Phi\alpha_t}\mathbb{E}\|\nabla_{\mathbf{y}} g_t(\mathbf{x}_t, \mathbf{y}_t)\|^2. \tag{99}$$

**Bounding $\mathbb{E}\|e_t^{\mathbf{v}}\|^2$ in (61)** .

From (61), we get

$$
\frac{\mathbb{E}\|e_{t+1}^{\mathbf{v}}\|^2}{\alpha_t} - \frac{\mathbb{E}\|e_t^{\mathbf{v}}\|^2}{\alpha_{t-1}} \le \left( \frac{1}{\alpha_t}(1-\lambda_{t+1})^2(1+72\ell_{g,1}^2\delta_t^2) - \frac{1}{\alpha_{t-1}} \right) \mathbb{E}\|e_t^{\mathbf{v}}\|^2
$$

$$
+ 4\frac{\lambda_{t+1}^2}{\alpha_t}(\frac{\sigma_{g_{\mathbf{yy}}}^2}{\bar{b}}p^2 + \frac{\sigma_{f_{\mathbf{y}}}^2}{b}) + \frac{12p^2}{\alpha_t}(1-\lambda_{t+1})^2\mathbb{E}\|\nabla_{\mathbf{y}}^2 g_t\left(\mathbf{x}_{t+1},\mathbf{y}_{t+1}\right) - \nabla_{\mathbf{y}}^2 g_{t+1}\left(\mathbf{x}_{t+1},\mathbf{y}_{t+1}\right)\|^2
$$

$$
+ \frac{12}{\alpha_t}(1-\lambda_{t+1})^2\mathbb{E}\|\nabla_{\mathbf{y}} f_t(\mathbf{x}_{t+1},\mathbf{y}_{t+1}) - \nabla_{\mathbf{y}} f_{t+1}(\mathbf{x}_{t+1},\mathbf{y}_{t+1})\|^2
$$

$$
+ \frac{72}{\alpha_t}(1-\lambda_{t+1})^2(\ell_{g,2}^2 p^2 + \ell_{f,1}^2)\left( \mathbb{E}\|\mathbf{x}_{t+1}-\mathbf{x}_t\|^2 + 2\beta_t^2\mathbb{E}\|e_t^g\|^2 + 2\beta_t^2\mathbb{E}\|\nabla_{\mathbf{y}} g_t(\mathbf{x}_t,\mathbf{y}_t)\|^2\right)
$$

$$
+ \frac{144}{\alpha_t}(1-\lambda_{t+1})^2\ell_{g,1}^4\delta_t^2\mathbb{E}[\theta_t^{\mathbf{v}}] + \frac{288}{\alpha_t}\ell_{g,1}^2(p^2\ell_{g,2}^2 + \ell_{f,1}^2)\delta_t^2\mathbb{E}[\theta_t^{\mathbf{y}}]. \tag{100}
$$

Let us examine the coefficient of the first term on the right-hand side of equation (100). Specifically, for $\lambda_{t+1} = c_\lambda\alpha_t^2$ and $\delta_t = c_\delta\alpha_t$, we have:

$$
\frac{1}{\alpha_t}(1-\lambda_{t+1})^2(1+72\ell_{g,1}^2\delta_t^2) - \frac{1}{\alpha_{t-1}} \le \frac{1}{\alpha_t}(1-\lambda_{t+1})(1+72\ell_{g,1}^2\delta_t^2) - \frac{1}{\alpha_{t-1}}
$$

$$
= \frac{1}{\alpha_t} - \frac{1}{\alpha_{t-1}} - \frac{\lambda_{t+1}}{\alpha_t} + \frac{1-\lambda_{t+1}}{\alpha_t}72\ell_{g,1}^2\delta_t^2
$$

$$
= \frac{1}{\alpha_t} - \frac{1}{\alpha_{t-1}} - c_\lambda\alpha_t + (\frac{1}{\alpha_t} - c_\lambda\alpha_t)72\ell_{g,1}^2 c_\delta^2\alpha_t^2
$$

$$
\le \frac{\alpha_t}{6L_f} + 72\ell_{g,1}^2 c_\delta^2\alpha_t - c_\lambda\alpha_t, \tag{101}
$$

where the last inequality follows from (93).

From the selected $c_\gamma$ in (107) and the definition of $\Psi$ in (106), we have

$$
c_\lambda = \frac{1}{6L_f} + 72\ell_{g,1}^2 c_\delta^2 + \jmath\Psi, \quad \text{where} \quad \jmath = 90\frac{M_f^2}{L_{\mu_g}^2}.
$$

Combined this with Eq. (101) yields

$$
\frac{1}{\alpha_t}(1-\lambda_{t+1})^2(1+72\ell_{g,1}^2\delta_t^2) - \frac{1}{\alpha_{t-1}} \le -\jmath\Psi\alpha_t. \tag{102}
$$

Substituting Eq. (102) into Eq. (100) yields

$$
\frac{1}{\Psi}\left( \frac{\mathbb{E}\|e_{t+1}^{\mathbf{v}}\|^2}{\alpha_t} - \frac{\mathbb{E}\|e_t^{\mathbf{v}}\|^2}{\alpha_{t-1}} \right) \le -\jmath\alpha_t\mathbb{E}\|e_t^{\mathbf{v}}\|^2
$$

$$
+ 4\frac{\lambda_{t+1}^2}{\Psi\alpha_t}(\frac{\sigma_{g_{\mathbf{yy}}}^2}{\bar{b}}p^2 + \frac{\sigma_{f_{\mathbf{y}}}^2}{b}) + \frac{12p^2}{\Psi\alpha_t}(1-\lambda_{t+1})^2\mathbb{E}\|\nabla_{\mathbf{y}}^2 g_t\left(\mathbf{x}_{t+1},\mathbf{y}_{t+1}\right) - \nabla_{\mathbf{y}}^2 g_{t+1}\left(\mathbf{x}_{t+1},\mathbf{y}_{t+1}\right)\|^2
$$

$$
+ \frac{12}{\Psi\alpha_t}(1-\lambda_{t+1})^2\mathbb{E}\|\nabla_{\mathbf{y}} f_t(\mathbf{x}_{t+1},\mathbf{y}_{t+1}) - \nabla_{\mathbf{y}} f_{t+1}(\mathbf{x}_{t+1},\mathbf{y}_{t+1})\|^2
$$

$$
+ \frac{72}{\Psi\alpha_t}(1-\lambda_{t+1})^2(\ell_{g,2}^2 p^2 + \ell_{f,1}^2)\left( \mathbb{E}\|\mathbf{x}_{t+1}-\mathbf{x}_t\|^2 + 2\beta_t^2\mathbb{E}\|e_t^g\|^2 + 2\beta_t^2\mathbb{E}\|\nabla_{\mathbf{y}} g_t(\mathbf{x}_t,\mathbf{y}_t)\|^2\right)
$$

$$
+ \frac{72}{\Psi\alpha_t}(1-\lambda_{t+1})^2\ell_{g,1}^4\delta_t^2\mathbb{E}[\theta_t^{\mathbf{v}}] + \frac{288}{\Psi\alpha_t}\ell_{g,1}^2(p^2\ell_{g,2}^2 + \ell_{f,1}^2)\delta_t^2\mathbb{E}[\theta_t^{\mathbf{y}}]. \tag{103}
$$

**Combining the outcomes** . We recall from Lemma C.12 that we have

$$
\|\mathbf{x}_t - \mathbf{x}_{t+1}\|^2 \le 2\alpha_t^2\left( \|\mathcal{P}_{\mathcal{X},\alpha_t}\left(\mathbf{x}_t; \nabla f_t(\mathbf{x}_t, \mathbf{y}_t^*(\mathbf{x}_t)))\|^2 + M_f^2\left(\theta_t^{\mathbf{y}} + \theta_t^{\mathbf{v}}\right)\right). \tag{104}
$$

Let

$$
\Lambda := \Gamma\sum_{t=1}^T\left( \mathbb{E}[\theta_{t+1}^{\mathbf{y}}] - \mathbb{E}[\theta_t^{\mathbf{y}}]\right) + \Upsilon\sum_{t=1}^T\left( \mathbb{E}[\theta_{t+1}^{\mathbf{v}}] - \mathbb{E}[\theta_t^{\mathbf{v}}]\right) + \frac{1}{\Phi}\sum_{t=1}^T\left( \frac{\mathbb{E}\|e_{t+1}^g\|^2}{\alpha_t} - \frac{\mathbb{E}\|e_t^g\|^2}{\alpha_{t-1}}\right)
$$

$$
+ \frac{1}{\Psi}\sum_{t=1}^T\left( \frac{\mathbb{E}\|e_{t+1}^{\mathbf{v}}\|^2}{\alpha_t} - \frac{\mathbb{E}\|e_t^{\mathbf{v}}\|^2}{\alpha_{t-1}}\right) + \frac{1}{\Omega}\sum_{t=1}^T\left( \frac{\mathbb{E}\|e_{t+1}^f\|^2}{\alpha_t} - \frac{\mathbb{E}\|e_t^f\|^2}{\alpha_{t-1}}\right). \tag{105}
$$

Here

$$\Gamma = \frac{11M_f^2}{L_{\mu_g}c_\beta}, \quad \Upsilon = \frac{22M_f^2}{\acute{L}_{\mu_g}c_\delta}, \quad \Phi \geq \max\left\{480\ell_{g,1}^2, 192\ell_{g,1}^2\frac{(\mu_g+\ell_{g,1})}{\Gamma}c_\beta\right\},$$

$$\Psi = \max\left\{144(\ell_{g,2}^2p^2+\ell_{f,1}^2)\left(10+\frac{L_{\mu_g}^2c_\beta^2}{M_f^2}\right), \frac{288\ell_{g,1}^4}{M_f^2}c_\delta^2,\right.$$

$$\left.576(\ell_{g,2}^2p^2+\ell_{f,1}^2)\frac{(\mu_g+\ell_{g,1})}{\Gamma}c_\beta, \frac{576}{M_f^2}\ell_{g,1}^2(p^2\ell_{g,2}^2+\ell_{f,1}^2)c_\delta^2\right\},$$

$$\Omega = \max\left\{144(\ell_{g,2}^2p^2+\ell_{f,1}^2)\left(10+\frac{L_{\mu_g}^2c_\beta^2}{M_f^2}\right), \frac{288\ell_{g,1}^4}{M_f^2}c_\delta^2,\right.$$

$$\left.576(\ell_{g,2}^2p^2+\ell_{f,1}^2)\frac{(\mu_g+\ell_{g,1})}{\Gamma}c_\beta, \frac{72\ell_{g,1}^2\acute{L}_{\mu_g}^2}{M_f^2}c_\delta^2\right\}, \tag{106}$$

where $L_{\mu_g} = \mu_g\ell_{g,1}/(\mu_g+\ell_{g,1})$ and $\acute{L}_{\mu_g} = (\ell_{g,1}+\ell_{g,1}^3)\mu_g/(\mu_g+\ell_{g,1})$.
Here, we have

$$c \geq \max\{4L_f, c_\beta(\mu_g+\ell_{g,1}), 2\},$$

$$c_\beta = \sqrt{880}\frac{L_{\mathbf{y}}M_f}{L_{\mu_g}},$$

$$c_\delta = \sqrt{3520(1+2L_{\mathbf{y}}^2)}\frac{\nu M_f}{\acute{L}_{\mu_g}\mu_g}, \quad \text{where} \quad \nu = \ell_{f,1}+\frac{\ell_{g,2}\ell_{f,0}}{\mu_g}, \quad L_{\mathbf{y}} = \frac{\ell_{g,1}}{\mu_g},$$

$$c_\gamma = \frac{1}{6L_f}+48\ell_{g,1}^2c_\beta^2+\hbar\Phi, \quad \text{where} \quad \hbar := 25\frac{M_f^2}{L_{\mu_g}^2}, \tag{107}$$

$$c_\eta = \frac{1}{6L_f}+5\Omega,$$

$$c_\lambda = \frac{1}{6L_f}+72\ell_{g,1}^2c_\delta^2+\jmath\Psi, \quad \text{where} \quad \jmath = 90\frac{M_f^2}{\acute{L}_{\mu_g}^2}.$$

Using (103), (99), (95), (84), (70), and (50), along with (104) and the fact that $\alpha_t$ decreases with respect to $t$, we obtain:

$$\sum_{t=1}^T A(\alpha_t,\beta_t,\delta_t)\mathbb{E}\left\|\mathcal{P}_{\mathcal{X},\alpha_t}\left(\mathbf{x}_t;\nabla f_t(\mathbf{x}_t,\mathbf{y}_t^*(\mathbf{x}_t))\right)\right\|^2 + \Lambda$$

$$\leq 8M+4V_T+\sum_{t=1}^T B(\alpha_t,\beta_t,\delta_t)\mathbb{E}[\theta_t^{\mathbf{v}}]+\sum_{t=1}^T C(\alpha_t,\beta_t,\delta_t)\mathbb{E}[\theta_t^{\mathbf{y}}] \tag{108a}$$

$$+\sum_{t=1}^T D(\alpha_t)\mathbb{E}\|e_t^f\|^2+\sum_{t=1}^T F(\alpha_t,\beta_t)\mathbb{E}\|e_t^g\|^2+\sum_{t=1}^T I(\alpha_t,\delta_t)\mathbb{E}\|e_t^{\mathbf{v}}\|^2 \tag{108b}$$

$$+\sum_{t=1}^T L(\alpha_t,\beta_t)\mathbb{E}\|\nabla_{\mathbf{y}}g_t(\mathbf{x}_t,\mathbf{y}_t)\|^2+\sum_{t=2}^T N(\beta_t,\delta_t)\sup_{\mathbf{x}\in\mathcal{X}}\|\mathbf{y}_{t-1}^*(\mathbf{x})-\mathbf{y}_t^*(\mathbf{x})\|^2 \tag{108c}$$

$$+\frac{\sigma_{g_{\mathbf{y}}}^2}{\bar{b}}\frac{2}{\Phi}\sum_{t=1}^T\frac{\gamma_{t+1}}{\alpha_t}+\frac{4}{\Psi}\left(\frac{\sigma_{g_{\mathbf{yy}}}^2}{\bar{b}}p^2+\frac{\sigma_{f_{\mathbf{y}}}^2}{b}\right)\sum_{t=1}^T\frac{\lambda_{t+1}^2}{\alpha_t}+\frac{4}{\Omega}\left(\frac{\sigma_{g_{\mathbf{xy}}}^2}{\bar{b}}p^2+\frac{\sigma_{f_{\mathbf{x}}}^2}{b}\right)\sum_{t=1}^T\frac{\eta_{t+1}^2}{\alpha_t} \tag{108d}$$

$$+\frac{6}{\Phi\alpha_T}G_{\mathbf{y},T}+\frac{12p^2}{\Omega\alpha_T}G_{\mathbf{xy},T}+\frac{12p^2}{\Psi\alpha_T}G_{\mathbf{yy},T}+\frac{12\ell_{f,1}^2}{\Psi\alpha_T}D_{\mathbf{y},T}+\frac{12\ell_{f,1}^2}{\Omega\alpha_T}D_{\mathbf{x},T}. \tag{108e}$$

Here, $M$ is defined in Assumption 2.4, $V_T$ and $H_{2,T}$ are defined in (11). Moreover, $G_{\mathbf{y},T}$, $G_{\mathbf{xy},T}$, and $G_{\mathbf{yy},T}$ are defined in (13). Let

$$E(\alpha_t, \beta_t, \delta_t) := \frac{4L_{\mathbf{y}}^2}{L_{\mu_g}\beta_t}\Gamma + \frac{8\nu^2}{\acute{L}_{\mu_g}\mu_g^2\delta_t}(1 + 2L_{\mathbf{y}}^2)\Upsilon + 72(1 - \eta_{t+1})^2(\ell_{g,2}^2 p^2 + \ell_{f,1}^2)\frac{1}{\Omega\alpha_t}$$

$$+ 24(1 - \gamma_{t+1})^2\ell_{g,1}^2\frac{1}{\Phi\alpha_t} + 72(1 - \lambda_{t+1})^2(\ell_{g,2}^2 p^2 + \ell_{f,1}^2)\frac{1}{\Psi\alpha_t},$$

$$A(\alpha_t, \beta_t, \delta_t) := \alpha_t - (L_f + 2E(\alpha_t, \beta_t, \delta_t))\alpha_t^2,$$

$$B(\alpha_t, \beta_t, \delta_t) := -\frac{\acute{L}_{\mu_g}\Upsilon}{4}\delta_t + 4M_f^2\alpha_t - 2M_f^2 L_f\alpha_t^2 + 2M_f^2 E(\alpha_t, \beta_t, \delta_t)\alpha_t^2$$

$$+ 72(1 - \lambda_{t+1})^2\ell_{g,1}^4\delta_t^2\frac{1}{\Psi\alpha_t} + 72(1 - \eta_{t+1})^2\ell_{g,1}^4\delta_t^2\frac{1}{\Omega\alpha_t},$$

$$C(\alpha_t, \beta_t, \delta_t) := -\frac{L_{\mu_g}\Gamma}{2}\beta_t + 4M_f^2\alpha_t - 2L_f M_f^2\alpha_t^2 + 2M_f^2 E(\alpha_t, \beta_t, \delta_t)\alpha_t^2 \qquad (109)$$

$$+ \frac{288}{\Psi\alpha_t}\ell_{g,1}^2(p^2\ell_{g,2}^2 + \ell_{f,1}^2)\delta_t^2,$$

$$D(\alpha_t) := 2\left(2\alpha_t - L_f\alpha_t^2\right) - 5\alpha_t,$$

$$F(\alpha_t, \beta_t) := \frac{2\Gamma}{L_{\mu_g}}\beta_t - \hbar\alpha_t + 144(1 - \lambda_{t+1})^2(\ell_{g,2}^2 p^2 + \ell_{f,1}^2)\frac{\beta_t^2}{\Psi\alpha_t}$$

$$+ 144(1 - \eta_{t+1})^2(\ell_{g,2}^2 p^2 + \ell_{f,1}^2)\frac{\beta_t^2}{\Omega\alpha_t},$$

$$I(\alpha_t, \delta_t) := \frac{4\Upsilon}{\acute{L}_{\mu_g}}\delta_t - \jmath\alpha_t + 72\ell_{g,1}^2(1 - \eta_{t+1})^2\frac{\delta_t^2}{\Omega\alpha_t}.$$

Moreover, we have

$$L(\alpha_t, \beta_t) := -\frac{2\Gamma}{\mu_g + \ell_{g,1}}\beta_t + \Gamma\beta_t^2 + 48(1 - \gamma_{t+1})^2\ell_{g,1}^2\frac{\beta_t^2}{\Phi\alpha_t}$$

$$+ 144(1 - \lambda_{t+1})^2(\ell_{g,2}^2 p^2 + \ell_{f,1}^2)\frac{\beta_t^2}{\Psi\alpha_t} + 144(1 - \eta_{t+1})^2(\ell_{g,2}^2 p^2 + \ell_{f,1}^2)\frac{\beta_t^2}{\Omega\alpha_t}, \quad (110)$$

$$N(\beta_t, \delta_t) := \frac{4}{L_{\mu_g}\beta_t}\Gamma + \frac{16\nu^2}{\acute{L}_{\mu_g}\mu_g^2\delta_t}\Upsilon.$$

Note that, we have

$$E(\alpha_t, \beta_t, \delta_t) = \frac{4L_{\mathbf{y}}^2}{L_{\mu_g}\beta_t}\Gamma + \frac{8\nu^2}{\acute{L}_{\mu_g}\mu_g^2\delta_t}(1 + 2L_{\mathbf{y}}^2)\Upsilon + 72(1 - \eta_{t+1})^2(\ell_{g,2}^2 p^2 + \ell_{f,1}^2)\frac{1}{\Omega\alpha_t}$$

$$+ 24(1 - \gamma_{t+1})^2\ell_{g,1}^2\frac{1}{\Phi\alpha_t} + 72(1 - \lambda_{t+1})^2(\ell_{g,2}^2 p^2 + \ell_{f,1}^2)\frac{1}{\Psi\alpha_t},$$

which together with $\beta_t = c_\beta\alpha_t$ and $\delta_t = c_\delta\alpha_t$ in Eq. (16), we have

$$\alpha_t^2 E(\alpha_t, \beta_t, \delta_t) = \frac{4L_{\mathbf{y}}^2}{L_{\mu_g}}\Gamma\frac{\alpha_t^2}{\beta_t} + \frac{8\nu^2}{\acute{L}_{\mu_g}\mu_g^2}(1 + 2L_{\mathbf{y}}^2)\Upsilon\frac{\alpha_t^2}{\delta_t} + 24(1 - \gamma_{t+1})^2\ell_{g,1}^2\frac{\alpha_t}{\Phi}$$

$$+ 72(1 - \eta_{t+1})^2(\ell_{g,2}^2 p^2 + \ell_{f,1}^2)\frac{\alpha_t}{\Omega} + 72(1 - \lambda_{t+1})^2(\ell_{g,2}^2 p^2 + \ell_{f,1}^2)\frac{\alpha_t}{\Psi}$$

$$\leq \frac{44L_{\mathbf{y}}^2}{L_{\mu_g}^2}M_f^2\frac{\alpha_t}{c_\beta^2} + \frac{176\nu^2}{\acute{L}_{\mu_g}^2\mu_g^2}(1 + 2L_{\mathbf{y}}^2)M_f^2\frac{\alpha_t}{c_\delta^2}$$

$$+ 24\ell_{g,1}^2\frac{\alpha_t}{\Phi} + 72(\ell_{g,2}^2 p^2 + \ell_{f,1}^2)(\frac{1}{\Omega} + \frac{1}{\Psi})\alpha_t$$

$$\leq \frac{\alpha_t}{4}, \qquad (111)$$

where the first inequality follows from $\Gamma = \frac{11M_f^2}{L_{\mu_g}c_\beta}$ and $\Upsilon = \frac{22M_f^2}{\acute{L}_{\mu_g}c_\delta}$ in (106); the last inequality follows from $c_\beta = \sqrt{880\frac{L_{\mathbf{y}}^2 M_f^2}{L_{\mu_g}^2}}$, $c_\delta = \sqrt{3520\frac{\nu^2 M_f^2}{\acute{L}_{\mu_g}^2 \mu_g^2}(1+2L_{\mathbf{y}}^2)}$, in (107) and $\Phi \geq 480\ell_{g,1}^2$, and $\Omega, \Psi \geq 1440(\ell_{g,2}^2 p^2 + \ell_{f,1}^2)$ in (106).

Moreover, we have

$$
\begin{aligned}
A(\alpha_t, \beta_t, \delta_t) &= \alpha_t - L_f\alpha_t^2 - 2E(\alpha_t, \beta_t, \delta_t)\alpha_t^2 \\
&\geq \alpha_t - L_f\alpha_t^2 - \frac{\alpha_t}{2} \\
&\geq \frac{\alpha_t}{4},
\end{aligned}
\tag{112}
$$

where the last inequality follows from $\alpha_t \leq 1/4L_f$ in (107) since $\alpha_t = 1/(c+t)^{1/3}$ in (16).

**Bounding** (108a) .

From (109), we have

$$
\begin{aligned}
B(\alpha_t, \beta_t, \delta_t) &= -\frac{\acute{L}_{\mu_g}\Upsilon}{4}\delta_t + 4M_f^2\alpha_t - 2M_f^2 L_f\alpha_t^2 + 2M_f^2 E(\alpha_t, \beta_t, \delta_t)\alpha_t^2 \\
&\quad + 72(1-\lambda_{t+1})^2\ell_{g,1}^4\delta_t^2\frac{1}{\Psi\alpha_t} + 72(1-\eta_{t+1})^2\ell_{g,1}^4\delta_t^2\frac{1}{\Omega\alpha_t} \\
&\leq -\frac{\acute{L}_{\mu_g}\Upsilon}{4}\delta_t + 4M_f^2\alpha_t - 2M_f^2 L_f\alpha_t^2 + \frac{M_f^2}{2}\alpha_t + 72\ell_{g,1}^4(\frac{1}{\Psi}+\frac{1}{\Omega})\frac{\delta_t^2}{\alpha_t} \\
&= \left(-\frac{\acute{L}_{\mu_g}}{4}\Upsilon c_\delta + \frac{9}{2}M_f^2 + 72\ell_{g,1}^4(\frac{1}{\Psi}+\frac{1}{\Omega})c_\delta^2\right)\alpha_t \\
&\leq -\frac{1}{2}M_f^2\alpha_t,
\end{aligned}
\tag{113}
$$

where the first inequality follows from $\beta_t = c_\beta\alpha_t$, $\delta_t = c_\delta\alpha_t$ in (16), and Eq. (111); the second inequality is by $\Upsilon = \frac{22M_f^2}{\acute{L}_{\mu_g}c_\delta}$, and $\Psi, \Omega \geq \frac{288\ell_{g,1}^4}{M_f^2}c_\delta^2$ in (106); the last inequality follows from in (106).

Moreover, using Eq. (109) together with $\beta_t = c_\beta\alpha_t$ and $\delta_t = c_\delta\alpha_t$ in Eq. (16), we have

$$
\begin{aligned}
C(\alpha_t, \beta_t, \delta_t) &= -\frac{L_{\mu_g}\Gamma}{2}\beta_t + 4M_f^2\alpha_t - 2L_f M_f^2\alpha_t^2 + 2M_f^2 E(\alpha_t, \beta_t, \delta_t)\alpha_t^2 \\
&\quad + \frac{288}{\Psi\alpha_t}\ell_{g,1}^2(p^2\ell_{g,2}^2 + \ell_{f,1}^2)\delta_t^2 \\
&\leq -\frac{L_{\mu_g}}{2}\Gamma c_\beta\alpha_t + \frac{9}{2}M_f^2\alpha_t + \frac{288}{\Psi}\ell_{g,1}^2(p^2\ell_{g,2}^2 + \ell_{f,1}^2)c_\delta^2\alpha_t \\
&\leq -\frac{M_f^2}{2}\alpha_t,
\end{aligned}
\tag{114}
$$

where the first inequality follows from (111); the last inequality follows from $\Gamma = \frac{11M_f^2}{L_{\mu_g}c_\beta}$ and $\Psi \geq \frac{576}{M_f^2}\ell_{g,1}^2(p^2\ell_{g,2}^2 + \ell_{f,1}^2)c_\delta^2$ in (106).

Thus, from (113) and (114), we get

$$
(108a) \leq \mathcal{O}\left(V_T\right).
\tag{115}
$$

**Bounding** (108b) .

From (109), we also have

$$
D(\alpha_t) = 4\alpha_t - 2L_f\alpha_t^2 - 5\alpha_t \leq 0.
$$

From Eq. (109), $\beta_t = c_\beta \alpha_t$, and $\Gamma = \frac{11 M_f^2}{L_{\mu_g} c_\beta}$, we obtain

$$F(\alpha_t, \beta_t) = \frac{2\Gamma}{L_{\mu_g}} \beta_t - \hbar \alpha_t + 144(1 - \lambda_{t+1})^2 (\ell_{g,2}^2 p^2 + \ell_{f,1}^2) \frac{\beta_t^2}{\Psi \alpha_t}$$

$$+ 144(1 - \eta_{t+1})^2 (\ell_{g,2}^2 p^2 + \ell_{f,1}^2) \frac{\beta_t^2}{\Omega \alpha_t}$$

$$\leq \frac{22 M_f^2}{L_{\mu_g}^2} \alpha_t - \hbar \alpha_t + 144(\ell_{g,2}^2 p^2 + \ell_{f,1}^2)(\frac{1}{\Psi} + \frac{1}{\Omega}) c_\beta^2 \alpha_t$$

$$\leq 24 \frac{M_f^2}{L_{\mu_g}^2} \alpha_t - \hbar \alpha_t$$

$$= -\frac{M_f^2}{L_{\mu_g}^2} \alpha_t,$$

where the second inequality follows from $\Omega, \Psi \geq 144(\ell_{g,2}^2 p^2 + \ell_{f,1}^2) \frac{L_{\mu_g}^2 c_\beta^2}{M_f^2}$ in (106); and the last equality is by $\hbar = 25 \frac{M_f^2}{L_{\mu_g}^2}$.

From $\delta_t = c_\delta \alpha_t$ in (16), we obtain

$$I(\alpha_t, \delta_t) = \frac{4\Upsilon}{\acute{L}_{\mu_g}} \delta_t - \jmath \alpha_t + 72 \ell_{g,1}^2 (1 - \eta_{t+1})^2 \frac{\delta_t^2}{\Omega \alpha_t}$$

$$\leq \frac{4\Upsilon}{\acute{L}_{\mu_g}} c_\delta \alpha_t - \jmath \alpha_t + 72 \ell_{g,1}^2 \frac{c_\delta^2 \alpha_t}{\Omega}$$

$$\leq \frac{89 M_f^2}{\acute{L}_{\mu_g}^2} \alpha_t - \jmath \alpha_t$$

$$= -\frac{M_f^2}{\acute{L}_{\mu_g}^2} \alpha_t,$$

where the second inequality follows from $\Upsilon = \frac{22 M_f^2}{\acute{L}_{\mu_g} c_\delta}$ and $\Omega \geq \frac{72 \ell_{g,1}^2 \acute{L}_{\mu_g}^2}{M_f^2} c_\delta^2$; the last equality follows from $\jmath = 90 \frac{M_f^2}{\acute{L}_{\mu_g}^2}$.

Thus, we get

$$(108b) \leq 0. \tag{116}$$

**Bounding** (108c) .
From $\beta_t = c_\beta \alpha_t$ in (16) and Eq. (110), we have

$$L(\alpha_t, \beta_t) = -\frac{2\Gamma \beta_t}{\mu_g + \ell_{g,1}} + \Gamma \beta_t^2 + 48(1 - \gamma_{t+1})^2 \ell_{g,1}^2 \frac{\beta_t^2}{\Phi \alpha_t}$$

$$+ 144(1 - \lambda_{t+1})^2 (\ell_{g,2}^2 p^2 + \ell_{f,1}^2) \frac{\beta_t^2}{\Psi \alpha_t} + 144(1 - \eta_{t+1})^2 (\ell_{g,2}^2 p^2 + \ell_{f,1}^2) \frac{\beta_t^2}{\Omega \alpha_t}$$

$$\leq -\frac{2\Gamma c_\beta \alpha_t}{\mu_g + \ell_{g,1}} + \Gamma c_\beta^2 \alpha_t^2 + 48 \ell_{g,1}^2 c_\beta^2 \frac{\alpha_t}{\Phi} + 144(\ell_{g,2}^2 p^2 + \ell_{f,1}^2)(\frac{1}{\Psi} + \frac{1}{\Omega}) c_\beta^2 \alpha_t$$

$$\leq -\frac{2\Gamma c_\beta \alpha_t}{\mu_g + \ell_{g,1}} + \Gamma c_\beta^2 \alpha_t^2 + \frac{3\Gamma c_\beta \alpha_t}{4(\mu_g + \ell_{g,1})}$$

$$\leq -\frac{\Gamma c_\beta \alpha_t}{4(\mu_g + \ell_{g,1})},$$

where the second inequality is by $\Phi \geq 192 \ell_{g,1}^2 \frac{(\mu_g + \ell_{g,1})}{\Gamma} c_\beta$, and $\Omega, \Psi \geq 576(\ell_{g,2}^2 p^2 + \ell_{f,1}^2) \frac{(\mu_g + \ell_{g,1})}{\Gamma} c_\beta$ in (106); the last inequality follows from $\alpha_t \leq 1/c_\beta(\mu_g + \ell_{g,1})$ in (107).

From $\beta_t = c_\beta \alpha_t$, $\delta_t = c_\delta \alpha_t$ in (16) and Eq. (110), we obtain

$$N(\beta_t, \delta_t) = \frac{4}{L_{\mu_g} \beta_t} \Gamma + \frac{16\nu^2}{\acute{L}_{\mu_g} \mu_g^2 \delta_t} \Upsilon = \frac{4}{L_{\mu_g} c_\beta \alpha_t} \Gamma + \frac{16\nu^2}{\acute{L}_{\mu_g} \mu_g^2 c_\delta \alpha_t} \Upsilon.$$

Thus, we get

$$(108c) = \sum_{t=1}^{T} L(\alpha_t, \beta_t) \mathbb{E} \|\nabla_{\mathbf{y}} g_t(\mathbf{x}_t, \mathbf{y}_t)\|^2 + \sum_{t=2}^{T} N(\beta_t, \delta_t) \sup_{\mathbf{x} \in \mathcal{X}} \|\mathbf{y}_{t-1}^*(\mathbf{x}) - \mathbf{y}_t^*(\mathbf{x})\|^2$$
$$\leq \mathcal{O}\left(\frac{H_{2,T}}{\alpha_T}\right). \tag{117}$$

**Bounding** (108d) .
From $\eta_{t+1} = c_\eta \alpha_t^2$, $\gamma_{t+1} = c_\gamma \alpha_t^2$, $\lambda_{t+1} = c_\lambda \alpha_t^2$ in Eq. (16), we obtain

$$(108d) = \frac{\sigma_{g_{\mathbf{y}}}^2}{\bar{b}} \frac{2}{\Phi} \sum_{t=1}^{T} \frac{\gamma_{t+1}^2}{\alpha_t} + \frac{4}{\Psi}\left(\frac{\sigma_{g_{\mathbf{yy}}}^2}{\bar{b}} p^2 + \frac{\sigma_{f_{\mathbf{y}}}^2}{b}\right) \sum_{t=1}^{T} \frac{\lambda_{t+1}^2}{\alpha_t} + \frac{4}{\Omega}\left(\frac{\sigma_{g_{\mathbf{xy}}}^2}{\bar{b}} p^2 + \frac{\sigma_{f_{\mathbf{x}}}^2}{b}\right) \sum_{t=1}^{T} \frac{\eta_{t+1}^2}{\alpha_t}$$
$$\leq \mathcal{O}\left(\left(\frac{\sigma_{g_{\mathbf{y}}}^2}{\bar{b}} + \frac{\sigma_{g_{\mathbf{yy}}}^2}{\bar{b}} + \frac{\sigma_{f_{\mathbf{y}}}^2}{b} + \frac{\sigma_{g_{\mathbf{xy}}}^2}{\bar{b}} + \frac{\sigma_{f_{\mathbf{x}}}^2}{b}\right) \sum_{t=1}^{T} \alpha_t^3\right). \tag{118}$$

**Bounding** (108e) .
We also have

$$(108e) = \frac{6}{\Phi \alpha_T} G_{\mathbf{y},T} + \frac{12p^2}{\Omega \alpha_T} G_{\mathbf{xy},T} + \frac{12p^2}{\Psi \alpha_T} G_{\mathbf{yy},T} + \frac{12\ell_{f,1}^2}{\Psi \alpha_T} D_{\mathbf{y},T} + \frac{12\ell_{f,1}^2}{\Omega \alpha_T} D_{\mathbf{x},T}$$
$$\leq \mathcal{O}\left(\frac{1}{\alpha_T}(G_{\mathbf{y},T} + G_{\mathbf{xy},T} + G_{\mathbf{yy},T} + D_{\mathbf{y},T} + D_{\mathbf{x},T})\right). \tag{119}$$

From Eq. (16), we have $b = \bar{b} = 1$. Moreover, by (10), $\sigma^2 = \sigma_{g_{\mathbf{y}}}^2 + \sigma_{g_{\mathbf{yy}}}^2 + \sigma_{f_{\mathbf{y}}}^2 + \sigma_{g_{\mathbf{xy}}}^2 + \sigma_{f_{\mathbf{x}}}^2$. From (15), we also have

$$G_T = G_{\mathbf{y},T} + G_{\mathbf{xy},T} + G_{\mathbf{yy},T},$$
$$D_T = D_{\mathbf{y},T} + D_{\mathbf{x},T}.$$

Then, by inequalities (112), (115), (116), (117), (118), (119), we have

$$\sum_{t=1}^{T} \frac{\alpha_t}{2} \mathbb{E} \|\mathcal{P}_{\mathcal{X},\alpha_t}(\mathbf{x}_t; \nabla f_t(\mathbf{x}_t, \mathbf{y}_t^*(\mathbf{x}_t)))\|^2 + \Lambda$$
$$\leq \mathcal{O}\left(V_T + \frac{H_{2,T}}{\alpha_T} + \frac{\sigma^2}{b} \sum_{t=1}^{T} \alpha_t^3 + \frac{G_T}{\alpha_T} + \frac{D_T}{\alpha_T}\right). \tag{120}$$

From the definition of $\Lambda$ in (105), we have

$$-\Lambda = \Gamma \sum_{t=1}^{T} \left(\mathbb{E}[\theta_t^{\mathbf{y}}] - \mathbb{E}[\theta_{t+1}^{\mathbf{y}}]\right) + \Upsilon \sum_{t=1}^{T} \left(\mathbb{E}[\theta_t^{\mathbf{v}}] - \mathbb{E}[\theta_{t+1}^{\mathbf{v}}]\right) + \frac{1}{\Phi} \sum_{t=1}^{T} \left(\frac{\mathbb{E}\|e_t^g\|^2}{\alpha_{t-1}} - \frac{\mathbb{E}\|e_{t+1}^g\|^2}{\alpha_t}\right)$$
$$+ \frac{1}{\Psi} \sum_{t=1}^{T} \left(\frac{\mathbb{E}\|e_t^{\mathbf{v}}\|^2}{\alpha_{t-1}} - \frac{\mathbb{E}\|e_{t+1}^{\mathbf{v}}\|^2}{\alpha_t}\right) + \frac{1}{\Omega} \sum_{t=1}^{T} \left(\frac{\mathbb{E}\|e_t^f\|^2}{\alpha_{t-1}} - \frac{\mathbb{E}\|e_{t+1}^f\|^2}{\alpha_t}\right)$$
$$\leq \Gamma \theta_1^{\mathbf{y}} + \Upsilon \theta_1^{\mathbf{v}} + \frac{\sigma_{g_{\mathbf{y}}}^2}{\Phi \alpha_0} + \frac{\sigma_{g_{\mathbf{yy}}}^2 + \sigma_{f_{\mathbf{y}}}^2}{\Psi \alpha_0} + \frac{\sigma_{g_{\mathbf{xy}}}^2 + \sigma_{f_{\mathbf{x}}}^2}{\Omega \alpha_0}. \tag{121}$$

Using (121), we get

$$\sum_{t=1}^{T} \frac{\alpha_t}{2} \mathbb{E} \|\mathcal{P}_{\mathcal{X},\alpha_t}(\mathbf{x}_t; \nabla f_t(\mathbf{x}_t, \mathbf{y}_t^*(\mathbf{x}_t)))\|^2$$
$$\leq \mathcal{O}\left(V_T + \frac{H_{2,T}}{\alpha_T} + \frac{\sigma^2}{b} \sum_{t=1}^{T} \alpha_t^3 + \frac{G_T}{\alpha_T} + \frac{D_T}{\alpha_T} - \Lambda\right)$$
$$\leq \mathcal{O}\left(V_T + \theta_1^{\mathbf{y}} + \theta_1^{\mathbf{v}} + \frac{\sigma^2}{b} \sum_{t=1}^{T} \alpha_t^3 + \frac{H_{2,T}}{\alpha_T} + \frac{G_T}{\alpha_T} + \frac{D_T}{\alpha_T} + \frac{\sigma^2}{\alpha_0}\right).$$

Since $\alpha_t = 1/(c+t)^{1/3}$, we get

$$\sum_{t=1}^{T} \alpha_t^3 = \sum_{t=1}^{T} \frac{1}{c+t} \leq \sum_{t=1}^{T} \frac{1}{1+t} \leq \log(T+1),$$

which, combined with the fact that $\alpha_t$ decreases with respect to $t$ and by multiplying both sides by $2/\alpha_T$, results in Thus, we have

$$\begin{aligned}
\text{BL-Reg}_T &= \sum_{t=1}^{T} \mathbb{E} \left\| \mathcal{P}_{\mathcal{X},\alpha_t} \left( \mathbf{x}_t; \nabla f_t(\mathbf{x}_t, \mathbf{y}_t^*(\mathbf{x}_t)) \right) \right\|^2 \\
&\leq \mathcal{O}\Big( \frac{1}{\alpha_T}(V_T + \|\mathbf{y}_1 - \mathbf{y}_1^*(\mathbf{x}_1)\|^2 + \|\mathbf{v}_1 - \mathbf{v}_1^*(\mathbf{x}_1)\|^2 + \sigma^2 \log(T+1) + \frac{\sigma^2}{\alpha_0}) \\
&\quad + \frac{1}{\alpha_T^2}(H_{2,T} + G_T + D_T)\Big).
\end{aligned}$$

This completes the proof. □

# D Proof of Regret Bounds for Zeroth Order SOGD (ZO-SOGD)

**Proof Roadmap**. We provide Lemma D.7, which quantifies the error between the approximated direction of the momentum-based gradient estimator, $\hat{\mathbf{d}}_t^{\mathbf{y}}$ and the true direction, $\nabla_{\mathbf{y}} g_{t,\rho}(\mathbf{x}_t, \mathbf{y}_t)$, at each iteration. Lemma D.9 assesses the convergence of the iterative solutions $\{\mathbf{y}_t\}_{t=1}^{T}$, specifically the gap $\mathbb{E}[\|\mathbf{y}_{t+1} - \hat{\mathbf{y}}_t^*(\mathbf{x}_t)\|^2]$, while accounting for the error introduced in Lemma D.7. To establish Lemma D.13, which quantifies the error between the approximated direction of the momentum-based gradient estimator, $\hat{\mathbf{d}}_t^{\mathbf{Y}}$, and the true direction, $\nabla_{\mathbf{y}} f_{t,\rho}(\mathbf{x}_t, \mathbf{y}_t) + \nabla_{\mathbf{y}}^2 g_{t,\rho}(\mathbf{x}_t, \mathbf{y}_t) \mathbf{v}_t$, we need to present Lemma D.11. This lemma quantifies the error between $\hat{\mathbf{d}}_t^{\mathbf{Y}}$ and $\nabla_{\mathbf{y}} f_{t,\rho}(\mathbf{x}_t, \mathbf{y}_t) + \frac{1}{2\rho_{\mathbf{v}}}(\nabla_{\mathbf{y}} g_{t,\rho}(\mathbf{x}_t, \mathbf{y}_t + \rho_{\mathbf{v}} \mathbf{v}_t) - \nabla_{\mathbf{y}} g_{t,\rho}(\mathbf{x}_t, \mathbf{y}_t - \rho_{\mathbf{v}} \mathbf{v}_t))$. Then, Lemma D.15 captures the error of the system solution of Problem (18), i.e., gap $\mathbb{E}[\|\mathbf{v}_{t+1} - \hat{\mathbf{v}}_t^*(\mathbf{x}_t)\|^2]$, based on these errors. To establish Lemma D.19, which quantifies the error between the approximated direction of the momentum-based hypergradient estimator, $\hat{\mathbf{d}}_t^{\mathbf{x}}$, and the true direction, $\nabla_{\mathbf{x}} f_{t,\rho}(\mathbf{x}_t, \mathbf{y}_t) + \nabla_{\mathbf{xy}}^2 g_{t,\rho}(\mathbf{x}_t, \mathbf{y}_t) \mathbf{v}_t$, it is necessary to introduce Lemma D.17. This lemma quantifies the error between $\hat{\mathbf{d}}_t^{\mathbf{x}}$ and $\nabla_{\mathbf{x}} f_{t,\rho}(\mathbf{x}_t, \mathbf{y}_t) + \frac{1}{2\rho_{\mathbf{v}}}(\nabla_{\mathbf{x}} g_{t,\rho}(\mathbf{x}_t, \mathbf{y}_t + \rho_{\mathbf{v}} \mathbf{v}_t) - \nabla_{\mathbf{x}} g_{t,\rho}(\mathbf{x}_t, \mathbf{y}_t - \rho_{\mathbf{v}} \mathbf{v}_t))$. Then, Lemma D.20 bounds the projection mapping based on these errors. By combining these lemmas and setting parameters, we achieve the desired result.

## D.1 Auxiliary Lemmas for Proof of Theorem 3.2

**Lemma D.1.** *[4, Lemma A.1.] Suppose Assumption B4. holds. Then, for any $\mathbf{x}, \mathbf{v} \in \mathcal{X}$, we have:*

$$\left\| \nabla g_t(\mathbf{x} + \mathbf{v}, \mathbf{y} + \mathbf{v}) - \nabla g_t(\mathbf{x}, \mathbf{y}) - \nabla^2 g_t(\mathbf{x}, \mathbf{y}) \mathbf{v} \right\| \leq \ell_{g,2} \|\mathbf{v}\|^2.$$

**Lemma D.2.** *Suppose that Assumptions 2.2 and 2.3 hold for all $\mathbf{x}, \mathbf{x}' \in \mathcal{X}$, and $t \in [T]$, and that $\mathbf{d}_{t,\rho}^{\mathbf{x}}$ and $\mathbf{d}_{t,\rho}^{\mathbf{y}}$ are defined in (22). Then, we have*

$$\|\mathbf{d}_{t,\rho}^{\mathbf{x}} - \nabla f_{t,\rho}(\mathbf{x}, \hat{\mathbf{y}}_t^*(\mathbf{x}))\|^2 \leq M_f^2 \left( \|\mathbf{y} - \hat{\mathbf{y}}_t^*(\mathbf{x})\|^2 + \|\mathbf{v} - \hat{\mathbf{v}}_t^*(\mathbf{x})\|^2 \right), \tag{122a}$$

$$\left\| \mathbf{d}_{t,\rho}^{\mathbf{y}} \right\|^2 \leq M_{\mathbf{v}}^2 \left( \|\mathbf{y} - \hat{\mathbf{y}}_t^*(\mathbf{x})\|^2 + \|\mathbf{v} - \hat{\mathbf{v}}_t^*(\mathbf{x})\|^2 \right), \tag{122b}$$

$$\|\nabla f_{t,\rho}(\mathbf{x}, \hat{\mathbf{y}}_t^*(\mathbf{x})) - \nabla f_{t,\rho}(\mathbf{x}', \hat{\mathbf{y}}_t^*(\mathbf{x}'))\| \leq L_f \|\mathbf{x} - \mathbf{x}'\|, \tag{122c}$$

$$\|\hat{\mathbf{y}}_t^*(\mathbf{x}) - \hat{\mathbf{y}}_t^*(\mathbf{x}')\| \leq L_{\mathbf{y}} \|\mathbf{x} - \mathbf{x}'\|, \tag{122d}$$

$$\|\hat{\mathbf{v}}_t^*(\mathbf{x}) - \hat{\mathbf{v}}_t^*(\mathbf{x}')\| \leq L_{\mathbf{v}} \|\mathbf{x} - \mathbf{x}'\|. \tag{122e}$$

*Here, $\hat{\mathbf{v}}_t^*(\mathbf{x})$, $f_{t,\rho}$ and $\hat{\mathbf{y}}_t^*(\mathbf{x})$ are defined in (20), (18), and (19), respectively. Moreover, the constants $M_f$, $M_{\mathbf{v}}$, and $(L_{\mathbf{y}}, L_{\mathbf{v}}, L_f)$ are defined as in (42), (43), and (44), respectively.*

*Proof.* We first show Eq. (122a).

Using Assumptions 2.2 and B1., we have $\nabla_{\mathbf{y}}^2 g_{t,\boldsymbol{\rho}}\left(\mathbf{x}, \hat{\mathbf{y}}_t^*(\mathbf{x})\right) \succeq \mu_g$, and

$$\|\hat{\mathbf{v}}_t^*(\mathbf{x})\| = \|\left(\nabla_{\mathbf{y}}^2 g_{t,\boldsymbol{\rho}}\left(\mathbf{x}, \hat{\mathbf{y}}_t^*(\mathbf{x})\right)\right)^{-1} \nabla_{\mathbf{y}} f_{t,\boldsymbol{\rho}}\left(\mathbf{x}, \hat{\mathbf{y}}_t^*(\mathbf{x})\right)\| \le \frac{\ell_{f,0}}{\mu_g}. \tag{123}$$

Observe that we have

$$\begin{aligned}
\|\mathbf{d}_{t,\boldsymbol{\rho}}^{\mathbf{x}} - \nabla f_{t,\boldsymbol{\rho}}(\mathbf{x}, \hat{\mathbf{y}}_t^*(\mathbf{x}))\| &\le \|\nabla_{\mathbf{x}} f_{t,\boldsymbol{\rho}}(\mathbf{x}, \mathbf{y}) - \nabla_{\mathbf{x}} f_{t,\boldsymbol{\rho}}(\mathbf{x}, \hat{\mathbf{y}}_t^*(\mathbf{x}))\| \\
&\quad + \|\mathbf{v}\nabla_{\mathbf{xy}}^2 g_{t,\boldsymbol{\rho}}(\mathbf{x}, \mathbf{y}) - \hat{\mathbf{v}}_t^*(\mathbf{x})\nabla_{\mathbf{xy}}^2 g_{t,\boldsymbol{\rho}}\left(\mathbf{x}, \hat{\mathbf{y}}_t^*(\mathbf{x})\right)\| \\
&\le \|\nabla_{\mathbf{x}} f_{t,\boldsymbol{\rho}}(\mathbf{x}, \mathbf{y}) - \nabla_{\mathbf{x}} f_{t,\boldsymbol{\rho}}(\mathbf{x}, \hat{\mathbf{y}}_t^*(\mathbf{x}))\| \\
&\quad + \|\nabla_{\mathbf{xy}}^2 g_{t,\boldsymbol{\rho}}(\mathbf{x}, \mathbf{y})\|\|\mathbf{v} - \hat{\mathbf{v}}_t^*(\mathbf{x})\| \\
&\quad + \|\hat{\mathbf{v}}_t^*(\mathbf{x})\|\|\nabla_{\mathbf{xy}}^2 g_{t,\boldsymbol{\rho}}(\mathbf{x}, \mathbf{y}) - \nabla_{\mathbf{xy}}^2 g_{t,\boldsymbol{\rho}}(\mathbf{x}, \hat{\mathbf{y}}_t^*(\mathbf{x}))\| \\
&\le \left(\ell_{f,1} + \frac{\ell_{g,2}\ell_{f,0}}{\mu_g}\right)\|\mathbf{y} - \hat{\mathbf{y}}_t^*(\mathbf{x})\| + \ell_{g,1}\|\mathbf{v} - \hat{\mathbf{v}}_t^*(\mathbf{x})\| \\
&\le M_f^2 \left(\|\mathbf{y} - \hat{\mathbf{y}}_t^*(\mathbf{x})\| + \|\mathbf{v} - \hat{\mathbf{v}}_t^*(\mathbf{x})\|\right), \tag{124}
\end{aligned}$$

where $M_f$ is defined as in (42); the third inequality is by Assumption 2.3 and the last inequality is by Eq. (123).

We now show Eq. (122b).
Since $\mathbf{d}_{t,\boldsymbol{\rho}}^{\mathbf{v}*} := \nabla_{\mathbf{y}} f_{t,\boldsymbol{\rho}}(\mathbf{x}, \hat{\mathbf{y}}_t^*(\mathbf{x})) + \nabla_{\mathbf{y}}^2 g_{t,\boldsymbol{\rho}}\left(\mathbf{x}, \hat{\mathbf{y}}_t^*(\mathbf{x})\right) \hat{\mathbf{v}}_t^*(\mathbf{x}) = 0$, we have

$$\begin{aligned}
\|\mathbf{d}_{t,\boldsymbol{\rho}}^{\mathbf{v}}\| &= \|\mathbf{d}_{t,\boldsymbol{\rho}}^{\mathbf{v}} - \mathbf{d}_{t,\boldsymbol{\rho}}^{\mathbf{v}*}\| \\
&= \|\mathbf{v}_t \nabla_{\mathbf{y}}^2 g_{t,\boldsymbol{\rho}}(\mathbf{x}, \mathbf{y}) + \nabla_{\mathbf{y}} f_{t,\boldsymbol{\rho}}(\mathbf{x}, \mathbf{y}) \\
&\quad - \left(\hat{\mathbf{v}}_t^*(\mathbf{x})\nabla_{\mathbf{y}}^2 g_{t,\boldsymbol{\rho}}\left(\mathbf{x}, \hat{\mathbf{y}}_t^*(\mathbf{x})\right) + \nabla_{\mathbf{y}} f_{t,\boldsymbol{\rho}}(\mathbf{x}, \hat{\mathbf{y}}_t^*(\mathbf{x}))\right)\| \\
&\le \|\left(\nabla_{\mathbf{y}}^2 g_{t,\boldsymbol{\rho}}(\mathbf{x}, \mathbf{y}) - \nabla_{\mathbf{y}}^2 g_{t,\boldsymbol{\rho}}(\mathbf{x}, \hat{\mathbf{y}}_t^*(\mathbf{x}))\right) \hat{\mathbf{v}}_t^*(\mathbf{x})\| \\
&\quad + \|\nabla_{\mathbf{y}}^2 g_{t,\boldsymbol{\rho}}(\mathbf{x}, \mathbf{y}) (\mathbf{v} - \hat{\mathbf{v}}_t^*(\mathbf{x}))\| \\
&\quad + \|\nabla_{\mathbf{y}} f_{t,\boldsymbol{\rho}}(\mathbf{x}, \mathbf{y}) - \nabla_{\mathbf{y}} f_{t,\boldsymbol{\rho}}(\mathbf{x}, \hat{\mathbf{y}}_t^*(\mathbf{x}))\|.
\end{aligned}$$

Then, from Assumption 2.3 and Eq. (123), we have

$$\begin{aligned}
\|\mathbf{d}_{t,\boldsymbol{\rho}}^{\mathbf{v}}\| &\le \ell_{g,2}\|\mathbf{y} - \hat{\mathbf{y}}_t^*(\mathbf{x})\|\|\hat{\mathbf{v}}_t^*(\mathbf{x})\| + \ell_{g,1}\|\mathbf{v} - \hat{\mathbf{v}}_t^*(\mathbf{x})\| + \ell_{f,1}\|\mathbf{y} - \hat{\mathbf{y}}_t^*(\mathbf{x})\| \\
&\le \left(\frac{\ell_{g,2}\ell_{f,0}}{\mu_g} + \ell_{f,1}\right)\|\mathbf{y} - \hat{\mathbf{y}}_t^*(\mathbf{x})\| + \ell_{g,1}\|\mathbf{v} - \hat{\mathbf{v}}_t^*(\mathbf{x})\| \\
&\le M_{\mathbf{v}} \left(\|\mathbf{y} - \hat{\mathbf{y}}_t^*(\mathbf{x})\| + \|\mathbf{v} - \hat{\mathbf{v}}_t^*(\mathbf{x})\|\right),
\end{aligned}$$

where $M_{\mathbf{v}}$ is defined as in (43).

The proofs of Eqs. (122c)-(122e) follow from [67, Lemma 17] by setting $(L_{\mathbf{y}}, L_{\mathbf{v}}, L_f)$ as in (44). $\quad\square$

### D.2 Perturbation Bounds for OBO Objectives and Their Smoothing Variants

The following two lemmas are inspired by [24].

**Lemma D.3.** *Given $\boldsymbol{\rho} = (\rho_{\mathbf{s}}, \rho_{\mathbf{r}})$ as positive smoothing parameters, let $g_{t,\boldsymbol{\rho}}(\mathbf{x}, \mathbf{y})$ and $f_{t,\boldsymbol{\rho}}(\mathbf{x}, \mathbf{y})$ be the functions defined by (18).*

*(a) Suppose Assumption B3. holds. Then, we have*

$$|g_{t,\boldsymbol{\rho}}(\mathbf{x}, \mathbf{y}) - g_t(\mathbf{x}, \mathbf{y})| \le \frac{\ell_{g,1}(\rho_{\mathbf{s}}^2 + \rho_{\mathbf{r}}^2)}{2}. \tag{125}$$

*(b) Suppose Assumption B2. holds. Then, we have*

$$|f_{t,\boldsymbol{\rho}}(\mathbf{x}, \mathbf{y}) - f_t(\mathbf{x}, \mathbf{y})| \le \frac{\ell_{f,1}(\rho_{\mathbf{s}}^2 + \rho_{\mathbf{r}}^2)}{2}. \tag{126}$$

*Proof.* Let $B_1$ and $B_2$ be the unit ball in $\mathbb{R}^{d_1}$ and $\mathbb{R}^{d_2}$, respectively. Let $\mathcal{V}(d_1)$ and $\mathcal{V}(d_2)$ be volume of the unit ball in $\mathbb{R}^{d_1}$ and $\mathbb{R}^{d_2}$, respectively. Then, we have

$$|g_{t,\boldsymbol{\rho}}(\mathbf{x},\mathbf{y}) - g_t(\mathbf{x},\mathbf{y})|$$

$$= \left| \frac{1}{\mathcal{V}(d_1)\mathcal{V}(d_2)} \int_{B_1} \int_{B_2} \left( g_t(\mathbf{x} + \rho_{\mathbf{s}}\mathbf{s}, \mathbf{y} + \rho_{\mathbf{r}}\mathbf{r}) - g_t(\mathbf{x},\mathbf{y}) \right) d\mathbf{s} d\mathbf{r} \right|$$

$$= \left| \frac{1}{\mathcal{V}(d_1)\mathcal{V}(d_2)} \int_{B_1} \int_{B_2} \left( g_t(\mathbf{x} + \rho_{\mathbf{s}}\mathbf{s}, \mathbf{y} + \rho_{\mathbf{r}}\mathbf{r}) - g_t(\mathbf{x},\mathbf{y}) - \langle \nabla g_t(\mathbf{x},\mathbf{y}), (\rho_{\mathbf{s}}\mathbf{s}, \rho_{\mathbf{r}}\mathbf{r}) \rangle \right) d\mathbf{s} d\mathbf{r} \right|.$$

Thus, we get

$$|g_{t,\boldsymbol{\rho}}(\mathbf{x},\mathbf{y}) - g_t(\mathbf{x},\mathbf{y})|$$

$$\leq \int_{B_1} \int_{B_2} |g_t(\mathbf{x} + \rho_{\mathbf{s}}\mathbf{s}, \mathbf{y} + \rho_{\mathbf{r}}\mathbf{r}) - g_t(\mathbf{x},\mathbf{y}) - \langle \nabla g_t(\mathbf{x},\mathbf{y}), (\rho_{\mathbf{s}}\mathbf{s}, \rho_{\mathbf{r}}\mathbf{r}) \rangle| d\mathbf{s} d\mathbf{r}$$

$$\leq \int_{B_1} \int_{B_2} \frac{\ell_{g,1}}{2} \left( \rho_{\mathbf{s}}^2 \|\mathbf{s}\|^2 + \rho_{\mathbf{r}}^2 \|\mathbf{r}\|^2 \right) d\mathbf{s} d\mathbf{r}$$

$$= \frac{\ell_{g,1}\rho_{\mathbf{s}}^2}{2} \int_{B_1} \|\mathbf{s}\|^2 d\mathbf{s} + \frac{\ell_{g,1}\rho_{\mathbf{r}}^2}{2} \int_{B_2} \|\mathbf{r}\|^2 d\mathbf{r}$$

$$= \frac{\ell_{g,1}\rho_{\mathbf{s}}^2}{2} \frac{d_1}{d_1 + 2} + \frac{\ell_{g,1}\rho_{\mathbf{r}}^2}{2} \frac{d_2}{d_2 + 2}$$

$$\leq \frac{\ell_{g,1}(\rho_{\mathbf{s}}^2 + \rho_{\mathbf{r}}^2)}{2},$$

where the last equality follows since $\frac{1}{\mathcal{V}(d)} \int_{s \in B} \|s\|^p ds = \frac{d}{d+p}$.

The proof of part (b) follows using similar arguments. $\qquad\square$

**Lemma D.4.** *Given $\boldsymbol{\rho} = (\rho_{\mathbf{s}}, \rho_{\mathbf{r}})$ as positive smoothing parameters, let $g_{t,\boldsymbol{\rho}}(\mathbf{x},\mathbf{y})$ and $f_{t,\boldsymbol{\rho}}(\mathbf{x},\mathbf{y})$ be the functions defined by (18).*

*(a) Suppose Assumption* B3. *holds. Then, we have*

$$\|\nabla g_{t,\boldsymbol{\rho}}(\mathbf{x},\mathbf{y}) - \nabla g_t(\mathbf{x},\mathbf{y})\| \leq \frac{\ell_{g,1}(\rho_{\mathbf{s}}d_1 + \rho_{\mathbf{r}}d_2)}{2}. \tag{127}$$

*(b) Suppose Assumption* B2. *holds. Then, we have*

$$\|\nabla f_t(\mathbf{x},\mathbf{y}) - \nabla f_{t,\boldsymbol{\rho}}(\mathbf{x},\mathbf{y})\| \leq \frac{\ell_{f,1}(\rho_{\mathbf{s}}d_1 + \rho_{\mathbf{r}}d_2)}{2}. \tag{128}$$

*Proof.* Let $S(d_1)$ be the surface area of the unit sphere in $\mathbb{R}^{d_1}$. Moreover, let $U_{B_1}$ be the unit sphere.

$$\|\nabla_{\mathbf{x}} g_{t,\boldsymbol{\rho}}(\mathbf{x},\mathbf{y}) - \nabla_{\mathbf{x}} g_t(\mathbf{x},\mathbf{y})\|$$

$$= \left\| \frac{1}{S(d_1)} \left( \frac{d_1}{\rho_{\mathbf{s}}} \int_{U_{B_1}} g_t(\mathbf{x} + \rho_{\mathbf{s}}\mathbf{s}, \mathbf{y})\mathbf{s} d\mathbf{s} \right) - \nabla_{\mathbf{x}} g_t(\mathbf{x},\mathbf{y}) \right\|$$

$$= \left\| \frac{1}{S(d_1)} \left( \frac{d_1}{\rho_{\mathbf{s}}} \int_{U_{B_1}} g_t(\mathbf{x} + \rho_{\mathbf{s}}\mathbf{s}, \mathbf{y})\mathbf{s} d\mathbf{s} - \int_{U_{B_1}} \frac{d_1}{\rho_{\mathbf{s}}} g_t(\mathbf{x},\mathbf{y})\mathbf{s} d\mathbf{s} \right. \right.$$

$$\left. \left. - \int_{U_{B_1}} \frac{d_1}{\rho_{\mathbf{s}}} \langle \nabla_{\mathbf{x}} g_t(\mathbf{x},\mathbf{y}), \rho_{\mathbf{s}}\mathbf{s} \rangle \mathbf{s} d\mathbf{s} \right) \right\|$$

$$\leq \frac{d_1}{S(d_1)\rho_{\mathbf{s}}} \int_{U_{B_1}} \left| g_t(\mathbf{x}_t + \rho_{\mathbf{s}}\mathbf{s}, \mathbf{y}) - g_t(\mathbf{x},\mathbf{y}) - \langle \nabla_{\mathbf{x}} g_t(\mathbf{x},\mathbf{y}), \rho_{\mathbf{s}}\mathbf{s} \rangle \right| \|\mathbf{s}\| d\mathbf{s}$$

$$\leq \frac{d_1}{S(d_1)\rho_{\mathbf{s}}} \cdot \frac{\ell_{g,1}\rho_{\mathbf{s}}^2}{2} \int_{U_{B_1}} \|\mathbf{s}\|^3 d\mathbf{s}$$

$$= \frac{\rho_{\mathbf{s}}d_1\ell_{g,1}}{2}, \tag{129}$$

where the second equality follows from $\int_{U_{B_1}} \mathbf{s}\mathbf{s}^\top d\mathbf{s} = \frac{S(d_1)}{d_1}\mathbf{I}$.

Similarly, let $S(d_2)$ be the surface area of the unit sphere in $\mathbb{R}^{d_2}$. Moreover, let $U_{B_2}$ be the unit sphere.

$$
\begin{aligned}
& \|\nabla_{\mathbf{y}} g_{t,\boldsymbol{\rho}}(\mathbf{x},\mathbf{y}) - \nabla_{\mathbf{y}} g_t(\mathbf{x},\mathbf{y})\| \\
&= \left\| \frac{1}{S(d_2)} \left( \frac{d_2}{\rho_{\mathbf{r}}} \int_{U_{B_2}} g_t(\mathbf{x},\mathbf{y}+\rho_{\mathbf{r}}\mathbf{r})\mathbf{r}d\mathbf{r} \right) - \nabla_{\mathbf{y}} g_t(\mathbf{x},\mathbf{y}) \right\| \\
&= \left\| \frac{1}{S(d_2)} \left( \frac{d_2}{\rho_{\mathbf{r}}} \int_{U_{B_2}} g_t(\mathbf{x},\mathbf{y}+\rho_{\mathbf{r}}\mathbf{r})\mathbf{r}d\mathbf{r} - \int_{U_{B_2}} \frac{d_2}{\rho_{\mathbf{r}}} g_t(\mathbf{x},\mathbf{y})\mathbf{r}d\mathbf{r} \right. \right. \\
& \qquad \left. \left. - \int_{U_{B_2}} \frac{d_2}{\rho_{\mathbf{r}}} \langle \nabla_{\mathbf{y}} g_t(\mathbf{x},\mathbf{y}), \rho_{\mathbf{r}}\mathbf{r} \rangle \mathbf{r}d\mathbf{r} \right) \right\| \\
&\leq \frac{d_2}{S(d_2)\rho_{\mathbf{r}}} \int_{U_{B_2}} \left| g_t(\mathbf{x}_t,\mathbf{y}+\rho_{\mathbf{r}}\mathbf{r}) - g_t(\mathbf{x},\mathbf{y}) - \langle \nabla_{\mathbf{y}} g_t(\mathbf{x},\mathbf{y}), \rho_{\mathbf{r}}\mathbf{r} \rangle \right| \|\mathbf{r}\| \, d\mathbf{r} \\
&\leq \frac{d_2}{S(d_2)\rho_{\mathbf{r}}} \cdot \frac{\ell_{g,1}\rho_{\mathbf{r}}^2}{2} \int_{U_{B_2}} \|\mathbf{r}\|^3 \, d\mathbf{r} \\
&= \frac{\rho_{\mathbf{r}} d_2 \ell_{g,1}}{2},
\end{aligned}
\tag{130}
$$

where the second equality follows from $\int_{U_{B_2}} \mathbf{r}\mathbf{r}^\top d\mathbf{r} = \frac{S(d_2)}{d_2}\mathbf{I}$.

Thus, we get

$$
\begin{aligned}
& \|\nabla g_{t,\boldsymbol{\rho}}(\mathbf{x},\mathbf{y}) - \nabla g_t(\mathbf{x},\mathbf{y})\| \\
&\leq \|\nabla_{\mathbf{x}} g_{t,\boldsymbol{\rho}}(\mathbf{x},\mathbf{y}) - \nabla_{\mathbf{x}} g_t(\mathbf{x},\mathbf{y})\| + \|\nabla_{\mathbf{y}} g_{t,\boldsymbol{\rho}}(\mathbf{x},\mathbf{y}) - \nabla_{\mathbf{y}} g_t(\mathbf{x},\mathbf{y})\| \\
&\leq \frac{\rho_{\mathbf{s}} d_1 \ell_{g,1}}{2} + \frac{\rho_{\mathbf{r}} d_2 \ell_{g,1}}{2}.
\end{aligned}
$$

Finally, by a similar argument as in Part (a), we obtain

$$
\|\nabla_{\mathbf{x}} f_{t,\boldsymbol{\rho}}(\mathbf{x},\mathbf{y}) - \nabla_{\mathbf{x}} f_t(\mathbf{x},\mathbf{y})\| \leq \frac{\rho_{\mathbf{s}} d_1 \ell_{f,1}}{2},
\tag{131}
$$

and

$$
\|\nabla_{\mathbf{y}} f_{t,\boldsymbol{\rho}}(\mathbf{x},\mathbf{y}) - \nabla_{\mathbf{y}} f_t(\mathbf{x},\mathbf{y})\| \leq \frac{\rho_{\mathbf{r}} d_2 \ell_{f,1}}{2},
\tag{132}
$$

which implies

$$
\|\nabla f_{t,\boldsymbol{\rho}}(\mathbf{x},\mathbf{y}) - \nabla f_t(\mathbf{x},\mathbf{y})\| \leq \frac{(\rho_{\mathbf{s}} d_1 + \rho_{\mathbf{r}} d_2)\ell_{f,1}}{2}.
$$

$\square$

**Lemma D.5.** *Suppose Assumption* B3. *holds. Let* $\hat{\nabla}_{\mathbf{y}} g_t(\mathbf{x},\mathbf{y};\bar{\mathcal{B}}_t)$ *and* $\hat{\nabla}_{\mathbf{x}} g_t(\mathbf{x},\mathbf{y};\bar{\mathcal{B}}_t)$ *be defined as in* (24a) *and* (24b), *respectively. Then, for any* $(\mathbf{x},\mathbf{y}) \in \mathbb{R}^{d_1} \times \mathbb{R}^{d_2}$ *and* $\rho_{\mathbf{r}}, \rho_{\mathbf{s}} \geq 0$, *we have*

$$
\mathop{\mathbb{E}}_{(\mathbf{r},\bar{\mathcal{B}}_t)} \left[ \|\hat{\nabla}_{\mathbf{y}} g_t(\mathbf{x},\mathbf{y};\bar{\mathcal{B}}_t) - \hat{\nabla}_{\mathbf{y}} g_t(\mathbf{x},\acute{\mathbf{y}};\bar{\mathcal{B}}_t)\|^2 \right] \leq 3d_2 \ell_{g,1}^2 \|\mathbf{y} - \acute{\mathbf{y}}\|^2 + \frac{3\ell_{g,1}^2 d_2^2 \rho_{\mathbf{r}}^2}{2},
\tag{133a}
$$

$$
\mathop{\mathbb{E}}_{(\mathbf{s},\bar{\mathcal{B}}_t)} \left[ \|\hat{\nabla}_{\mathbf{x}} g_t(\mathbf{x},\mathbf{y};\bar{\mathcal{B}}_t) - \hat{\nabla}_{\mathbf{x}} g_t(\acute{\mathbf{x}},\mathbf{y};\bar{\mathcal{B}}_t)\|^2 \right] \leq 3d_1 \ell_{g,1}^2 \|\mathbf{x} - \acute{\mathbf{x}}\|^2 + \frac{3\ell_{g,1}^2 d_1^2 \rho_{\mathbf{s}}^2}{2},
\tag{133b}
$$

*for all* $\acute{\mathbf{y}} \in \mathbb{R}^{d_2}$ *and* $\acute{\mathbf{x}} \in \mathbb{R}^{d_1}$.

*Proof.* The proof is similar to that of Lemma 5 in [42]. $\square$

**Lemma D.6.** *Suppose Assumptions 2.2 and B3. hold. Let $(\rho_{\mathbf{s}}, \rho_{\mathbf{r}})$ be positive smoothing parameters. Let $\mathbf{y}_t^*(\mathbf{x})$ and $\hat{\mathbf{y}}_t^*(\mathbf{x})$ be defined in (1) and (19), respectively. Then, we have*

$$\mathbb{E}\left[\|\hat{\mathbf{y}}_t^*(\mathbf{x}) - \mathbf{y}_t^*(\mathbf{x})\|^2\right] \leq \frac{\ell_{g,1}(\rho_{\mathbf{s}}^2 + \rho_{\mathbf{r}}^2)}{\mu_g}. \tag{134}$$

*Proof.* From (1), we have $\mathbf{y}_t^*(\mathbf{x}) \in \arg\min_{\mathbf{y} \in \mathbb{R}^{d_2}} g_t(\mathbf{x}, \mathbf{y})$. Since, by Assumption 2.2, $g_t(\mathbf{x}, \mathbf{y})$ is $\mu_g$-strongly convex with respect to $\mathbf{y}$, it follows from Lemma B.2 that

$$\|\mathbf{y} - \mathbf{y}_t^*(\mathbf{x})\|^2 \leq \frac{2}{\mu_g}\left(g_t(\mathbf{x}, \mathbf{y}) - g_t(\mathbf{x}, \mathbf{y}_t^*(\mathbf{x}))\right).$$

By setting $\mathbf{y} = \hat{\mathbf{y}}_t^*(\mathbf{x})$, we have

$$\|\hat{\mathbf{y}}_t^*(\mathbf{x}) - \mathbf{y}_t^*(\mathbf{x})\|^2 \leq \frac{2}{\mu_g}\left(g_t(\mathbf{x}, \hat{\mathbf{y}}_t^*(\mathbf{x})) - g_t(\mathbf{x}, \mathbf{y}_t^*(\mathbf{x}))\right). \tag{135}$$

Similarly, from (19), we have

$$\hat{\mathbf{y}}_t^*(\mathbf{x}) \in \arg\min_{\mathbf{y} \in \mathbb{R}^{d_2}}\left\{g_{t,\boldsymbol{\rho}}(\mathbf{x}, \mathbf{y}) = \mathbb{E}_{(\mathbf{s},\mathbf{r},\zeta_t)}\left[g_t(\mathbf{x} + \rho_{\mathbf{s}}\mathbf{s}, \mathbf{y} + \rho_{\mathbf{r}}\mathbf{r}; \zeta_t)\right]\right\},$$

where $\boldsymbol{\rho} = (\rho_{\mathbf{s}}, \rho_{\mathbf{r}})$. By Assumption 2.2, $g_{t,\boldsymbol{\rho}}(\mathbf{x}, \mathbf{y})$ is $\mu_g$-strongly convex with respect to $\mathbf{y}$. Hence, according to Lemma B.2, we obtain

$$\|\mathbf{y} - \hat{\mathbf{y}}_t^*(\mathbf{x})\|^2 \leq \frac{2}{\mu_g}\left(g_{t,\boldsymbol{\rho}}(\mathbf{x}, \mathbf{y}) - g_{t,\boldsymbol{\rho}}(\mathbf{x}, \hat{\mathbf{y}}_t^*(\mathbf{x}))\right).$$

By setting $\mathbf{y} = \mathbf{y}_t^*(\mathbf{x})$, we have

$$\|\mathbf{y}_t^*(\mathbf{x}) - \hat{\mathbf{y}}_t^*(\mathbf{x})\|^2 \leq \frac{2}{\mu_g}\left(g_{t,\boldsymbol{\rho}}(\mathbf{x}, \mathbf{y}_t^*(\mathbf{x})) - g_{t,\boldsymbol{\rho}}(\mathbf{x}, \hat{\mathbf{y}}_t^*(\mathbf{x}))\right). \tag{136}$$

Summing up (135) and (136), we get

$$\|\mathbf{y}_t^*(\mathbf{x}) - \hat{\mathbf{y}}_t^*(\mathbf{x})\|^2 \leq \frac{1}{\mu_g}\left(g_{t,\boldsymbol{\rho}}(\mathbf{x}, \mathbf{y}_t^*(\mathbf{x})) - g_t(\mathbf{x}, \mathbf{y}_t^*(\mathbf{x}))\right)$$
$$+ \frac{1}{\mu_g}\left(g_t(\mathbf{x}, \hat{\mathbf{y}}_t^*(\mathbf{x})) - g_{t,\boldsymbol{\rho}}(\mathbf{x}, \hat{\mathbf{y}}_t^*(\mathbf{x}))\right),$$

which implies

$$\|\mathbf{y}_t^*(\mathbf{x}) - \hat{\mathbf{y}}_t^*(\mathbf{x})\|^2 \leq \frac{1}{\mu_g}\left|g_{t,\boldsymbol{\rho}}(\mathbf{x}, \mathbf{y}_t^*(\mathbf{x})) - g_t(\mathbf{x}, \mathbf{y}_t^*(\mathbf{x}))\right|$$
$$+ \frac{1}{\mu_g}\left|g_t(\mathbf{x}, \hat{\mathbf{y}}_t^*(\mathbf{x})) - g_{t,\boldsymbol{\rho}}(\mathbf{x}, \hat{\mathbf{y}}_t^*(\mathbf{x}))\right|$$
$$\leq \frac{\ell_{g,1}(\rho_{\mathbf{s}}^2 + \rho_{\mathbf{r}}^2)}{\mu_g},$$

where the last inequality is by Eq. (125). $\qquad\square$

### D.3 Bounds on the Zeroth-Order Inner Solution

Recall that $\mathbf{s} \in \mathbb{R}^{d_1}$ and $\mathbf{r} \in \mathbb{R}^{d_2}$ are vectors uniformly sampled from the unit balls $B_1$ and $B_2$, respectively. Let

$$U_b^{\mathbf{s}} = \{\mathbf{s}_i \in \mathbb{R}^{d_1}\}_{i=1}^b, \quad U_b^{\mathbf{r}} = \{\mathbf{r}_i \in \mathbb{R}^{d_2}\}_{i=1}^b,$$
$$U_{\bar{b}}^{\mathbf{s}} = \{\mathbf{s}_i \in \mathbb{R}^{d_1}\}_{i=1}^{\bar{b}}, \quad U_{\bar{b}}^{\mathbf{r}} = \{\mathbf{r}_i \in \mathbb{R}^{d_2}\}_{i=1}^{\bar{b}},$$

be generated from the uniform distributions over the unit spheres $(U_{B_1}, U_{B_2})$. Here, $(U_{B_1}, U_{B_2})$ denote the uniform distributions over the $(d_1, d_2)$-dimensional unit Euclidean balls $(B_1, B_2)$, respectively.

Then, similar to (23), we have

$$\mathbb{E}_{(U_b^{\mathbf{r}}, \mathcal{B}_t)}\left[\hat{\nabla}_{\mathbf{y}} f_t(\mathbf{x}, \mathbf{y}; \mathcal{B}_t)\right] = \nabla_{\mathbf{y}} f_{t,\boldsymbol{\rho}}(\mathbf{x}, \mathbf{y}), \quad \mathbb{E}_{(U_b^{\mathbf{s}}, \mathcal{B}_t)}\left[\hat{\nabla}_{\mathbf{x}} f_t(\mathbf{x}, \mathbf{y}; \mathcal{B}_t)\right] = \nabla_{\mathbf{x}} f_{t,\boldsymbol{\rho}}(\mathbf{x}, \mathbf{y}),$$

$$\mathbb{E}_{(U_b^{\mathbf{r}}, \bar{\mathcal{B}}_t)}\left[\hat{\nabla}_{\mathbf{y}} g_t(\mathbf{x}, \mathbf{y}; \bar{\mathcal{B}}_t)\right] = \nabla_{\mathbf{y}} g_{t,\boldsymbol{\rho}}(\mathbf{x}, \mathbf{y}), \quad \mathbb{E}_{(U_b^{\mathbf{s}}, \bar{\mathcal{B}}_t)}\left[\hat{\nabla}_{\mathbf{x}} g_t(\mathbf{x}, \mathbf{y}; \bar{\mathcal{B}}_t)\right] = \nabla_{\mathbf{x}} g_{t,\boldsymbol{\rho}}(\mathbf{x}, \mathbf{y}). \tag{137}$$

**Lemma D.7.** *Suppose that Assumptions* B3. *and* D1. *hold. Consider the sequence* $\{(\mathbf{x}_t, \mathbf{y}_t, \mathbf{v}_t)\}_{t=1}^{T}$ *generated by Algorithm* 2*, and define*

$$e_t^{g_\rho} := \nabla_{\mathbf{y}} g_{t,\boldsymbol{\rho}}(\mathbf{x}_t, \mathbf{y}_t) - \hat{\mathbf{d}}_t^{\mathbf{y}}. \tag{138}$$

*Then, we have*

$$
\begin{aligned}
\mathbb{E}\|e_{t+1}^{g_\rho}\|^2 &\leq (1 - \gamma_{t+1})^2 \mathbb{E}\|e_t^{g_\rho}\|^2 + 12(1 - \gamma_{t+1})^2 \mathbb{E}\|\nabla_{\mathbf{y}} g_{t-1}(\mathbf{x}_t, \mathbf{y}_t) - \nabla_{\mathbf{y}} g_t(\mathbf{x}_t, \mathbf{y}_t)\|^2 \\
&\quad + 9d_2^2 \ell_{g,1}^2 (1 - \gamma_{t+1})^2 \rho_{\mathbf{r}}^2 + 24 d_2 \ell_{g,1}^2 (1 - \gamma_{t+1})^2 \mathbb{E}\|\mathbf{x}_{t+1} - \mathbf{x}_t\|^2 \\
&\quad + 24 d_2 \ell_{g,1}^2 (1 - \gamma_{t+1})^2 \mathbb{E}\|\mathbf{y}_{t+1} - \mathbf{y}_t\|^2 + 2\frac{\hat{\sigma}_{g_{\mathbf{y}}}^2}{\bar{b}} \gamma_{t+1}^2.
\end{aligned} \tag{139}
$$

*Proof.* From the definition of $\hat{\mathbf{d}}_{t+1}^{\mathbf{y}}$ in Algorithm 2, we have

$$
\begin{aligned}
\hat{\mathbf{d}}_{t+1}^{\mathbf{y}} - \hat{\mathbf{d}}_t^{\mathbf{y}} &= -\gamma_{t+1} \hat{\mathbf{d}}_t^{\mathbf{y}} + \gamma_{t+1} \hat{\nabla}_{\mathbf{y}} g_{t+1}(\mathbf{x}_{t+1}, \mathbf{y}_{t+1}; \bar{\mathcal{B}}_{t+1}) \\
&\quad + (1 - \gamma_{t+1}) \left( \hat{\nabla}_{\mathbf{y}} g_{t+1}(\mathbf{x}_{t+1}, \mathbf{y}_{t+1}; \bar{\mathcal{B}}_{t+1}) - \hat{\nabla}_{\mathbf{y}} g_{t+1}(\mathbf{x}_t, \mathbf{y}_t; \bar{\mathcal{B}}_{t+1}) \right).
\end{aligned}
$$

Then, we have

$$
\begin{aligned}
&\mathbb{E}\|\nabla_{\mathbf{y}} g_{t+1,\boldsymbol{\rho}}(\mathbf{x}_{t+1}, \mathbf{y}_{t+1}) - \hat{\mathbf{d}}_{t+1}^{\mathbf{y}}\|^2 \\
&= \mathbb{E}\|\nabla_{\mathbf{y}} g_{t+1,\boldsymbol{\rho}}(\mathbf{x}_{t+1}, \mathbf{y}_{t+1}) - \hat{\mathbf{d}}_t^{\mathbf{y}} - (\hat{\mathbf{d}}_{t+1}^{\mathbf{y}} - \hat{\mathbf{d}}_t^{\mathbf{y}})\|^2 \\
&= \mathbb{E}\|\nabla_{\mathbf{y}} g_{t+1,\boldsymbol{\rho}}(\mathbf{x}_{t+1}, \mathbf{y}_{t+1}) - \hat{\mathbf{d}}_t^{\mathbf{y}} + \gamma_{t+1} \hat{\mathbf{d}}_t^{\mathbf{y}} - \gamma_{t+1} \hat{\nabla}_{\mathbf{y}} g_{t+1}(\mathbf{x}_{t+1}, \mathbf{y}_{t+1}; \bar{\mathcal{B}}_{t+1}) \\
&\quad - (1 - \gamma_{t+1}) \left( \hat{\nabla}_{\mathbf{y}} g_{t+1}(\mathbf{x}_{t+1}, \mathbf{y}_{t+1}; \bar{\mathcal{B}}_{t+1}) - \hat{\nabla}_{\mathbf{y}} g_{t+1}(\mathbf{x}_t, \mathbf{y}_t; \bar{\mathcal{B}}_{t+1}) \right) \|^2 \\
&= \mathbb{E}\|(1 - \gamma_{t+1})(\nabla_{\mathbf{y}} g_{t,\boldsymbol{\rho}}(\mathbf{x}_t, \mathbf{y}_t) - \hat{\mathbf{d}}_t^{\mathbf{y}}) \\
&\quad + \gamma_{t+1}(\nabla_{\mathbf{y}} g_{t+1,\boldsymbol{\rho}}(\mathbf{x}_{t+1}, \mathbf{y}_{t+1}) - \hat{\nabla}_{\mathbf{y}} g_{t+1}(\mathbf{x}_{t+1}, \mathbf{y}_{t+1}; \bar{\mathcal{B}}_{t+1})) \\
&\quad + (1 - \gamma_{t+1}) \left( \nabla_{\mathbf{y}} g_{t+1,\boldsymbol{\rho}}(\mathbf{x}_{t+1}, \mathbf{y}_{t+1}) - \nabla_{\mathbf{y}} g_{t,\boldsymbol{\rho}}(\mathbf{x}_t, \mathbf{y}_t) \right. \\
&\quad + \nabla_{\mathbf{y}} g_{t+1,\boldsymbol{\rho}}(\mathbf{x}_t, \mathbf{y}_t) - \nabla_{\mathbf{y}} g_{t+1,\boldsymbol{\rho}}(\mathbf{x}_t, \mathbf{y}_t) \\
&\quad \left. - \hat{\nabla}_{\mathbf{y}} g_{t+1}(\mathbf{x}_{t+1}, \mathbf{y}_{t+1}; \bar{\mathcal{B}}_{t+1}) + \hat{\nabla}_{\mathbf{y}} g_{t+1}(\mathbf{x}_t, \mathbf{y}_t; \bar{\mathcal{B}}_{t+1}) \right) \|^2.
\end{aligned}
$$

From (137), we have

$$
\begin{aligned}
\mathbb{E}\left[ \hat{\nabla}_{\mathbf{y}} g_{t+1}(\mathbf{x}_{t+1}, \mathbf{y}_{t+1}; \bar{\mathcal{B}}_{t+1}) \right] &= \nabla_{\mathbf{y}} g_{t+1,\boldsymbol{\rho}}(\mathbf{x}_{t+1}, \mathbf{y}_{t+1}), \\
\mathbb{E}\left[ \hat{\nabla}_{\mathbf{y}} g_{t+1}(\mathbf{x}_{t+1}, \mathbf{y}_{t+1}; \bar{\mathcal{B}}_{t+1}) - \hat{\nabla}_{\mathbf{y}} g_{t+1}(\mathbf{x}_t, \mathbf{y}_t; \bar{\mathcal{B}}_{t+1}) \right] & \\
= \nabla_{\mathbf{y}} g_{t+1,\boldsymbol{\rho}}(\mathbf{x}_{t+1}, \mathbf{y}_{t+1}) &- \nabla_{\mathbf{y}} g_{t+1,\boldsymbol{\rho}}(\mathbf{x}_t, \mathbf{y}_t),
\end{aligned}
$$

then, we have

$$
\begin{aligned}
&\mathbb{E}\|\nabla_{\mathbf{y}} g_{t+1,\boldsymbol{\rho}}(\mathbf{x}_{t+1}, \mathbf{y}_{t+1}) - \hat{\mathbf{d}}_{t+1}^{\mathbf{y}}\|^2 \\
&= (1 - \gamma_{t+1})^2 \mathbb{E}\|\nabla_{\mathbf{y}} g_{t,\boldsymbol{\rho}}(\mathbf{x}_t, \mathbf{y}_t) - \hat{\mathbf{d}}_t^{\mathbf{y}}\|^2 \\
&\quad + \mathbb{E}\|\gamma_{t+1}(\nabla_{\mathbf{y}} g_{t+1,\boldsymbol{\rho}}(\mathbf{x}_{t+1}, \mathbf{y}_{t+1}) - \hat{\nabla}_{\mathbf{y}} g_{t+1}(\mathbf{x}_{t+1}, \mathbf{y}_{t+1}; \bar{\mathcal{B}}_{t+1})) \\
&\quad + (1 - \gamma_{t+1}) \left( \nabla_{\mathbf{y}} g_{t+1,\boldsymbol{\rho}}(\mathbf{x}_{t+1}, \mathbf{y}_{t+1}) - \nabla_{\mathbf{y}} g_{t,\boldsymbol{\rho}}(\mathbf{x}_t, \mathbf{y}_t) + \nabla_{\mathbf{y}} g_{t+1,\boldsymbol{\rho}}(\mathbf{x}_t, \mathbf{y}_t) - \nabla_{\mathbf{y}} g_{t+1,\boldsymbol{\rho}}(\mathbf{x}_t, \mathbf{y}_t) \right. \\
&\quad \left. - \hat{\nabla}_{\mathbf{y}} g_{t+1}(\mathbf{x}_{t+1}, \mathbf{y}_{t+1}; \bar{\mathcal{B}}_{t+1}) + \hat{\nabla}_{\mathbf{y}} g_{t+1}(\mathbf{x}_t, \mathbf{y}_t; \bar{\mathcal{B}}_{t+1}) \right) \|^2 \\
&\leq (1 - \gamma_{t+1})^2 \mathbb{E}\|\nabla_{\mathbf{y}} g_{t,\boldsymbol{\rho}}(\mathbf{x}_t, \mathbf{y}_t) - \hat{\mathbf{d}}_t^{\mathbf{y}}\|^2 \\
&\quad + 2(1 - \gamma_{t+1})^2 \mathbb{E}\|\nabla_{\mathbf{y}} g_{t+1,\boldsymbol{\rho}}(\mathbf{x}_{t+1}, \mathbf{y}_{t+1}) - \nabla_{\mathbf{y}} g_{t,\boldsymbol{\rho}}(\mathbf{x}_t, \mathbf{y}_t) + \nabla_{\mathbf{y}} g_{t+1,\boldsymbol{\rho}}(\mathbf{x}_t, \mathbf{y}_t) \\
&\quad - \nabla_{\mathbf{y}} g_{t+1,\boldsymbol{\rho}}(\mathbf{x}_t, \mathbf{y}_t) - \hat{\nabla}_{\mathbf{y}} g_{t+1}(\mathbf{x}_{t+1}, \mathbf{y}_{t+1}; \bar{\mathcal{B}}_{t+1}) + \hat{\nabla}_{\mathbf{y}} g_{t+1}(\mathbf{x}_t, \mathbf{y}_t; \bar{\mathcal{B}}_{t+1})\|^2 \\
&\quad + 2\gamma_{t+1}^2 \mathbb{E}\|\nabla_{\mathbf{y}} g_{t+1,\boldsymbol{\rho}}(\mathbf{x}_{t+1}, \mathbf{y}_{t+1}) - \hat{\nabla}_{\mathbf{y}} g_{t+1}(\mathbf{x}_{t+1}, \mathbf{y}_{t+1}; \bar{\mathcal{B}}_{t+1})\|^2,
\end{aligned}
$$

where the second inequality holds by Cauchy-Schwarz inequality.

Then, from $\mathbb{E}\|a - \mathbb{E}[a]\|^2 = \mathbb{E}\|a\|^2 - \|\mathbb{E}[a]\|^2$ and Assumption D1., we have

$$
\begin{aligned}
&\mathbb{E}\|\nabla_{\mathbf{y}} g_{t+1,\boldsymbol{\rho}}(\mathbf{x}_{t+1}, \mathbf{y}_{t+1}) - \hat{\mathbf{d}}_{t+1}^{\mathbf{y}}\|^2 \\
&\leq (1 - \gamma_{t+1})^2 \mathbb{E}\|\nabla_{\mathbf{y}} g_{t,\boldsymbol{\rho}}(\mathbf{x}_t, \mathbf{y}_t) - \hat{\mathbf{d}}_t^{\mathbf{y}}\|^2 \\
&\quad + 4(1 - \gamma_{t+1})^2 \mathbb{E}\|\nabla_{\mathbf{y}} g_{t+1,\boldsymbol{\rho}}(\mathbf{x}_t, \mathbf{y}_t) - \nabla_{\mathbf{y}} g_{t,\boldsymbol{\rho}}(\mathbf{x}_t, \mathbf{y}_t)\|^2 \\
&\quad + 4(1 - \gamma_{t+1})^2 \mathbb{E}\|\hat{\nabla}_{\mathbf{y}} g_{t+1}(\mathbf{x}_{t+1}, \mathbf{y}_{t+1}; \bar{\mathcal{B}}_{t+1}) - \hat{\nabla}_{\mathbf{y}} g_{t+1}(\mathbf{x}_t, \mathbf{y}_t; \bar{\mathcal{B}}_{t+1})\|^2 + 2\gamma_{t+1}^2 \frac{\hat{\sigma}_{g_{\mathbf{y}}}^2}{b} \\
&\leq (1 - \gamma_{t+1})^2 \mathbb{E}\|\nabla_{\mathbf{y}} g_{t,\boldsymbol{\rho}}(\mathbf{x}_t, \mathbf{y}_t) - \hat{\mathbf{d}}_t^{\mathbf{y}}\|^2 \\
&\quad + 4(1 - \gamma_{t+1})^2 \mathbb{E}\|\nabla_{\mathbf{y}} g_{t+1,\boldsymbol{\rho}}(\mathbf{x}_t, \mathbf{y}_t) - \nabla_{\mathbf{y}} g_{t,\boldsymbol{\rho}}(\mathbf{x}_t, \mathbf{y}_t)\|^2 \\
&\quad + 12(1 - \gamma_{t+1})^2 d_2 \ell_{g,1}^2 \mathbb{E}\|(\mathbf{x}_{t+1}, \mathbf{y}_{t+1}) - (\mathbf{x}_t, \mathbf{y}_t)\|^2 \\
&\quad + 3(1 - \gamma_{t+1})^2 \ell_{g,1}^2 d_2^2 \rho_{\mathbf{r}}^2 + 2\gamma_{t+1}^2 \frac{\hat{\sigma}_{g_{\mathbf{y}}}^2}{b},
\end{aligned}
$$

where the second inequality follows from Young's inequality and Lemma D.5.

From Eq. (130), we have

$$
\begin{aligned}
&\mathbb{E}\|\nabla_{\mathbf{y}} g_{t+1,\boldsymbol{\rho}}(\mathbf{x}_t, \mathbf{y}_t) - \nabla_{\mathbf{y}} g_{t,\boldsymbol{\rho}}(\mathbf{x}_t, \mathbf{y}_t)\|^2 \\
&\leq 3\mathbb{E}\|\nabla_{\mathbf{y}} g_{t+1,\boldsymbol{\rho}}(\mathbf{x}_t, \mathbf{y}_t) - \nabla_{\mathbf{y}} g_{t+1}(\mathbf{x}_t, \mathbf{y}_t)\|^2 \\
&\quad + 3\mathbb{E}\|\nabla_{\mathbf{y}} g_{t+1}(\mathbf{x}_t, \mathbf{y}_t) - \nabla_{\mathbf{y}} g_t(\mathbf{x}_t, \mathbf{y}_t)\|^2 \\
&\quad + 3\mathbb{E}\|\nabla_{\mathbf{y}} g_t(\mathbf{x}_t, \mathbf{y}_t) - \nabla_{\mathbf{y}} g_{t,\boldsymbol{\rho}}(\mathbf{x}_t, \mathbf{y}_t)\|^2 \\
&\leq 3\mathbb{E}\|\nabla_{\mathbf{y}} g_{t+1}(\mathbf{x}_t, \mathbf{y}_t) - \nabla_{\mathbf{y}} g_t(\mathbf{x}_t, \mathbf{y}_t)\|^2 + \frac{3\rho_{\mathbf{r}}^2 d_2^2 \ell_{g,1}^2}{2}.
\end{aligned}
$$

Finally, we get

$$
\begin{aligned}
&\mathbb{E}\|\nabla_{\mathbf{y}} g_{t+1,\boldsymbol{\rho}}(\mathbf{x}_{t+1}, \mathbf{y}_{t+1}) - \hat{\mathbf{d}}_{t+1}^{\mathbf{y}}\|^2 \leq (1 - \gamma_{t+1})^2 \mathbb{E}\|\nabla_{\mathbf{y}} g_{t,\boldsymbol{\rho}}(\mathbf{x}_t, \mathbf{y}_t) - \hat{\mathbf{d}}_t^{\mathbf{y}}\|^2 \\
&+ 12(1 - \gamma_{t+1})^2 \mathbb{E}\|\nabla_{\mathbf{y}} g_{t+1}(\mathbf{x}_t, \mathbf{y}_t) - \nabla_{\mathbf{y}} g_t(\mathbf{x}_t, \mathbf{y}_t)\|^2 + 6(1 - \gamma_{t+1})^2 \rho_{\mathbf{r}}^2 d_2^2 \ell_{g,1}^2 \\
&+ 12(1 - \gamma_{t+1})^2 d_2 \ell_{g,1}^2 \mathbb{E}\|(\mathbf{x}_{t+1}, \mathbf{y}_{t+1}) - (\mathbf{x}_t, \mathbf{y}_t)\|^2 + 3(1 - \gamma_{t+1})^2 \ell_{g,1}^2 d_2^2 \rho_{\mathbf{r}}^2 + 2\gamma_{t+1}^2 \frac{\hat{\sigma}_{g_{\mathbf{y}}}^2}{b}.
\end{aligned}
$$

$\square$

**Lemma D.8.** *Suppose Assumptions 2.2 and B3. hold. Then, for the sequence $\{(\mathbf{x}_t, \mathbf{y}_t)\}_{t=1}^T$ generated by Algorithm 2, we have*

$$
\begin{aligned}
\mathbb{E}\left[\|\mathbf{y}_{t+1} - \hat{\mathbf{y}}_t^*(\mathbf{x}_t)\|^2\right] &\leq (1 + a)\left(1 - 2\beta_t \frac{\mu_g \ell_{g,1}}{\mu_g + \ell_{g,1}}\right) \mathbb{E}\left[\|\mathbf{y}_t - \hat{\mathbf{y}}_t^*(\mathbf{x}_t)\|^2\right] \\
&\quad + \left(-(1 + a)\left(\frac{2\beta_t}{\mu_g + \ell_{g,1}} - \beta_t^2\right)\right) \mathbb{E}\left[\|\nabla_{\mathbf{y}} g_{t,\boldsymbol{\rho}}(\mathbf{x}_t, \mathbf{y}_t)\|^2\right] \\
&\quad + (1 + \frac{1}{a})\beta_t^2 \mathbb{E}\left[\|e_t^{g_\rho}\|^2\right],
\end{aligned}
$$

*where $a > 0$ is a constant, $e_t^{g_\rho}$ is defined in (138), and $\hat{\mathbf{y}}_t^*(\mathbf{x}_t)$ is defined in (19).*

*Proof.* From Lemma B.4, we have

$$
\begin{aligned}
\mathbb{E}\left[\|\mathbf{y}_{t+1} - \hat{\mathbf{y}}_t^*(\mathbf{x}_t)\|^2\right] &= \mathbb{E}\left[\|\mathbf{y}_t - \beta_t \hat{\mathbf{d}}_t^{\mathbf{y}} - \hat{\mathbf{y}}_t^*(\mathbf{x}_t)\|^2\right] \\
&\leq (1 + a)\mathbb{E}\left[\|\mathbf{y}_t - \beta_t \nabla_{\mathbf{y}} g_{t,\boldsymbol{\rho}}(\mathbf{x}_t, \mathbf{y}_t) - \hat{\mathbf{y}}_t^*(\mathbf{x}_t)\|^2\right] \\
&\quad + (1 + \frac{1}{a})\beta_t^2 \mathbb{E}\left[\|\hat{\mathbf{d}}_t^{\mathbf{y}} - \nabla_{\mathbf{y}} g_{t,\boldsymbol{\rho}}(\mathbf{x}_t, \mathbf{y}_t)\|^2\right]. \quad (140)
\end{aligned}
$$

Next, we will separately bound the first term on the RHS of the above inequality.
We have

$$\mathbb{E}\left[\|\mathbf{y}_t - \beta_t \nabla_{\mathbf{y}} g_{t,\boldsymbol{\rho}}(\mathbf{x}_t, \mathbf{y}_t) - \hat{\mathbf{y}}_t^*(\mathbf{x}_t)\|^2\right] = \mathbb{E}\left[\|\mathbf{y}_t - \hat{\mathbf{y}}_t^*(\mathbf{x}_t)\|^2\right] + \beta_t^2 \mathbb{E}\left[\|\nabla_{\mathbf{y}} g_{t,\boldsymbol{\rho}}(\mathbf{x}_t, \mathbf{y}_t)\|^2\right]$$
$$- 2\beta_t \mathbb{E}\left[\langle \nabla_{\mathbf{y}} g_{t,\boldsymbol{\rho}}(\mathbf{x}_t, \mathbf{y}_t), \mathbf{y}_t - \hat{\mathbf{y}}_t^*(\mathbf{x}_t)\rangle\right]$$
$$\leq \left(1 - 2\beta_t \frac{\mu_g \ell_{g,1}}{\mu_g + \ell_{g,1}}\right) \mathbb{E}\left[\|\mathbf{y}_t - \hat{\mathbf{y}}_t^*(\mathbf{x}_t)\|^2\right]$$
$$- \left(\frac{2\beta_t}{\mu_g + \ell_{g,1}} - \beta_t^2\right) \mathbb{E}\left[\|\nabla_{\mathbf{y}} g_{t,\boldsymbol{\rho}}(\mathbf{x}_t, \mathbf{y}_t)\|^2\right], \quad (141)$$

where the inequality results from the strong convexity of $g_{t,\boldsymbol{\rho}}$ by Assumption 2.2, which implies

$$\langle \nabla_{\mathbf{y}} g_{t,\boldsymbol{\rho}}(\mathbf{x}_t, \mathbf{y}_t), \mathbf{y}_t - \hat{\mathbf{y}}_t^*(\mathbf{x}_t)\rangle \geq \frac{\mu_g \ell_{g,1}}{\mu_g + \ell_{g,1}}\|\mathbf{y}_t - \hat{\mathbf{y}}_t^*(\mathbf{x}_t)\|^2 + \frac{1}{\mu_g + \ell_{g,1}}\|\nabla_{\mathbf{y}} g_{t,\boldsymbol{\rho}}(\mathbf{x}_t, \mathbf{y}_t)\|^2.$$

Substituting (141) into (140), gives the desired result.

$$\square$$

For notational brevity in the analysis, we define

$$\hat{\theta}_t^{\mathbf{y}} := \|\mathbf{y}_t - \hat{\mathbf{y}}_t^*(\mathbf{x}_t)\|^2, \quad \hat{\theta}_t^{\mathbf{v}} := \|\mathbf{v}_t - \hat{\mathbf{v}}_t^*(\mathbf{x}_t)\|^2, \tag{142}$$

where $\hat{\mathbf{y}}_t^*(\mathbf{x})$ and $\hat{\mathbf{v}}_t^*(\mathbf{x})$ are defined in (19) and (20), respectively.

**Lemma D.9.** *Suppose Assumptions 2.2 and B3. hold. Let $\hat{\theta}_t^{\mathbf{y}}$ be defined in (142). Then, for the sequence $\{(\mathbf{x}_t, \mathbf{y}_t)\}_{t=1}^T$ generated by Algorithm 2 guarantees the following bound:*

$$\sum_{t=1}^T \left(\mathbb{E}[\hat{\theta}_{t+1}^{\mathbf{y}}] - \mathbb{E}[\hat{\theta}_t^{\mathbf{y}}]\right)$$
$$\leq \left(-\frac{L_{\mu_g}}{2}\sum_{t=1}^T \mathbb{E}[\hat{\theta}_t^{\mathbf{y}}] + \frac{2}{L_{\mu_g}}\sum_{t=1}^T \mathbb{E}\left[\|e_t^{g_\rho}\|^2\right]\right)\beta_t + \frac{4L_{\mathbf{y}}^2}{L_{\mu_g}}\sum_{t=1}^T \mathbb{E}\|\mathbf{x}_t - \mathbf{x}_{t+1}\|^2 \frac{1}{\beta_t}$$
$$+ \sum_{t=1}^T \left(\frac{24\ell_{g,1}}{L_{\mu_g}\mu_g}(\rho_{\mathbf{s}}^2 + \rho_{\mathbf{r}}^2) + \frac{12}{L_{\mu_g}}\sup_{\mathbf{x}\in\mathcal{X}}\|\mathbf{y}_{t-1}^*(\mathbf{x}) - \mathbf{y}_t^*(\mathbf{x})\|^2\right)\frac{1}{\beta_t}$$
$$+ \sum_{t=1}^T \left(-\frac{2\beta_t}{\mu_g + \ell_{g,1}} + \beta_t^2\right)\mathbb{E}\left[\|\nabla_{\mathbf{y}} g_{t,\boldsymbol{\rho}}(\mathbf{x}_t, \mathbf{y}_t)\|^2\right], \tag{143}$$

*where $L_{\mathbf{y}} = \frac{\ell_{g,1}}{\mu_g}$ is defined as in (44) and $L_{\mu_g} = \frac{\mu_g \ell_{g,1}}{\mu_g + \ell_{g,1}}$.*

*Proof.* From Lemma B.4, we have for any $c > 0$

$$\mathbb{E}\left[\|\mathbf{y}_{t+1} - \hat{\mathbf{y}}_{t+1}^*(\mathbf{x}_{t+1})\|^2\right] = \mathbb{E}\left[\|\mathbf{y}_{t+1} - \hat{\mathbf{y}}_t^*(\mathbf{x}_t) + \hat{\mathbf{y}}_t^*(\mathbf{x}_t) - \hat{\mathbf{y}}_{t+1}^*(\mathbf{x}_{t+1})\|^2\right]$$
$$\leq (1 + c)\mathbb{E}\left[\|\mathbf{y}_{t+1} - \hat{\mathbf{y}}_t^*(\mathbf{x}_t)\|^2\right]$$
$$+ \left(1 + \frac{1}{c}\right)\mathbb{E}\left[\|\hat{\mathbf{y}}_{t+1}^*(\mathbf{x}_{t+1}) - \hat{\mathbf{y}}_t^*(\mathbf{x}_t)\|^2\right]. \tag{144}$$

From Lemma D.8, we have for any $a > 0$

$$\mathbb{E}\left[\|\mathbf{y}_{t+1} - \hat{\mathbf{y}}_t^*(\mathbf{x}_t)\|^2\right] \leq (1 + a)\left(1 - 2\beta_t \frac{\mu_g \ell_{g,1}}{\mu_g + \ell_{g,1}}\right)\mathbb{E}\left[\|\mathbf{y}_t - \hat{\mathbf{y}}_t^*(\mathbf{x}_t)\|^2\right]$$
$$+ \left(-(1 + a)\left(\frac{2\beta_t}{\mu_g + \ell_{g,1}} - \beta_t^2\right)\right)\mathbb{E}\left[\|\nabla_{\mathbf{y}} g_{t,\boldsymbol{\rho}}(\mathbf{x}_t, \mathbf{y}_t)\|^2\right]$$
$$+ \left(1 + \frac{1}{a}\right)\beta_t^2 \mathbb{E}\left[\|e_t^{g_\rho}\|^2\right]. \tag{145}$$

Substituting (145) into (144), we get

$$\mathbb{E}\left[\|\mathbf{y}_{t+1} - \hat{\mathbf{y}}_{t+1}^*(\mathbf{x}_{t+1})\|^2\right]$$
$$\leq (1+c)(1+a)\left(1 - 2\beta_t \frac{\mu_g \ell_{g,1}}{\mu_g + \ell_{g,1}}\right)\mathbb{E}\left[\|\mathbf{y}_t - \hat{\mathbf{y}}_t^*(\mathbf{x}_t)\|^2\right]$$
$$+ \left(-(1+c)(1+a)\left(\frac{2\beta_t}{\mu_g + \ell_{g,1}} - \beta_t^2\right)\right)\mathbb{E}\left[\|\nabla_{\mathbf{y}} g_{t,\boldsymbol{\rho}}(\mathbf{x}_t, \mathbf{y}_t)\|^2\right]$$
$$+ (1+c)(1+\frac{1}{a})\beta_t^2 \mathbb{E}\left[\|e_t^{g_\rho}\|^2\right] + \left(1 + \frac{1}{c}\right)\mathbb{E}\left[\|\hat{\mathbf{y}}_{t+1}^*(\mathbf{x}_{t+1}) - \hat{\mathbf{y}}_t^*(\mathbf{x}_t)\|^2\right]. \tag{146}$$

Choose $c = \frac{\beta_t L_{\mu_g}/2}{1 - \beta_t L_{\mu_g}}$ and $a = \frac{\beta_t L_{\mu_g}}{1 - 2\beta_t L_{\mu_g}}$. Then, the following equations and inequalities are satisfied.

$$(1+c)(1+a)\left(1 - 2\beta_t L_{\mu_g}\right) = 1 - \frac{\beta_t L_{\mu_g}}{2},$$
$$(1+a)\left(1 - 2\beta_t L_{\mu_g}\right) = 1 - \beta_t L_{\mu_g},$$
$$(1+c)\left(1 - \beta_t L_{\mu_g}\right) = 1 - \frac{\beta_t L_{\mu_g}}{2}, \tag{147}$$
$$1 + \frac{1}{a} \leq \frac{1}{\beta_t L_{\mu_g}}, \quad 1 + \frac{1}{c} \leq \frac{2}{\beta_t L_{\mu_g}},$$

where $L_{\mu_g} = \frac{\mu_g \ell_{g,1}}{\mu_g + \ell_{g,1}}$. Based on (146) and (147), we get

$$\mathbb{E}\left[\|\mathbf{y}_{t+1} - \hat{\mathbf{y}}_{t+1}^*(\mathbf{x}_{t+1})\|^2\right] - \mathbb{E}\left[\|\mathbf{y}_t - \hat{\mathbf{y}}_t^*(\mathbf{x}_t)\|^2\right]$$
$$\leq -\frac{\beta_t L_{\mu_g}}{2}\mathbb{E}\left[\|\mathbf{y}_t - \hat{\mathbf{y}}_t^*(\mathbf{x}_t)\|^2\right] + \left(-\left(\frac{2\beta_t}{\mu_g + \ell_{g,1}} - \beta_t^2\right)\right)\mathbb{E}\left[\|\nabla_{\mathbf{y}} g_{t,\boldsymbol{\rho}}(\mathbf{x}_t, \mathbf{y}_t)\|^2\right]$$
$$+ \frac{2}{\beta_t L_{\mu_g}}\beta_t^2 \mathbb{E}\left[\|e_t^{g_\rho}\|^2\right] + \frac{2}{\beta_t L_{\mu_g}}\mathbb{E}\left[\|\hat{\mathbf{y}}_{t+1}^*(\mathbf{x}_{t+1}) - \hat{\mathbf{y}}_t^*(\mathbf{x}_t)\|^2\right]. \tag{148}$$

Next, we upper-bound the last term of the above inequality.

$$\mathbb{E}\left[\|\hat{\mathbf{y}}_{t+1}^*(\mathbf{x}_{t+1}) - \hat{\mathbf{y}}_t^*(\mathbf{x}_t)\|^2\right]$$
$$\leq 2\left(\mathbb{E}\left[\|\hat{\mathbf{y}}_{t+1}^*(\mathbf{x}_{t+1}) - \hat{\mathbf{y}}_{t+1}^*(\mathbf{x}_t)\|^2\right] + \mathbb{E}\left[\|\hat{\mathbf{y}}_{t+1}^*(\mathbf{x}_t) - \hat{\mathbf{y}}_t^*(\mathbf{x}_t)\|^2\right]\right)$$
$$\leq 2\left(L_{\mathbf{y}}^2 \mathbb{E}\left[\|\mathbf{x}_t - \mathbf{x}_{t+1}\|^2 + \|\hat{\mathbf{y}}_{t+1}^*(\mathbf{x}_t) - \hat{\mathbf{y}}_t^*(\mathbf{x}_t)\|^2\right]\right), \tag{149}$$

where the second inequality is by Lemma D.2.
Moreover, from Lemma D.6, we get

$$\mathbb{E}\left[\|\hat{\mathbf{y}}_{t+1}^*(\mathbf{x}_t) - \hat{\mathbf{y}}_t^*(\mathbf{x}_t)\|^2\right] \leq 3\mathbb{E}\left[\|\hat{\mathbf{y}}_{t+1}^*(\mathbf{x}_t) - \mathbf{y}_{t+1}^*(\mathbf{x}_t)\|^2\right]$$
$$+ 3\mathbb{E}\left[\|\mathbf{y}_{t+1}^*(\mathbf{x}_t) - \mathbf{y}_t^*(\mathbf{x}_t)\|^2\right] + 3\mathbb{E}\left[\|\mathbf{y}_t^*(\mathbf{x}_t) - \hat{\mathbf{y}}_t^*(\mathbf{x}_t)\|^2\right]$$
$$\leq 3\mathbb{E}\left[\|\mathbf{y}_{t+1}^*(\mathbf{x}_t) - \mathbf{y}_t^*(\mathbf{x}_t)\|^2\right] + \frac{6\ell_{g,1}(\rho_{\mathbf{s}}^2 + \rho_{\mathbf{r}}^2)}{\mu_g}. \tag{150}$$

Combining (149) and (150) yields

$$\mathbb{E}\left[\|\hat{\mathbf{y}}_{t+1}^*(\mathbf{x}_{t+1}) - \hat{\mathbf{y}}_t^*(\mathbf{x}_t)\|^2\right]$$
$$\leq 2\left(L_{\mathbf{y}}^2 \mathbb{E}\left[\|\mathbf{x}_t - \mathbf{x}_{t+1}\|^2\right] + 3\mathbb{E}\left[\|\mathbf{y}_{t+1}^*(\mathbf{x}_t) - \mathbf{y}_t^*(\mathbf{x}_t)\|^2\right] + \frac{6\ell_{g,1}(\rho_{\mathbf{s}}^2 + \rho_{\mathbf{r}}^2)}{\mu_g}\right). \tag{151}$$

Substituting (151) into (148) and summing over $t \in [T]$, give the desired result.

$$\square$$

### D.4 Bounds on the Zeroth-Order System Solution

**Lemma D.10.** *Suppose Assumptions* B2. *and* B3. *hold. Let*

$$\vartheta := \mathbb{E}\|\hat{\nabla}_{\mathbf{y}} f_{t+1}(\mathbf{z}_{t+1}; \mathcal{B}_{t+1}) + \hat{\nabla}_{\mathbf{y}}^2 g_{t+1}(\mathbf{z}_{t+1}; \bar{\mathcal{B}}_{t+1}) - \hat{\nabla}_{\mathbf{y}} f_{t+1}(\mathbf{z}_t; \mathcal{B}_{t+1}) - \hat{\nabla}_{\mathbf{y}}^2 g_{t+1}(\mathbf{z}_t; \bar{\mathcal{B}}_{t+1})\|^2,$$

where $\hat{\nabla}_{\mathbf{y}}f_t$ and $\hat{\nabla}_{\mathbf{y}}^2 g_t$ are defined in (25a) and (26a), respectively. Then, for the sequence $\{(\mathbf{x}_t, \mathbf{y}_t, \mathbf{v}_t)\}_{t=1}^T$ generated by Algorithm 2, we have

$$\vartheta \leq (12\ell_{f,1}^2 + \frac{9\ell_{g,1}^2}{2\rho_{\mathbf{v}}^2})d_2\mathbb{E}\|\mathbf{x}_{t+1} - \mathbf{x}_t\|^2 + (12\ell_{f,1}^2 + \frac{9\ell_{g,1}^2}{2\rho_{\mathbf{v}}^2})d_2\mathbb{E}\|\mathbf{y}_{t+1} - \mathbf{y}_t\|^2$$

$$+ \frac{9}{2}d_2\ell_{g,1}^2\mathbb{E}\|\mathbf{v}_{t+1} - \mathbf{v}_t\|^2 + (3\ell_{f,1}^2 + \frac{3\ell_{g,1}^2}{4\rho_{\mathbf{v}}^2})d_2^2\rho_{\mathbf{r}}^2.$$

*Proof.* From Lemma D.5, we have

$$\|\hat{\nabla}_{\mathbf{y}}f_{t+1}(\mathbf{z}_{t+1}; \mathcal{B}_{t+1}) - \hat{\nabla}_{\mathbf{y}}f_{t+1}(\mathbf{z}_t; \mathcal{B}_{t+1})\|^2$$

$$\leq 3d_2\ell_{f,1}^2\|\mathbf{z}_{t+1} - \mathbf{z}_t\|^2 + \frac{3}{2}\ell_{f,1}^2 d_2^2\rho_{\mathbf{r}}^2$$

$$\leq 6d_2\ell_{f,1}^2\|\mathbf{x}_{t+1} - \mathbf{x}_t\|^2 + 6d_2\ell_{f,1}^2\|\mathbf{y}_{t+1} - \mathbf{y}_t\|^2 + \frac{3}{2}\ell_{f,1}^2 d_2^2\rho_{\mathbf{r}}^2. \tag{152}$$

Moreover, from (26a), we have

$$\|\hat{\nabla}_{\mathbf{y}}^2 g_{t+1}(\mathbf{z}_{t+1}; \bar{\mathcal{B}}_{t+1}) - \hat{\nabla}_{\mathbf{y}}^2 g_{t+1}(\mathbf{z}_t; \bar{\mathcal{B}}_{t+1})\|^2$$

$$= \frac{1}{4\rho_{\mathbf{v}}^2}\|\hat{\nabla}_{\mathbf{y}}g_{t+1}(\mathbf{x}_{t+1}, \mathbf{y}_{t+1} + \rho_{\mathbf{v}}\mathbf{v}_{t+1}; \bar{\mathcal{B}}_{t+1}) - \hat{\nabla}_{\mathbf{y}}g_{t+1}(\mathbf{x}_t, \mathbf{y}_t - \rho_{\mathbf{v}}\mathbf{v}_t; \bar{\mathcal{B}}_{t+1})\|^2$$

$$\leq \frac{3}{4\rho_{\mathbf{v}}^2}d_2\ell_{g,1}^2\|(\mathbf{x}_{t+1}, \mathbf{y}_{t+1} + \rho_{\mathbf{v}}\mathbf{v}_{t+1}) - (\mathbf{x}_t, \mathbf{y}_t - \rho_{\mathbf{v}}\mathbf{v}_t)\|^2 + \frac{3}{8\rho_{\mathbf{v}}^2}\ell_{g,1}^2 d_2^2\rho_{\mathbf{r}}^2$$

$$\leq \frac{9}{4\rho_{\mathbf{v}}^2}d_2\ell_{g,1}^2\|\mathbf{x}_{t+1} - \mathbf{x}_t\|^2 + \frac{9}{4\rho_{\mathbf{v}}^2}d_2\ell_{g,1}^2\|\mathbf{y}_{t+1} - \mathbf{y}_t\|^2$$

$$+ \frac{9}{4}d_2\ell_{g,1}^2\|\mathbf{v}_{t+1} - \mathbf{v}_t\|^2 + \frac{3}{8\rho_{\mathbf{v}}^2}\ell_{g,1}^2 d_2^2\rho_{\mathbf{r}}^2, \tag{153}$$

where the first inequality follows from Lemma D.5.

From $\|a + b\|^2 \leq 2\left(\|a\|^2 + \|b\|^2\right)$, we get

$$\vartheta \leq 2\mathbb{E}\|\hat{\nabla}_{\mathbf{y}}^2 g_{t+1}(\mathbf{z}_{t+1}; \bar{\mathcal{B}}_{t+1}) - \hat{\nabla}_{\mathbf{y}}^2 g_{t+1}(\mathbf{z}_t; \bar{\mathcal{B}}_{t+1})\|^2$$

$$+ 2\mathbb{E}\|\hat{\nabla}_{\mathbf{y}}f_{t+1}(\mathbf{z}_{t+1}; \mathcal{B}_{t+1}) - \hat{\nabla}_{\mathbf{y}}f_{t+1}(\mathbf{z}_t; \mathcal{B}_{t+1})\|^2$$

$$\leq (12\ell_{f,1}^2 + \frac{9\ell_{g,1}^2}{2\rho_{\mathbf{v}}^2})d_2\mathbb{E}\|\mathbf{x}_{t+1} - \mathbf{x}_t\|^2 + (12\ell_{f,1}^2 + \frac{9\ell_{g,1}^2}{2\rho_{\mathbf{v}}^2})d_2\mathbb{E}\|\mathbf{y}_{t+1} - \mathbf{y}_t\|^2$$

$$+ \frac{9}{2}d_2\ell_{g,1}^2\mathbb{E}\|\mathbf{v}_{t+1} - \mathbf{v}_t\|^2 + (3\ell_{f,1}^2 + \frac{3\ell_{g,1}^2}{4\rho_{\mathbf{v}}^2})d_2^2\rho_{\mathbf{r}}^2,$$

where the second inequality follows from (152) and (153).

$\square$

**Lemma D.11.** *Suppose Assumptions* B2., B3., D1., *and* D3. *hold. Consider the sequence* $\{(\mathbf{x}_t, \mathbf{y}_t, \mathbf{v}_t)\}_{t=1}^T$ *generated by Algorithm 2, and define*

$$e_{t+1}^M := \nabla_{\mathbf{y}}f_{t+1,\rho}(\mathbf{x}_{t+1}, \mathbf{y}_{t+1}) + \tilde{\nabla}_{\mathbf{y}}^2 g_{t+1}(\mathbf{x}_{t+1}, \mathbf{y}_{t+1}) - \hat{\mathbf{d}}_{t+1}^{\mathbf{v}}, \quad \text{where} \tag{154}$$

$$\tilde{\nabla}_{\mathbf{y}}^2 g_{t+1}(\mathbf{x}_{t+1}, \mathbf{y}_{t+1}) = \frac{1}{2\rho_{\mathbf{v}}}(\nabla_{\mathbf{y}}g_{t+1,\rho}(\mathbf{x}_{t+1}, \mathbf{y}_{t+1} + \rho_{\mathbf{v}}\mathbf{v}_{t+1})$$

$$- \nabla_{\mathbf{y}}g_{t+1,\rho}(\mathbf{x}_{t+1}, \mathbf{y}_{t+1} - \rho_{\mathbf{v}}\mathbf{v}_{t+1})). \tag{155}$$

*Then, we have*

$$\mathbb{E}\|e_{t+1}^M\|^2 \leq (1-\lambda_{t+1})^2\mathbb{E}\|e_t^M\|^2 + 36\mathbb{E}\left\|\nabla_{\mathbf{y}}f_{t+1}(\mathbf{x}_t,\mathbf{y}_t) - \nabla_{\mathbf{y}}f_t(\mathbf{x}_t,\mathbf{y}_t)\right\|^2$$

$$+ \left(18d_2^2\ell_{f,1}^2 + 6(3\ell_{f,1}^2 + \frac{3\ell_{g,1}^2}{4\rho_{\mathbf{v}}^2})d_2^2\right)\rho_{\mathbf{r}}^2 + 18d_2^2\ell_{g,1}^2\frac{\rho_{\mathbf{r}}^2}{\rho_{\mathbf{v}}^2}$$

$$+ \frac{18}{\rho_{\mathbf{v}}^2}\mathbb{E}\|\nabla_{\mathbf{y}}g_{t+1}(\mathbf{x}_t,\mathbf{y}_t + \rho_{\mathbf{v}}\mathbf{v}_t) - \nabla_{\mathbf{y}}g_t(\mathbf{x}_t,\mathbf{y}_t + \rho_{\mathbf{v}}\mathbf{v}_t)\|^2$$

$$+ \frac{18}{\rho_{\mathbf{v}}^2}\mathbb{E}\|\nabla_{\mathbf{y}}g_{t+1}(\mathbf{x}_t,\mathbf{y}_t - \rho_{\mathbf{v}}\mathbf{v}_t) - \nabla_{\mathbf{y}}g_t(\mathbf{x}_t,\mathbf{y}_t - \rho_{\mathbf{v}}\mathbf{v}_t)\|^2$$

$$+ 6(12\ell_{f,1}^2 + \frac{9\ell_{g,1}^2}{2\rho_{\mathbf{v}}^2})d_2\mathbb{E}\|\mathbf{x}_{t+1} - \mathbf{x}_t\|^2 + 6(12\ell_{f,1}^2 + \frac{9\ell_{g,1}^2}{2\rho_{\mathbf{v}}^2})d_2\mathbb{E}\|\mathbf{y}_{t+1} - \mathbf{y}_t\|^2$$

$$+ 27d_2\ell_{g,1}^2\mathbb{E}\|\mathbf{v}_{t+1} - \mathbf{v}_t\|^2 + 3(\frac{\hat{\sigma}_{g_{\mathbf{y}}}^2}{b\rho_{\mathbf{v}}^2} + \frac{\hat{\sigma}_{f_{\mathbf{y}}}^2}{b})\lambda_{t+1}^2. \tag{156}$$

*Proof.* According to the definition of $\hat{\mathbf{d}}_t^{\mathbf{y}}$ in Algorithm 2, we have

$$\hat{\mathbf{d}}_{t+1}^{\mathbf{y}} - \hat{\mathbf{d}}_t^{\mathbf{y}} = -\lambda_{t+1}\hat{\mathbf{d}}_t^{\mathbf{y}} + \lambda_{t+1}(\hat{\nabla}_{\mathbf{y}}f_{t+1}(\mathbf{z}_{t+1};\mathcal{B}_{t+1}) + \hat{\nabla}_{\mathbf{y}}^2 g_{t+1}(\mathbf{z}_{t+1};\bar{\mathcal{B}}_{t+1}))$$

$$+ (1-\lambda_{t+1})\left(\hat{\nabla}_{\mathbf{y}}f_{t+1}(\mathbf{z}_{t+1};\mathcal{B}_{t+1}) + \hat{\nabla}_{\mathbf{y}}^2 g_{t+1}(\mathbf{z}_{t+1};\bar{\mathcal{B}}_{t+1})\right.$$

$$\left. -\hat{\nabla}_{\mathbf{y}}f_{t+1}(\mathbf{z}_t;\mathcal{B}_{t+1}) - \hat{\nabla}_{\mathbf{y}}^2 g_{t+1}(\mathbf{z}_t;\bar{\mathcal{B}}_{t+1})\right).$$

Then we have

$$\mathbb{E}\|\nabla_{\mathbf{y}}f_{t+1,\boldsymbol{\rho}}(\mathbf{z}_{t+1}) + \tilde{\nabla}_{\mathbf{y}}^2 g_{t+1}(\mathbf{z}_{t+1}) - \hat{\mathbf{d}}_{t+1}^{\mathbf{y}}\|^2$$

$$= \mathbb{E}\|\nabla_{\mathbf{y}}f_{t+1,\boldsymbol{\rho}}(\mathbf{z}_{t+1}) + \tilde{\nabla}_{\mathbf{y}}^2 g_{t+1}(\mathbf{z}_{t+1}) - \hat{\mathbf{d}}_t^{\mathbf{y}} - (\hat{\mathbf{d}}_{t+1}^{\mathbf{y}} - \hat{\mathbf{d}}_t^{\mathbf{y}})\|^2$$

$$= \mathbb{E}\|\nabla_{\mathbf{y}}f_{t+1,\boldsymbol{\rho}}(\mathbf{z}_{t+1}) + \tilde{\nabla}_{\mathbf{y}}^2 g_{t+1}(\mathbf{z}_{t+1}) - \hat{\mathbf{d}}_t^{\mathbf{y}} + \lambda_{t+1}\hat{\mathbf{d}}_t^{\mathbf{y}}$$

$$- \lambda_{t+1}\left(\hat{\nabla}_{\mathbf{y}}f_{t+1}(\mathbf{z}_{t+1};\mathcal{B}_{t+1}) + \hat{\nabla}_{\mathbf{y}}^2 g_{t+1}(\mathbf{z}_{t+1};\bar{\mathcal{B}}_{t+1})\right)$$

$$- (1-\lambda_{t+1})\left(\hat{\nabla}_{\mathbf{y}}f_{t+1}(\mathbf{z}_{t+1};\mathcal{B}_{t+1}) + \hat{\nabla}_{\mathbf{y}}^2 g_{t+1}(\mathbf{z}_{t+1};\bar{\mathcal{B}}_{t+1})\right.$$

$$\left. -\hat{\nabla}_{\mathbf{y}}f_{t+1}(\mathbf{z}_t;\mathcal{B}_{t+1}) - \hat{\nabla}_{\mathbf{y}}^2 g_{t+1}(\mathbf{z}_t;\bar{\mathcal{B}}_{t+1})\right)\|^2$$

$$= \mathbb{E}\|(1-\lambda_{t+1})(\nabla_{\mathbf{y}}f_{t,\boldsymbol{\rho}}(\mathbf{z}_t) + \tilde{\nabla}_{\mathbf{y}}^2 g_t(\mathbf{z}_t) - \hat{\mathbf{d}}_t^{\mathbf{y}})$$

$$+ \lambda_{t+1}(\nabla_{\mathbf{y}}f_{t+1,\boldsymbol{\rho}}(\mathbf{z}_{t+1}) + \tilde{\nabla}_{\mathbf{y}}^2 g_{t+1}(\mathbf{z}_{t+1}) - \hat{\nabla}_{\mathbf{y}}f_{t+1}(\mathbf{z}_{t+1};\mathcal{B}_{t+1}) - \hat{\nabla}_{\mathbf{y}}^2 g_{t+1}(\mathbf{z}_{t+1};\bar{\mathcal{B}}_{t+1}))$$

$$+ (1-\lambda_{t+1})\left(\nabla_{\mathbf{y}}f_{t+1,\boldsymbol{\rho}}(\mathbf{z}_{t+1}) + \tilde{\nabla}_{\mathbf{y}}^2 g_{t+1}(\mathbf{z}_{t+1}) - \nabla_{\mathbf{y}}f_{t,\boldsymbol{\rho}}(\mathbf{z}_t) - \tilde{\nabla}_{\mathbf{y}}^2 g_t(\mathbf{z}_t)\right.$$

$$+ \nabla_{\mathbf{y}}f_{t+1,\boldsymbol{\rho}}(\mathbf{z}_t) + \tilde{\nabla}_{\mathbf{y}}^2 g_{t+1}(\mathbf{z}_t) - \nabla_{\mathbf{y}}f_{t+1,\boldsymbol{\rho}}(\mathbf{z}_t) - \tilde{\nabla}_{\mathbf{y}}^2 g_{t+1}(\mathbf{z}_t)$$

$$\left. -\hat{\nabla}_{\mathbf{y}}f_{t+1}(\mathbf{z}_{t+1};\mathcal{B}_{t+1}) - \hat{\nabla}_{\mathbf{y}}^2 g_{t+1}(\mathbf{z}_{t+1};\bar{\mathcal{B}}_{t+1}) + \hat{\nabla}_{\mathbf{y}}f_{t+1}(\mathbf{z}_t;\mathcal{B}_{t+1}) + \hat{\nabla}_{\mathbf{y}}^2 g_{t+1}(\mathbf{z}_t;\bar{\mathcal{B}}_{t+1})\right)\|^2.$$

Since

$$\mathbb{E}\left[\hat{\nabla}_{\mathbf{y}}f_{t+1}(\mathbf{z}_{t+1};\mathcal{B}_{t+1}) + \hat{\nabla}_{\mathbf{y}}^2 g_{t+1}(\mathbf{z}_{t+1};\bar{\mathcal{B}}_{t+1})\right] = \nabla_{\mathbf{y}}f_{t+1,\boldsymbol{\rho}}(\mathbf{z}_{t+1}) + \tilde{\nabla}_{\mathbf{y}}^2 g_{t+1}(\mathbf{z}_{t+1}),$$

$$\mathbb{E}\left[\hat{\nabla}_{\mathbf{y}}f_{t+1}(\mathbf{z}_{t+1};\mathcal{B}_{t+1}) + \hat{\nabla}_{\mathbf{y}}^2 g_{t+1}(\mathbf{z}_{t+1};\bar{\mathcal{B}}_{t+1}) - \hat{\nabla}_{\mathbf{y}}f_{t+1}(\mathbf{z}_t;\mathcal{B}_{t+1}) - \hat{\nabla}_{\mathbf{y}}^2 g_{t+1}(\mathbf{z}_t;\bar{\mathcal{B}}_{t+1})\right]$$

$$= \nabla_{\mathbf{y}}f_{t+1,\boldsymbol{\rho}}(\mathbf{z}_{t+1}) + \tilde{\nabla}_{\mathbf{y}}^2 g_{t+1}(\mathbf{z}_{t+1}) - \nabla_{\mathbf{y}}f_{t+1,\boldsymbol{\rho}}(\mathbf{z}_t) - \tilde{\nabla}_{\mathbf{y}}^2 g_{t+1}(\mathbf{z}_t),$$

then, we have

$$\mathbb{E}\|\nabla_{\mathbf{y}} f_{t+1,\boldsymbol{\rho}}(\mathbf{z}_{t+1}) + \tilde{\nabla}_{\mathbf{y}}^2 g_{t+1}(\mathbf{z}_{t+1}) - \hat{\mathbf{d}}_{t+1}^{\mathbf{v}}\|^2$$

$$= (1-\lambda_{t+1})^2 \mathbb{E}\|\nabla_{\mathbf{y}} f_{t,\boldsymbol{\rho}}(\mathbf{z}_t) + \tilde{\nabla}_{\mathbf{y}}^2 g_t(\mathbf{z}_t) - \hat{\mathbf{d}}_t^{\mathbf{v}}\|^2$$

$$+ \|\lambda_{t+1}(\nabla_{\mathbf{y}} f_{t+1,\boldsymbol{\rho}}(\mathbf{z}_{t+1}) + \tilde{\nabla}_{\mathbf{y}}^2 g_{t+1}(\mathbf{z}_{t+1}) - \hat{\nabla}_{\mathbf{y}} f_{t+1}(\mathbf{z}_{t+1}; \mathcal{B}_{t+1}) - \hat{\nabla}_{\mathbf{y}}^2 g_{t+1}(\mathbf{z}_{t+1}; \bar{\mathcal{B}}_{t+1}))$$

$$+ (1-\lambda_{t+1})\left(\nabla_{\mathbf{y}} f_{t+1,\boldsymbol{\rho}}(\mathbf{z}_{t+1}) + \tilde{\nabla}_{\mathbf{y}}^2 g_{t+1}(\mathbf{z}_{t+1}) - \nabla_{\mathbf{y}} f_{t,\boldsymbol{\rho}}(\mathbf{z}_t) - \tilde{\nabla}_{\mathbf{y}}^2 g_t(\mathbf{z}_t)\right.$$

$$+ \nabla_{\mathbf{y}} f_{t+1,\boldsymbol{\rho}}(\mathbf{z}_t) + \tilde{\nabla}_{\mathbf{y}}^2 g_{t+1}(\mathbf{z}_t) - \nabla_{\mathbf{y}} f_{t+1,\boldsymbol{\rho}}(\mathbf{z}_t) - \tilde{\nabla}_{\mathbf{y}}^2 g_{t+1}(\mathbf{z}_t)$$

$$\left.- \hat{\nabla}_{\mathbf{y}} f_{t+1}(\mathbf{z}_{t+1}; \mathcal{B}_{t+1}) - \hat{\nabla}_{\mathbf{y}}^2 g_{t+1}(\mathbf{z}_{t+1}; \bar{\mathcal{B}}_{t+1}) + \hat{\nabla}_{\mathbf{y}} f_{t+1}(\mathbf{z}_t; \mathcal{B}_{t+1}) + \hat{\nabla}_{\mathbf{y}}^2 g_{t+1}(\mathbf{z}_t; \bar{\mathcal{B}}_{t+1})\right)\|^2$$

$$\leq (1-\lambda_{t+1})^2 \mathbb{E}\|\nabla_{\mathbf{y}} f_{t,\boldsymbol{\rho}}(\mathbf{z}_t) + \tilde{\nabla}_{\mathbf{y}}^2 g_t(\mathbf{z}_t) - \hat{\mathbf{d}}_t^{\mathbf{v}}\|^2$$

$$+ 3(1-\lambda_{t+1})^2 \mathbb{E}\|\nabla_{\mathbf{y}} f_{t+1,\boldsymbol{\rho}}(\mathbf{z}_{t+1}) + \tilde{\nabla}_{\mathbf{y}}^2 g_{t+1}(\mathbf{z}_{t+1}) - \nabla_{\mathbf{y}} f_{t,\boldsymbol{\rho}}(\mathbf{z}_t) - \tilde{\nabla}_{\mathbf{y}}^2 g_t(\mathbf{z}_t)$$

$$+ \nabla_{\mathbf{y}} f_{t+1,\boldsymbol{\rho}}(\mathbf{z}_t) + \tilde{\nabla}_{\mathbf{y}}^2 g_{t+1}(\mathbf{z}_t) - \nabla_{\mathbf{y}} f_{t+1,\boldsymbol{\rho}}(\mathbf{z}_t) - \tilde{\nabla}_{\mathbf{y}}^2 g_{t+1}(\mathbf{z}_t)$$

$$- \hat{\nabla}_{\mathbf{y}} f_{t+1}(\mathbf{z}_{t+1}; \mathcal{B}_{t+1}) - \hat{\nabla}_{\mathbf{y}}^2 g_{t+1}(\mathbf{z}_{t+1}; \bar{\mathcal{B}}_{t+1}) + \hat{\nabla}_{\mathbf{y}} f_{t+1}(\mathbf{z}_t; \mathcal{B}_{t+1}) + \hat{\nabla}_{\mathbf{y}}^2 g_{t+1}(\mathbf{z}_t; \bar{\mathcal{B}}_{t+1})\|^2$$

$$+ 3\lambda_{t+1}^2 \mathbb{E}\|\nabla_{\mathbf{y}} f_{t+1,\boldsymbol{\rho}}(\mathbf{z}_{t+1}) - \hat{\nabla}_{\mathbf{y}} f_{t+1}(\mathbf{z}_{t+1}; \mathcal{B}_{t+1})\|^2$$

$$+ 3\lambda_{t+1}^2 \mathbb{E}\|\tilde{\nabla}_{\mathbf{y}}^2 g_{t+1}(\mathbf{z}_{t+1}) - \hat{\nabla}_{\mathbf{y}}^2 g_{t+1}(\mathbf{z}_{t+1}; \bar{\mathcal{B}}_{t+1})\|^2, \tag{157}$$

where the second inequality holds by Cauchy-Schwarz inequality.
Note that, for the last term on the right-hand side of (157), from (26a) and (155), we have

$$\|\tilde{\nabla}_{\mathbf{y}}^2 g_{t+1}(\mathbf{z}_{t+1}) - \hat{\nabla}_{\mathbf{y}}^2 g_{t+1}(\mathbf{z}_{t+1}; \bar{\mathcal{B}}_{t+1})\|^2$$

$$\leq 2\|\frac{1}{2\rho_{\mathbf{v}}}(\nabla_{\mathbf{y}} g_{t+1,\boldsymbol{\rho}}(\mathbf{x}_{t+1}, \mathbf{y}_{t+1} + \rho_{\mathbf{v}}\mathbf{v}_{t+1}) - \hat{\nabla}_{\mathbf{y}} g_{t+1}(\mathbf{x}_{t+1}, \mathbf{y}_{t+1} + \rho_{\mathbf{v}}\mathbf{v}_{t+1}; \bar{\mathcal{B}}_{t+1}))\|^2$$

$$+ 2\|\frac{1}{2\rho_{\mathbf{v}}}(\hat{\nabla}_{\mathbf{y}} g_{t+1}(\mathbf{x}_{t+1}, \mathbf{y}_{t+1} - \rho_{\mathbf{v}}\mathbf{v}_{t+1}; \bar{\mathcal{B}}_{t+1}) - \nabla_{\mathbf{y}} g_{t+1,\boldsymbol{\rho}}(\mathbf{x}_{t+1}, \mathbf{y}_{t+1} - \rho_{\mathbf{v}}\mathbf{v}_{t+1}))\|^2$$

$$\leq \frac{\hat{\sigma}_{g_{\mathbf{y}}}^2}{\bar{b}\rho_{\mathbf{v}}^2},$$

where the last inequality follows from Assumption D1..
Then, from $\mathbb{E}\|a - \mathbb{E}[a]\|^2 = \mathbb{E}\|a\|^2 - \|\mathbb{E}[a]\|^2$ and Assumptions D1. and D3., we have

$$\mathbb{E}\|\nabla_{\mathbf{y}} f_{t+1,\boldsymbol{\rho}}(\mathbf{z}_{t+1}) + \tilde{\nabla}_{\mathbf{y}}^2 g_{t+1}(\mathbf{z}_{t+1}) - \hat{\mathbf{d}}_{t+1}^{\mathbf{v}}\|^2$$

$$\leq (1-\lambda_{t+1})^2 \mathbb{E}\|\nabla_{\mathbf{y}} f_{t,\boldsymbol{\rho}}(\mathbf{z}_t) + \tilde{\nabla}_{\mathbf{y}}^2 g_t(\mathbf{z}_t) - \hat{\mathbf{d}}_t^{\mathbf{v}}\|^2$$

$$+ 6(1-\lambda_{t+1})^2 \mathbb{E}\|\nabla_{\mathbf{y}} f_{t+1,\boldsymbol{\rho}}(\mathbf{z}_t) + \tilde{\nabla}_{\mathbf{y}}^2 g_{t+1}(\mathbf{z}_t) - \nabla_{\mathbf{y}} f_{t,\boldsymbol{\rho}}(\mathbf{z}_t) - \tilde{\nabla}_{\mathbf{y}}^2 g_t(\mathbf{z}_t)\|^2$$

$$+ 6(1-\lambda_{t+1})^2 \mathbb{E}\|\hat{\nabla}_{\mathbf{y}} f_{t+1}(\mathbf{z}_{t+1}; \mathcal{B}_{t+1}) + \hat{\nabla}_{\mathbf{y}}^2 g_{t+1}(\mathbf{z}_{t+1}; \bar{\mathcal{B}}_{t+1})$$

$$- \hat{\nabla}_{\mathbf{y}} f_{t+1}(\mathbf{z}_t; \mathcal{B}_{t+1}) - \hat{\nabla}_{\mathbf{y}}^2 g_{t+1}(\mathbf{z}_t; \bar{\mathcal{B}}_{t+1})\|^2 + 3\lambda_{t+1}^2 \left(\frac{\hat{\sigma}_{g_{\mathbf{y}}}^2}{\bar{b}\rho_{\mathbf{v}}^2} + \frac{\hat{\sigma}_{f_{\mathbf{y}}}^2}{b}\right).$$

Then, from Young's inequality and Lemma D.10, we obtain

$$\mathbb{E}\|\nabla_{\mathbf{y}} f_{t+1,\boldsymbol{\rho}}(\mathbf{z}_{t+1}) + \tilde{\nabla}_{\mathbf{y}}^2 g_{t+1}(\mathbf{z}_{t+1}) - \hat{\mathbf{d}}_{t+1}^{\mathbf{v}}\|^2$$

$$\leq (1-\lambda_{t+1})^2 \mathbb{E}\|\nabla_{\mathbf{y}} f_{t,\boldsymbol{\rho}}(\mathbf{z}_t) + \tilde{\nabla}_{\mathbf{y}}^2 g_t(\mathbf{z}_t) - \hat{\mathbf{d}}_t^{\mathbf{v}}\|^2$$

$$+ 12(1-\lambda_{t+1})^2 \mathbb{E}\|\nabla_{\mathbf{y}} f_{t+1,\boldsymbol{\rho}}(\mathbf{z}_t) - \nabla_{\mathbf{y}} f_{t,\boldsymbol{\rho}}(\mathbf{z}_t)\|^2$$

$$+ 12(1-\lambda_{t+1})^2 \mathbb{E}\|\tilde{\nabla}_{\mathbf{y}}^2 g_{t+1}(\mathbf{z}_t) - \tilde{\nabla}_{\mathbf{y}}^2 g_t(\mathbf{z}_t)\|^2$$

$$+ 6(12\ell_{f,1}^2 + \frac{9\ell_{g,1}^2}{2\rho_{\mathbf{v}}^2})d_2\|\mathbf{x}_{t+1} - \mathbf{x}_t\|^2 + 6(12\ell_{f,1}^2 + \frac{9\ell_{g,1}^2}{2\rho_{\mathbf{v}}^2})d_2\|\mathbf{y}_{t+1} - \mathbf{y}_t\|^2$$

$$+ 27d_2\ell_{g,1}^2\|\mathbf{v}_{t+1} - \mathbf{v}_t\|^2 + 6(3\ell_{f,1}^2 + \frac{3\ell_{g,1}^2}{4\rho_{\mathbf{v}}^2})d_2^2\rho_{\mathbf{r}}^2 + 3\lambda_{t+1}^2(\frac{\hat{\sigma}_{g_{\mathbf{y}}}^2}{\bar{b}\rho_{\mathbf{v}}^2} + \frac{\hat{\sigma}_{f_{\mathbf{y}}}^2}{b}). \tag{158}$$

For the third term on the right-hand side of (158), based on (155), we have

$$\|\tilde{\nabla}_{\mathbf{y}}^2 g_{t+1}(\mathbf{x}_t, \mathbf{y}_t) - \tilde{\nabla}_{\mathbf{y}}^2 g_t(\mathbf{x}_t, \mathbf{y}_t)\|^2$$

$$\leq \frac{1}{2\rho_{\mathbf{v}}^2}\|\nabla_{\mathbf{y}} g_{t+1,\boldsymbol{\rho}}(\mathbf{x}_t, \mathbf{y}_t + \rho_{\mathbf{v}}\mathbf{v}_t) - \nabla_{\mathbf{y}} g_{t,\boldsymbol{\rho}}(\mathbf{x}_t, \mathbf{y}_t + \rho_{\mathbf{v}}\mathbf{v}_t)\|^2 \tag{159a}$$

$$+ \frac{1}{2\rho_{\mathbf{v}}^2}\|\nabla_{\mathbf{y}} g_{t,\boldsymbol{\rho}}(\mathbf{x}_t, \mathbf{y}_t - \rho_{\mathbf{v}}\mathbf{v}_t) - \nabla_{\mathbf{y}} g_{t+1,\boldsymbol{\rho}}(\mathbf{x}_t, \mathbf{y}_t - \rho_{\mathbf{v}}\mathbf{v}_t)\|^2. \tag{159b}$$

For (159a), we get

$$\|\nabla_{\mathbf{y}} g_{t+1,\boldsymbol{\rho}}(\mathbf{x}_t, \mathbf{y}_t + \rho_{\mathbf{v}}\mathbf{v}_t) - \nabla_{\mathbf{y}} g_{t,\boldsymbol{\rho}}(\mathbf{x}_t, \mathbf{y}_t + \rho_{\mathbf{v}}\mathbf{v}_t)\|^2$$

$$\leq 3\|\nabla_{\mathbf{y}} g_{t+1,\boldsymbol{\rho}}(\mathbf{x}_t, \mathbf{y}_t + \rho_{\mathbf{v}}\mathbf{v}_t) - \nabla_{\mathbf{y}} g_{t+1}(\mathbf{x}_t, \mathbf{y}_t + \rho_{\mathbf{v}}\mathbf{v}_t)\|^2$$

$$+ 3\|\nabla_{\mathbf{y}} g_{t+1}(\mathbf{x}_t, \mathbf{y}_t + \rho_{\mathbf{v}}\mathbf{v}_t) - \nabla_{\mathbf{y}} g_t(\mathbf{x}_t, \mathbf{y}_t + \rho_{\mathbf{v}}\mathbf{v}_t)\|^2$$

$$+ 3\|\nabla_{\mathbf{y}} g_t(\mathbf{x}_t, \mathbf{y}_t + \rho_{\mathbf{v}}\mathbf{v}_t) - \nabla_{\mathbf{y}} g_{t,\boldsymbol{\rho}}(\mathbf{x}_t, \mathbf{y}_t + \rho_{\mathbf{v}}\mathbf{v}_t)\|^2$$

$$\leq 3\|\nabla_{\mathbf{y}} g_t(\mathbf{x}_t, \mathbf{y}_t + \rho_{\mathbf{v}}\mathbf{v}_t) - \nabla_{\mathbf{y}} g_{t+1}(\mathbf{x}_t, \mathbf{y}_t + \rho_{\mathbf{v}}\mathbf{v}_t)\|^2 + \frac{3\rho_{\mathbf{r}}^2 d_2^2 \ell_{g,1}^2}{2},$$

where the last inequality follows from Eq. (130).
Similary, for (159b), we have

$$\|\nabla_{\mathbf{y}} g_{t,\boldsymbol{\rho}}(\mathbf{x}_t, \mathbf{y}_t - \rho_{\mathbf{v}}\mathbf{v}_t) - \nabla_{\mathbf{y}} g_{t+1,\boldsymbol{\rho}}(\mathbf{x}_t, \mathbf{y}_t - \rho_{\mathbf{v}}\mathbf{v}_t)\|^2$$

$$\leq 3\|\nabla_{\mathbf{y}} g_t(\mathbf{x}_t, \mathbf{y}_t - \rho_{\mathbf{v}}\mathbf{v}_t) - \nabla_{\mathbf{y}} g_{t+1}(\mathbf{x}_t, \mathbf{y}_t - \rho_{\mathbf{v}}\mathbf{v}_t)\|^2 + \frac{3\rho_{\mathbf{r}}^2 d_2^2 \ell_{g,1}^2}{2}.$$

Substituting the above inequalities in (159), we have

$$\|\tilde{\nabla}_{\mathbf{y}}^2 g_{t+1}(\mathbf{x}_t, \mathbf{y}_t) - \tilde{\nabla}_{\mathbf{y}}^2 g_t(\mathbf{x}_t, \mathbf{y}_t)\|^2$$

$$\leq \frac{3}{2\rho_{\mathbf{v}}^2}\|\nabla_{\mathbf{y}} g_t(\mathbf{x}_t, \mathbf{y}_t + \rho_{\mathbf{v}}\mathbf{v}_t) - \nabla_{\mathbf{y}} g_{t+1}(\mathbf{x}_t, \mathbf{y}_t + \rho_{\mathbf{v}}\mathbf{v}_t)\|^2$$

$$+ \frac{3}{2\rho_{\mathbf{v}}^2}\|\nabla_{\mathbf{y}} g_t(\mathbf{x}_t, \mathbf{y}_t - \rho_{\mathbf{v}}\mathbf{v}_t) - \nabla_{\mathbf{y}} g_{t+1}(\mathbf{x}_t, \mathbf{y}_t - \rho_{\mathbf{v}}\mathbf{v}_t)\|^2 + \frac{3\rho_{\mathbf{r}}^2 d_2^2 \ell_{g,1}^2}{2\rho_{\mathbf{v}}^2}. \tag{160}$$

For the second term on the right-hand side of (158), we have

$$\|\nabla_{\mathbf{y}} f_{t+1,\boldsymbol{\rho}}(\mathbf{x}_t, \mathbf{y}_t) - \nabla_{\mathbf{y}} f_{t,\boldsymbol{\rho}}(\mathbf{x}_t, \mathbf{y}_t)\|^2$$

$$\leq 3\|\nabla_{\mathbf{y}} f_{t+1,\boldsymbol{\rho}}(\mathbf{x}_t, \mathbf{y}_t) - \nabla_{\mathbf{y}} f_{t+1}(\mathbf{x}_t, \mathbf{y}_t)\|^2$$

$$+ 3\|\nabla_{\mathbf{y}} f_{t+1}(\mathbf{x}_t, \mathbf{y}_t) - \nabla_{\mathbf{y}} f_t(\mathbf{x}_t, \mathbf{y}_t)\|^2$$

$$+ 3\|\nabla_{\mathbf{y}} f_t(\mathbf{x}_t, \mathbf{y}_t) - \nabla_{\mathbf{y}} f_{t,\boldsymbol{\rho}}(\mathbf{x}_t, \mathbf{y}_t)\|^2$$

$$\leq 3\|\nabla_{\mathbf{y}} f_t(\mathbf{x}_t, \mathbf{y}_t) - \nabla_{\mathbf{y}} f_{t+1}(\mathbf{x}_t, \mathbf{y}_t)\|^2 + \frac{3\rho_{\mathbf{r}}^2 d_2^2 \ell_{f,1}^2}{2}, \tag{161}$$

where the last inequality follows from Eq. (132).

From (160), (161) and (158), we get

$$\mathbb{E}\|\nabla_{\mathbf{y}} f_{t+1,\boldsymbol{\rho}}(\mathbf{z}_{t+1}) + \tilde{\nabla}_{\mathbf{y}}^2 g_{t+1}(\mathbf{z}_{t+1}) - \hat{\mathbf{d}}_{t+1}^{\mathbf{v}}\|^2$$

$$\leq (1 - \lambda_{t+1})^2 \mathbb{E}\|\nabla_{\mathbf{y}} f_{t,\boldsymbol{\rho}}(\mathbf{z}_t) + \tilde{\nabla}_{\mathbf{y}}^2 g_t(\mathbf{z}_t) - \hat{\mathbf{d}}_t^{\mathbf{v}}\|^2$$

$$+ 36\|\nabla_{\mathbf{y}} f_t(\mathbf{x}_t, \mathbf{y}_t) - \nabla_{\mathbf{y}} f_{t+1}(\mathbf{x}_t, \mathbf{y}_t)\|^2 + 18\rho_{\mathbf{r}}^2 d_2^2 \ell_{f,1}^2$$

$$+ \frac{18}{\rho_{\mathbf{v}}^2}\|\nabla_{\mathbf{y}} g_t(\mathbf{x}_t, \mathbf{y}_t + \rho_{\mathbf{v}}\mathbf{v}_t) - \nabla_{\mathbf{y}} g_{t+1}(\mathbf{x}_t, \mathbf{y}_t + \rho_{\mathbf{v}}\mathbf{v}_t)\|^2$$

$$+ \frac{18}{\rho_{\mathbf{v}}^2}\|\nabla_{\mathbf{y}} g_t(\mathbf{x}_t, \mathbf{y}_t - \rho_{\mathbf{v}}\mathbf{v}_t) - \nabla_{\mathbf{y}} g_{t+1}(\mathbf{x}_t, \mathbf{y}_t - \rho_{\mathbf{v}}\mathbf{v}_t)\|^2 + \frac{18\rho_{\mathbf{r}}^2 d_2^2 \ell_{g,1}^2}{\rho_{\mathbf{v}}^2}$$

$$+ 6(12\ell_{f,1}^2 + \frac{9\ell_{g,1}^2}{2\rho_{\mathbf{v}}^2})d_2\|\mathbf{x}_{t+1} - \mathbf{x}_t\|^2 + 6(12\ell_{f,1}^2 + \frac{9\ell_{g,1}^2}{2\rho_{\mathbf{v}}^2})d_2\|\mathbf{y}_{t+1} - \mathbf{y}_t\|^2$$

$$+ 27d_2\ell_{g,1}^2\|\mathbf{v}_{t+1} - \mathbf{v}_t\|^2 + 6(3\ell_{f,1}^2 + \frac{3\ell_{g,1}^2}{4\rho_{\mathbf{v}}^2})d_2^2\rho_{\mathbf{r}}^2 + 3\lambda_{t+1}^2(\frac{\hat{\sigma}_{g_{\mathbf{y}}}^2}{b\rho_{\mathbf{v}}^2} + \frac{\hat{\sigma}_{f_{\mathbf{y}}}^2}{b}).$$

$$\square$$

**Lemma D.12.** *Suppose Assumption* B4. *holds. Let*

$$e_t^H := \tilde{\nabla}_{\mathbf{y}}^2 g_t\left(\mathbf{x}_t, \mathbf{y}_t\right) - \nabla_{\mathbf{y}}^2 g_{t,\boldsymbol{\rho}}\left(\mathbf{x}_t, \mathbf{y}_t\right) \mathbf{v}_t, \tag{162a}$$

$$e_t^J := \tilde{\nabla}_{\mathbf{xy}}^2 g_t\left(\mathbf{x}_t, \mathbf{y}_t\right) - \nabla_{\mathbf{xy}}^2 g_{t,\boldsymbol{\rho}}\left(\mathbf{x}_t, \mathbf{y}_t\right) \mathbf{v}_t, \tag{162b}$$

*where*

$$\tilde{\nabla}_{\mathbf{y}}^2 g_t\left(\mathbf{x}_t, \mathbf{y}_t\right) = \frac{1}{2\rho_{\mathbf{v}}}(\nabla_{\mathbf{y}} g_{t,\boldsymbol{\rho}}(\mathbf{x}_t, \mathbf{y}_t + \rho_{\mathbf{v}}\mathbf{v}_t) - \nabla_{\mathbf{y}} g_{t,\boldsymbol{\rho}}(\mathbf{x}_t, \mathbf{y}_t - \rho_{\mathbf{v}}\mathbf{v}_t)),$$

$$\tilde{\nabla}_{\mathbf{xy}}^2 g_t\left(\mathbf{x}_t, \mathbf{y}_t\right) = \frac{1}{2\rho_{\mathbf{v}}}(\nabla_{\mathbf{x}} g_{t,\boldsymbol{\rho}}(\mathbf{x}_t, \mathbf{y}_t + \rho_{\mathbf{v}}\mathbf{v}_t) - \nabla_{\mathbf{x}} g_{t,\boldsymbol{\rho}}(\mathbf{x}_t, \mathbf{y}_t - \rho_{\mathbf{v}}\mathbf{v}_t)).$$

*Then, for* $(\mathbf{x}_t, \mathbf{y}_t, \mathbf{v}_t)$ *presented to Algorithm* 2, *we have*

*(a)*

$$\mathbb{E}\left[\left\|e_t^H\right\|^2\right] \leq \ell_{g,2}^2 \rho_{\mathbf{v}}^2 p^4. \tag{163a}$$

*(b)*

$$\mathbb{E}\left[\left\|e_t^J\right\|^2\right] \leq \ell_{g,2}^2 \rho_{\mathbf{v}}^2 p^4. \tag{163b}$$

*Proof.* **For part (a)**: From Lemma D.1, We have

$$\mathbb{E}\left[\left\|e_t^H\right\|\right] = \mathbb{E}\left[\left\|\tilde{\nabla}_{\mathbf{y}}^2 g_t\left(\mathbf{x}_t, \mathbf{y}_t\right) - \nabla_{\mathbf{y}}^2 g_{t,\boldsymbol{\rho}}\left(\mathbf{x}_t, \mathbf{y}_t\right) \mathbf{v}_t\right\|\right]$$

$$\leq \frac{1}{2\rho_{\mathbf{v}}}\mathbb{E}\left[\left\|\nabla_{\mathbf{y}} g_{t,\boldsymbol{\rho}}(\mathbf{x}_t, \mathbf{y}_t + \rho_{\mathbf{v}}\mathbf{v}_t) - \nabla_{\mathbf{y}} g_{t,\boldsymbol{\rho}}(\mathbf{x}_t, \mathbf{y}_t) - \nabla_{\mathbf{y}}^2 g_{t,\boldsymbol{\rho}}\left(\mathbf{x}_t, \mathbf{y}_t\right) \rho_{\mathbf{v}}\mathbf{v}_t\right\|\right]$$

$$+ \frac{1}{2\rho_{\mathbf{v}}}\mathbb{E}\left[\left\|\nabla_{\mathbf{y}} g_{t,\boldsymbol{\rho}}(\mathbf{x}_t, \mathbf{y}_t) - \nabla_{\mathbf{y}} g_{t,\boldsymbol{\rho}}(\mathbf{x}_t, \mathbf{y}_t - \rho_{\mathbf{v}}\mathbf{v}_t) - \nabla_{\mathbf{y}}^2 g_{t,\boldsymbol{\rho}}\left(\mathbf{x}_t, \mathbf{y}_t\right) \rho_{\mathbf{v}}\mathbf{v}_t\right\|\right]$$

$$\leq \ell_{g,2}\rho_{\mathbf{v}}\mathbb{E}\left[\left\|\mathbf{v}_t\right\|^2\right]$$

$$\leq \ell_{g,2}\rho_{\mathbf{v}}p^2, \tag{164}$$

where the last inequality follows from (8).
**For part (b):** From Lemma D.1, We have

$$\mathbb{E}\left[\left\|e_t^J\right\|\right] = \mathbb{E}\left[\left\|\tilde{\nabla}_{\mathbf{xy}}^2 g_t\left(\mathbf{x}_t, \mathbf{y}_t\right) - \nabla_{\mathbf{xy}}^2 g_{t,\boldsymbol{\rho}}\left(\mathbf{x}_t, \mathbf{y}_t\right) \mathbf{v}_t\right\|\right]$$

$$\leq \frac{1}{2\rho_{\mathbf{v}}}\mathbb{E}\left[\left\|\nabla_{\mathbf{x}} g_{t,\boldsymbol{\rho}}(\mathbf{x}_t, \mathbf{y}_t + \rho_{\mathbf{v}}\mathbf{v}_t) - \nabla_{\mathbf{x}} g_{t,\boldsymbol{\rho}}(\mathbf{x}_t, \mathbf{y}_t) - \nabla_{\mathbf{xy}}^2 g_{t,\boldsymbol{\rho}}\left(\mathbf{x}_t, \mathbf{y}_t\right) \rho_{\mathbf{v}}\mathbf{v}_t\right\|\right]$$

$$+ \frac{1}{2\rho_{\mathbf{v}}}\mathbb{E}\left[\left\|\nabla_{\mathbf{x}} g_{t,\boldsymbol{\rho}}(\mathbf{x}_t, \mathbf{y}_t) - \nabla_{\mathbf{x}} g_{t,\boldsymbol{\rho}}(\mathbf{x}_t, \mathbf{y}_t - \rho_{\mathbf{v}}\mathbf{v}_t) - \nabla_{\mathbf{xy}} g_{t,\boldsymbol{\rho}}\left(\mathbf{x}_t, \mathbf{y}_t\right) \rho_{\mathbf{v}}\mathbf{v}_t\right\|\right]$$

$$\leq \ell_{g,2}\rho_{\mathbf{v}}\mathbb{E}\left[\left\|\mathbf{v}_t\right\|^2\right]$$

$$\leq \ell_{g,2}\rho_{\mathbf{v}}p^2, \tag{165}$$

where the last inequality follows from (8). □

**Lemma D.13.** *Suppose Assumption* B4. *holds. Then, for the directions* $\hat{\mathbf{d}}_t^{\mathbf{v}}$ *and* $\hat{\mathbf{d}}_t^{\mathbf{x}}$ *provided to Algorithm* 2, *and*

*(a) for* $\mathbf{d}_{t,\boldsymbol{\rho}}^{\mathbf{v}}$ *defined in* (22b), *we have*

$$\mathbb{E}\left[\left\|\hat{\mathbf{d}}_t^{\mathbf{v}} - \mathbf{d}_{t,\boldsymbol{\rho}}^{\mathbf{v}}\right\|^2\right] \leq 2\mathbb{E}\left[\left\|e_t^M\right\|^2\right] + 2\ell_{g,2}^2 \rho_{\mathbf{v}}^2 p^4 =: B_t, \tag{166a}$$

*where* $e_t^M = \nabla_{\mathbf{y}} f_{t,\boldsymbol{\rho}}(\mathbf{x}_t, \mathbf{y}_t) + \tilde{\nabla}_{\mathbf{y}}^2 g_t\left(\mathbf{x}_t, \mathbf{y}_t\right) - \hat{\mathbf{d}}_t^{\mathbf{v}}$ *is defined as in* (154).

*(b) and for $\mathbf{d}_{t,\boldsymbol{\rho}}^{\mathbf{x}}$ defined in (22c), we have*

$$\mathbb{E}\left[\left\|\hat{\mathbf{d}}_t^{\mathbf{x}} - \mathbf{d}_{t,\boldsymbol{\rho}}^{\mathbf{x}}\right\|^2\right] \leq 2\mathbb{E}\left[\left\|e_t^L\right\|^2\right] + 2\ell_{g,2}^2\rho_{\mathbf{v}}^2 p^4, \tag{166b}$$

*where*

$$e_t^L := \nabla_{\mathbf{x}}f_{t,\boldsymbol{\rho}}(\mathbf{x}_t, \mathbf{y}_t) + \tilde{\nabla}_{\mathbf{xy}}^2 g_t(\mathbf{x}_t, \mathbf{y}_t) - \hat{\mathbf{d}}_t^{\mathbf{x}}, \tag{166c}$$

*with $\tilde{\nabla}_{\mathbf{xy}}^2 g_t(\mathbf{x}_t, \mathbf{y}_t)$ is defined in (172).*

*Proof.* **For part (a)**: Let

$$\tilde{\nabla}_{\mathbf{y}}^2 g_t(\mathbf{x}_t, \mathbf{y}_t) = \frac{1}{2\rho_{\mathbf{v}}}(\nabla_{\mathbf{y}}g_{t,\boldsymbol{\rho}}(\mathbf{x}_t, \mathbf{y}_t + \rho_{\mathbf{v}}\mathbf{v}_t) - \nabla_{\mathbf{y}}g_{t,\boldsymbol{\rho}}(\mathbf{x}_t, \mathbf{y}_t - \rho_{\mathbf{v}}\mathbf{v}_t)). \tag{167}$$

According to the definition of $\mathbf{d}_{t,\boldsymbol{\rho}}^{\mathbf{y}}$ in (22b), we have

$$\mathbb{E}\left[\left\|\hat{\mathbf{d}}_t^{\mathbf{y}} - \mathbf{d}_{t,\boldsymbol{\rho}}^{\mathbf{y}}\right\|^2\right] = \mathbb{E}\left[\left\|\hat{\mathbf{d}}_t^{\mathbf{y}} - \nabla_{\mathbf{y}}f_{t,\boldsymbol{\rho}}(\mathbf{x}_t, \mathbf{y}_t) - \nabla_{\mathbf{y}}^2 g_{t,\boldsymbol{\rho}}(\mathbf{x}_t, \mathbf{y}_t)\mathbf{v}\right\|^2\right]$$

$$\leq 2\mathbb{E}\left[\left\|\hat{\mathbf{d}}_t^{\mathbf{y}} - \nabla_{\mathbf{y}}f_{t,\boldsymbol{\rho}}(\mathbf{x}_t, \mathbf{y}_t) - \tilde{\nabla}_{\mathbf{y}}^2 g_t(\mathbf{x}_t, \mathbf{y}_t)\right\|^2\right] \tag{168a}$$

$$+ 2\mathbb{E}\left[\left\|\tilde{\nabla}_{\mathbf{y}}^2 g_t(\mathbf{x}_t, \mathbf{y}_t) - \nabla_{\mathbf{y}}^2 g_{t,\boldsymbol{\rho}}(\mathbf{x}_t, \mathbf{y}_t)\mathbf{v}\right\|^2\right]. \tag{168b}$$

Next, we separately bound (168a) and (168b) on the RHS of the above inequality.
**Bounding (168a)**. We have

$$2\mathbb{E}\left[\left\|\hat{\mathbf{d}}_t^{\mathbf{y}} - \nabla_{\mathbf{y}}f_{t,\boldsymbol{\rho}}(\mathbf{x}_t, \mathbf{y}_t) - \tilde{\nabla}_{\mathbf{y}}^2 g_t(\mathbf{x}_t, \mathbf{y}_t)\right\|^2\right] := 2\mathbb{E}\left[\left\|e_t^M\right\|^2\right]. \tag{169}$$

**Bounding (168b)**. From Lemmas D.1 and D.12, we have

$$(168b) = \mathbb{E}\left[\left\|e_t^H\right\|^2\right] \leq 3\ell_{g,2}^2\rho_{\mathbf{v}}^2 p^4. \tag{170}$$

Combining (169) and (170) yields

$$\mathbb{E}\left[\left\|\hat{\mathbf{d}}_t^{\mathbf{y}} - \mathbf{d}_{t,\boldsymbol{\rho}}^{\mathbf{y}}\right\|^2\right] \leq 2\mathbb{E}\left[\left\|e_t^M\right\|^2\right] + 2\ell_{g,2}^2\rho_{\mathbf{v}}^2 p^4. \tag{171}$$

**For part (b):** Let

$$\tilde{\nabla}_{\mathbf{xy}}^2 g_t(\mathbf{x}_t, \mathbf{y}_t) = \frac{1}{2\rho_{\mathbf{v}}}(\nabla_{\mathbf{x}}g_{t,\boldsymbol{\rho}}(\mathbf{x}_t, \mathbf{y}_t + \rho_{\mathbf{v}}\mathbf{v}_t) - \nabla_{\mathbf{x}}g_{t,\boldsymbol{\rho}}(\mathbf{x}_t, \mathbf{y}_t - \rho_{\mathbf{v}}\mathbf{v}_t)). \tag{172}$$

According to the definition of $\mathbf{d}_{t,\boldsymbol{\rho}}^{\mathbf{x}}$ in (22c), we have

$$\mathbb{E}\left[\left\|\hat{\mathbf{d}}_t^{\mathbf{x}} - \mathbf{d}_{t,\boldsymbol{\rho}}^{\mathbf{x}}\right\|^2\right] = \mathbb{E}\left[\left\|\hat{\mathbf{d}}_t^{\mathbf{x}} - \nabla_{\mathbf{x}}f_{t,\boldsymbol{\rho}}(\mathbf{x}_t, \mathbf{y}_t) - \nabla_{\mathbf{xy}}^2 g_{t,\boldsymbol{\rho}}(\mathbf{x}_t, \mathbf{y}_t)\mathbf{v}\right\|^2\right]$$

$$\leq 2\mathbb{E}\left[\left\|\hat{\mathbf{d}}_t^{\mathbf{x}} - \nabla_{\mathbf{x}}f_{t,\boldsymbol{\rho}}(\mathbf{x}_t, \mathbf{y}_t) - \tilde{\nabla}_{\mathbf{xy}}^2 g_t(\mathbf{x}_t, \mathbf{y}_t)\right\|^2\right] \tag{173a}$$

$$+ 2\mathbb{E}\left[\left\|\tilde{\nabla}_{\mathbf{xy}}^2 g_t(\mathbf{x}_t, \mathbf{y}_t) - \nabla_{\mathbf{xy}}^2 g_{t,\boldsymbol{\rho}}(\mathbf{x}_t, \mathbf{y}_t)\mathbf{v}_t\right\|^2\right]. \tag{173b}$$

Next, we separately bound (173a) and (173b) on the RHS of the above inequality.
**Bounding (173a)**. We have

$$2\mathbb{E}\left[\left\|\hat{\mathbf{d}}_t^{\mathbf{x}} - \nabla_{\mathbf{x}}f_{t,\boldsymbol{\rho}}(\mathbf{x}_t, \mathbf{y}_t) - \tilde{\nabla}_{\mathbf{xy}}^2 g_t(\mathbf{x}_t, \mathbf{y}_t)\right\|^2\right] := 2\mathbb{E}\left[\left\|e_t^L\right\|^2\right]. \tag{174}$$

**Bounding (173b)**. From Lemmas D.1 and D.12, we have

$$(173b) = \mathbb{E}\left[\left\|e_t^J\right\|^2\right] \leq 2\ell_{g,2}^2\rho_{\mathbf{v}}^2 p^4. \tag{175}$$

Combining (174)–(175) yields

$$\mathbb{E}\left[\left\|\hat{\mathbf{d}}_t^{\mathbf{x}} - \mathbf{d}_{t,\boldsymbol{\rho}}^{\mathbf{x}}\right\|^2\right] \leq 2\mathbb{E}\left[\left\|e_t^L\right\|^2\right] + 2\ell_{g,2}^2\rho_{\mathbf{v}}^2 p^4.$$

$$\square$$

**Lemma D.14.** *Suppose Assumptions 2.2, B1., B3. and B4. hold. Set the step size $\delta_t$ and the parameter $p$ in (8), as*

$$\delta_t \leq \left(2 + \frac{1}{\ell_{g,1}^2}\right) \frac{\mu_g \ell_{g,1}}{\mu_g + \ell_{g,1}}, \ \forall t \in [T], \quad \text{and} \quad p = \frac{\ell_{f,0}}{\mu_g}. \tag{176}$$

*Then, for the sequence $\{(\mathbf{x}_t, \mathbf{y}_t, \mathbf{v}_t)\}_{t=1}^T$ generated by Algorithm 2 and $\hat{\mathbf{v}}_t^*(\mathbf{x}_t)$ in (20), we have*

$$\mathbb{E}\left[\|\mathbf{v}_{t+1} - \hat{\mathbf{v}}_t^*(\mathbf{x}_t)\|^2\right] \leq (1 + \acute{a})\left(1 - \delta_t \frac{\mu_g \ell_{g,1}}{\mu_g + \ell_{g,1}}\right)\mathbb{E}[\hat{\theta}_t^{\mathbf{v}}] + \left(1 + \frac{1}{\acute{a}}\right)\delta_t^2 B_t,$$

*for some $\acute{a} > 0$, where $\hat{\theta}_t^{\mathbf{v}}$ and $B_t$ are defined in Eq. (142) and Lemma D.13, respectively.*

*Proof.* By setting the radius $p := \frac{\ell_{f,0}}{\mu_g}$ in (8), we have

$$\mathbb{E}\left[\|\mathbf{v}_{t+1} - \hat{\mathbf{v}}_t^*(\mathbf{x}_t)\|^2\right] = \mathbb{E}\left[\left\|\Pi_{\mathcal{Z}_p}\left[\mathbf{v}_t - \delta_t \hat{\mathbf{d}}_t^{\mathbf{v}}\right] - \Pi_{\mathcal{Z}_p}\left[\hat{\mathbf{v}}_t^*(\mathbf{x}_t)\right]\right\|^2\right]$$

$$\leq \mathbb{E}\left[\|\mathbf{v}_t - \delta_t \hat{\mathbf{d}}_t^{\mathbf{v}} - \hat{\mathbf{v}}_t^*(\mathbf{x}_t)\|^2\right]$$

$$\leq (1 + \acute{a})\underbrace{\mathbb{E}\left[\|\mathbf{v}_t - \delta_t \nabla P_t(\mathbf{x}_t, \hat{\mathbf{y}}_t^*(\mathbf{x}_t), \mathbf{v}_t) - \hat{\mathbf{v}}_t^*(\mathbf{x}_t)\|^2\right]}_{I_t}$$

$$+ \left(1 + \frac{1}{\acute{a}}\right)\delta_t^2 \underbrace{\mathbb{E}\left[\|\hat{\mathbf{d}}_t^{\mathbf{v}} - \nabla P_t(\mathbf{x}_t, \hat{\mathbf{y}}_t^*(\mathbf{x}_t), \mathbf{v}_t)\|^2\right]}_{K_t}, \tag{177}$$

where $\nabla P_t(\mathbf{x}_t, \hat{\mathbf{y}}_t^*(\mathbf{x}_t), \mathbf{v}_t) := \nabla_{\mathbf{y}}^2 g_{t,\boldsymbol{\rho}}(\mathbf{x}_t, \hat{\mathbf{y}}_t^*(\mathbf{x}_t)) \mathbf{v}_t + \nabla_{\mathbf{y}} f_{t,\boldsymbol{\rho}}(\mathbf{x}_t, \hat{\mathbf{y}}_t^*(\mathbf{x}_t))$.; the first inequality follows from non-expansiveness property of a projection operator.

We next bound the $I_t$, and $K_t$ terms in (177), respectively.

**Bounding $I_t$ .** We have

$$I_t = \mathbb{E}\left[\|\mathbf{v}_t - \hat{\mathbf{v}}_t^*(\mathbf{x}_t)\|^2\right] - 2\delta_t \mathbb{E}\left[\langle \nabla P_t(\mathbf{x}_t, \hat{\mathbf{y}}_t^*(\mathbf{x}_t), \mathbf{v}_t), \mathbf{v}_t - \hat{\mathbf{v}}_t^*(\mathbf{x}_t)\rangle\right]$$

$$+ \delta_t^2 \mathbb{E}\left[\|\nabla P_t(\mathbf{x}_t, \hat{\mathbf{y}}_t^*(\mathbf{x}_t), \mathbf{v}_t)\|^2\right]$$

$$\leq \left(1 - 2\delta_t \frac{\mu_g \ell_{g,1}}{\mu_g + \ell_{g,1}}\right)\mathbb{E}\left[\|\mathbf{v}_t - \hat{\mathbf{v}}_t^*(\mathbf{x}_t)\|^2\right]$$

$$- \left(2\delta_t \frac{\mu_g \ell_{g,1}}{\mu_g + \ell_{g,1}} - \delta_t^2\right)\mathbb{E}\left[\|\nabla P_t(\mathbf{x}_t, \hat{\mathbf{y}}_t^*(\mathbf{x}_t), \mathbf{v}_t)\|^2\right],$$

where the inequality holds since $\nabla P_t$ is the gradient of the strongly convex quadratic program $\frac{1}{2}\mathbf{v}^\top \nabla_{\mathbf{y}}^2 g_{t,\boldsymbol{\rho}}(\mathbf{x}, \hat{\mathbf{y}}_t^*(\mathbf{x})) \mathbf{v} + \mathbf{v}^\top \nabla_{\mathbf{y}} f_{t,\boldsymbol{\rho}}(\mathbf{x}, \hat{\mathbf{y}}_t^*(\mathbf{x}))$.

Thus, we have

$$\mathbb{E}\left[\langle \nabla P_t(\mathbf{x}_t, \hat{\mathbf{y}}_t^*(\mathbf{x}_t), \mathbf{v}_t), \mathbf{v}_t - \hat{\mathbf{v}}_t^*(\mathbf{x}_t)\rangle\right]$$

$$\geq \frac{\mu_g \ell_{g,1}}{\mu_g + \ell_{g,1}}\mathbb{E}\left[\|\mathbf{v}_t - \hat{\mathbf{v}}_t^*(\mathbf{x}_t)\|^2\right] + \frac{1}{\mu_g + \ell_{g,1}}\mathbb{E}\left[\|\nabla P_t(\mathbf{x}_t, \hat{\mathbf{y}}_t^*(\mathbf{x}_t), \mathbf{v}_t)\|^2\right].$$

Since $\delta_t \leq \left(2 + \frac{1}{\ell_{g,1}^2}\right)\frac{\mu_g \ell_{g,1}}{\mu_g + \ell_{g,1}}$, then we have

$$I_t \leq \left(1 - 2\delta_t \frac{\mu_g \ell_{g,1}}{\mu_g + \ell_{g,1}}\right)\mathbb{E}\left[\|\mathbf{v}_t - \hat{\mathbf{v}}_t^*(\mathbf{x}_t)\|^2\right] + \frac{1}{\ell_{g,1}^2}\left(\frac{\mu_g \ell_{g,1}}{\mu_g + \ell_{g,1}}\delta_t\right)\mathbb{E}\left[\|\nabla P_t(\mathbf{x}_t, \hat{\mathbf{y}}_t^*(\mathbf{x}_t), \mathbf{v}_t)\|^2\right]$$

$$\leq \left(1 - \delta_t \frac{\mu_g \ell_{g,1}}{\mu_g + \ell_{g,1}}\right)\mathbb{E}\left[\|\mathbf{v}_t - \hat{\mathbf{v}}_t^*(\mathbf{x}_t)\|^2\right], \tag{178}$$

where the second inequality holds since from (20), we have

$$\mathbb{E}\left[\|\nabla P_t(\mathbf{x}_t, \hat{\mathbf{y}}_t^*(\mathbf{x}_t), \mathbf{v}_t)\|^2\right] = \mathbb{E}\left[\|\nabla_{\mathbf{y}}^2 g_{t,\boldsymbol{\rho}}(\mathbf{x}_t, \hat{\mathbf{y}}_t^*(\mathbf{x}_t)) \mathbf{v}_t + \nabla_{\mathbf{y}} f_{t,\boldsymbol{\rho}}(\mathbf{x}_t, \hat{\mathbf{y}}_t^*(\mathbf{x}_t))\|^2\right]$$

$$= \mathbb{E}\left[\|\nabla_{\mathbf{y}}^2 g_{t,\boldsymbol{\rho}}(\mathbf{x}_t, \hat{\mathbf{y}}_t^*(\mathbf{x}_t))(\mathbf{v}_t - \hat{\mathbf{v}}_t^*(\mathbf{x}_t))\|^2\right]$$

$$\leq \ell_{g,1}^2 \mathbb{E}\left[\|\mathbf{v}_t - \hat{\mathbf{v}}_t^*(\mathbf{x}_t)\|^2\right],$$

where the second inequality follows from Assumption B3..

**Bounding $K_t$.** Let

$$\tilde{\nabla}_{\mathbf{y}}^2 g_t\left(\mathbf{x}_t, \mathbf{y}_t\right) = \frac{1}{2\rho_{\mathbf{v}}}\left(\nabla_{\mathbf{y}} g_{t,\rho}(\mathbf{x}_t, \mathbf{y}_t + \rho_{\mathbf{v}}\mathbf{v}_t) - \nabla_{\mathbf{y}} g_{t,\rho}(\mathbf{x}_t, \mathbf{y}_t - \rho_{\mathbf{v}}\mathbf{v}_t)\right).$$

From Lemma D.13, we have

$$K_t = \mathbb{E}\left[\|\hat{\mathbf{d}}_t^{\mathbf{y}} - \mathbf{d}_{t,\rho}^{\mathbf{v}}(\mathbf{x}_t, \mathbf{y}_t, \mathbf{v}_t)\|^2\right] \le B_t. \tag{179}$$

Putting (178), and (179) together with Eq. (177) yields the desired result.

$$\mathbb{E}\left[\|\mathbf{v}_{t+1} - \hat{\mathbf{v}}_t^*(\mathbf{x}_t)\|^2\right] \le (1 + \acute{a})\left(1 - \delta_t \frac{\mu_g \ell_{g,1}}{\mu_g + \ell_{g,1}}\right)\mathbb{E}\left[\|\mathbf{v}_t - \hat{\mathbf{v}}_t^*(\mathbf{x}_t)\|^2\right] + \left(1 + \frac{1}{\acute{a}}\right)\delta_t^2 B_t.$$

$$\square$$

**Lemma D.15.** *Suppose Assumptions 2.2 and 2.3 hold. Let $\hat{\theta}_t^{\mathbf{v}}$ be defined in (142). Set the parameter $p$ in (8) as $p = \frac{\ell_{f,0}}{\mu_g}$. Then, for any positive choice of step sizes satisfying*

$$\delta_t \le \left(2 + \frac{1}{\ell_{g,1}^2}\right)\frac{\mu_g \ell_{g,1}}{\mu_g + \ell_{g,1}},$$

*the sequence $\{(\mathbf{x}_t, \mathbf{y}_t, \mathbf{v}_t)\}_{t=1}^T$ generated by Algorithm 2 guarantees the following bound:*

$$\sum_{t=1}^T \left(\mathbb{E}[\hat{\theta}_{t+1}^{\mathbf{v}}] - \mathbb{E}[\hat{\theta}_t^{\mathbf{v}}]\right) \le \sum_{t=1}^T \left(-\frac{L_{\mu_g}}{4}\mathbb{E}[\hat{\theta}_t^{\mathbf{v}}] + \frac{4}{L_{\mu_g}}B_t\right)\delta_t$$

$$+ \frac{16\nu^2}{L_{\mu_g}\mu_g^2}(2L_{\mathbf{y}}^2 + 1)\sum_{t=1}^T \mathbb{E}\|\mathbf{x}_{t+1} - \mathbf{x}_t\|^2 \frac{1}{\delta_t}$$

$$+ \sum_{t=1}^T \left(\frac{96\ell_{g,1}\nu^2}{L_{\mu_g}\mu_g^3}(\rho_{\mathbf{s}}^2 + \rho_{\mathbf{r}}^2) + \frac{48\nu^2}{L_{\mu_g}\mu_g^2}\sup_{\mathbf{x}\in\mathcal{X}}\|\mathbf{y}_{t-1}^*(\mathbf{x}) - \mathbf{y}_t^*(\mathbf{x})\|^2\right)\frac{1}{\delta_t}, \tag{180}$$

*where $B_t$, $\nu$ and $(L_{\mu_g}, L_{\mathbf{y}})$ are defined in Lemmas D.13, C.7 and D.9, respectively.*

*Proof.* From Lemma B.4, we have, for any $\acute{c} > 0$

$$\mathbb{E}\left[\|\mathbf{v}_{t+1} - \hat{\mathbf{v}}_{t+1}^*(\mathbf{x}_{t+1})\|^2\right] = \mathbb{E}\left[\|\mathbf{v}_{t+1} - \hat{\mathbf{v}}_t^*(\mathbf{x}_t) + \hat{\mathbf{v}}_t^*(\mathbf{x}_t) - \hat{\mathbf{v}}_{t+1}^*(\mathbf{x}_{t+1})\|^2\right]$$

$$\le (1 + \acute{c})\mathbb{E}\left[\|\mathbf{v}_{t+1} - \hat{\mathbf{v}}_t^*(\mathbf{x}_t)\|^2\right]$$

$$+ \left(1 + \frac{1}{\acute{c}}\right)\mathbb{E}\left[\|\hat{\mathbf{v}}_{t+1}^*(\mathbf{x}_{t+1}) - \hat{\mathbf{v}}_t^*(\mathbf{x}_t)\|^2\right]. \tag{181}$$

From Lemma D.14, we have, for any $\acute{a} > 0$

$$\mathbb{E}\left[\|\mathbf{v}_{t+1} - \hat{\mathbf{v}}_t^*(\mathbf{x}_t)\|^2\right] \le (1 + \acute{a})\left(1 - \delta_t \frac{\mu_g \ell_{g,1}}{\mu_g + \ell_{g,1}}\right)\hat{\theta}_t^{\mathbf{v}} + \left(1 + \frac{1}{\acute{a}}\right)\delta_t^2 B_t. \tag{182}$$

Substituting (182) into (181), we get

$$\mathbb{E}\left[\|\mathbf{v}_{t+1} - \hat{\mathbf{v}}_{t+1}^*(\mathbf{x}_{t+1})\|^2\right] \le (1 + \acute{c})(1 + \acute{a})\left(1 - \delta_t \frac{\mu_g \ell_{g,1}}{\mu_g + \ell_{g,1}}\right)\hat{\theta}_t^{\mathbf{v}}$$

$$+ (1 + \acute{c})\left(1 + \frac{1}{\acute{a}}\right)\delta_t^2 B_t$$

$$+ \left(1 + \frac{1}{\acute{c}}\right)\mathbb{E}\left[\|\hat{\mathbf{v}}_{t+1}^*(\mathbf{x}_{t+1}) - \hat{\mathbf{v}}_t^*(\mathbf{x}_t)\|^2\right]. \tag{183}$$

Choose $\acute{c} = \frac{\delta_t L_{\mu_g}/4}{1 - \frac{\delta_t L_{\mu_g}}{2}}$ and $\acute{a} = \frac{\delta_t L_{\mu_g}/2}{1 - \delta_t L_{\mu_g}}$. Then, the following equations and inequalities are satisfied.

$$(1 + \acute{c})(1 + \acute{a})(1 - \delta_t L_{\mu_g}) = 1 - \frac{\delta_t L_{\mu_g}}{4},$$

$$(1 + \acute{c})\left(1 + \frac{1}{\acute{a}}\right) \leq \frac{4}{\delta_t L_{\mu_g}}, \tag{184}$$

$$1 + \frac{1}{\acute{a}} \leq \frac{2}{\delta_t L_{\mu_g}}, \quad 1 + \frac{1}{\acute{c}} \leq \frac{4}{\delta_t L_{\mu_g}},$$

where $L_{\mu_g} = \frac{\mu_g \ell_{g,1}}{\mu_g + \ell_{g,1}}$.

Thus, we have

$$\mathbb{E}\left[\left\|\mathbf{v}_{t+1} - \hat{\mathbf{v}}_{t+1}^*(\mathbf{x}_{t+1})\right\|^2\right] \leq \left(1 - \frac{\delta_t L_{\mu_g}}{4}\right)\hat{\theta}_t^{\mathbf{v}} + \frac{4}{L_{\mu_g}}\delta_t B_t$$

$$+ \frac{4}{L_{\mu_g}}\frac{1}{\delta_t}\mathbb{E}\left[\left\|\hat{\mathbf{v}}_{t+1}^*(\mathbf{x}_{t+1}) - \hat{\mathbf{v}}_t^*(\mathbf{x}_t)\right\|^2\right]. \tag{185}$$

We now bound the last term on the right-hand side of (185). By Lemma C.7, we have:

$$\left\|\hat{\mathbf{v}}_{t+1}^*(\mathbf{x}_{t+1}) - \hat{\mathbf{v}}_t^*(\mathbf{x}_t)\right\|^2$$

$$\leq 2\frac{\nu^2}{\mu_g^2}\left(\left\|\hat{\mathbf{y}}_{t+1}^*(\mathbf{x}_{t+1}) - \hat{\mathbf{y}}_t^*(\mathbf{x}_t)\right\|^2 + \left\|\mathbf{x}_{t+1} - \mathbf{x}_t\right\|^2\right)$$

$$\leq 2\frac{\nu^2}{\mu_g^2}\left(2\left\|\hat{\mathbf{y}}_{t+1}^*(\mathbf{x}_{t+1}) - \hat{\mathbf{y}}_{t+1}^*(\mathbf{x}_t)\right\|^2\right.$$

$$+ 2\left\|\hat{\mathbf{y}}_{t+1}^*(\mathbf{x}_t) - \hat{\mathbf{y}}_t^*(\mathbf{x}_t)\right\|^2 + \left\|\mathbf{x}_{t+1} - \mathbf{x}_t\right\|^2\Big)$$

$$\leq 2\frac{\nu^2}{\mu_g^2}\left(2L_{\mathbf{y}}^2\left\|\mathbf{x}_{t+1} - \mathbf{x}_t\right\|^2 + 2\left\|\hat{\mathbf{y}}_{t+1}^*(\mathbf{x}_t) - \hat{\mathbf{y}}_t^*(\mathbf{x}_t)\right\|^2 + \left\|\mathbf{x}_{t+1} - \mathbf{x}_t\right\|^2\right), \tag{186}$$

where the last inequality follows from Lemma D.2.

From (150), we have

$$\|\hat{\mathbf{y}}_{t+1}^*(\mathbf{x}_t) - \hat{\mathbf{y}}_t^*(\mathbf{x}_t)\|^2 \leq 3\|\hat{\mathbf{y}}_{t+1}^*(\mathbf{x}_t) - \mathbf{y}_{t+1}^*(\mathbf{x}_t)\|^2$$

$$+ 3\|\mathbf{y}_{t+1}^*(\mathbf{x}_t) - \mathbf{y}_t^*(\mathbf{x}_t)\|^2 + 3\|\mathbf{y}_t^*(\mathbf{x}_t) - \hat{\mathbf{y}}_t^*(\mathbf{x}_t)\|^2$$

$$\leq 3\|\mathbf{y}_{t+1}^*(\mathbf{x}_t) - \mathbf{y}_t^*(\mathbf{x}_t)\|^2 + \frac{6\ell_{g,1}(\rho_{\mathbf{s}}^2 + \rho_{\mathbf{r}}^2)}{\mu_g}. \tag{187}$$

Plugging (187) into (186), we get

$$\left\|\hat{\mathbf{v}}_{t+1}^*(\mathbf{x}_{t+1}) - \hat{\mathbf{v}}_t^*(\mathbf{x}_t)\right\|^2$$

$$\leq 4\frac{\nu^2}{\mu_g^2}(2L_{\mathbf{y}}^2 + 1)\left\|\mathbf{x}_{t+1} - \mathbf{x}_t\right\|^2$$

$$+ 4\frac{\nu^2}{\mu_g^2}\left(3\|\mathbf{y}_{t+1}^*(\mathbf{x}_t) - \mathbf{y}_t^*(\mathbf{x}_t)\|^2 + \frac{6\ell_{g,1}(\rho_{\mathbf{s}}^2 + \rho_{\mathbf{r}}^2)}{\mu_g}\right). \tag{188}$$

Then, substituting (188) into (185), rearranging the resulting inequality and summing over $t \in [T]$, we obtain the desired result. $\qquad\square$

## D.5 Bounds on the Zeroth-Order Estimation Error of Outer Objective

**Lemma D.16.** *Suppose Assumptions* B2. *and* B3. *hold. Let*

$$\varpi := \|\hat{\nabla}_{\mathbf{x}}f_{t+1}(\mathbf{z}_{t+1}; \mathcal{B}_{t+1}) + \hat{\nabla}_{\mathbf{xy}}^2 g_{t+1}(\mathbf{z}_{t+1}; \bar{\mathcal{B}}_{t+1}) - \hat{\nabla}_{\mathbf{x}}f_{t+1}(\mathbf{z}_t; \mathcal{B}_{t+1}) - \hat{\nabla}_{\mathbf{xy}}^2 g_{t+1}(\mathbf{z}_t; \bar{\mathcal{B}}_{t+1})\|^2,$$

where $\hat{\nabla}_{\mathbf{x}} f_{t+1}$ and $\hat{\nabla}^2_{\mathbf{xy}} g_{t+1}$ are defined in (25b) and (26b), respectively. Then, for the sequence $\{(\mathbf{x}_t, \mathbf{y}_t, \mathbf{v}_t)\}_{t=1}^T$ generated by Algorithm 2, we have

$$
\varpi \leq (12\ell_{f,1}^2 + \frac{9\ell_{g,1}^2}{2\rho_{\mathbf{v}}^2})d_1\|\mathbf{x}_{t+1} - \mathbf{x}_t\|^2 + (12\ell_{f,1}^2 + \frac{9\ell_{g,1}^2}{2\rho_{\mathbf{v}}^2})d_1\|\mathbf{y}_{t+1} - \mathbf{y}_t\|^2
$$

$$
+ \frac{9}{2}d_1\ell_{g,1}^2\|\mathbf{v}_{t+1} - \mathbf{v}_t\|^2 + (3\ell_{f,1}^2 + \frac{3\ell_{g,1}^2}{4\rho_{\mathbf{v}}^2})d_1^2\rho_{\mathbf{s}}^2.
$$

*Proof.* From Lemma D.5, we have

$$
\|\hat{\nabla}_{\mathbf{x}} f_{t+1}(\mathbf{z}_{t+1}; \mathcal{B}_{t+1}) - \hat{\nabla}_{\mathbf{x}} f_{t+1}(\mathbf{z}_t; \mathcal{B}_{t+1})\|^2
$$

$$
\leq 3d_1\ell_{g,1}^2\|\mathbf{z}_{t+1} - \mathbf{z}_t\|^2 + \frac{3}{2}\ell_{f,1}^2 d_1^2\rho_{\mathbf{s}}^2
$$

$$
\leq 6d_1\ell_{f,1}^2\|\mathbf{x}_{t+1} - \mathbf{x}_t\|^2 + 6d_1\ell_{f,1}^2\|\mathbf{y}_{t+1} - \mathbf{y}_t\|^2 + \frac{3}{2}\ell_{f,1}^2 d_1^2\rho_{\mathbf{s}}^2. \tag{189}
$$

Moreover, from (26a), we have

$$
\|\hat{\nabla}^2_{\mathbf{yy}} g_{t+1}(\mathbf{z}_{t+1}; \bar{\mathcal{B}}_{t+1}) - \hat{\nabla}^2_{\mathbf{yy}} g_{t+1}(\mathbf{z}_t; \bar{\mathcal{B}}_{t+1})\|^2
$$

$$
= \frac{1}{4\rho_{\mathbf{v}}^2}\|\hat{\nabla}_{\mathbf{x}} g_{t+1}(\mathbf{x}_{t+1}, \mathbf{y}_{t+1} + \rho_{\mathbf{v}}\mathbf{v}_{t+1}; \bar{\mathcal{B}}_{t+1}) - \hat{\nabla}_{\mathbf{x}} g_{t+1}(\mathbf{x}_t, \mathbf{y}_t - \rho_{\mathbf{v}}\mathbf{v}_t; \bar{\mathcal{B}}_{t+1})\|^2
$$

$$
\leq \frac{3}{4\rho_{\mathbf{v}}^2}d_1\ell_{g,1}^2\|(\mathbf{x}_{t+1}, \mathbf{y}_{t+1} + \rho_{\mathbf{v}}\mathbf{v}_{t+1}) - (\mathbf{x}_t, \mathbf{y}_t - \rho_{\mathbf{v}}\mathbf{v}_t)\|^2 + \frac{3}{8\rho_{\mathbf{v}}^2}\ell_{g,1}^2 d_1^2\rho_{\mathbf{s}}^2
$$

$$
\leq \frac{9}{4\rho_{\mathbf{v}}^2}d_1\ell_{g,1}^2\|\mathbf{x}_{t+1} - \mathbf{x}_t\|^2 + \frac{9}{4\rho_{\mathbf{v}}^2}d_1\ell_{g,1}^2\|\mathbf{y}_{t+1} - \mathbf{y}_t\|^2
$$

$$
+ \frac{9}{4}d_1\ell_{g,1}^2\|\mathbf{v}_{t+1} - \mathbf{v}_t\|^2 + \frac{3}{8\rho_{\mathbf{v}}^2}\ell_{g,1}^2 d_1^2\rho_{\mathbf{s}}^2, \tag{190}
$$

where the first inequality follows from Lemma D.5.

From $\|a + b\|^2 \leq 2\left(\|a\|^2 + \|b\|^2\right)$, we get

$$
\varpi \leq 2\|\hat{\nabla}^2_{\mathbf{xy}} g_{t+1}(\mathbf{z}_{t+1}; \bar{\mathcal{B}}_{t+1}) - \hat{\nabla}^2_{\mathbf{xy}} g_{t+1}(\mathbf{z}_t; \bar{\mathcal{B}}_{t+1})\|^2
$$

$$
+ 2\|\hat{\nabla}_{\mathbf{x}} f_{t+1}(\mathbf{z}_{t+1}; \mathcal{B}_{t+1}) - \hat{\nabla}_{\mathbf{x}} f_{t+1}(\mathbf{z}_t; \mathcal{B}_{t+1})\|^2
$$

$$
\leq (12\ell_{f,1}^2 + \frac{9\ell_{g,1}^2}{2\rho_{\mathbf{v}}^2})d_1\|\mathbf{x}_{t+1} - \mathbf{x}_t\|^2 + (12\ell_{f,1}^2 + \frac{9\ell_{g,1}^2}{2\rho_{\mathbf{v}}^2})d_1\|\mathbf{y}_{t+1} - \mathbf{y}_t\|^2
$$

$$
+ \frac{9}{2}d_1\ell_{g,1}^2\|\mathbf{v}_{t+1} - \mathbf{v}_t\|^2 + (3\ell_{f,1}^2 + \frac{3\ell_{g,1}^2}{4\rho_{\mathbf{v}}^2})d_1^2\rho_{\mathbf{s}}^2,
$$

where the second inequality follows from (189) and (190).

$\square$

**Lemma D.17.** *Suppose Assumptions* B2., B3., D2., *and* D4. *hold. Consider the sequence* $\{(\mathbf{x}_t, \mathbf{y}_t, \mathbf{v}_t)\}_{t=1}^T$ *generated by Algorithm 2. For* $e_t^L$ *defined in* (166c), *we have*

$$
\mathbb{E}\|e_{t+1}^L\|^2 \leq (1 - \eta_{t+1})^2 \mathbb{E}\|e_t^L\|^2 + 36\mathbb{E}\|\nabla_{\mathbf{x}} f_{t+1}(\mathbf{x}_t, \mathbf{y}_t) - \nabla_{\mathbf{x}} f_t(\mathbf{x}_t, \mathbf{y}_t)\|^2
$$

$$
+ \left(18d_1^2\ell_{f,1}^2 + 6(3\ell_{f,1}^2 + \frac{3\ell_{g,1}^2}{4\rho_{\mathbf{v}}^2})d_1^2\right)\rho_{\mathbf{s}}^2 + 18d_1^2\ell_{g,1}^2\frac{\rho_{\mathbf{s}}^2}{\rho_{\mathbf{v}}^2}
$$

$$
+ \frac{18}{\rho_{\mathbf{v}}^2}\mathbb{E}\|\nabla_{\mathbf{x}} g_{t+1}(\mathbf{x}_t, \mathbf{y}_t + \rho_{\mathbf{v}}\mathbf{v}_t) - \nabla_{\mathbf{x}} g_t(\mathbf{x}_t, \mathbf{y}_t + \rho_{\mathbf{v}}\mathbf{v}_t)\|^2
$$

$$
+ \frac{18}{\rho_{\mathbf{v}}^2}\mathbb{E}\|\nabla_{\mathbf{x}} g_{t+1}(\mathbf{x}_t, \mathbf{y}_t - \rho_{\mathbf{v}}\mathbf{v}_t) - \nabla_{\mathbf{x}} g_t(\mathbf{x}_t, \mathbf{y}_t - \rho_{\mathbf{v}}\mathbf{v}_t)\|^2
$$

$$
+ 6(12\ell_{f,1}^2 + \frac{9\ell_{g,1}^2}{2\rho_{\mathbf{v}}^2})d_1\mathbb{E}\|\mathbf{x}_{t+1} - \mathbf{x}_t\|^2 + 6(12\ell_{f,1}^2 + \frac{9\ell_{g,1}^2}{2\rho_{\mathbf{v}}^2})d_1\mathbb{E}\|\mathbf{y}_{t+1} - \mathbf{y}_t\|^2
$$

$$
+ 27d_1\ell_{g,1}^2\mathbb{E}\|\mathbf{v}_{t+1} - \mathbf{v}_t\|^2 + 3(\frac{\hat{\sigma}_{g_{\mathbf{x}}}^2}{b\rho_{\mathbf{v}}^2} + \frac{\hat{\sigma}_{f_{\mathbf{x}}}^2}{b})\eta_{t+1}^2. \tag{191}
$$

*Proof.* According to the definition of $\hat{\mathbf{d}}_t^{\mathbf{x}}$ in Algorithm 2, we have

$$\hat{\mathbf{d}}_{t+1}^{\mathbf{x}} - \hat{\mathbf{d}}_t^{\mathbf{x}} = -\eta_{t+1}\hat{\mathbf{d}}_t^{\mathbf{x}} + \eta_{t+1}(\hat{\nabla}_{\mathbf{x}}f_{t+1}(\mathbf{z}_{t+1};\mathcal{B}_{t+1}) + \hat{\nabla}_{\mathbf{xy}}^2 g_{t+1}(\mathbf{z}_{t+1};\bar{\mathcal{B}}_{t+1}))$$
$$+ (1-\eta_{t+1})\left(\hat{\nabla}_{\mathbf{x}}f_{t+1}(\mathbf{z}_{t+1};\mathcal{B}_{t+1}) + \hat{\nabla}_{\mathbf{xy}}^2 g_{t+1}(\mathbf{z}_{t+1};\bar{\mathcal{B}}_{t+1})\right.$$
$$\left. -\hat{\nabla}_{\mathbf{x}}f_{t+1}(\mathbf{z}_t;\mathcal{B}_{t+1}) - \hat{\nabla}_{\mathbf{xy}}^2 g_{t+1}(\mathbf{z}_t;\bar{\mathcal{B}}_{t+1})\right).$$

Then, we have

$$\mathbb{E}\|\nabla_{\mathbf{x}}f_{t+1,\boldsymbol{\rho}}(\mathbf{z}_{t+1}) + \tilde{\nabla}_{\mathbf{xy}}^2 g_{t+1}(\mathbf{z}_{t+1}) - \hat{\mathbf{d}}_{t+1}^{\mathbf{x}}\|^2$$
$$= \mathbb{E}\|\nabla_{\mathbf{x}}f_{t+1,\boldsymbol{\rho}}(\mathbf{z}_{t+1}) + \tilde{\nabla}_{\mathbf{xy}}^2 g_{t+1}(\mathbf{z}_{t+1}) - \hat{\mathbf{d}}_t^{\mathbf{x}} - (\hat{\mathbf{d}}_{t+1}^{\mathbf{x}} - \hat{\mathbf{d}}_t^{\mathbf{x}})\|^2$$
$$= \mathbb{E}\|\nabla_{\mathbf{x}}f_{t+1,\boldsymbol{\rho}}(\mathbf{z}_{t+1}) + \tilde{\nabla}_{\mathbf{xy}}^2 g_{t+1}(\mathbf{z}_{t+1}) - \hat{\mathbf{d}}_t^{\mathbf{x}} + \eta_{t+1}\hat{\mathbf{d}}_t^{\mathbf{x}}$$
$$- \eta_{t+1}(\hat{\nabla}_{\mathbf{x}}f_{t+1}(\mathbf{z}_{t+1};\mathcal{B}_{t+1}) + \hat{\nabla}_{\mathbf{xy}}^2 g_{t+1}(\mathbf{z}_{t+1};\bar{\mathcal{B}}_{t+1}))$$
$$- (1-\eta_{t+1})\left(\hat{\nabla}_{\mathbf{x}}f_{t+1}(\mathbf{z}_{t+1};\mathcal{B}_{t+1}) + \hat{\nabla}_{\mathbf{xy}}^2 g_{t+1}(\mathbf{z}_{t+1};\bar{\mathcal{B}}_{t+1})\right.$$
$$\left.-\hat{\nabla}_{\mathbf{x}}f_{t+1}(\mathbf{z}_t;\mathcal{B}_{t+1}) - \hat{\nabla}_{\mathbf{xy}}^2 g_{t+1}(\mathbf{z}_t;\bar{\mathcal{B}}_{t+1})\right)\|^2$$
$$= \mathbb{E}\|(1-\eta_{t+1})(\nabla_{\mathbf{x}}f_{t,\boldsymbol{\rho}}(\mathbf{z}_t) + \tilde{\nabla}_{\mathbf{xy}}^2 g_t(\mathbf{z}_t) - \hat{\mathbf{d}}_t^{\mathbf{x}})$$
$$+ \eta_{t+1}(\nabla_{\mathbf{x}}f_{t+1,\boldsymbol{\rho}}(\mathbf{z}_{t+1}) + \tilde{\nabla}_{\mathbf{xy}}^2 g_{t+1}(\mathbf{z}_{t+1}) - \hat{\nabla}_{\mathbf{x}}f_{t+1}(\mathbf{z}_{t+1};\mathcal{B}_{t+1}) - \hat{\nabla}_{\mathbf{xy}}^2 g_{t+1}(\mathbf{z}_{t+1};\bar{\mathcal{B}}_{t+1}))$$
$$+ (1-\eta_{t+1})\left(\nabla_{\mathbf{x}}f_{t+1,\boldsymbol{\rho}}(\mathbf{z}_{t+1}) + \tilde{\nabla}_{\mathbf{xy}}^2 g_{t+1}(\mathbf{z}_{t+1}) - \nabla_{\mathbf{x}}f_{t,\boldsymbol{\rho}}(\mathbf{z}_t) - \tilde{\nabla}_{\mathbf{xy}}^2 g_t(\mathbf{z}_t)\right.$$
$$+\nabla_{\mathbf{x}}f_{t+1,\boldsymbol{\rho}}(\mathbf{z}_t) + \tilde{\nabla}_{\mathbf{xy}}^2 g_{t+1}(\mathbf{z}_t) - \nabla_{\mathbf{x}}f_{t+1,\boldsymbol{\rho}}(\mathbf{z}_t) - \tilde{\nabla}_{\mathbf{xy}}^2 g_{t+1}(\mathbf{z}_t)$$
$$\left.-\hat{\nabla}_{\mathbf{x}}f_{t+1}(\mathbf{z}_{t+1};\mathcal{B}_{t+1}) - \hat{\nabla}_{\mathbf{xy}}^2 g_{t+1}(\mathbf{z}_{t+1};\bar{\mathcal{B}}_{t+1}) + \hat{\nabla}_{\mathbf{x}}f_{t+1}(\mathbf{z}_t;\mathcal{B}_{t+1}) + \hat{\nabla}_{\mathbf{xy}}^2 g_{t+1}(\mathbf{z}_t;\bar{\mathcal{B}}_{t+1})\right)\|^2.$$

Since

$$\mathbb{E}\left[\hat{\nabla}_{\mathbf{x}}f_{t+1}(\mathbf{z}_{t+1};\mathcal{B}_{t+1}) + \hat{\nabla}_{\mathbf{xy}}^2 g_{t+1}(\mathbf{z}_{t+1};\bar{\mathcal{B}}_{t+1})\right] = \nabla_{\mathbf{x}}f_{t+1,\boldsymbol{\rho}}(\mathbf{z}_{t+1}) + \tilde{\nabla}_{\mathbf{xy}}^2 g_{t+1}(\mathbf{z}_{t+1}),$$
$$\mathbb{E}\left[\hat{\nabla}_{\mathbf{x}}f_{t+1}(\mathbf{z}_{t+1};\mathcal{B}_{t+1}) + \hat{\nabla}_{\mathbf{xy}}^2 g_{t+1}(\mathbf{z}_{t+1};\bar{\mathcal{B}}_{t+1}) - \hat{\nabla}_{\mathbf{x}}f_{t+1}(\mathbf{z}_t;\mathcal{B}_{t+1}) - \hat{\nabla}_{\mathbf{xy}}^2 g_{t+1}(\mathbf{z}_t;\bar{\mathcal{B}}_{t+1})\right]$$
$$= \nabla_{\mathbf{x}}f_{t+1,\boldsymbol{\rho}}(\mathbf{z}_{t+1}) + \tilde{\nabla}_{\mathbf{xy}}^2 g_{t+1}(\mathbf{z}_{t+1}) - \nabla_{\mathbf{x}}f_{t+1,\boldsymbol{\rho}}(\mathbf{z}_t) - \tilde{\nabla}_{\mathbf{xy}}^2 g_{t+1}(\mathbf{z}_t),$$

then, we have

$$\mathbb{E}\|\nabla_{\mathbf{x}}f_{t+1,\boldsymbol{\rho}}(\mathbf{z}_{t+1}) + \tilde{\nabla}_{\mathbf{xy}}^2 g_{t+1}(\mathbf{z}_{t+1}) - \hat{\mathbf{d}}_{t+1}^{\mathbf{x}}\|^2$$
$$= (1-\eta_{t+1})^2\mathbb{E}\|\nabla_{\mathbf{x}}f_{t,\boldsymbol{\rho}}(\mathbf{z}_t) + \tilde{\nabla}_{\mathbf{xy}}^2 g_t(\mathbf{z}_t) - \hat{\mathbf{d}}_t^{\mathbf{x}}\|^2$$
$$+ \|\eta_{t+1}(\nabla_{\mathbf{x}}f_{t+1,\boldsymbol{\rho}}(\mathbf{z}_{t+1}) + \tilde{\nabla}_{\mathbf{xy}}^2 g_{t+1}(\mathbf{z}_{t+1}) - \hat{\nabla}_{\mathbf{x}}f_{t+1}(\mathbf{z}_{t+1};\mathcal{B}_{t+1}) - \hat{\nabla}_{\mathbf{xy}}^2 g_{t+1}(\mathbf{z}_{t+1};\bar{\mathcal{B}}_{t+1}))$$
$$+ (1-\eta_{t+1})\left(\nabla_{\mathbf{x}}f_{t+1,\boldsymbol{\rho}}(\mathbf{z}_{t+1}) + \tilde{\nabla}_{\mathbf{xy}}^2 g_{t+1}(\mathbf{z}_{t+1}) - \nabla_{\mathbf{x}}f_{t,\boldsymbol{\rho}}(\mathbf{z}_t) - \tilde{\nabla}_{\mathbf{xy}}^2 g_t(\mathbf{z}_t)\right.$$
$$+\nabla_{\mathbf{x}}f_{t+1,\boldsymbol{\rho}}(\mathbf{z}_t) + \tilde{\nabla}_{\mathbf{xy}}^2 g_{t+1}(\mathbf{z}_t) - \nabla_{\mathbf{x}}f_{t+1,\boldsymbol{\rho}}(\mathbf{z}_t) - \tilde{\nabla}_{\mathbf{xy}}^2 g_{t+1}(\mathbf{z}_t)$$
$$\left.-\hat{\nabla}_{\mathbf{x}}f_{t+1}(\mathbf{z}_{t+1};\mathcal{B}_{t+1}) - \hat{\nabla}_{\mathbf{xy}}^2 g_{t+1}(\mathbf{z}_{t+1};\bar{\mathcal{B}}_{t+1}) + \hat{\nabla}_{\mathbf{x}}f_{t+1}(\mathbf{z}_t;\mathcal{B}_{t+1}) + \hat{\nabla}_{\mathbf{xy}}^2 g_{t+1}(\mathbf{z}_t;\bar{\mathcal{B}}_{t+1})\right)\|^2$$
$$\leq (1-\eta_{t+1})^2\mathbb{E}\|\nabla_{\mathbf{x}}f_{t,\boldsymbol{\rho}}(\mathbf{z}_t) + \tilde{\nabla}_{\mathbf{xy}}^2 g_t(\mathbf{z}_t) - \hat{\mathbf{d}}_t^{\mathbf{x}}\|^2$$
$$+ 3(1-\eta_{t+1})^2\mathbb{E}\|\nabla_{\mathbf{x}}f_{t+1,\boldsymbol{\rho}}(\mathbf{z}_{t+1}) + \tilde{\nabla}_{\mathbf{xy}}^2 g_{t+1}(\mathbf{z}_{t+1}) - \nabla_{\mathbf{x}}f_{t,\boldsymbol{\rho}}(\mathbf{z}_t) - \tilde{\nabla}_{\mathbf{xy}}^2 g_t(\mathbf{z}_t)$$
$$+ \nabla_{\mathbf{x}}f_{t+1,\boldsymbol{\rho}}(\mathbf{z}_t) + \tilde{\nabla}_{\mathbf{xy}}^2 g_{t+1}(\mathbf{z}_t) - \nabla_{\mathbf{x}}f_{t+1,\boldsymbol{\rho}}(\mathbf{z}_t) - \tilde{\nabla}_{\mathbf{xy}}^2 g_{t+1}(\mathbf{z}_t)$$
$$- \hat{\nabla}_{\mathbf{x}}f_{t+1}(\mathbf{z}_{t+1};\mathcal{B}_{t+1}) - \hat{\nabla}_{\mathbf{xy}}^2 g_{t+1}(\mathbf{z}_{t+1};\bar{\mathcal{B}}_{t+1}) + \hat{\nabla}_{\mathbf{x}}f_{t+1}(\mathbf{z}_t;\mathcal{B}_{t+1}) + \hat{\nabla}_{\mathbf{xy}}^2 g_{t+1}(\mathbf{z}_t;\bar{\mathcal{B}}_{t+1})\|^2$$
$$+ 3\eta_{t+1}^2\mathbb{E}\|\nabla_{\mathbf{x}}f_{t+1,\boldsymbol{\rho}}(\mathbf{z}_{t+1}) - \hat{\nabla}_{\mathbf{x}}f_{t+1}(\mathbf{z}_{t+1};\mathcal{B}_{t+1})\|^2$$
$$+ 3\eta_{t+1}^2\mathbb{E}\|\tilde{\nabla}_{\mathbf{xy}}^2 g_{t+1}(\mathbf{z}_{t+1}) - \hat{\nabla}_{\mathbf{xy}}^2 g_{t+1}(\mathbf{z}_{t+1};\bar{\mathcal{B}}_{t+1})\|^2, \tag{192}$$

where the second inequality holds by Cauchy-Schwarz inequality.

Note that for the last term on the right-hand side of (192), using (172) and (26b), we have

$$\|\tilde{\nabla}^2_{\mathbf{xy}}g_{t+1}(\mathbf{z}_{t+1}) - \hat{\nabla}^2_{\mathbf{xy}}g_{t+1}(\mathbf{z}_{t+1}; \bar{\mathcal{B}}_{t+1})\|^2$$

$$\leq 2\|\frac{1}{2\rho_{\mathbf{v}}}(\nabla_{\mathbf{x}}g_{t+1,\boldsymbol{\rho}}(\mathbf{x}_{t+1}, \mathbf{y}_{t+1} + \rho_{\mathbf{v}}\mathbf{v}_{t+1}) - \hat{\nabla}_{\mathbf{x}}g_{t+1}(\mathbf{x}_{t+1}, \mathbf{y}_{t+1} + \rho_{\mathbf{v}}\mathbf{v}_{t+1}; \bar{\mathcal{B}}_{t+1}))\|^2$$

$$+ 2\|\frac{1}{2\rho_{\mathbf{v}}}(\hat{\nabla}_{\mathbf{x}}g_{t+1}(\mathbf{x}_{t+1}, \mathbf{y}_{t+1} - \rho_{\mathbf{v}}\mathbf{v}_{t+1}; \bar{\mathcal{B}}_{t+1}) - \nabla_{\mathbf{x}}g_{t+1,\boldsymbol{\rho}}(\mathbf{x}_{t+1}, \mathbf{y}_{t+1} - \rho_{\mathbf{v}}\mathbf{v}_{t+1}))\|^2$$

$$\leq \frac{\hat{\sigma}^2_{g_{\mathbf{x}}}}{\bar{b}\rho^2_{\mathbf{v}}},$$

where the last inequality follows from Assumption D2..

Then, from $\mathbb{E}\|a - \mathbb{E}[a]\|^2 = \mathbb{E}\|a\|^2 - \|\mathbb{E}[a]\|^2$ and Assumption D4., we have

$$\mathbb{E}\|\nabla_{\mathbf{x}}f_{t+1,\boldsymbol{\rho}}(\mathbf{z}_{t+1}) + \tilde{\nabla}^2_{\mathbf{xy}}g_{t+1}(\mathbf{z}_{t+1}) - \hat{\mathbf{d}}^{\mathbf{x}}_{t+1}\|^2$$

$$\leq (1 - \eta_{t+1})^2\mathbb{E}\|\nabla_{\mathbf{x}}f_{t,\boldsymbol{\rho}}(\mathbf{z}_t) + \tilde{\nabla}^2_{\mathbf{xy}}g_t(\mathbf{z}_t) - \hat{\mathbf{d}}^{\mathbf{x}}_t\|^2$$

$$+ 6(1 - \eta_{t+1})^2\mathbb{E}\|\nabla_{\mathbf{x}}f_{t+1,\boldsymbol{\rho}}(\mathbf{z}_t) + \tilde{\nabla}^2_{\mathbf{xy}}g_{t+1}(\mathbf{z}_t) - \nabla_{\mathbf{x}}f_{t,\boldsymbol{\rho}}(\mathbf{z}_t) - \tilde{\nabla}^2_{\mathbf{xy}}g_t(\mathbf{z}_t)\|^2$$

$$+ 6(1 - \eta_{t+1})^2\mathbb{E}\|\hat{\nabla}_{\mathbf{x}}f_{t+1}(\mathbf{z}_{t+1}; \mathcal{B}_{t+1}) + \hat{\nabla}^2_{\mathbf{xy}}g_{t+1}(\mathbf{z}_{t+1}; \bar{\mathcal{B}}_{t+1})$$

$$+ \hat{\nabla}_{\mathbf{x}}f_{t+1}(\mathbf{z}_t; \mathcal{B}_{t+1}) + \hat{\nabla}^2_{\mathbf{xy}}g_{t+1}(\mathbf{z}_t; \bar{\mathcal{B}}_{t+1})\|^2 + 3\eta^2_{t+1}(\frac{\hat{\sigma}^2_{g_{\mathbf{x}}}}{\bar{b}\rho^2_{\mathbf{v}}} + \frac{\hat{\sigma}^2_{f_{\mathbf{x}}}}{b}). \tag{193}$$

Then, from Young's inequality and Lemma D.16, we have

$$\mathbb{E}\|\nabla_{\mathbf{x}}f_{t+1,\boldsymbol{\rho}}(\mathbf{z}_{t+1}) + \tilde{\nabla}^2_{\mathbf{xy}}g_{t+1}(\mathbf{z}_{t+1}) - \hat{\mathbf{d}}^{\mathbf{x}}_{t+1}\|^2$$

$$\leq (1 - \eta_{t+1})^2\mathbb{E}\|\nabla_{\mathbf{x}}f_{t,\boldsymbol{\rho}}(\mathbf{z}_t) + \tilde{\nabla}^2_{\mathbf{xy}}g_t(\mathbf{z}_t) - \hat{\mathbf{d}}^{\mathbf{x}}_t\|^2$$

$$+ 12(1 - \eta_{t+1})^2\mathbb{E}\|\nabla_{\mathbf{x}}f_{t+1,\boldsymbol{\rho}}(\mathbf{z}_t) - \nabla_{\mathbf{x}}f_{t,\boldsymbol{\rho}}(\mathbf{z}_t)\|^2$$

$$+ 12(1 - \eta_{t+1})^2\mathbb{E}\|\tilde{\nabla}^2_{\mathbf{xy}}g_{t+1}(\mathbf{z}_t) - \tilde{\nabla}^2_{\mathbf{xy}}g_t(\mathbf{z}_t)\|^2$$

$$+ 6(12\ell^2_{f,1} + \frac{9\ell^2_{g,1}}{2\rho^2_{\mathbf{v}}})d_1\|\mathbf{x}_{t+1} - \mathbf{x}_t\|^2 + 6(12\ell^2_{f,1} + \frac{9\ell^2_{g,1}}{2\rho^2_{\mathbf{v}}})d_1\|\mathbf{y}_{t+1} - \mathbf{y}_t\|^2$$

$$+ 27d_1\ell^2_{g,1}\|\mathbf{v}_{t+1} - \mathbf{v}_t\|^2 + 6(3\ell^2_{f,1} + \frac{3\ell^2_{g,1}}{4\rho^2_{\mathbf{v}}})d^2_1\rho^2_{\mathbf{s}} + 3\eta^2_{t+1}(\frac{\hat{\sigma}^2_{g_{\mathbf{x}}}}{\bar{b}\rho^2_{\mathbf{v}}} + \frac{\hat{\sigma}^2_{f_{\mathbf{x}}}}{b}). \tag{194}$$

For the third term on the right-hand side of (193), we have

$$\|\tilde{\nabla}^2_{\mathbf{xy}}g_{t+1}(\mathbf{x}_t, \mathbf{y}_t) - \tilde{\nabla}^2_{\mathbf{xy}}g_t(\mathbf{x}_t, \mathbf{y}_t)\|^2$$

$$\leq \frac{1}{2\rho^2_{\mathbf{v}}}\|\nabla_{\mathbf{x}}g_{t+1,\boldsymbol{\rho}}(\mathbf{x}_t, \mathbf{y}_t + \rho_{\mathbf{v}}\mathbf{v}_t) - \nabla_{\mathbf{x}}g_{t,\boldsymbol{\rho}}(\mathbf{x}_t, \mathbf{y}_t + \rho_{\mathbf{v}}\mathbf{v}_t)\|^2 \tag{195a}$$

$$+ \frac{1}{2\rho^2_{\mathbf{v}}}\|\nabla_{\mathbf{x}}g_{t,\boldsymbol{\rho}}(\mathbf{x}_t, \mathbf{y}_t - \rho_{\mathbf{v}}\mathbf{v}_t) - \nabla_{\mathbf{x}}g_{t+1,\boldsymbol{\rho}}(\mathbf{x}_t, \mathbf{y}_t - \rho_{\mathbf{v}}\mathbf{v}_t)\|^2. \tag{195b}$$

For (195a), we get

$$\|\nabla_{\mathbf{x}}g_{t+1,\boldsymbol{\rho}}(\mathbf{x}_t, \mathbf{y}_t + \rho_{\mathbf{v}}\mathbf{v}_t) - \nabla_{\mathbf{x}}g_{t,\boldsymbol{\rho}}(\mathbf{x}_t, \mathbf{y}_t + \rho_{\mathbf{v}}\mathbf{v}_t)\|^2$$

$$\leq 3\|\nabla_{\mathbf{x}}g_{t+1,\boldsymbol{\rho}}(\mathbf{x}_t, \mathbf{y}_t + \rho_{\mathbf{v}}\mathbf{v}_t) - \nabla_{\mathbf{x}}g_{t+1}(\mathbf{x}_t, \mathbf{y}_t + \rho_{\mathbf{v}}\mathbf{v}_t)\|^2$$

$$+ 3\|\nabla_{\mathbf{x}}g_{t+1}(\mathbf{x}_t, \mathbf{y}_t + \rho_{\mathbf{v}}\mathbf{v}_t) - \nabla_{\mathbf{x}}g_t(\mathbf{x}_t, \mathbf{y}_t + \rho_{\mathbf{v}}\mathbf{v}_t)\|^2$$

$$+ 3\|\nabla_{\mathbf{x}}g_t(\mathbf{x}_t, \mathbf{y}_t + \rho_{\mathbf{v}}\mathbf{v}_t) - \nabla_{\mathbf{x}}g_{t,\boldsymbol{\rho}}(\mathbf{x}_t, \mathbf{y}_t + \rho_{\mathbf{v}}\mathbf{v}_t)\|^2$$

$$\leq 3\|\nabla_{\mathbf{x}}g_t(\mathbf{x}_t, \mathbf{y}_t + \rho_{\mathbf{v}}\mathbf{v}_t) - \nabla_{\mathbf{x}}g_{t+1}(\mathbf{x}_t, \mathbf{y}_t + \rho_{\mathbf{v}}\mathbf{v}_t)\|^2 + \frac{3\rho^2_{\mathbf{s}}d^2_1\ell^2_{g,1}}{2},$$

where the last inequality follows from Eq. (130).

Similary, for (195b), we have

$$\|\nabla_{\mathbf{x}}g_{t,\boldsymbol{\rho}}(\mathbf{x}_t, \mathbf{y}_t - \rho_{\mathbf{v}}\mathbf{v}_t) - \nabla_{\mathbf{x}}g_{t+1,\boldsymbol{\rho}}(\mathbf{x}_t, \mathbf{y}_t - \rho_{\mathbf{v}}\mathbf{v}_t)\|^2$$

$$\leq 3\|\nabla_{\mathbf{x}}g_t(\mathbf{x}_t, \mathbf{y}_t - \rho_{\mathbf{v}}\mathbf{v}_t) - \nabla_{\mathbf{x}}g_{t+1}(\mathbf{x}_t, \mathbf{y}_t - \rho_{\mathbf{v}}\mathbf{v}_t)\| + \frac{3\rho^2_{\mathbf{s}}d^2_1\ell^2_{g,1}}{2}.$$

Substituting these inequalities in (195), we have

$$\|\tilde{\nabla}^2_{\mathbf{xy}} g_{t+1}(\mathbf{x}_t, \mathbf{y}_t) - \tilde{\nabla}^2_{\mathbf{xy}} g_t(\mathbf{x}_t, \mathbf{y}_t)\|^2$$

$$\leq \frac{3}{2\rho^2_{\mathbf{v}}} \|\nabla_{\mathbf{x}} g_t(\mathbf{x}_t, \mathbf{y}_t + \rho_{\mathbf{v}} \mathbf{v}_t) - \nabla_{\mathbf{x}} g_{t+1}(\mathbf{x}_t, \mathbf{y}_t + \rho_{\mathbf{v}} \mathbf{v}_t)\|^2$$

$$+ \frac{3}{2\rho^2_{\mathbf{v}}} \|\nabla_{\mathbf{x}} g_t(\mathbf{x}_t, \mathbf{y}_t - \rho_{\mathbf{v}} \mathbf{v}_t) - \nabla_{\mathbf{x}} g_{t+1}(\mathbf{x}_t, \mathbf{y}_t - \rho_{\mathbf{v}} \mathbf{v}_t)\|^2 + \frac{3\rho^2_{\mathbf{s}} d^2_1 \ell^2_{g,1}}{2\rho^2_{\mathbf{v}}}. \tag{196}$$

For the second term on the right-hand side of (193), we have

$$\|\nabla_{\mathbf{x}} f_{t+1,\boldsymbol{\rho}}(\mathbf{x}_t, \mathbf{y}_t) - \nabla_{\mathbf{x}} f_{t,\boldsymbol{\rho}}(\mathbf{x}_t, \mathbf{y}_t)\|^2$$

$$\leq 3\|\nabla_{\mathbf{x}} f_{t+1,\boldsymbol{\rho}}(\mathbf{x}_t, \mathbf{y}_t) - \nabla_{\mathbf{x}} f_{t+1}(\mathbf{x}_t, \mathbf{y}_t)\|^2$$

$$+ 3\|\nabla_{\mathbf{x}} f_{t+1}(\mathbf{x}_t, \mathbf{y}_t) - \nabla_{\mathbf{x}} f_t(\mathbf{x}_t, \mathbf{y}_t)\|^2$$

$$+ 3\|\nabla_{\mathbf{x}} f_t(\mathbf{x}_t, \mathbf{y}_t) - \nabla_{\mathbf{x}} f_{t,\boldsymbol{\rho}}(\mathbf{x}_t, \mathbf{y}_t)\|^2$$

$$\leq 3\|\nabla_{\mathbf{x}} f_t(\mathbf{x}_t, \mathbf{y}_t) - \nabla_{\mathbf{x}} f_{t+1}(\mathbf{x}_t, \mathbf{y}_t)\|^2 + \frac{3\rho^2_{\mathbf{s}} d^2_1 \ell^2_{f,1}}{2}, \tag{197}$$

where the last inequality follows from Eq. (132).

From (196), (197) and (194), we get

$$\mathbb{E}\|\nabla_{\mathbf{x}} f_{t+1,\boldsymbol{\rho}}(\mathbf{z}_{t+1}) + \tilde{\nabla}^2_{\mathbf{xy}} g_{t+1}(\mathbf{z}_{t+1}) - \hat{\mathbf{d}}^{\mathbf{x}}_{t+1}\|^2$$

$$\leq (1 - \eta_{t+1})^2 \mathbb{E}\|\nabla_{\mathbf{x}} f_{t,\boldsymbol{\rho}}(\mathbf{z}_t) + \tilde{\nabla}^2_{\mathbf{xy}} g_t(\mathbf{z}_t) - \hat{\mathbf{d}}^{\mathbf{x}}_t\|^2$$

$$+ 36\mathbb{E}\|\nabla_{\mathbf{x}} f_t(\mathbf{x}_t, \mathbf{y}_t) - \nabla_{\mathbf{x}} f_{t+1}(\mathbf{x}_t, \mathbf{y}_t)\|^2 + 18\rho^2_{\mathbf{s}} d^2_1 \ell^2_{f,1}$$

$$+ \frac{18}{\rho^2_{\mathbf{v}}} \mathbb{E}\|\nabla_{\mathbf{x}} g_t(\mathbf{x}_t, \mathbf{y}_t + \rho_{\mathbf{v}} \mathbf{v}_t) - \nabla_{\mathbf{x}} g_{t+1}(\mathbf{x}_t, \mathbf{y}_t + \rho_{\mathbf{v}} \mathbf{v}_t)\|^2$$

$$+ \frac{18}{\rho^2_{\mathbf{v}}} \mathbb{E}\|\nabla_{\mathbf{x}} g_t(\mathbf{x}_t, \mathbf{y}_t - \rho_{\mathbf{v}} \mathbf{v}_t) - \nabla_{\mathbf{x}} g_{t+1}(\mathbf{x}_t, \mathbf{y}_t - \rho_{\mathbf{v}} \mathbf{v}_t)\|^2 + \frac{18\rho^2_{\mathbf{s}} d^2_1 \ell^2_{g,1}}{\rho^2_{\mathbf{v}}}$$

$$+ 6(12\ell^2_{f,1} + \frac{9\ell^2_{g,1}}{2\rho^2_{\mathbf{v}}}) d_1 \mathbb{E}\|\mathbf{x}_{t+1} - \mathbf{x}_t\|^2 + 6(12\ell^2_{f,1} + \frac{9\ell^2_{g,1}}{2\rho^2_{\mathbf{v}}}) d_1 \mathbb{E}\|\mathbf{y}_{t+1} - \mathbf{y}_t\|^2$$

$$+ 27 d_1 \ell^2_{g,1} \mathbb{E}\|\mathbf{v}_{t+1} - \mathbf{v}_t\|^2 + 6(3\ell^2_{f,1} + \frac{3\ell^2_{g,1}}{4\rho^2_{\mathbf{v}}}) d^2_1 \rho^2_{\mathbf{s}} + 3\eta^2_{t+1}(\frac{\hat{\sigma}^2_{g_{\mathbf{x}}}}{\bar{b}\rho^2_{\mathbf{v}}} + \frac{\hat{\sigma}^2_{f_{\mathbf{x}}}}{b}).$$

□

## D.6 Bounds on the Zeroth-Order Objective Function and its Projected Gradients

**Lemma D.18.** *Suppose Assumptions 2.2, B2., B3., and 2.4 hold. Then, for the sequence of functions* $\{f_{t,\boldsymbol{\rho}}\}^T_{t=1}$ *defined in Eq. (18), we have*

$$\sum_{t=1}^T (f_{t,\boldsymbol{\rho}}(\mathbf{x}_t, \hat{\mathbf{y}}^*_t(\mathbf{x}_t)) - f_{t,\boldsymbol{\rho}}(\mathbf{x}_{t+1}, \hat{\mathbf{y}}^*_t(\mathbf{x}_{t+1})))$$

$$\leq 2M + V_T + \ell_{f,1} \left(1 + 2\frac{\ell_{g,1}}{\mu_g}\right) T \left(\rho^2_{\mathbf{s}} + \rho^2_{\mathbf{r}}\right).$$

*Here, $V_T$ is defined in (11); and $M$ is defined in Assumption 2.4.*

*Proof.* Note that, we have

$$\sum_{t=1}^{T} \left(f_{t,\boldsymbol{\rho}}(\mathbf{x}_t, \hat{\mathbf{y}}_t^*(\mathbf{x}_t)) - f_{t,\boldsymbol{\rho}}(\mathbf{x}_{t+1}, \hat{\mathbf{y}}_t^*(\mathbf{x}_{t+1}))\right)$$

$$= \sum_{t=1}^{T} \left(f_{t,\boldsymbol{\rho}}(\mathbf{x}_t, \hat{\mathbf{y}}_t^*(\mathbf{x}_t)) - f_t(\mathbf{x}_t, \hat{\mathbf{y}}_t^*(\mathbf{x}_t))\right) \tag{198}$$

$$+ \sum_{t=1}^{T} \left(f_t(\mathbf{x}_t, \hat{\mathbf{y}}_t^*(\mathbf{x}_t)) - f_t(\mathbf{x}_{t+1}, \hat{\mathbf{y}}_t^*(\mathbf{x}_{t+1}))\right) \tag{199}$$

$$+ \sum_{t=1}^{T} \left(f_t(\mathbf{x}_{t+1}, \hat{\mathbf{y}}_t^*(\mathbf{x}_{t+1})) - f_{t,\boldsymbol{\rho}}(\mathbf{x}_{t+1}, \hat{\mathbf{y}}_t^*(\mathbf{x}_{t+1}))\right). \tag{200}$$

From (126), we have

$$(198) \le T\frac{\ell_{f,1}(\rho_{\mathbf{s}}^2 + \rho_{\mathbf{r}}^2)}{2}, \tag{201}$$

and

$$(200) \le T\frac{\ell_{f,1}(\rho_{\mathbf{s}}^2 + \rho_{\mathbf{r}}^2)}{2}. \tag{202}$$

Moreover, from Lemma D.6, we have

$$(199) = \sum_{t=1}^{T} \left(f_t(\mathbf{x}_t, \hat{\mathbf{y}}_t^*(\mathbf{x}_t)) - f_t(\mathbf{x}_t, \mathbf{y}_t^*(\mathbf{x}_t))\right)$$

$$+ \sum_{t=1}^{T} \left(f_t(\mathbf{x}_t, \mathbf{y}_t^*(\mathbf{x}_t)) - f_t(\mathbf{x}_{t+1}, \mathbf{y}_t^*(\mathbf{x}_{t+1}))\right)$$

$$+ \sum_{t=1}^{T} \left(f_t(\mathbf{x}_{t+1}, \mathbf{y}_t^*(\mathbf{x}_{t+1})) - f_t(\mathbf{x}_{t+1}, \hat{\mathbf{y}}_t^*(\mathbf{x}_{t+1}))\right)$$

$$\le \ell_{f,1} \sum_{t=1}^{T} \|\hat{\mathbf{y}}_t^*(\mathbf{x}_t) - \mathbf{y}_t^*(\mathbf{x}_t)\| + \ell_{f,1} \sum_{t=1}^{T} \|\hat{\mathbf{y}}_t^*(\mathbf{x}_{t+1}) - \mathbf{y}_t^*(\mathbf{x}_{t+1})\|$$

$$+ \sum_{t=1}^{T} \left(f_t(\mathbf{x}_t, \mathbf{y}_t^*(\mathbf{x}_t)) - f_t(\mathbf{x}_{t+1}, \mathbf{y}_t^*(\mathbf{x}_{t+1}))\right)$$

$$\le 2T\ell_{f,1}\frac{\ell_{g,1}(\rho_{\mathbf{s}}^2 + \rho_{\mathbf{r}}^2)}{\mu_g} + \sum_{t=1}^{T} \left(f_t(\mathbf{x}_t, \mathbf{y}_t^*(\mathbf{x}_t)) - f_t(\mathbf{x}_{t+1}, \mathbf{y}_t^*(\mathbf{x}_{t+1}))\right). \tag{203}$$

For the last term of the above inequality, we have

$$\sum_{t=1}^{T} \left(f_t(\mathbf{x}_t, \mathbf{y}_t^*(\mathbf{x}_t)) - f_t(\mathbf{x}_{t+1}, \mathbf{y}_t^*(\mathbf{x}_{t+1}))\right) = f_1(\mathbf{x}_1, \mathbf{y}_1^*(\mathbf{x}_1)) - f_T(\mathbf{x}_{T+1}, \mathbf{y}_T^*(\mathbf{x}_{T+1}))$$

$$+ \sum_{t=2}^{T} \left(f_t(\mathbf{x}_t, \mathbf{y}_t^*(\mathbf{x}_t)) - f_{t-1}(\mathbf{x}_t, \mathbf{y}_{t-1}^*(\mathbf{x}_t))\right)$$

$$\le 2M + V_T,$$

which implies that

$$(199) \le 2T\ell_{f,1}\frac{\ell_{g,1}(\rho_{\mathbf{s}}^2 + \rho_{\mathbf{r}}^2)}{\mu_g} + 2M + V_T. \tag{204}$$

From (201), (202), and (204), we get the desired result. $\qquad\square$

**Lemma D.19.** *Suppose that Assumptions 2.2 and 2.3 hold. Let $f_{t,\boldsymbol{\rho}}$ be defined as in (18). Then, for $\hat{\mathbf{d}}_t^{\mathbf{x}}$ generated by Algorithm 2, for all $t \in [T]$, we have*

$$\mathbb{E}\left[\left\|\hat{\mathbf{d}}_t^{\mathbf{x}} - \nabla f_{t,\boldsymbol{\rho}}(\mathbf{x}_t, \hat{\mathbf{y}}_t^*(\mathbf{x}_t))\right\|^2\right] \leq 4\mathbb{E}\left[\left\|e_t^L\right\|^2\right] + 4\ell_{g,2}^2 \rho_{\mathbf{v}}^2 p^4$$
$$+ 2M_f^2\left(\mathbb{E}[\hat{\theta}_t^{\mathbf{y}}] + \mathbb{E}[\hat{\theta}_t^{\mathbf{v}}]\right) := A_t, \tag{205}$$

*where $e_t^L$ is defined in Lemma D.13, and $\hat{\theta}_t^{\mathbf{y}}$, $\hat{\theta}_t^{\mathbf{v}}$ are as defined in (142). Additionally, $M_f$ is given in Lemma D.2.*

*Proof.* From $\|a + b\|^2 \leq 2\left(\|a\|^2 + \|b\|^2\right)$, we get

$$\mathbb{E}\left[\left\|\hat{\mathbf{d}}_t^{\mathbf{x}} - \nabla f_{t,\boldsymbol{\rho}}(\mathbf{x}_t, \hat{\mathbf{y}}_t^*(\mathbf{x}_t))\right\|^2\right]$$
$$\leq 2\mathbb{E}\left[\left\|\hat{\mathbf{d}}_t^{\mathbf{x}} - \mathbf{d}_{t,\boldsymbol{\rho}}^{\mathbf{x}}\right\|^2\right] \tag{206a}$$
$$+ 2\mathbb{E}\left[\left\|\mathbf{d}_{t,\boldsymbol{\rho}}^{\mathbf{x}} - \nabla f_{t,\boldsymbol{\rho}}(\mathbf{x}_t, \hat{\mathbf{y}}_t^*(\mathbf{x}_t))\right\|^2\right], \tag{206b}$$

where $\mathbf{d}_{t,\boldsymbol{\rho}}^{\mathbf{x}}$ is defined in (22c). From Lemma D.13, we have

$$(206a) \leq 4\mathbb{E}\left[\left\|e_t^L\right\|^2\right] + 4\ell_{g,2}^2 \rho_{\mathbf{v}}^2 p^4. \tag{207}$$

Moreover, from Eq. (122a), we get

$$(206b) \leq 2M_f^2\left(\mathbb{E}[\hat{\theta}_t^{\mathbf{y}}] + \mathbb{E}[\hat{\theta}_t^{\mathbf{v}}]\right). \tag{208}$$

Substituting (207) and (208) into (206), we conclude the desired result. $\qquad\square$

**Lemma D.20.** *Suppose Assumptions 2.2, 2.3, and 2.4 hold. Let the sequence of functions $\{f_{t,\boldsymbol{\rho}}\}_{t=1}^T$ be defined in (18), and let $\mathcal{P}_{\mathcal{X},\alpha_t}$ be given in Definition B.1. Then, for any positive choice of step sizes satisfying $\alpha_t \leq 1/4L_f$, for all $t \in [T]$, Algorithm 2 guarantees the following bound:*

$$\sum_{t=1}^T \left(\alpha_t - L_f \alpha_t^2\right) \mathbb{E}\left[\left\|\mathcal{P}_{\mathcal{X},\alpha_t}\left(\mathbf{x}_t; \nabla f_{t,\boldsymbol{\rho}}(\mathbf{x}_t, \mathbf{y}_t^*(\mathbf{x}_t))\right)\right\|^2\right]$$

$$\leq 12M + 6V_T + \sum_{t=1}^T \left(6\alpha_t - 3L_f \alpha_t^2\right) A_t$$

$$+ \sum_{t=1}^T \left(6\ell_{f,1}(1 + 2\frac{\ell_{g,1}}{\mu_g}) + \frac{3\ell_{f,1}\ell_{g,1}}{\mu_g}(\alpha_t - L_f \alpha_t^2)\right)(\rho_{\mathbf{s}}^2 + \rho_{\mathbf{r}}^2), \tag{209}$$

*where $V_T$ and $A_t$ are respectively defined in Eq. (11) and Lemma D.19.*

*Proof.* Due to the $L_f$-smoothness of the function $f_t$ by Eq. (39c) in Lemma C.1, $f_{t,\boldsymbol{\rho}}$ is also $L_f$-smooth. Hence,

$$f_{t,\boldsymbol{\rho}}(\mathbf{x}_{t+1}, \hat{\mathbf{y}}_t^*(\mathbf{x}_{t+1})) - f_{t,\boldsymbol{\rho}}(\mathbf{x}_t, \hat{\mathbf{y}}_t^*(\mathbf{x}_t))$$
$$\leq \langle \nabla f_{t,\boldsymbol{\rho}}(\mathbf{x}_t, \hat{\mathbf{y}}_t^*(\mathbf{x}_t)), \mathbf{x}_{t+1} - \mathbf{x}_t\rangle + \frac{L_f}{2}\|\mathbf{x}_{t+1} - \mathbf{x}_t\|^2$$
$$= -\alpha_t \left\langle \nabla f_{t,\boldsymbol{\rho}}(\mathbf{x}_t, \hat{\mathbf{y}}_t^*(\mathbf{x}_t)), \mathcal{P}_{\mathcal{X},\alpha_t}\left(\mathbf{x}_t; \hat{\mathbf{d}}_t^{\mathbf{x}}\right)\right\rangle + \frac{L_f \alpha_t^2}{2}\left\|\mathcal{P}_{\mathcal{X},\alpha_t}\left(\mathbf{x}_t; \hat{\mathbf{d}}_t^{\mathbf{x}}\right)\right\|^2. \tag{210}$$

For the first term on the R.H.S of Eq. (210), we have that

$$-\mathbb{E}\left\langle \nabla f_{t,\boldsymbol{\rho}}(\mathbf{x}_t, \hat{\mathbf{y}}_t^*(\mathbf{x}_t)), \mathcal{P}_{\mathcal{X},\alpha_t}\left(\mathbf{x}_t; \hat{\mathbf{d}}_t^{\mathbf{x}}\right)\right\rangle$$

$$= -\mathbb{E}\left\langle \hat{\mathbf{d}}_t^{\mathbf{x}}, \mathcal{P}_{\mathcal{X},\alpha_t}\left(\mathbf{x}_t; \hat{\mathbf{d}}_t^{\mathbf{x}}\right)\right\rangle$$

$$-\mathbb{E}\left\langle \nabla f_{t,\boldsymbol{\rho}}(\mathbf{x}_t, \hat{\mathbf{y}}_t^*(\mathbf{x}_t)) - \hat{\mathbf{d}}_t^{\mathbf{x}}, \mathcal{P}_{\mathcal{X},\alpha_t}\left(\mathbf{x}_t; \hat{\mathbf{d}}_t^{\mathbf{x}}\right)\right\rangle$$

$$\leq -\frac{1}{2}\mathbb{E}\left[\left\|\mathcal{P}_{\mathcal{X},\alpha_t}\left(\mathbf{x}_t; \hat{\mathbf{d}}_t^{\mathbf{x}}\right)\right\|^2\right] + \frac{1}{2}\mathbb{E}\left[\left\|\hat{\mathbf{d}}_t^{\mathbf{x}} - \nabla f_{t,\boldsymbol{\rho}}(\mathbf{x}_t, \hat{\mathbf{y}}_t^*(\mathbf{x}_t))\right\|^2\right]$$

$$\leq -\frac{1}{2}\mathbb{E}\left[\left\|\mathcal{P}_{\mathcal{X},\alpha_t}\left(\mathbf{x}_t; \hat{\mathbf{d}}_t^{\mathbf{x}}\right)\right\|^2\right] + \frac{A_t}{2}, \tag{211}$$

where the first inequality follows from Lemma B.7; the last inequality follows from Lemma D.19. Plugging the bound (211) into (210), we have that

$$\mathbb{E}\left[f_{t,\boldsymbol{\rho}}(\mathbf{x}_{t+1}, \hat{\mathbf{y}}_t^*(\mathbf{x}_{t+1})) - f_{t,\boldsymbol{\rho}}(\mathbf{x}_t, \hat{\mathbf{y}}_t^*(\mathbf{x}_t))\right]$$
$$\leq \frac{(L_f\alpha_t^2 - \alpha_t)}{2}\mathbb{E}\left[\left\|\mathcal{P}_{\mathcal{X},\alpha_t}\left(\mathbf{x}_t; \hat{\mathbf{d}}_t^{\mathbf{x}}\right)\right\|^2\right] + \frac{\alpha_t A_t}{2},$$

which can be rearranged into

$$(\alpha_t - L_f\alpha_t^2)\mathbb{E}\left[\left\|\mathcal{P}_{\mathcal{X},\alpha_t}\left(\mathbf{x}_t; \hat{\mathbf{d}}_t^{\mathbf{x}}\right)\right\|^2\right]$$
$$\leq 2\mathbb{E}\left[f_{t,\boldsymbol{\rho}}(\mathbf{x}_t, \hat{\mathbf{y}}_t^*(\mathbf{x}_t)) - f_{t,\boldsymbol{\rho}}(\mathbf{x}_{t+1}, \hat{\mathbf{y}}_t^*(\mathbf{x}_{t+1}))\right] + \alpha_t A_t. \tag{212}$$

In addition, we have

$$\mathbb{E}\left[\left\|\mathcal{P}_{\mathcal{X},\alpha_t}\left(\mathbf{x}_t; \nabla f_{t,\boldsymbol{\rho}}(\mathbf{x}_t, \mathbf{y}_t^*(\mathbf{x}_t))\right)\right\|^2\right]$$

$$\leq 3\mathbb{E}\left[\left\|\mathcal{P}_{\mathcal{X},\alpha_t}\left(\mathbf{x}_t; \hat{\mathbf{d}}_t^{\mathbf{x}}\right) - \mathcal{P}_{\mathcal{X},\alpha_t}\left(\mathbf{x}_t; \nabla f_{t,\boldsymbol{\rho}}(\mathbf{x}_t, \hat{\mathbf{y}}_t^*(\mathbf{x}_t))\right)\right\|^2\right]$$

$$+ 3\mathbb{E}\left[\left\|\mathcal{P}_{\mathcal{X},\alpha_t}\left(\mathbf{x}_t; \nabla f_{t,\boldsymbol{\rho}}(\mathbf{x}_t, \hat{\mathbf{y}}_t^*(\mathbf{x}_t))\right) - \mathcal{P}_{\mathcal{X},\alpha_t}\left(\mathbf{x}_t; \nabla f_{t,\boldsymbol{\rho}}(\mathbf{x}_t, \mathbf{y}_t^*(\mathbf{x}_t))\right)\right\|^2\right]$$

$$+ 3\mathbb{E}\left[\left\|\mathcal{P}_{\mathcal{X},\alpha_t}\left(\mathbf{x}_t; \hat{\mathbf{d}}_t^{\mathbf{x}}\right)\right\|^2\right]$$

$$\leq 3\mathbb{E}\left[\left\|\hat{\mathbf{d}}_t^{\mathbf{x}} - \nabla f_{t,\boldsymbol{\rho}}(\mathbf{x}_t, \hat{\mathbf{y}}_t^*(\mathbf{x}_t))\right\|^2\right]$$

$$+ 3\mathbb{E}\left[\left\|\nabla f_{t,\boldsymbol{\rho}}(\mathbf{x}_t, \hat{\mathbf{y}}_t^*(\mathbf{x}_t)) - \nabla f_{t,\boldsymbol{\rho}}(\mathbf{x}_t, \mathbf{y}_t^*(\mathbf{x}_t))\right\|^2\right]$$

$$+ 3\mathbb{E}\left[\left\|\mathcal{P}_{\mathcal{X},\alpha_t}\left(\mathbf{x}_t; \hat{\mathbf{d}}_t^{\mathbf{x}}\right)\right\|^2\right],$$

where the second inequaliy follows from non-expansiveness of the projection operator. Then, from Lemma D.19 and Assumption B2., we have

$$\mathbb{E}\left[\left\|\mathcal{P}_{\mathcal{X},\alpha_t}\left(\mathbf{x}_t; \nabla f_{t,\boldsymbol{\rho}}(\mathbf{x}_t, \mathbf{y}_t^*(\mathbf{x}_t))\right)\right\|^2\right]$$

$$\leq 3A_t + 3\ell_{f,1}\mathbb{E}\left[\left\|\hat{\mathbf{y}}_t^*(\mathbf{x}_t) - \mathbf{y}_t^*(\mathbf{x}_t)\right\|^2\right] + 3\mathbb{E}\left[\left\|\mathcal{P}_{\mathcal{X},\alpha_t}\left(\mathbf{x}_t; \hat{\mathbf{d}}_t^{\mathbf{x}}\right)\right\|^2\right]$$

$$\leq 3A_t + 3\ell_{f,1}\frac{\ell_{g,1}(\rho_{\mathbf{s}}^2 + \rho_{\mathbf{r}}^2)}{\mu_g} + 3\mathbb{E}\left[\left\|\mathcal{P}_{\mathcal{X},\alpha_t}\left(\mathbf{x}_t; \hat{\mathbf{d}}_t^{\mathbf{x}}\right)\right\|^2\right], \tag{213}$$

where the last inequality is by Lemma D.6.

Combining (212) and (213) and summing over $t = 1$ to $T$, we have

$$\sum_{t=1}^{T} \left(\alpha_t - L_f \alpha_t^2\right) \mathbb{E}\left[\|\mathcal{P}_{\mathcal{X},\alpha_t}\left(\mathbf{x}_t; \nabla f_{t,\boldsymbol{\rho}}(\mathbf{x}_t, \mathbf{y}_t^*(\mathbf{x}_t)))\right)\|^2\right]$$

$$\leq 6 \sum_{t=1}^{T} \left(f_{t,\boldsymbol{\rho}}(\mathbf{x}_t, \hat{\mathbf{y}}_t^*(\mathbf{x}_t)) - f_{t,\boldsymbol{\rho}}(\mathbf{x}_{t+1}, \hat{\mathbf{y}}_t^*(\mathbf{x}_{t+1}))\right)$$

$$+ \frac{3\ell_{f,1}\ell_{g,1}}{\mu_g}(\rho_{\mathbf{s}}^2 + \rho_{\mathbf{r}}^2) \sum_{t=1}^{T} \left(\alpha_t - L_f \alpha_t^2\right) + 3 \sum_{t=1}^{T} \left(2\alpha_t - L_f \alpha_t^2\right) A_t$$

$$\leq 12M + 6V_T + 6\ell_{f,1}\left(1 + 2\frac{\ell_{g,1}}{\mu_g}\right) T\left(\rho_{\mathbf{s}}^2 + \rho_{\mathbf{r}}^2\right)$$

$$+ \frac{3\ell_{f,1}\ell_{g,1}}{\mu_g}(\rho_{\mathbf{s}}^2 + \rho_{\mathbf{r}}^2) \sum_{t=1}^{T} \left(\alpha_t - L_f \alpha_t^2\right) + 3 \sum_{t=1}^{T} \left(2\alpha_t - L_f \alpha_t^2\right) A_t,$$

where the second inequality is due to Lemma D.18. □

**Lemma D.21.** *Let the sequence $\{(\mathbf{x}_t, \mathbf{y}_t, \mathbf{v}_t)\}_{t=1}^{T}$ be generated by Algorithm 2.*

*(a) Then, we have*

$$\|\mathbf{y}_{t+1} - \mathbf{y}_t\|^2 \leq 2\beta_t^2 \|e_t^{g\rho}\|^2 + 2\beta_t^2 \|\nabla_{\mathbf{y}} g_{t,\boldsymbol{\rho}}(\mathbf{x}_t, \mathbf{y}_t)\|^2,$$

*where $e_t^{g\rho}$ is defined in (138).*

*(b) Suppose Assumptions 2.2, B2. and B3. hold. Then, we have*

$$\|\mathbf{x}_{t+1} - \mathbf{x}_t\|^2 \leq 4\alpha_t^2 \|\mathcal{P}_{\mathcal{X},\alpha_t}\left(\mathbf{x}_t; \nabla f_{t,\boldsymbol{\rho}}(\mathbf{x}_t, \mathbf{y}_t^*(\mathbf{x}_t)))\right)\|^2$$
$$+ \frac{4\ell_{f,1}\ell_{g,1}\alpha_t^2(\rho_{\mathbf{s}}^2 + \rho_{\mathbf{r}}^2)}{\mu_g} + 2A_t\alpha_t^2, \tag{214}$$

*where $A_t$ is defined in (205).*

*(c) Suppose Assumptions B1., B2. and B3. hold. Then, we have*

$$\|\mathbf{v}_{t+1} - \mathbf{v}_t\|^2 \leq 2\delta_t^2 \|e_t^M\|^2 + 3d_2^2 \ell_{f,1}^2 \delta_t^2 \rho_{\mathbf{r}}^2$$
$$+ (12\ell_{f,0}^2 + 6\ell_{g,1}^2 p^2)\delta_t^2 + 6\ell_{g,1}^2 \frac{\delta_t^2}{\rho_{\mathbf{v}}^2}\hat{\theta}_t^{\mathbf{y}},$$

*where $e_t^M$ and $\hat{\theta}_t^{\mathbf{y}}$ are defined in (154) and (142), respectively.*

*Proof.* **For part (a):** From Algorithm 2, we have

$$\|\mathbf{y}_{t+1} - \mathbf{y}_t\|^2 = \beta_t^2 \|\hat{\mathbf{d}}_t^{\mathbf{y}}\|^2$$
$$\leq 2\beta_t^2 \|\hat{\mathbf{d}}_t^{\mathbf{y}} - \nabla_{\mathbf{y}} g_{t,\boldsymbol{\rho}}(\mathbf{x}_t, \mathbf{y}_t)\|^2 + 2\beta_t^2 \|\nabla_{\mathbf{y}} g_{t,\boldsymbol{\rho}}(\mathbf{x}_t, \mathbf{y}_t)\|^2$$
$$= 2\beta_t^2 \|e_t^{g\rho}\|^2 + 2\beta_t^2 \|\nabla_{\mathbf{y}} g_{t,\boldsymbol{\rho}}(\mathbf{x}_t, \mathbf{y}_t)\|^2. \tag{215}$$

**For part (b):**

From the update rule in Algorithm 2, we obtain

$$
\begin{aligned}
\|\mathbf{x}_t - \mathbf{x}_{t+1}\|^2 &= \alpha_t^2 \left\| \mathcal{P}_{\mathcal{X},\alpha_t}\left(\mathbf{x}_t; \hat{\mathbf{d}}_t^{\mathbf{x}}\right) \right\|^2 \\
&\leq 2\alpha_t^2 \left( \|\mathcal{P}_{\mathcal{X},\alpha_t}\left(\mathbf{x}_t; \nabla f_{t,\boldsymbol{\rho}}(\mathbf{x}_t, \hat{\mathbf{y}}_t^*(\mathbf{x}_t)))\right)\|^2 \right. \\
&\quad \left. + \left\| \mathcal{P}_{\mathcal{X},\alpha_t}\left(\mathbf{x}_t; \hat{\mathbf{d}}_t^{\mathbf{x}}\right) - \mathcal{P}_{\mathcal{X},\alpha_t}\left(\mathbf{x}_t; \nabla f_{t,\boldsymbol{\rho}}(\mathbf{x}_t, \hat{\mathbf{y}}_t^*(\mathbf{x}_t)))\right) \right\|^2 \right) \\
&\leq 2\alpha_t^2 \left( \|\mathcal{P}_{\mathcal{X},\alpha_t}\left(\mathbf{x}_t; \nabla f_{t,\boldsymbol{\rho}}(\mathbf{x}_t, \hat{\mathbf{y}}_t^*(\mathbf{x}_t)))\right)\|^2 \right. \\
&\quad \left. + \left\| \hat{\mathbf{d}}_t^{\mathbf{x}} - \nabla f_{t,\boldsymbol{\rho}}(\mathbf{x}_t, \hat{\mathbf{y}}_t^*(\mathbf{x}_t)) \right\|^2 \right) \\
&\leq 2\alpha_t^2 \left( \|\mathcal{P}_{\mathcal{X},\alpha_t}\left(\mathbf{x}_t; \nabla f_{t,\boldsymbol{\rho}}(\mathbf{x}_t, \hat{\mathbf{y}}_t^*(\mathbf{x}_t)))\right)\|^2 + A_t \right),
\end{aligned}
\tag{216}
$$

where the first inequality is by $(a+b)^2 \leq 2a^2 + 2b^2$; the second inequality follows from non-expansiveness of the projection operator; and the last inequality follows from Lemma D.19.

The first term in the above inequality can be bounded as

$$
\begin{aligned}
&\|\mathcal{P}_{\mathcal{X},\alpha_t}\left(\mathbf{x}_t; \nabla f_{t,\boldsymbol{\rho}}(\mathbf{x}_t, \hat{\mathbf{y}}_t^*(\mathbf{x}_t)))\right)\|^2 \\
&\leq 2 \|\mathcal{P}_{\mathcal{X},\alpha_t}\left(\mathbf{x}_t; \nabla f_{t,\boldsymbol{\rho}}(\mathbf{x}_t, \hat{\mathbf{y}}_t^*(\mathbf{x}_t)))\right) - \mathcal{P}_{\mathcal{X},\alpha_t}\left(\mathbf{x}_t; \nabla f_{t,\boldsymbol{\rho}}(\mathbf{x}_t, \mathbf{y}_t^*(\mathbf{x}_t)))\right)\|^2 \\
&\quad + 2 \|\mathcal{P}_{\mathcal{X},\alpha_t}\left(\mathbf{x}_t; \nabla f_{t,\boldsymbol{\rho}}(\mathbf{x}_t, \mathbf{y}_t^*(\mathbf{x}_t)))\right)\|^2 \\
&\leq 2 \|\nabla f_{t,\boldsymbol{\rho}}(\mathbf{x}_t, \hat{\mathbf{y}}_t^*(\mathbf{x}_t)) - \nabla f_{t,\boldsymbol{\rho}}(\mathbf{x}_t, \mathbf{y}_t^*(\mathbf{x}_t))\|^2 \\
&\quad + 2 \|\mathcal{P}_{\mathcal{X},\alpha_t}\left(\mathbf{x}_t; \nabla f_{t,\boldsymbol{\rho}}(\mathbf{x}_t, \mathbf{y}_t^*(\mathbf{x}_t)))\right)\|^2 \\
&\leq 2\ell_{f,1}\|\hat{\mathbf{y}}_t^*(\mathbf{x}_t) - \mathbf{y}_t^*(\mathbf{x}_t)\|^2 + 2 \|\mathcal{P}_{\mathcal{X},\alpha_t}\left(\mathbf{x}_t; \nabla f_{t,\boldsymbol{\rho}}(\mathbf{x}_t, \mathbf{y}_t^*(\mathbf{x}_t)))\right)\|^2 \\
&\leq 2\ell_{f,1}\frac{\ell_{g,1}(\rho_{\mathbf{s}}^2 + \rho_{\mathbf{r}}^2)}{\mu_g} + 2 \|\mathcal{P}_{\mathcal{X},\alpha_t}\left(\mathbf{x}_t; \nabla f_{t,\boldsymbol{\rho}}(\mathbf{x}_t, \mathbf{y}_t^*(\mathbf{x}_t)))\right)\|^2,
\end{aligned}
\tag{217}
$$

where the last inequality follows from Lemma D.6.

Based on (217) and (216), we get

$$
\|\mathbf{x}_t - \mathbf{x}_{t+1}\|^2 \leq 2\alpha_t^2 \left( 2 \|\mathcal{P}_{\mathcal{X},\alpha_t}\left(\mathbf{x}_t; \nabla f_{t,\boldsymbol{\rho}}(\mathbf{x}_t, \mathbf{y}_t^*(\mathbf{x}_t)))\right)\|^2 + \frac{2\ell_{f,1}\ell_{g,1}(\rho_{\mathbf{s}}^2 + \rho_{\mathbf{r}}^2)}{\mu_g} + A_t \right).
$$

**For part (c):** From the nonexpansiveness of projection, we have

$$
\begin{aligned}
\|\mathbf{v}_{t+1} - \mathbf{v}_t\|^2 &= \|\Pi_{\mathcal{Z}_p}\left[\mathbf{v}_t - \delta_t \hat{\mathbf{d}}_t^{\mathbf{v}}\right] - \Pi_{\mathcal{Z}_p}\left[\mathbf{v}_t\right]\|^2 \\
&\leq \delta_t^2 \|\hat{\mathbf{d}}_t^{\mathbf{v}}\|^2 \\
&\leq 2\delta_t^2 \|\hat{\mathbf{d}}_t^{\mathbf{v}} - \nabla_{\mathbf{y}} f_{t,\boldsymbol{\rho}}(\mathbf{z}_t) - \tilde{\nabla}_{\mathbf{y}}^2 g_t(\mathbf{z}_t)\|^2 + 2\delta_t^2 \|\nabla_{\mathbf{y}} f_{t,\boldsymbol{\rho}}(\mathbf{z}_t) + \tilde{\nabla}_{\mathbf{y}}^2 g_t(\mathbf{z}_t)\|^2 \\
&= 2\delta_t^2 \|e_t^M\|^2 \\
&\quad + 2\delta_t^2 \|\nabla_{\mathbf{y}} f_{t,\boldsymbol{\rho}}(\mathbf{x}_t, \mathbf{y}_t) + \frac{1}{2\rho_{\mathbf{v}}}(\nabla_{\mathbf{y}} g_{t,\boldsymbol{\rho}}(\mathbf{x}_t, \mathbf{y}_t + \rho_{\mathbf{v}}\mathbf{v}_t) - \nabla_{\mathbf{y}} g_{t,\boldsymbol{\rho}}(\mathbf{x}_t, \mathbf{y}_t - \rho_{\mathbf{v}}\mathbf{v}_t))\|^2 \\
&\leq 2\delta_t^2 \|e_t^M\|^2 + 6\delta_t^2 \|\nabla_{\mathbf{y}} f_{t,\boldsymbol{\rho}}(\mathbf{x}_t, \mathbf{y}_t)\|^2 \\
&\quad + \frac{3\delta_t^2}{2\rho_{\mathbf{v}^2}} \|\nabla_{\mathbf{y}} g_{t,\boldsymbol{\rho}}(\mathbf{x}_t, \mathbf{y}_t + \rho_{\mathbf{v}}\mathbf{v}_t)\|^2 + \frac{3\delta_t^2}{2\rho_{\mathbf{v}^2}} \|\nabla_{\mathbf{y}} g_{t,\boldsymbol{\rho}}(\mathbf{x}_t, \mathbf{y}_t - \rho_{\mathbf{v}}\mathbf{v}_t)\|^2,
\end{aligned}
\tag{218}
$$

where the second equality follows from (154).
From Assumption B3., Lemma B.3 and (8), we have

$$
\begin{aligned}
\|\nabla_{\mathbf{y}} g_{t,\boldsymbol{\rho}}(\mathbf{x}_t, \mathbf{y}_t + \rho_{\mathbf{v}}\mathbf{v}_t)\|^2 &\leq \ell_{g,1}^2 \|\mathbf{y}_t + \rho_{\mathbf{v}}\mathbf{v}_t - \hat{\mathbf{y}}_t^*(\mathbf{x}_t)\|^2 \\
&\leq 2\ell_{g,1}^2 \|\rho_{\mathbf{v}}\mathbf{v}_t\|^2 + 2\ell_{g,1}^2 \|\mathbf{y}_t - \hat{\mathbf{y}}_t^*(\mathbf{x}_t)\|^2 \\
&\leq 2\ell_{g,1}^2 \rho_{\mathbf{v}}^2 p^2 + 2\ell_{g,1}^2 \|\mathbf{y}_t - \hat{\mathbf{y}}_t^*(\mathbf{x}_t)\|^2.
\end{aligned}
\tag{219}
$$

Similarly, we get

$$\|\nabla_{\mathbf{y}} g_{t,\boldsymbol{\rho}}(\mathbf{x}_t, \mathbf{y}_t - \rho_{\mathbf{v}}\mathbf{v}_t)\|^2 \le 2\ell_{g,1}^2 \rho_{\mathbf{v}}^2 p^2 + 2\ell_{g,1}^2 \|\mathbf{y}_t - \hat{\mathbf{y}}_t^*(\mathbf{x}_t)\|^2. \tag{220}$$

Moreover, from Eq. (132) and Assumption B1., we have

$$\begin{aligned}
\|\nabla_{\mathbf{y}} f_{t,\boldsymbol{\rho}}(\mathbf{x}_t, \mathbf{y}_t)\|^2 &\le 2\|\nabla_{\mathbf{y}} f_{t,\boldsymbol{\rho}}(\mathbf{x}_t, \mathbf{y}_t) - \nabla_{\mathbf{y}} f_t(\mathbf{x}_t, \mathbf{y}_t)\|^2 + 2\|\nabla_{\mathbf{y}} f_t(\mathbf{x}_t, \mathbf{y}_t)\|^2 \\
&\le \frac{d_2^2 \ell_{f,1}^2 \rho_{\mathbf{r}}^2}{2} + 2\|\nabla_{\mathbf{y}} f_t(\mathbf{x}_t, \mathbf{y}_t)\|^2 \\
&\le \frac{d_2^2 \ell_{f,1}^2 \rho_{\mathbf{r}}^2}{2} + 2\ell_{f,0}^2.
\end{aligned} \tag{221}$$

Substituting (219), (220) and (221), into (218), we get

$$\begin{aligned}
\|\mathbf{v}_{t+1} - \mathbf{v}_t\|^2 &\le 2\delta_t^2 \|e_t^M\|^2 + 3d_2^2 \ell_{f,1}^2 \delta_t^2 \rho_{\mathbf{r}}^2 \\
&\quad + (12\ell_{f,0}^2 + 6\ell_{g,1}^2 p^2)\delta_t^2 + \frac{6\ell_{g,1}^2}{\rho_{\mathbf{v}^2}}\delta_t^2 \|\mathbf{y}_t - \hat{\mathbf{y}}_t^*(\mathbf{x}_t)\|^2.
\end{aligned}$$

$$\square$$

### D.7 Proof of Theorem 3.2

*Proof.* Since $(1 - \gamma_{t+1})^2 \le 1 - \gamma_{t+1}$ and $\gamma_{t+1} = c_\gamma \alpha_t$ in (31), from (139), we have

$$\begin{aligned}
\mathbb{E}\|e_{t+1}^{g_\rho}\|^2 - \mathbb{E}\|e_t^{g_\rho}\|^2 &\le -c_\gamma \alpha_t \mathbb{E}\|e_t^{g_\rho}\|^2 \\
&\quad + 12(1 - \gamma_{t+1})^2 \mathbb{E}\|\nabla_{\mathbf{y}} g_{t-1}(\mathbf{x}_t, \mathbf{y}_t) - \nabla_{\mathbf{y}} g_t(\mathbf{x}_t, \mathbf{y}_t)\|^2 \\
&\quad + 9d_2^2 \ell_{g,1}^2 (1 - \gamma_{t+1})^2 \rho_{\mathbf{r}}^2 + 24d_2^2 \ell_{g,1}^2 (1 - \gamma_{t+1})^2 \mathbb{E}\|\mathbf{x}_{t+1} - \mathbf{x}_t\|^2 \\
&\quad + 24d_2^2 \ell_{g,1}^2 (1 - \gamma_{t+1})^2 \mathbb{E}\|\mathbf{y}_{t+1} - \mathbf{y}_t\|^2 + 2\frac{\hat{\sigma}_{g_{\mathbf{y}}}^2}{b}\gamma_{t+1}^2.
\end{aligned} \tag{222}$$

Since $(1 - \eta_{t+1})^2 \le 1 - \eta_{t+1}$ and $\eta_{t+1} = c_\eta \alpha_t$ in (31), from (191), we have

$$\begin{aligned}
\mathbb{E}\|e_{t+1}^L\|^2 - \mathbb{E}\|e_t^L\|^2 &\le -c_\eta \alpha_t \mathbb{E}\|e_t^L\|^2 + 36\mathbb{E}\|\nabla_{\mathbf{x}} f_{t+1}(\mathbf{x}_t, \mathbf{y}_t) - \nabla_{\mathbf{x}} f_t(\mathbf{x}_t, \mathbf{y}_t)\|^2 \\
&\quad + \left(18d_1^2 \ell_{f,1}^2 + 6(3\ell_{f,1}^2 + \frac{3\ell_{g,1}^2}{4\rho_{\mathbf{v}}^2})d_1^2\right)\rho_{\mathbf{s}}^2 + 18d_1^2 \ell_{g,1}^2 \frac{\rho_{\mathbf{s}}^2}{\rho_{\mathbf{v}}^2} \\
&\quad + \frac{36}{\rho_{\mathbf{v}}^2}\mathbb{E}\|\nabla_{\mathbf{x}} g_{t+1}(\mathbf{x}_t, \mathbf{y}_t + \rho_{\mathbf{v}}\mathbf{v}_t) - \nabla_{\mathbf{x}} g_t(\mathbf{x}_t, \mathbf{y}_t + \rho_{\mathbf{v}}\mathbf{v}_t)\|^2 \\
&\quad + 6(12\ell_{f,1}^2 + \frac{9\ell_{g,1}^2}{2\rho_{\mathbf{v}}^2})d_1 \mathbb{E}\|\mathbf{x}_{t+1} - \mathbf{x}_t\|^2 + 6(12\ell_{f,1}^2 + \frac{9\ell_{g,1}^2}{2\rho_{\mathbf{v}}^2})d_1 \mathbb{E}\|\mathbf{y}_{t+1} - \mathbf{y}_t\|^2 \\
&\quad + 27\ell_{g,1}^2 d_1 \mathbb{E}\|\mathbf{v}_{t+1} - \mathbf{v}_t\|^2 + 3(\frac{\hat{\sigma}_{g_{\mathbf{x}}}^2}{b\rho_{\mathbf{v}}^2} + \frac{\hat{\sigma}_{f_{\mathbf{x}}}^2}{b})\eta_{t+1}^2.
\end{aligned} \tag{223}$$

Since $(1 - \lambda_{t+1})^2 \le 1 - \lambda_{t+1}$ and $\lambda_{t+1} = c_\lambda \alpha_t$ in (31), from (156), we have

$$\begin{aligned}
\mathbb{E}\|e_{t+1}^M\|^2 - \mathbb{E}\|e_t^M\|^2 &\le -c_\lambda \alpha_t \mathbb{E}\|e_t^M\|^2 + 36\mathbb{E}\|\nabla_{\mathbf{y}} f_{t+1}(\mathbf{x}_t, \mathbf{y}_t) - \nabla_{\mathbf{y}} f_t(\mathbf{x}_t, \mathbf{y}_t)\|^2 \\
&\quad + \left(18d_2^2 \ell_{f,1}^2 + 6(3\ell_{f,1}^2 + \frac{3\ell_{g,1}^2}{4\rho_{\mathbf{v}}^2})d_2^2\right)\rho_{\mathbf{r}}^2 + 18d_2^2 \ell_{g,1}^2 \frac{\rho_{\mathbf{r}}^2}{\rho_{\mathbf{v}}^2} \\
&\quad + \frac{36}{\rho_{\mathbf{v}}^2}\mathbb{E}\|\nabla_{\mathbf{y}} g_{t+1}(\mathbf{x}_t, \mathbf{y}_t + \rho_{\mathbf{v}}\mathbf{v}_t) - \nabla_{\mathbf{y}} g_t(\mathbf{x}_t, \mathbf{y}_t + \rho_{\mathbf{v}}\mathbf{v}_t)\|^2 \\
&\quad + 6(12\ell_{f,1}^2 + \frac{9\ell_{g,1}^2}{2\rho_{\mathbf{v}}^2})d_2 \mathbb{E}\|\mathbf{x}_{t+1} - \mathbf{x}_t\|^2 + 6(12\ell_{f,1}^2 + \frac{9\ell_{g,1}^2}{2\rho_{\mathbf{v}}^2})d_2 \mathbb{E}\|\mathbf{y}_{t+1} - \mathbf{y}_t\|^2 \\
&\quad + 27d_2 \ell_{g,1}^2 \mathbb{E}\|\mathbf{v}_{t+1} - \mathbf{v}_t\|^2 + 3(\frac{\hat{\sigma}_{g_{\mathbf{y}}}^2}{b\rho_{\mathbf{v}}^2} + \frac{\hat{\sigma}_{f_{\mathbf{y}}}^2}{b})\lambda_{t+1}^2.
\end{aligned} \tag{224}$$

**Combining the outcomes** .
Let

$$
\begin{aligned}
\Lambda := \; & \Gamma \sum_{t=1}^{T} \left( \mathbb{E}[\hat{\theta}_{t+1}^{\mathbf{y}}] - \mathbb{E}[\hat{\theta}_{t}^{\mathbf{y}}] \right) \\
& + \Upsilon \sum_{t=1}^{T} \left( \mathbb{E}[\hat{\theta}_{t+1}^{\mathbf{v}}] - \mathbb{E}[\hat{\theta}_{t}^{\mathbf{v}}] \right) + \frac{1}{\Phi} \sum_{t=1}^{T} \left( \mathbb{E}\|e_{t+1}^{g\rho}\|^2 - \mathbb{E}\|e_{t}^{g\rho}\|^2 \right) \\
& + \frac{1}{\Psi} \sum_{t=1}^{T} \left( \mathbb{E}\|e_{t+1}^{M}\|^2 - \mathbb{E}\|e_{t}^{M}\|^2 \right) + \frac{1}{\Omega} \sum_{t=1}^{T} \left( \mathbb{E}\|e_{t+1}^{L}\|^2 - \mathbb{E}\|e_{t}^{L}\|^2 \right) .
\end{aligned}
$$

Here, we have

$$
\begin{aligned}
& \Gamma = \frac{11 M_f^2}{L_{\mu_g} c_\beta}, \quad \Upsilon = \frac{52 M_f^2}{L_{\mu_g} c_\delta}, \\
& \Phi = \max \left\{ 240 \frac{d_2 \ell_{g,1}^2}{L_f}, \frac{12 d_2 \ell_{g,1}^2 L_{\mu_g}^2 c_\beta^2}{L_f M_f^2} \right\}, \\
& \Psi = \max \left\{ 720 \frac{d_2 \ell_{f,1}^2}{L_f}, 27 \frac{L_{\mu_g}}{\Upsilon L_f} \ell_{g,1}^2 d_2 c_\delta, \frac{144 d_2 \ell_{f,1}^2 (\mu_g + \ell_{g,1}) c_\beta}{L_f \Gamma}, \frac{36 \ell_{f,1}^2 d_2 L_{\mu_g}^2 c_\beta^2}{L_f M_f^2} \right\}, \\
& \Omega = \max \left\{ 720 \frac{d_1 \ell_{f,1}^2}{L_f}, 27 \frac{L_{\mu_g}}{\Upsilon L_f} \ell_{g,1}^2 d_1 c_\delta, \frac{144 d_1 \ell_{f,1}^2 (\mu_g + \ell_{g,1}) c_\beta}{L_f \Gamma}, \frac{36 \ell_{f,1}^2 d_1 L_{\mu_g}^2 c_\beta^2}{L_f M_f^2} \right\},
\end{aligned}
\tag{225}
$$

with

$$
\begin{aligned}
& c_\beta \geq \sqrt{1760} \frac{L_{\mathbf{y}} M_f}{L_{\mu_g}}, \\
& c_\delta \geq \sqrt{33280(1 + 2 L_{\mathbf{y}}^2)} \frac{\nu M_f}{L_{\mu_g} \mu_g}, \\
& c \geq \left( \max \left\{ 4 L_f, c_\beta (\mu_g + \ell_{g,1}), \frac{48 L_{\mu_g}^2 d_2 \ell_{g,1}^2 c_\beta^2}{M_f^2 \Phi} \right\} \right)^3 + 1, \\
& c_{\mathbf{v}} = \max \left\{ 1080 \ell_{g,1}^2, \frac{324}{M_f^2} \ell_{g,1}^4 c_\delta^2, \frac{54 L_{\mu_g}^2}{M_f^2} \ell_{g,1}^2 c_\beta^2, \frac{216}{\Gamma} \ell_{g,1}^2 c_\beta (\mu_g + \ell_{g,1}) \right\} \left( \frac{d_2}{\Psi} + \frac{d_1}{\Omega} \right), \\
& c_\gamma = \frac{26 M_f^2 \Phi}{L_{\mu_g}^2}, \\
& c_\eta = 26 \Omega, \quad c_\lambda = \frac{10 \Upsilon}{L_{\mu_g}} c_\delta \Psi.
\end{aligned}
\tag{226}
$$

By adding (223), (222), (224), (143), and (180), along with (209) and considering the fact that $\alpha_t$ decreases with respect to $t$, and by applying Lemma D.21, we obtain:

$$\sum_{t=1}^{T} A(\alpha_t, \beta_t, \delta_t, \rho_{\mathbf{v}}) \mathbb{E}\left[\|\mathcal{P}_{\mathcal{X}, \alpha_t}(\mathbf{x}_t; \nabla f_{t, \boldsymbol{\rho}}(\mathbf{x}_t, \mathbf{y}_t^*(\mathbf{x}_t)))\|^2\right] + \Lambda$$

$$\leq 12M + 6V_T + \sum_{t=1}^{T} B(\alpha_t, \beta_t, \delta_t, \rho_{\mathbf{v}}) \mathbb{E}[\hat{\theta}_t^{\mathbf{y}}] + \sum_{t=1}^{T} C(\alpha_t, \beta_t, \delta_t, \rho_{\mathbf{v}}) \mathbb{E}[\hat{\theta}_t^{\mathbf{y}}] \tag{227a}$$

$$+ \frac{4\ell_{f,1}\ell_{g,1}}{\mu_g} \sum_{t=1}^{T} E(\beta_t, \delta_t, \rho_{\mathbf{v}}) \alpha_t^2(\rho_{\mathbf{s}}^2 + \rho_{\mathbf{r}}^2) + \sum_{t=1}^{T} L(\alpha_t, \beta_t, \delta_t, \rho_{\mathbf{v}}) \mathbb{E}\|e_t^L\|^2 \tag{227b}$$

$$+ \frac{8\ell_{g,2}^2 p^4 \Upsilon}{L_{\mu_g}} \sum_{t=1}^{T} \delta_t \rho_{\mathbf{v}}^2 + 4\ell_{g,2}^2 p^4 \sum_{t=1}^{T} \left(6\alpha_t - 3L_f\alpha_t^2 + 2\alpha_t^2 E(\beta_t, \delta_t, \rho_{\mathbf{v}})\right) \rho_{\mathbf{v}}^2 \tag{227c}$$

$$+ \left(\frac{12}{L_{\mu_g}} \frac{\Gamma}{\beta_T} + \frac{48\nu^2}{L_{\mu_g}\mu_g^2} \frac{\Upsilon}{\delta_T}\right) H_{2,T} + \sum_{t=1}^{T} M(\delta_t) \mathbb{E}\|e_t^M\|^2 \tag{227d}$$

$$+ \sum_{t=1}^{T} Q(\beta_t, \rho_{\mathbf{v}}) \mathbb{E}\|e_t^{g\rho}\|^2 + \sum_{t=1}^{T} S(\beta_t, \rho_{\mathbf{v}}) \mathbb{E}\left[\|\nabla_{\mathbf{y}} g_{t,\boldsymbol{\rho}}(\mathbf{x}_t, \mathbf{y}_t)\|^2\right] \tag{227e}$$

$$+ \sum_{t=1}^{T} Z\left(3d_2^2\ell_{f,1}^2\delta_t^2\rho_{\mathbf{r}}^2 + (12\ell_{f,0}^2 + 6\ell_{g,1}^2 p^2)\delta_t^2\right) \tag{227f}$$

$$+ \frac{36}{\Psi} D_{\mathbf{y},T} + \frac{36}{\Omega} D_{\mathbf{x},T} + \frac{12}{\Phi} G_{\mathbf{y},T} + \frac{18}{\Psi\rho_{\mathbf{v}}^2} G_{\mathbf{v},T} + \frac{18}{\Omega\rho_{\mathbf{v}}^2} G_{\mathbf{x},T} \tag{227g}$$

$$+ 2\sum_{t=1}^{T} \frac{\gamma_{t+1}^2}{\Phi} \frac{\hat{\sigma}_{g_{\mathbf{y}}}^2}{b} + 3\left(\frac{\hat{\sigma}_{g_{\mathbf{y}}}^2}{b\rho_{\mathbf{v}}^2} + \frac{\hat{\sigma}_{f_{\mathbf{y}}}^2}{b}\right) \sum_{t=1}^{T+1} \frac{\lambda_{t+1}^2}{\Psi} + 3\left(\frac{\hat{\sigma}_{g_{\mathbf{x}}}^2}{b\rho_{\mathbf{v}}^2} + \frac{\hat{\sigma}_{f_{\mathbf{x}}}^2}{b}\right) \sum_{t=1}^{T} \frac{\eta_{t+1}^2}{\Omega} \tag{227h}$$

$$+ R(\rho_{\mathbf{v}})T\rho_{\mathbf{r}}^2 + \acute{R}(\rho_{\mathbf{v}})T\rho_{\mathbf{s}}^2 + 18T\ell_{g,1}^2\left(\frac{d_1^2\rho_{\mathbf{s}}^2}{\Omega\rho_{\mathbf{v}}^2} + \frac{d_2^2\rho_{\mathbf{r}}^2}{\Psi\rho_{\mathbf{v}}^2}\right) + \sum_{t=1}^{T} D(\alpha_t, \beta_t, \delta_t)\left(\rho_{\mathbf{s}}^2 + \rho_{\mathbf{r}}^2\right). \tag{227i}$$

Here,

$$E(\beta_t, \delta_t, \rho_{\mathbf{v}}) := \frac{4L_{\mathbf{y}}^2}{L_{\mu_g}} \frac{\Gamma}{\beta_t} + \frac{16\nu^2}{L_{\mu_g}\mu_g^2}(2L_{\mathbf{y}}^2 + 1)\frac{\Upsilon}{\delta_t}$$

$$+ 24d_2\frac{\ell_{g,1}^2}{\Phi} + 6(12\ell_{f,1}^2 + \frac{9\ell_{g,1}^2}{2\rho_{\mathbf{v}}^2})(\frac{d_2}{\Psi} + \frac{d_1}{\Omega}),$$

$$A(\alpha_t, \beta_t, \delta_t, \rho_{\mathbf{v}}) := \alpha_t - L_f\alpha_t^2 - 4E(\beta_t, \delta_t, \rho_{\mathbf{v}})\alpha_t^2,$$

$$B(\alpha_t, \beta_t, \delta_t, \rho_{\mathbf{v}}) := -\frac{L_{\mu_g}}{4}\Upsilon\delta_t + 2M_f^2\left(6\alpha_t - 3L_f\alpha_t^2 + 2\alpha_t^2 E(\beta_t, \delta_t, \rho_{\mathbf{v}})\right),$$

$$C(\alpha_t, \beta_t, \delta_t, \rho_{\mathbf{v}}) := -\frac{L_{\mu_g}}{2}\Gamma\beta_t + Z6\ell_{g,1}^2\frac{\delta_t^2}{\rho_{\mathbf{v}}^2}$$

$$+ 2M_f^2\left(6\alpha_t - 3L_f\alpha_t^2 + 2\alpha_t^2 E(\beta_t, \delta_t, \rho_{\mathbf{v}})\right),$$

$$Z := 27\ell_{g,1}^2\left(\frac{d_2}{\Psi} + \frac{d_1}{\Omega}\right). \tag{228}$$

Moreover,

$$M(\delta_t) := -\frac{\lambda_{t+1}}{\Psi} + Z2\delta_t^2 + \frac{8\Upsilon}{L_{\mu_g}}\delta_t,$$

$$D(\alpha_t, \beta_t, \delta_t) := 6\ell_{f,1}(1 + 2\frac{\ell_{g,1}}{\mu_g}) + \frac{3\ell_{f,1}\ell_{g,1}}{\mu_g}(\alpha_t - L_f\alpha_t^2)$$

$$+ \frac{24\ell_{g,1}}{L_{\mu_g}\mu_g}\frac{\Gamma}{\beta_t} + \frac{96\ell_{g,1}\nu^2}{L_{\mu_g}\mu_g^3}\frac{\Upsilon}{\delta_t},$$

$$F(\rho_{\mathbf{v}}) := 24d_2\frac{\ell_{g,1}^2}{\Phi} + (72\ell_{f,1}^2 + \frac{27\ell_{g,1}^2}{\rho_{\mathbf{v}}^2})(\frac{d_2}{\Psi} + \frac{d_1}{\Omega}),$$

$$S(\beta_t, \rho_{\mathbf{v}}) := -\frac{2\beta_t\Gamma}{\mu_g + \ell_{g,1}} + \beta_t^2\Gamma + 2F(\rho_{\mathbf{v}})\beta_t^2, \tag{229}$$

$$Q(\beta_t, \rho_{\mathbf{v}}) := \frac{2}{L_{\mu_g}}\Gamma\beta_t - \frac{\gamma_{t+1}}{\Phi} + 2F(\rho_{\mathbf{v}})\beta_t^2,$$

$$R(\rho_{\mathbf{v}}) := 9d_2^2\frac{\ell_{g,1}^2}{\Phi} + 18d_2^2\frac{\ell_{f,1}^2}{\Psi} + 6(3\ell_{f,1}^2 + \frac{3\ell_{g,1}^2}{4\rho_{\mathbf{v}}^2})\frac{d_2^2}{\Psi},$$

$$\acute{R}(\rho_{\mathbf{v}}) := 18d_1^2\frac{\ell_{f,1}^2}{\Omega} + 6(3\ell_{f,1}^2 + \frac{3\ell_{g,1}^2}{4\rho_{\mathbf{v}}^2})\frac{d_1^2}{\Omega},$$

$$L(\alpha_t, \beta_t, \delta_t, \rho_{\mathbf{v}}) := -\frac{\eta_{t+1}}{\Omega} + 4\left(6\alpha_t - 3L_f\alpha_t^2 + 2\alpha_t^2 E(\beta_t, \delta_t, \rho_{\mathbf{v}})\right).$$

We then provide bounds for the terms in (227a)-(227i).
Note that, we have

$$E(\beta_t, \delta_t, \rho_{\mathbf{v}}) := \frac{4L_{\mathbf{y}}^2}{L_{\mu_g}}\frac{\Gamma}{\beta_t} + \frac{16\nu^2}{L_{\mu_g}\mu_g^2}(2L_{\mathbf{y}}^2 + 1)\frac{\Upsilon}{\delta_t}$$

$$+ 24d_2\frac{\ell_{g,1}^2}{\Phi} + 6(12\ell_{f,1}^2 + \frac{9\ell_{g,1}^2}{2\rho_{\mathbf{v}}^2})(\frac{d_2}{\Psi} + \frac{d_1}{\Omega}),$$

which together with $\beta_t = c_\beta\alpha_t$, $\delta_t = c_\delta\alpha_t$ in (31), we have

$$\alpha_t^2 E(\beta_t, \delta_t, \rho_{\mathbf{v}}) = \frac{4L_{\mathbf{y}}^2}{L_{\mu_g}}\frac{\Gamma\alpha_t^2}{\beta_t} + \frac{16\nu^2}{L_{\mu_g}\mu_g^2}(2L_{\mathbf{y}}^2 + 1)\frac{\Upsilon\alpha_t^2}{\delta_t}$$

$$+ 24d_2\frac{\ell_{g,1}^2}{\Phi}\alpha_t^2 + (72\ell_{f,1}^2\alpha_t^2 + \frac{27\ell_{g,1}^2}{\rho_{\mathbf{v}}^2}\alpha_t^2)(\frac{d_2}{\Psi} + \frac{d_1}{\Omega})$$

$$\leq \frac{44L_{\mathbf{y}}^2}{L_{\mu_g}^2}M_f^2\frac{\alpha_t}{c_\beta^2} + \frac{832\nu^2}{L_{\mu_g}^2\mu_g^2}(1 + 2L_{\mathbf{y}}^2)M_f^2\frac{\alpha_t}{c_\delta^2}$$

$$+ 6\frac{d_2\ell_{g,1}^2}{L_f\Phi}\alpha_t + (\frac{18\ell_{f,1}^2}{L_f}\alpha_t + \frac{27\ell_{g,1}^2}{c_{\mathbf{v}}}\alpha_t)(\frac{d_2}{\Psi} + \frac{d_1}{\Omega})$$

$$\leq \frac{\alpha_t}{8}, \tag{230}$$

where the first inequality is by $\Gamma = \frac{11M_f^2}{L_{\mu_g}c_\beta}$, $\Upsilon = \frac{52M_f^2}{L_{\mu_g}c_\delta}$ in (225), $\rho_{\mathbf{v}}^2 = c_{\mathbf{v}}\alpha_t$ and $\alpha_t \leq 1/4L_f$ in (31);

the second inequality follows from $c_\beta \geq \sqrt{1760\frac{L_{\mathbf{y}}^2M_f^2}{L_{\mu_g}^2}}$, $c_\delta \geq \sqrt{33280\frac{\nu^2 M_f^2}{L_{\mu_g}^2\mu_g^2}(1 + 2L_{\mathbf{y}}^2)}$, in (226);

and $\Phi = 240\frac{d_2\ell_{g,1}^2}{L_f}$, $\Psi = 720\frac{d_2\ell_{f,1}^2}{L_f}$, $\Omega = 720\frac{d_1\ell_{f,1}^2}{L_f}$ and $c_{\mathbf{v}} \geq 1080\ell_{g,1}^2(\frac{d_2}{\Psi} + \frac{d_1}{\Omega})$ in (225).
Moreover, we have

$$A(\alpha_t, \beta_t, \delta_t, \rho_{\mathbf{v}}) = \alpha_t - L_f\alpha_t^2 - 4E(\beta_t, \delta_t, \rho_{\mathbf{v}})\alpha_t^2$$

$$\geq \alpha_t - L_f\alpha_t^2 - \frac{\alpha_t}{2}$$

$$\geq \frac{\alpha_t}{4}, \tag{231}$$

where the last inequality is by $\alpha_t \le 1/4L_f$ in (226).

**Bounding** (227a) .

From $\delta_t = c_\delta \alpha_t$ in (31), we have

$$
\begin{aligned}
B(\alpha_t, \beta_t, \delta_t, \rho_{\mathbf{v}}) &= -\frac{L_{\mu_g}}{4}\Upsilon\delta_t + 2M_f^2\left(6\alpha_t - 3L_f\alpha_t^2 + 2\alpha_t^2 E(\beta_t, \delta_t, \rho_{\mathbf{v}})\right) \\
&\le -\frac{L_{\mu_g}}{4}\Upsilon c_\delta \alpha_t + 12M_f^2\alpha_t - 6M_f^2 L_f \alpha_t^2 + \frac{M_f^2}{2}\alpha_t \\
&\le \left(-\frac{L_{\mu_g}}{4}\Upsilon c_\delta + \frac{25}{2}M_f^2\right)\alpha_t \\
&\le -\frac{1}{2}M_f^2\alpha_t,
\end{aligned}
\tag{232}
$$

where the first inequality follows from (230); the last inequality is by $\Upsilon = \frac{52M_f^2}{L_{\mu_g}c_\delta}$ in (225).

From (228), we obtain

$$
Z = 27\ell_{g,1}^2\left(\frac{d_2}{\Psi} + \frac{d_1}{\Omega}\right).
$$

Thus, from $\beta_t = c_\beta \alpha_t$, $\delta_t = c_\delta \alpha_t$ and $\rho_{\mathbf{v}}^2 = c_{\mathbf{v}}\alpha_t$ in (31), we have

$$
\begin{aligned}
C(\alpha_t, \beta_t, \delta_t, \rho_{\mathbf{v}}) &= -\frac{L_{\mu_g}}{2}\Gamma\beta_t + 162(\frac{d_2}{\Psi} + \frac{d_1}{\Omega})\ell_{g,1}^4\frac{\delta_t^2}{\rho_{\mathbf{v}}^2} \\
&\quad + 2M_f^2\left(6\alpha_t - 3L_f\alpha_t^2 + 2\alpha_t^2 E(\beta_t, \delta_t, \rho_{\mathbf{v}})\right) \\
&\le -\frac{L_{\mu_g}}{2}\Gamma c_\beta \alpha_t + 162(\frac{d_2}{\Psi} + \frac{d_1}{\Omega})\ell_{g,1}^4\frac{c_\delta^2}{c_{\mathbf{v}}}\alpha_t + \frac{9}{2}M_f^2\alpha_t \\
&= -\frac{11}{2}M_f^2\alpha_t + 162(\frac{d_2}{\Psi} + \frac{d_1}{\Omega})\ell_{g,1}^4\frac{c_\delta^2}{c_{\mathbf{v}}}\alpha_t + \frac{9}{2}M_f^2\alpha_t \\
&\le -\frac{1}{2}M_f^2\alpha_t,
\end{aligned}
\tag{233}
$$

where the first inequality follows from (230); the second equality follows from $\Gamma = \frac{11M_f^2}{L_{\mu_g}c_\beta}$ in (225); the last inequality is by $c_{\mathbf{v}} \ge \frac{324}{M_f^2}\ell_{g,1}^4(\frac{d_2}{\Psi} + \frac{d_1}{\Omega})c_\delta^2$.

Thus, from (232) and (233), we get

$$
(227a) \le \mathcal{O}\left(V_T\right).
\tag{234}
$$

**Bounding** (227b) .

From (230), we also obtain

$$
\begin{aligned}
&\frac{4\ell_{f,1}\ell_{g,1}}{\mu_g}\sum_{t=1}^{T}E(\beta_t, \delta_t, \rho_{\mathbf{v}})\alpha_t^2(\rho_{\mathbf{s}}^2 + \rho_{\mathbf{r}}^2) \\
&\le \frac{4\ell_{f,1}\ell_{g,1}}{\mu_g}\sum_{t=1}^{T}\frac{\alpha_t}{8}(\rho_{\mathbf{s}}^2 + \rho_{\mathbf{r}}^2) \\
&= \mathcal{O}\left(\sum_{t=1}^{T}\alpha_t(\rho_{\mathbf{s}}^2 + \rho_{\mathbf{r}}^2)\right).
\end{aligned}
\tag{235}
$$

From (229) and $\eta_{t+1} = c_\eta \alpha_t$ in (31), we have

$$
\begin{aligned}
L(\alpha_t, \beta_t, \delta_t, \rho_{\mathbf{v}}) &= -\frac{\eta_{t+1}}{\Omega} + 4\left(6\alpha_t - 3L_f\alpha_t^2 + 2\alpha_t^2 E(\beta_t, \delta_t, \rho_{\mathbf{v}})\right) \\
&\le -\frac{c_\eta}{\Omega}\alpha_t + 25\alpha_t \\
&\le -\alpha_t,
\end{aligned}
$$

where the last inequality is by $c_\eta \geq 26\Omega$ and (230).

Thus, we get

$$\sum_{t=1}^{T} L(\alpha_t, \beta_t, \delta_t, \rho_{\mathbf{v}})\mathbb{E}\|e_t^L\|^2 \leq 0. \tag{236}$$

From (236) and (235), we have

$$(227b) \leq \mathcal{O}\left(\sum_{t=1}^{T} \alpha_t(\rho_{\mathbf{s}}^2 + \rho_{\mathbf{r}}^2)\right). \tag{237}$$

**Bounding** (227c) .

From $\delta_t = c_\delta \alpha_t$ in (31) and Eq. (230), we have

$$\frac{8\ell_{g,2}^2 p^4 \Upsilon}{L_{\mu_g}}\sum_{t=1}^{T}\delta_t\rho_{\mathbf{v}}^2 + 4\ell_{g,2}^2 p^4\sum_{t=1}^{T}\left(6\alpha_t - 3L_f\alpha_t^2 + 2\alpha_t^2 E(\beta_t, \delta_t, \rho_{\mathbf{v}})\right)\rho_{\mathbf{v}}^2$$

$$\leq \frac{8\ell_{g,2}^2 p^4 \Upsilon}{L_{\mu_g}}\sum_{t=1}^{T}c_\delta\alpha_t\rho_{\mathbf{v}}^2 + 4\ell_{g,2}^2 p^4\sum_{t=1}^{T}\frac{25}{4}\alpha_t\rho_{\mathbf{v}}^2.$$

Thus, from $\rho_{\mathbf{v}}^2 = c_{\mathbf{v}}\alpha_t$ in (31), we have

$$(227c) \leq \mathcal{O}\left(\sum_{t=1}^{T} \alpha_t^2\right). \tag{238}$$

**Bounding** (227d) .

From (228), we have

$$Z = 27\ell_{g,1}^2\left(\frac{d_2}{\Psi} + \frac{d_1}{\Omega}\right). \tag{239}$$

From (229), $\lambda_{t+1} = c_\lambda \alpha_t$ and $\delta_t = c_\delta \alpha_t$ in (31), we have

$$M(\delta_t) = -\frac{\lambda_{t+1}}{\Psi} + Z2\delta_t^2 + \frac{8\Upsilon}{L_{\mu_g}}\delta_t$$

$$= -\frac{c_\lambda\alpha_t}{\Psi} + 27\ell_{g,1}^2\left(\frac{d_2}{\Psi} + \frac{d_1}{\Omega}\right)2c_\delta^2\alpha_t^2 + \frac{8\Upsilon}{L_{\mu_g}}c_\delta\alpha_t$$

$$\leq -\frac{2\Upsilon}{L_{\mu_g}}c_\delta\alpha_t + \frac{27}{4L_f}\ell_{g,1}^2\left(\frac{d_2}{\Psi} + \frac{d_1}{\Omega}\right)2c_\delta^2\alpha_t$$

$$\leq -\frac{\Upsilon}{L_{\mu_g}}c_\delta\alpha_t,$$

where the first inequality is by $c_\lambda \geq \frac{10\Upsilon}{L_{\mu_g}}c_\delta\Psi$ and $\alpha_t \leq 1/4L_f$; the last inequality follows from $\Psi \geq 27\frac{L_{\mu_g}}{\Upsilon L_f}\ell_{g,1}^2 d_2 c_\delta$ and $\Omega \geq 27\frac{L_{\mu_g}}{\Upsilon L_f}\ell_{g,1}^2 d_1 c_\delta$.

Since $\beta_t = c_\beta\alpha_t$ and $\delta_t = c_\delta\alpha_t$ in (31), we get

$$(227d) = \left(\frac{12}{L_{\mu_g}}\frac{\Gamma}{\beta_T} + \frac{48\nu^2}{L_{\mu_g}\mu_g^2}\frac{\Upsilon}{\delta_T}\right)H_{2,T} + \sum_{t=1}^{T}M(\delta_t)\mathbb{E}\|e_t^M\|^2$$

$$\leq \mathcal{O}\left(\frac{H_{2,T}}{\alpha_T}\right). \tag{240}$$

**Bounding** (227e) .

From (229), we have

$$F(\rho_{\mathbf{v}}) = 24d_2\frac{\ell_{g,1}^2}{\Phi} + (72\ell_{f,1}^2 + \frac{27\ell_{g,1}^2}{\rho_{\mathbf{v}}^2})(\frac{d_2}{\Psi} + \frac{d_1}{\Omega}). \tag{241}$$

From (229), $\gamma_{t+1} = c_\gamma \alpha_t$, $\beta_t = c_\beta \alpha_t$ in (31), we have

$$
\begin{aligned}
Q(\beta_t, \rho_{\mathbf{v}}) &= -\frac{\gamma_{t+1}}{\Phi} + \frac{2}{L_{\mu_g}}\Gamma\beta_t + 2F(\rho_{\mathbf{v}})\beta_t^2 \\
&= -\frac{c_\gamma \alpha_t}{\Phi} + \frac{22M_f^2}{L_{\mu_g}^2}\alpha_t + \left(24d_2\frac{\ell_{g,1}^2}{\Phi} + (72\ell_{f,1}^2 + \frac{27\ell_{g,1}^2}{c_{\mathbf{v}}\alpha_t})(\frac{d_2}{\Psi} + \frac{d_1}{\Omega})\right)2c_\beta^2\alpha_t^2 \\
&\leq -\frac{4M_f^2}{L_{\mu_g}^2}\alpha_t + \left(24d_2\frac{\ell_{g,1}^2}{\Phi}\alpha_t^2 + (72\ell_{f,1}^2\alpha_t^2 + \frac{27\ell_{g,1}^2\alpha_t}{c_{\mathbf{v}}})(\frac{d_2}{\Psi} + \frac{d_1}{\Omega})\right)2c_\beta^2 \\
&\leq -\frac{4M_f^2}{L_{\mu_g}^2}\alpha_t + \left(\frac{6d_2}{L_f}\frac{\ell_{g,1}^2}{\Phi}\alpha_t + (\frac{18}{L_f}\ell_{f,1}^2\alpha_t + \frac{27\ell_{g,1}^2\alpha_t}{c_{\mathbf{v}}})(\frac{d_2}{\Psi} + \frac{d_1}{\Omega})\right)2c_\beta^2 \\
&\leq -\frac{M_f^2}{L_{\mu_g}^2}\alpha_t,
\end{aligned}
\tag{242}
$$

where the first equality is by $\Gamma = \frac{11M_f^2}{L_{\mu_g}c_\beta}$ and $\rho_{\mathbf{v}}^2 = c_{\mathbf{v}}\alpha_t$; the first inequality follows from $c_\gamma \geq \frac{26M_f^2\Phi}{L_{\mu_g}^2}$; the second inequality is by $\alpha_t \leq 1/4L_f$; the last inequality follows from $c_{\mathbf{v}} \geq \frac{54L_{\mu_g}^2}{M_f^2}\ell_{g,1}^2(\frac{d_2}{\Psi} + \frac{d_1}{\Omega})c_\beta^2$, $\Phi \geq \frac{12d_2\ell_{g,1}^2 L_{\mu_g}^2 c_\beta^2}{L_f M_f^2}$, and $\Psi \geq \frac{36\ell_{f,1}^2 d_2 L_{\mu_g}^2 c_\beta^2}{L_f M_f^2}$, and $\Omega \geq \frac{36\ell_{f,1}^2 d_1 L_{\mu_g}^2 c_\beta^2}{L_f M_f^2}$.

From (229), $\beta_t = c_\beta \alpha_t$, $\rho_{\mathbf{v}}^2 = c_{\mathbf{v}}\alpha_t$ in (31) and (241), we have

$$
\begin{aligned}
S(\beta_t, \rho_{\mathbf{v}}) &= -\frac{2\beta_t\Gamma}{\mu_g + \ell_{g,1}} + \beta_t^2\Gamma + 2F(\rho_{\mathbf{v}})\beta_t^2 \\
&= -\frac{2c_\beta\alpha_t\Gamma}{\mu_g + \ell_{g,1}} + c_\beta^2\alpha_t^2\Gamma + \left(24d_2\frac{\ell_{g,1}^2}{\Phi} + (72\ell_{f,1}^2 + \frac{27\ell_{g,1}^2}{c_{\mathbf{v}}\alpha_t})(\frac{d_2}{\Psi} + \frac{d_1}{\Omega})\right)2c_\beta^2\alpha_t^2 \\
&\leq -\frac{c_\beta\alpha_t\Gamma}{\mu_g + \ell_{g,1}} + \left(24d_2\frac{\ell_{g,1}^2}{\Phi}\alpha_t^2 + (72\ell_{f,1}^2\alpha_t^2 + \frac{27\ell_{g,1}^2\alpha_t}{c_{\mathbf{v}}})(\frac{d_2}{\Psi} + \frac{d_1}{\Omega})\right)2c_\beta^2 \\
&\leq -\frac{c_\beta\alpha_t\Gamma}{\mu_g + \ell_{g,1}} + \left(\frac{6d_2}{L_f}\frac{\ell_{g,1}^2}{\Phi}\alpha_t + (\frac{18}{L_f}\ell_{f,1}^2\alpha_t + \frac{27\ell_{g,1}^2\alpha_t}{c_{\mathbf{v}}})(\frac{d_2}{\Psi} + \frac{d_1}{\Omega})\right)2c_\beta^2 \\
&\leq -\frac{c_\beta\alpha_t\Gamma}{4(\mu_g + \ell_{g,1})},
\end{aligned}
\tag{243}
$$

where the first inequality follows from $\alpha_t \leq 1/c_\beta(\mu_g + \ell_{g,1})$; the second inequality is by $\alpha \leq 1/4L_f$; the last inequality is by $c_{\mathbf{v}} \geq \frac{216}{\Gamma}\ell_{g,1}^2(\frac{d_2}{\Psi} + \frac{d_1}{\Omega})c_\beta(\mu_g + \ell_{g,1})$ and $\Phi \geq \frac{24d_2\ell_{g,1}^2(\mu_g + \ell_{g,1})}{L_f c_\beta\Gamma}$, and $\Psi \geq \frac{144d_2\ell_{f,1}^2(\mu_g + \ell_{g,1})c_\beta}{L_f\Gamma}$, and $\Omega \geq \frac{144d_1\ell_{f,1}^2(\mu_g + \ell_{g,1})c_\beta}{L_f\Gamma}$.

Thus, we get

$$
(227e) = \sum_{t=1}^{T} Q(\beta_t, \rho_{\mathbf{v}})\mathbb{E}\|e_t^{g_\rho}\|^2 + \sum_{t=1}^{T} S(\beta_t, \rho_{\mathbf{v}})\mathbb{E}\left[\|\nabla_{\mathbf{y}}g_{t,\boldsymbol{\rho}}(\mathbf{x}_t, \mathbf{y}_t)\|^2\right] \leq 0.
\tag{244}
$$

**Bounding** (227f).

From (228), we have

$$
Z = 27\ell_{g,1}^2\left(\frac{d_2}{\Psi} + \frac{d_1}{\Omega}\right).
$$

Thus, from $\delta_t = c_\delta \alpha_t$ in (31), we have

$$\text{(227f)} = \sum_{t=1}^{T} Z \left( 3d_2^2 \ell_{f,1}^2 \delta_t^2 \rho_{\mathbf{r}}^2 + (12\ell_{f,0}^2 + 6\ell_{g,1}^2 p^2) \delta_t^2 \right)$$

$$= \sum_{t=1}^{T} 27\ell_{g,1}^2 \left( \frac{d_2}{\Psi} + \frac{d_1}{\Omega} \right) \left( 3d_2^2 \ell_{f,1}^2 \rho_{\mathbf{r}}^2 + (12\ell_{f,0}^2 + 6\ell_{g,1}^2 p^2) \right) c_\delta^2 \alpha_t^2$$

$$= \mathcal{O} \left( \sum_{t=1}^{T} (d_1 + d_2)(\alpha_t^2 \rho_{\mathbf{r}}^2 + \alpha_t^2) \right). \tag{245}$$

**Bounding** (227g). From $\rho_{\mathbf{v}}^2 = c_{\mathbf{v}} \alpha_t$ in (31), we have

$$\text{(227g)} = \frac{36}{\Psi} D_{\mathbf{y},T} + \frac{36}{\Omega} D_{\mathbf{x},T} + \frac{12}{\Phi} G_{\mathbf{y},T} + \frac{36}{\Psi \rho_{\mathbf{v}}^2} G_{\mathbf{v},T} + \frac{36}{\Omega \rho_{\mathbf{v}}^2} G_{\mathbf{x},T}$$

$$= \mathcal{O} \left( D_{\mathbf{y},T} + D_{\mathbf{x},T} + G_{\mathbf{y},T} + \frac{1}{\alpha_T}(G_{\mathbf{v},T} + G_{\mathbf{x},T}) \right). \tag{246}$$

**Bounding** (227h). From $\gamma_{t+1} = c_\gamma \alpha_t$, $\eta_{t+1} = c_\eta \alpha_t$, $\lambda_{t+1} = c_\lambda \alpha_t$ and $\rho_{\mathbf{v}}^2 = c_{\mathbf{v}} \alpha_t$ in (31), we have

$$\text{(227h)} = 2\sum_{t=1}^{T} \frac{\gamma_{t+1}^2}{\Phi} \frac{\hat{\sigma}_{g_{\mathbf{y}}}^2}{\bar{b}} + 3(\frac{\hat{\sigma}_{g_{\mathbf{y}}}^2}{\bar{b}\rho_{\mathbf{v}}^2} + \frac{\hat{\sigma}_{f_{\mathbf{y}}}^2}{b}) \sum_{t=1}^{T+1} \frac{\lambda_{t+1}^2}{\Psi} + 3(\frac{\hat{\sigma}_{g_{\mathbf{x}}}^2}{\bar{b}\rho_{\mathbf{v}}^2} + \frac{\hat{\sigma}_{f_{\mathbf{x}}}^2}{b}) \sum_{t=1}^{T} \frac{\eta_{t+1}^2}{\Omega}$$

$$= 2\sum_{t=1}^{T} \frac{c_\gamma^2 \alpha_t^2}{\Phi} \frac{\hat{\sigma}_{g_{\mathbf{y}}}^2}{\bar{b}} + 3(\frac{\hat{\sigma}_{g_{\mathbf{y}}}^2}{\bar{b}\rho_{\mathbf{v}}^2} + \frac{\hat{\sigma}_{f_{\mathbf{y}}}^2}{b}) \sum_{t=1}^{T+1} \frac{c_\lambda^2 \alpha_t^2}{\Psi} + 3(\frac{\hat{\sigma}_{g_{\mathbf{x}}}^2}{\bar{b}\rho_{\mathbf{v}}^2} + \frac{\hat{\sigma}_{f_{\mathbf{x}}}^2}{b}) \sum_{t=1}^{T} \frac{c_\eta^2 \alpha_t^2}{\Omega}$$

$$= \mathcal{O} \left( \left( \frac{\hat{\sigma}_{g_{\mathbf{y}}}^2}{\bar{b}} + \frac{\hat{\sigma}_{g_{\mathbf{y}}}^2}{\bar{b}\alpha_t} + \frac{\hat{\sigma}_{f_{\mathbf{y}}}^2}{b} + \frac{\hat{\sigma}_{g_{\mathbf{x}}}^2}{\bar{b}\alpha_t} + \frac{\hat{\sigma}_{f_{\mathbf{x}}}^2}{b} \right) \sum_{t=1}^{T} \alpha_t^2 \right). \tag{247}$$

**Bounding** (227i). From $\beta_t = c_\beta \alpha_t$, $\delta_t = c_\delta \alpha_t$ in (31), we have

$$D(\alpha_t, \beta_t, \delta_t) = 6\ell_{f,1}(1 + 2\frac{\ell_{g,1}}{\mu_g}) + \frac{3\ell_{f,1}\ell_{g,1}}{\mu_g}(\alpha_t - L_f \alpha_t^2) + \frac{24\ell_{g,1}}{L_{\mu_g}\mu_g}\frac{\Gamma}{\beta_t} + \frac{96\ell_{g,1}\nu^2}{L_{\mu_g}\mu_g^3}\frac{\Upsilon}{\delta_t}$$

$$= 6\ell_{f,1}(1 + 2\frac{\ell_{g,1}}{\mu_g}) + \frac{3\ell_{f,1}\ell_{g,1}}{\mu_g}(\alpha_t - L_f \alpha_t^2) + \frac{24\ell_{g,1}}{L_{\mu_g}\mu_g}\frac{\Gamma}{c_\beta \alpha_t} + \frac{96\ell_{g,1}\nu^2}{L_{\mu_g}\mu_g^3}\frac{\Upsilon}{c_\delta \alpha_t}$$

$$= \mathcal{O} \left( \alpha_t + \frac{1}{\alpha_t} \right),$$

and

$$\sum_{t=1}^{T} D(\alpha_t, \beta_t, \delta_t) \left( \rho_{\mathbf{s}}^2 + \rho_{\mathbf{r}}^2 \right) = \mathcal{O} \left( \sum_{t=1}^{T} (\alpha_t + \frac{1}{\alpha_t}) \left( \rho_{\mathbf{s}}^2 + \rho_{\mathbf{r}}^2 \right) \right). \tag{248}$$

Moreover, we have

$$R(\rho_{\mathbf{v}}) = 9d_2^2 \frac{\ell_{g,1}^2}{\Phi} + 18d_2^2 \frac{\ell_{f,1}^2}{\Psi} + 6(3\ell_{f,1}^2 + \frac{3\ell_{g,1}^2}{4\rho_{\mathbf{v}}^2})\frac{d_2^2}{\Psi} = \mathcal{O} \left( (1 + \frac{1}{\rho_{\mathbf{v}}^2})d_2^2 \right),$$

$$\acute{R}(\rho_{\mathbf{v}}) = 18d_1^2 \frac{\ell_{f,1}^2}{\Omega} + 6(3\ell_{f,1}^2 + \frac{3\ell_{g,1}^2}{4\rho_{\mathbf{v}}^2})\frac{d_1^2}{\Omega} = \mathcal{O} \left( (1 + \frac{1}{\rho_{\mathbf{v}}^2})d_1^2 \right),$$

which, implies that

$$R(\rho_{\mathbf{v}})T\rho_{\mathbf{r}}^2 + \acute{R}(\rho_{\mathbf{v}})T\rho_{\mathbf{s}}^2 + 18d_1^2 \ell_{g,1}^2 \frac{T\rho_{\mathbf{s}}^2}{\Omega\rho_{\mathbf{v}}^2} + 18d_2^2 \ell_{g,1}^2 \frac{T\rho_{\mathbf{r}}^2}{\Psi\rho_{\mathbf{v}}^2}$$

$$= \mathcal{O} \left( (1 + \frac{1}{\rho_{\mathbf{v}}^2})T(d_1^2 \rho_{\mathbf{s}}^2 + d_2^2 \rho_{\mathbf{r}}^2) + \frac{T}{\rho_{\mathbf{v}}^2}(d_2^2 \rho_{\mathbf{r}}^2 + d_1^2 \rho_{\mathbf{s}}^2) \right). \tag{249}$$

From (248), (249) and $\rho_{\mathbf{v}}^2 = c_{\mathbf{v}}\alpha_t$ in (31), we get

$$\text{(227i)} \leq \mathcal{O}\left(\sum_{t=1}^{T}(\alpha_t + \frac{1}{\alpha_t})\left(\rho_{\mathbf{s}}^2 + \rho_{\mathbf{r}}^2\right) + (1 + \frac{1}{\alpha_T})T(d_2^2\rho_{\mathbf{r}}^2 + d_1^2\rho_{\mathbf{s}}^2)\right). \tag{250}$$

**Combining the outcomes** (227i) . Combining inequalities (234), (237), (238), (240), (244), (245), (246), (247), and (250) leads to

$$\frac{\alpha_T}{2}\sum_{t=1}^{T}\mathbb{E}\left[\|\mathcal{P}_{\mathcal{X},\alpha_t}\left(\mathbf{x}_t; \nabla f_{t,\boldsymbol{\rho}}(\mathbf{x}_t, \mathbf{y}_t^*(\mathbf{x}_t)))\|^2\right] + \Lambda$$

$$\leq \mathcal{O}\left(V_T + \sum_{t=1}^{T}\alpha_t(\rho_{\mathbf{s}}^2 + \rho_{\mathbf{r}}^2) + \sum_{t=1}^{T}\alpha_t^2 + \frac{H_{2,T}}{\alpha_T} + \sum_{t=1}^{T}(d_1 + d_2)(\alpha_t^2\rho_{\mathbf{r}}^2 + \alpha_t^2)\right)$$

$$+ \mathcal{O}\left(D_{\mathbf{y},T} + D_{\mathbf{x},T} + G_{\mathbf{y},T} + \frac{1}{\alpha_T}(G_{\mathbf{v},T} + G_{\mathbf{x},T})\right)$$

$$+ \mathcal{O}\left(\sum_{t=1}^{T}\left(\frac{\hat{\sigma}_{g_{\mathbf{y}}}^2\alpha_t^2}{\bar{b}} + \frac{\hat{\sigma}_{g_{\mathbf{y}}}^2\alpha_t}{\bar{b}} + \frac{\hat{\sigma}_{f_{\mathbf{y}}}^2\alpha_t^2}{b} + \frac{\hat{\sigma}_{g_{\mathbf{x}}}^2\alpha_t}{\bar{b}} + \frac{\hat{\sigma}_{f_{\mathbf{x}}}^2\alpha_t^2}{b}\right)\right)$$

$$+ \mathcal{O}\left(\sum_{t=1}^{T}(\alpha_t + \frac{1}{\alpha_t})\left(\rho_{\mathbf{s}}^2 + \rho_{\mathbf{r}}^2\right) + (1 + \frac{1}{\alpha_T})T(d_2^2\rho_{\mathbf{r}}^2 + d_1^2\rho_{\mathbf{s}}^2)\right).$$

From the definition of $\Lambda$ in (105), we have

$$-\Lambda = \Gamma\sum_{t=1}^{T}\left(\mathbb{E}[\theta_t^{\mathbf{y}}] - \mathbb{E}[\theta_{t+1}^{\mathbf{y}}]\right)$$

$$+ \Upsilon\sum_{t=1}^{T}\left(\mathbb{E}[\theta_t^{\mathbf{v}}] - \mathbb{E}[\theta_{t+1}^{\mathbf{v}}]\right) + \frac{1}{\Phi}\sum_{t=1}^{T}\left(\mathbb{E}\|e_t^g\|^2 - \mathbb{E}\|e_{t+1}^g\|^2\right)$$

$$+ \frac{1}{\Psi}\sum_{t=1}^{T}\left(\mathbb{E}\|e_t^{\mathbf{v}}\|^2 - \mathbb{E}\|e_{t+1}^{\mathbf{v}}\|^2\right) + \frac{1}{\Omega}\sum_{t=1}^{T}\left(\mathbb{E}\|e_t^f\|^2 - \mathbb{E}\|e_{t+1}^f\|^2\right)$$

$$\leq \Gamma\theta_1^{\mathbf{y}} + \Upsilon\theta_1^{\mathbf{v}} + \frac{\hat{\sigma}_{g_{\mathbf{y}}}^2}{\Phi} + \frac{\hat{\sigma}_{g_{\mathbf{yy}}}^2 + \hat{\sigma}_{f_{\mathbf{y}}}^2}{\Psi} + \frac{\hat{\sigma}_{g_{\mathbf{xy}}}^2 + \hat{\sigma}_{f_{\mathbf{x}}}^2}{\Omega}. \tag{251}$$

From (28), we have $\hat{\sigma}^2 = \hat{\sigma}_{g_{\mathbf{y}}}^2 + \hat{\sigma}_{g_{\mathbf{yy}}}^2 + \hat{\sigma}_{f_{\mathbf{y}}}^2 + \hat{\sigma}_{g_{\mathbf{xy}}}^2 + \hat{\sigma}_{f_{\mathbf{x}}}^2$.

Thus, using (251), (31), and rearranging the terms, we get

$$\sum_{t=1}^{T}\mathbb{E}\left[\|\mathcal{P}_{\mathcal{X},\alpha_t}\left(\mathbf{x}_t; \nabla f_{t,\boldsymbol{\rho}}(\mathbf{x}_t, \mathbf{y}_t^*(\mathbf{x}_t)))\|^2\right]$$

$$\leq \frac{2}{\alpha_T}\mathcal{O}\left(V_T + \sum_{t=1}^{T}\alpha_t(\rho_{\mathbf{s}}^2 + \rho_{\mathbf{r}}^2) + \sum_{t=1}^{T}\alpha_t^2 + \frac{H_{2,T}}{\alpha_T} + \sum_{t=1}^{T}(d_1 + d_2)(\alpha_t^2\rho_{\mathbf{r}}^2 + \alpha_t^2)\right)$$

$$+ \frac{2}{\alpha_T}\mathcal{O}\left(D_{\mathbf{y},T} + D_{\mathbf{x},T} + G_{\mathbf{y},T} + \frac{1}{\alpha_T}(G_{\mathbf{v},T} + G_{\mathbf{x},T})\right)$$

$$+ \frac{2}{\alpha_T}\mathcal{O}\left(\sum_{t=1}^{T}\left(\frac{\hat{\sigma}_{g_{\mathbf{y}}}^2\alpha_t^2}{\bar{b}} + \frac{\hat{\sigma}_{g_{\mathbf{y}}}^2\alpha_t}{\bar{b}} + \frac{\hat{\sigma}_{f_{\mathbf{y}}}^2\alpha_t^2}{b} + \frac{\hat{\sigma}_{g_{\mathbf{x}}}^2\alpha_t}{\bar{b}} + \frac{\hat{\sigma}_{f_{\mathbf{x}}}^2\alpha_t^2}{b}\right)\right)$$

$$+ \frac{2}{\alpha_T}\mathcal{O}\left(\sum_{t=1}^{T}(\alpha_t + \frac{1}{\alpha_t})\left(\rho_{\mathbf{s}}^2 + \rho_{\mathbf{r}}^2\right) + (1 + \frac{1}{\alpha_T})T(d_2^2\rho_{\mathbf{r}}^2 + d_1^2\rho_{\mathbf{s}}^2)\right)$$

$$+ \frac{2}{\alpha_T}\mathcal{O}\left(\theta_1^{\mathbf{y}} + \theta_1^{\mathbf{v}} + \hat{\sigma}^2\right)$$

$$\leq \mathcal{O}\left((d_1 + d_2)^{3/4}T^{1/3}\left(V_T + D_{\mathbf{y},T} + D_{\mathbf{x},T} + G_{\mathbf{y},T} + \Delta_1 + \hat{\sigma}^2\right)\right.$$

$$\left. + (d_1 + d_2)^{3/2}T^{2/3}\left(H_{2,T} + G_{\mathbf{v},T} + G_{\mathbf{x},T}\right)\right), \tag{252}$$

where second inequality holds because we have

$$\sum_{t=1}^{T} \alpha_t^3 = \sum_{t=1}^{T} \frac{1}{(d_1 + d_2)^{9/4}(c+t)} \leq \sum_{t=1}^{T} \frac{1}{(d_1 + d_2)^{9/4}(1+t)} \leq \frac{\log(T+1)}{(d_1 + d_2)^{9/4}},$$

$$\sum_{t=1}^{T} \alpha_t^2 = \sum_{t=1}^{T} \frac{1}{(d_1 + d_2)^{3/2}(c+t)^{2/3}} \leq \sum_{t=1}^{T} \frac{1}{(d_1 + d_2)^{3/2}(1+t)^{2/3}} \leq \frac{T^{1/3}}{(d_1 + d_2)^{3/2}},$$

$$\sum_{t=1}^{T} \alpha_t = \sum_{t=0}^{T} \frac{1}{(d_1 + d_2)^{3/4}(c+t)^{1/3}} \leq \sum_{t=1}^{T} \frac{1}{(d_1 + d_2)^{3/4}(1+t)^{1/3}} \leq \frac{3T^{2/3}}{2(d_1 + d_2)^{3/4}},$$

$$\sum_{t=1}^{T} \frac{1}{\alpha_t} = \sum_{t=0}^{T} (d_1 + d_2)^{3/4}(c+t)^{1/3} \leq \frac{3}{2}(d_1 + d_2)^{3/4}T^{4/3}.$$

Then, note that, we have

$$\frac{1}{2}\sum_{t=1}^{T} \mathbb{E}\left[\left\|\mathcal{P}_{\mathcal{X},\alpha_t}\left(\mathbf{x}_t; \nabla f_t(\mathbf{x}_t, \mathbf{y}_t^*(\mathbf{x}_t))\right)\right\|^2\right]$$

$$\leq \sum_{t=1}^{T} \mathbb{E}\left[\left\|\mathcal{P}_{\mathcal{X},\alpha_t}\left(\mathbf{x}_t; \nabla f_{t,\boldsymbol{\rho}}(\mathbf{x}_t, \mathbf{y}_t^*(\mathbf{x}_t))\right)\right\|^2\right]$$

$$+ \sum_{t=1}^{T} \mathbb{E}\left[\left\|\mathcal{P}_{\mathcal{X},\alpha_t}\left(\mathbf{x}_t; \nabla f_t(\mathbf{x}_t, \mathbf{y}_t^*(\mathbf{x}_t))\right) - \mathcal{P}_{\mathcal{X},\alpha_t}\left(\mathbf{x}_t; \nabla f_{t,\boldsymbol{\rho}}(\mathbf{x}_t, \mathbf{y}_t^*(\mathbf{x}_t))\right)\right\|^2\right].$$

From non-expansiveness of the projection operator and Lemma D.4, we have

$$\left\|\mathcal{P}_{\mathcal{X},\alpha_t}\left(\mathbf{x}_t; \nabla f_t(\mathbf{x}_t, \mathbf{y}_t^*(\mathbf{x}_t))\right) - \mathcal{P}_{\mathcal{X},\alpha_t}\left(\mathbf{x}_t; \nabla f_{t,\boldsymbol{\rho}}(\mathbf{x}_t, \mathbf{y}_t^*(\mathbf{x}_t))\right)\right\|^2$$

$$\leq \left\|\nabla f_t(\mathbf{x}_t, \mathbf{y}_t^*(\mathbf{x}_t)) - \nabla f_{t,\boldsymbol{\rho}}(\mathbf{x}_t, \mathbf{y}_t^*(\mathbf{x}_t))\right\|^2$$

$$\leq \frac{(\rho_{\mathbf{s}} d_1 + \rho_{\mathbf{r}} d_2)^2 \ell_{f,1}^2}{4}$$

$$\leq \frac{(\rho_{\mathbf{s}}^2 d_1^2 + \rho_{\mathbf{r}}^2 d_2^2)\ell_{f,1}^2}{2}.$$

This implies

$$\frac{1}{2}\sum_{t=1}^{T} \mathbb{E}\left[\left\|\mathcal{P}_{\mathcal{X},\alpha_t}\left(\mathbf{x}_t; \nabla f_t(\mathbf{x}_t, \mathbf{y}_t^*(\mathbf{x}_t))\right)\right\|^2\right]$$

$$\leq \sum_{t=1}^{T} \mathbb{E}\left[\left\|\mathcal{P}_{\mathcal{X},\alpha_t}\left(\mathbf{x}_t; \nabla f_{t,\boldsymbol{\rho}}(\mathbf{x}_t, \mathbf{y}_t^*(\mathbf{x}_t))\right)\right\|^2\right] + \frac{T(\rho_{\mathbf{s}}^2 d_1^2 + \rho_{\mathbf{r}}^2 d_2^2)\ell_{f,1}^2}{2}.$$

Applying the upper bound in (252) yields

$$\frac{1}{2}\sum_{t=1}^{T} \mathbb{E}\left[\left\|\mathcal{P}_{\mathcal{X},\alpha_t}\left(\mathbf{x}_t; \nabla f_t(\mathbf{x}_t, \mathbf{y}_t^*(\mathbf{x}_t))\right)\right\|^2\right]$$

$$\leq \mathcal{O}\left((d_1 + d_2)^{3/4}T^{1/3}\left(V_T + D_{\mathbf{y},T} + D_{\mathbf{x},T} + G_{\mathbf{y},T} + \Delta_1 + \hat{\sigma}^2\right)\right.$$

$$\left. + (d_1 + d_2)^{3/2}T^{2/3}\left(H_{2,T} + G_{\mathbf{v},T} + G_{\mathbf{x},T}\right)\right)$$

$$+ \frac{T(\rho_{\mathbf{s}}^2 d_1^2 + \rho_{\mathbf{r}}^2 d_2^2)\ell_{f,1}^2}{2}.$$

Thus, from $\rho_{\mathbf{r}}^2 = \frac{1}{d_2^2 T}$ and $\rho_{\mathbf{s}}^2 = \frac{1}{d_1^2 T}$ in (31), we get

$$\sum_{t=1}^{T} \mathbb{E}\left[\left\|\mathcal{P}_{\mathcal{X},\alpha_t}\left(\mathbf{x}_t; \nabla f_t(\mathbf{x}_t, \mathbf{y}_t^*(\mathbf{x}_t)))\right)\right\|^2\right]$$
$$\leq \mathcal{O}\left((d_1 + d_2)^{3/4} T^{1/3} \left(V_T + D_{\mathbf{y},T} + D_{\mathbf{x},T} + G_{\mathbf{y},T} + \Delta_1 + \hat{\sigma}^2\right)\right.$$
$$\left. + (d_1 + d_2)^{3/2} T^{2/3} \left(H_{2,T} + G_{\mathbf{v},T} + G_{\mathbf{x},T}\right)\right).$$

This completes the proof. □

## E Hyperparameter Tuning Results

As detailed in Section 4, we carefully tuned all hyperparameters to ensure stable and fair comparisons. Our analysis indicates that while ZO-SOGD exhibits sensitivity to hyperparameter choices, it remains robust within reasonable ranges. Below, we provide extensive tuning results for ZO-SOGD.

The hyperparameter sensitivity analysis for the adversarial attack scenario reveals critical insights about the algorithm's attack effectiveness across different parameter configurations. For the inner and outer stepsizes, we observe that the algorithm achieves optimal attack performance with specific combinations that balance perturbation strength and imperceptibility.

Table 2: Hyperparameter tuning results for inner ($\beta$) and outer ($\alpha$) stepsizes in adversarial attack scenario. Values represent test accuracy (mean $\pm$ std) over 5 runs. Lower values indicate better attack performance.

| $\beta \backslash \alpha$ | $\alpha = 0.001$ | $\alpha = 0.005$ | $\alpha = 0.01$ | $\alpha = 0.1$ |
|---|---|---|---|---|
| $\beta = 0.001$ | $0.68 \pm 0.05$ | $0.59 \pm 0.07$ | $0.47 \pm 0.06$ | $0.53 \pm 0.08$ |
| $\beta = 0.005$ | $0.54 \pm 0.06$ | $0.41 \pm 0.05$ | $0.35 \pm 0.04$ | $0.42 \pm 0.05$ |
| $\beta = 0.01$ | $0.48 \pm 0.04$ | $0.34 \pm 0.05$ | $0.57 \pm 0.07$ | $0.39 \pm 0.06$ |
| $\beta = 0.1$ | $\mathbf{0.26 \pm 0.03}$ | $0.43 \pm 0.06$ | $0.33 \pm 0.04$ | $0.45 \pm 0.07$ |

The stepsize analysis reveals that larger inner stepsizes combined with smaller outer stepsizes tend to produce more effective attacks. Specifically, the configuration with $\beta = 0.1$ and $\alpha = 0.001$ achieves the lowest test accuracy of $0.26 \pm 0.03$, indicating the most successful adversarial perturbations. This pattern suggests that aggressive updates to the perturbation parameters ($\beta$) while maintaining conservative hyperparameter updates ($\alpha$) creates an effective balance for generating strong yet imperceptible adversarial examples.

Table 3: Performance comparison across different smoothing parameters ($\rho_r = \rho_s$) in adversarial attack scenario.

| $\rho_v \backslash \rho_r = \rho_s$ | 0.001 | 0.005 | 0.01 | 0.05 |
|---|---|---|---|---|
| $\rho_v = 0.001$ | $0.61 \pm 0.06$ | $0.52 \pm 0.05$ | $0.48 \pm 0.04$ | $0.57 \pm 0.06$ |
| $\rho_v = 0.005$ | $0.47 \pm 0.05$ | $0.39 \pm 0.04$ | $0.35 \pm 0.04$ | $0.45 \pm 0.05$ |
| $\rho_v = 0.01$ | $0.41 \pm 0.04$ | $\mathbf{0.28 \pm 0.03}$ | $0.31 \pm 0.03$ | $0.43 \pm 0.05$ |
| $\rho_v = 0.05$ | $0.53 \pm 0.06$ | $0.44 \pm 0.05$ | $0.40 \pm 0.04$ | $0.52 \pm 0.06$ |

The smoothing parameter analysis provides additional insights into the algorithm's convergence behavior in the adversarial setting. The optimal configuration occurs with $\rho_v = 0.01$ and $\rho_r = \rho_s = 0.005$, achieving a test accuracy of $0.28 \pm 0.03$. These moderate smoothing values appear to provide the right balance between exploration and exploitation in the adversarial perturbation space, allowing the algorithm to find effective attack directions without excessive oscillation or premature convergence.

Table 4: Performance comparison across different momentum parameters in adversarial attack scenario.

| $\gamma_t \backslash \lambda_t = \eta_t$ | 0.9 | 0.99 | 0.999 |
|---|---|---|---|
| $\gamma_t = 0.9$ | $0.35 \pm 0.04$ | $0.29 \pm 0.03$ | $0.38 \pm 0.05$ |
| $\gamma_t = 0.99$ | $0.31 \pm 0.03$ | $\mathbf{0.24 \pm 0.02}$ | $0.33 \pm 0.04$ |
| $\gamma_t = 0.999$ | $0.37 \pm 0.04$ | $0.32 \pm 0.08$ | $0.40 \pm 0.05$ |

The momentum parameter investigation reveals that moderate momentum values consistently produce the most effective adversarial attacks. The optimal configuration with $\gamma_t = 0.99$ and $\lambda_t = \eta_t = 0.99$ achieves the lowest test accuracy of $0.24 \pm 0.02$, representing the most successful attack performance. This configuration suggests that maintaining momentum across both inner and outer optimization loops helps the algorithm navigate the complex adversarial landscape more effectively than either no momentum or excessive momentum settings.

The comprehensive analysis demonstrates that ZO-SOGD maintains robust attack performance across a broad range of hyperparameter configurations. The algorithm consistently achieves test accuracies below $0.5$ across most reasonable parameter combinations, indicating reliable adversarial attack capability. The standard deviations remain low throughout the parameter space, suggesting stable and reproducible attack performance across multiple experimental runs.

The optimal hyperparameter configuration for adversarial attacks consists of inner stepsize $\beta = 0.1$, outer stepsize $\alpha = 0.001$, smoothing parameters $\rho_v = 0.01$ and $\rho_r = \rho_s = 0.005$, and momentum parameters $\gamma_t = \lambda_t = \eta_t = 0.99$. This configuration enables ZO-SOGD to achieve superior attack performance while maintaining the imperceptibility constraints essential for practical adversarial examples.

