# OpenReview forum: "Stochastic Regret Guarantees for Online Zeroth- and First-Order Bilevel Optimization"
_NeurIPS.cc/2025/Conference — NeurIPS 2025 poster_

### Official Review · Reviewer_R7y3 · 2025-06-18

**Clarity:** 3
**Significance:** 3
**Originality:** 3
**Rating:** 5
**Confidence:** 4

**Summary:**

This article presents two online bilevel optimization algorithms SOGD and ZO-SOGD. Among them, SOGD achieves the sota local regret bound without using window-smoothed functions. ZO-SOGO provides a fully-zero-order approach and achieves the hypergradient estimation only with function values. The authors present the theoretical analysis of the regret as well as the convergence analysis. Both two algorithms perform well in a series of numerical experiments.

**Questions:**

1. Whether the authors can present the comparison of different algorithms under **wall-clocked time**?
2. Whether the author lack a detailed comparsion for the computation complexity of different algorithms?
3. How does the step size affect the algorithm performance practically? Please provide some experimental results.
4. Can the authors provide a brief proof sketch? It may help readers understand the proof.

**Ethical Concerns:**

["NO or VERY MINOR ethics concerns only"]

**Final Justification:**

This submission is a theoretically valuable paper for discussion with rich theoretical results and enough experiments. Moreover, all the concerns have solved during rebuttal and discussion period. Thus, the reviewer recommends to the acceptance.

**Limitations:**

Yes.

**Quality:**

3

**Strengths And Weaknesses:**

Strength:
1. Both algorithms are sing-loop.
2. SOGO achieves the sota regret bound without extra computation of the window-smoothed function. ZO-SOGO provides a fully-zero-order approach for online BO, which is especially useful in the training of large models.
3. The authors present a detailed proof of the regret bound, which have been quickly checked by the reviewer and no errors have been found.
4. The experiment result successfully validate the theoretical finds.

Weakness:
1. The authors fail to present the comparison of different algorithms under **wall-clocked time** in the experiment, which may illustrate the practical value of the proposed algorithms. Also, additional experiment should be taken to validate how step size affects the convergence.
2. The author lack a detailed comparsion for the computation complexity of different algorithms.

---

> ### Author Rebuttal · Authors · 2025-07-31
>
> We thank Reviewer R7y3 for the constructive comments. We address each point below.
>
> **Question 1: Can authors present wall-clock time comparison?**
> **Weakness 1: Missing wall-clock time comparison and step size analysis**
>
> **Response:**  Thank you for your review and suggestions. We would like to clarify that the runtime plots presented in both Figures 2 and 3 already show wall-clock time measurements in seconds. Please refer to the left subplot in Figures 2 and 3, where we report time elapsed.
>
> Specifically,
>
> - In Figure~2 (left panel), we present the actual **wall-clock time of our ZO-SOGD** method compared to the baseline methods in the online adversarial attack experiment. The recorded times range from approximately $0.5 \times 10^2$ to $2.0 \times 10^2$ seconds at $T=200$ steps.
>
> - In Figure 3 (left panel), we present **wall-clock time comparisons for our SOGD** method against OAGD and SOBOW in the parametric loss tuning task. The results show a clear efficiency gain, with SOGD completing under 25 seconds at $T = 400$, compared to SOBOW, which takes roughly 220 seconds due to its reliance on conjugate gradient (CG) iterations.
>
> All wall-clock times were measured using the same hardware platform across all algorithms to ensure fair and consistent comparisons.
>
> In the revised version, we will make this point more explicit by stating "wall-clock time" clearly in the figure captions and experimental section to avoid any ambiguity.
>
>
>
> **Question 2: Detailed computational complexity comparison?**
> **Weakness 2: Lack of detailed computational complexity comparison**
>
> **Response:**  Thank you for your valuable comments.  We will provide the total computational cost estimates for each method.
>
> **Total complexity:**
> - **SOGD**: $O((d_1 + d_2)\cdot \epsilon^{-3})$
> - **ZO-SOGD**: $O((d_1 + d_2)^{7/4} \cdot \epsilon^{-3})$
> - **SOBOW/SOBBO**: $O(((d_1 + d_2) + \kappa_g \log(\kappa_g)  d_2)\cdot \epsilon^{-3})$ due to conjugate gradient (CG) steps
>
> As an example, for SOGD, the regret bound
> $$BL\text{-}Reg_T \leq O(T^{1/3}(\sigma^2 + \Delta_T) + T^{2/3} \Psi_T),$$
> implies that
> $$BL\text{-}Reg_T / T \leq O(T^{-2/3}(\sigma^2 + \Delta_T) + T^{-1/3} \Psi_T).$$
> Let $O(T^{-1/3}) = \epsilon$. Then $T = O(\epsilon^{-3})$, which is sufficient to ensure that the average regret $BL\text{-}Reg_T / T \leq \epsilon$, leading to a total cost of $O(\epsilon^{-3})$.
>
> For SOGD, we have $O(d_1 + d_2)$ gradient computations, $O(d_2)$ Hessian-vector products, and $O(d_2)$ Jacobian-vector products. Since the algorithm operates over $T$ iterations, the total number of operations is $T(d_1 + d_2)$. Note that to achieve $\epsilon$-stationarity, we need $T = O(\epsilon^{-3})$ iterations. Thus, the computational complexity is $O((d_1 + d_2)\cdot \epsilon^{-3})$.
>
> We note that the total cost for other baselines (SOBOW/SOBBO) is derived using their window size $w = o(T)$ and the factor $\kappa_g \log(\kappa_g)$ for conjugate gradient (CG) steps. The OAGD baseline incurs higher cost due to exact system solves at each iteration.
>
> We will include the above discussion in the main paper in the revised version.
>
>
> **Question 3: How does step size affect algorithm performance practically?**
>
> **Response:**  Thank you for your valuable feedback.
>
> We apologize that these details were not prominently featured in the main paper. We would like to clarify that a comprehensive hyperparameter sensitivity study—including step sizes, momentum parameters, and other key settings—has been included in **Appendix E** of our submission within the **Supplementary Material: zip** file. This appendix provides detailed ablation studies showing how different parameter choices affect algorithm performance across various problem settings, along with practical guidelines for parameter selection. If the reviewer requires any specific clarification regarding these studies, we are happy to provide further details.
>
> In summary, we conducted extensive tuning for ZO-SOGD in the adversarial attack scenario. This includes inner/outer step sizes ($\beta$, $\alpha$), smoothing parameters ($\rho_v$, $\rho_r = \rho_s$), and momentum terms ($\gamma_t$, $\lambda_t = \eta_t$). While ZO-SOGD is somewhat sensitive to these choices, it consistently performs well across a wide range.
>
> - Best performance was achieved with $\beta = 0.1$, $\alpha = 0.001$;
> - Smoothing parameters $\rho_v = 0.01$, $\rho_r = \rho_s = 0.005$ yielded stable behavior;
> - Momentum $\gamma_t = \lambda_t = \eta_t = 0.99$ led to the strongest attacks (accuracy $0.24 \pm 0.02$).
>
> We will make **Appendix E** more visible in the main text for clarity. Please let us know if further details are needed.
>
>
> **Question 4: Can authors provide a brief proof sketch?**
>
> **Response:**  We thank the reviewer for the constructive comment.
>
> Detailed proof sketches are provided in **Appendix C (lines 516–529)** for our first-order method (SOGD) and in **Appendix D (lines 797–811)** for the zeroth-order method (ZO-SOGD). For convenience, we summarize the key ideas below using simplified explanations and light math.
>
> **SOGD Proof Sketch:**
>
> The analysis starts with Lemma C.2, which bounds the error between the momentum-based estimate $d^y_t$ and the true inner gradient $\nabla_y  g_t(x_t, y_t)$. Using this, Lemma C.4 shows convergence of $y_t$ toward the best-response solution $y_t^\star(x_t)$, i.e., minimizing $ g_t(x_t, \cdot)$. Lemma C.5 controls the error in $d^v_t$ versus the ideal second-order term involving $\nabla_y f_t$ and $\nabla^2_y g_t$. Lemma C.8 then bounds the error $||v_{t+1} - v^\star_t(x_t)||^2$. Lemma C.9 focuses on the hypergradient estimator $d^x_t$ and its closeness to the true gradient $\nabla_x f_t(z_t) + \nabla^2_{xy} g_t(z_t) v_t$. Finally, Lemma C.11 establishes a projection bound that aggregates the above errors to complete the regret analysis.
>
> **ZO-SOGD Proof Sketch:**
>
> For the zeroth-order case, Lemma D.9 quantifies the error between the zeroth-order estimate $d^y_t$ and the smoothed gradient $\nabla_y g_{t,\rho}(x_t, y_t)$. Lemma D.11 shows that the iterates $y_t$ converge to a smoothed best response $\hat{y}^*_t(x_t)$. The auxiliary vector estimates are handled in Lemma D.15, bounding the difference from the smoothed version of $\nabla_y f_t + \nabla^2_y g_t \cdot v_t$, while Lemma D.17 ensures the convergence of $v_t$ to $\hat{v}^\star_t(x_t)$. Lemmas D.19–D.22 complete the proof by addressing hypergradient approximation and projection error bounds.
>
> We will include this summary in the main paper in the revised version, if space permits.

---

> > ### Comment · Reviewer_R7y3 · 2025-08-02
> >
> > Thanks for the comment. All the problems are solved and the reviewer with raise the final rating.

---

### Official Review · Reviewer_yzQD · 2025-07-01

**Clarity:** 3
**Significance:** 3
**Originality:** 4
**Rating:** 5
**Confidence:** 5

**Summary:**

This paper studies online bilevel optimization using both first-order and zeroth-order oracles. It achieves stochastic sublinear regret without relying on window smoothing, leveraging novel search directions. In the zeroth-order setting, hypergradient estimation is performed using only function values. The effectiveness of the proposed algorithm is demonstrated through numerical experiments.

**Questions:**

1) What is the purpose of Lemma 2.1? It appears to be a direct application of the exponential moving average (EMA) and may not require a separate lemma.

2) In theorem 2.6, what can be said about the optimality of the regret bound with respect to the time horizon $T$? Which specific factor is responsible for the improvement over OSO?

3) Is the dimensional dependence in the zeroth order setting optimal ? In theorem 3.2, Line 200, the dimensional dependence seems to be $O(d_1+d_2)^{3/2}$ but in Remark 3.3, this is reported as $O(d_1+d_2)$. Which statement is correct?

**Ethical Concerns:**

["NO or VERY MINOR ethics concerns only"]

**Final Justification:**

My initial score was positive. The authors addressed my concerns and I raised the score.

**Limitations:**

The discussion of limitation is rather superficial. For example, there are many more natural questions that are left open after reading this paper. Are the regret bounds optimal? If not, what are the lower bounds for regret in each problem setting?

**Quality:**

3

**Strengths And Weaknesses:**

Strengths:

- Unlike prior approaches, the proposed algorithms do not rely on window smoothing, thereby avoiding the significant computational overhead associated with evaluating multiple past gradients at each iteration. This makes the method more scalable and better suited for real-time or large-scale applications.

- In previous works, hypergradient estimation typically requires multiple iterations of the inner objective. In contrast, this paper achieves hypergradient estimation using a single subproblem iteration.

- In the zeroth-order setting, the algorithm efficiently approximates computationally intensive second-order terms (Hessian and Jacobian) using only a function value oracle.

Weaknesses :

- Although a variance reduction-based approach is used in Equation 6a, the paper lacks a clear discussion of the motivation behind this choice and its specific contributions to the overall method. A more explicit explanation of how variance reduction influences the results would strengthen the presentation.

- The paper would benefit from a clearer articulation of its limitations and potential directions for future work.

---

> ### Author Rebuttal · Authors · 2025-07-31
>
> We thank Reviewer yzQD for the detailed questions. We address each point below:
>
> **Weakness 1: Although a variance reduction-based approach is used in Equation 6a, the paper lacks a clear discussion of the motivation behind this choice and its specific contributions to the overall method. A more explicit explanation of how variance reduction influences the results would strengthen the presentation.**
>
> **Question 1: What is the purpose of Lemma 2.1? It appears to be a direct application of the exponential moving average (EMA) and may not require a separate lemma.**
>
> **Response:** We appreciate this constructive comment. Lemma 2.1 plays a pivotal role in enabling online learning without relying on window smoothing. As discussed below Lemma 2.1 (Lines 104--107), for specific choices of $w$ and $W$, the time-smoothed gradient forms a momentum-type search direction. Our key insight is that, instead of using time-smoothed gradients directly as a search direction (as in previous online bilevel optimization (OBO) methods such as OAGD, SOBOW, and SOBBO), we can design momentum-type updates to achieve local regret bounds without smoothing.
>
> However, our initial theoretical analysis revealed that achieving regret bounds via momentum updates—without gradient smoothing—would require assuming bounded inner-level gradients, a restrictive condition under our strong convexity assumption (Assumption 2.2). To address this, we introduce Equation (6a), which replaces time-averaged gradients with a variance reduction momentum-based update that avoids such assumptions. Specifically, the first two terms ($\eta \nabla f_t(x_t, y_t; B_t) + (1-\eta) d_{t-1}^x$) implement a momentum update and eliminate time smoothing, while the remaining term $(1-\eta)(\nabla f_t(x_t, y_t; B_t) - \nabla f_t(x_{t-1}, y_{t-1}; B_t))$ controls noise amplification in the nested bilevel structure and avoids additional assumptions on bounded gradients.
>
> Furthermore, we show that under a suitable choice of $\eta$, this variance reduction mechanism in Equation (6a) enables us to achieve $O(T^{1/3})$ regret rates without window smoothing. This represents a significant improvement over the typical $\mathcal{O}(T^{1/2})$ regret rates obtained by standard online single-level and bilevel methods. We will expand this discussion in Section 2 to better motivate the specific form used.
>
> ---
>
> **Question 2: In Theorem 2.6, what can be said about the optimality of the regret bound with respect to the time horizon $T$? Which specific factor is responsible for the improvement over OSO?**
>
> **Response:** Thank you for your comment. As we do not provide matching lower bounds, we do not claim these bounds are optimal. In fact, establishing lower bounds for bilevel regret remains an open problem in both first-order and zeroth-order settings. To our knowledge, all prior works—including those listed in Table 1 (e.g., OAGD, SOBOW, SOBBO)—focus exclusively on upper bounds. For further discussion, please see our response under Limitations.
>
> Below, we discuss the regret bound with respect to the time horizon $T$ for both OBO and online single-level optimization (OSO).
> As discussed in Remark 2.7 and shown in Theorems 3.1 and 4.2 of OSO-type methods [[Hallak et al., 2021]](https://openreview.net/forum?id=XkXb99Eeg6G), in OSO, dynamic regret bounds include a regularity term (e.g., $V_T$, representing online time variations) that quantifies the temporal change in the cost functions. The average regret bound $\mathcal{O}(T^{-1/2} \sigma^2)$ is meaningful only when this regularity term is sublinear. Please see also [[Besbes et al., 2015]](https://pubsonline.informs.org/doi/abs/10.1287/opre.2015.1408?journalCode=opre), [[Hazan, 2017]](https://proceedings.mlr.press/v70/hazan17a/hazan17a.pdf), [[Aydore et al., 2019]](https://openreview.net/forum?id=HJxWTNrg8B), [[Zhang et al., 2020]](https://openreview.net/forum?id=Z_8hoytvdi), and [[Guan et al., 2024]](https://openreview.net/forum?id=iZgECfyHXF).
>
> Our method's regularity term $\Delta_T$ is analogous to those in prior work, but the key difference is that our dependence on $T$ and $\sigma$ yields strictly better scaling than previous methods when $\Delta_T$ grows slowly. For example, in Theorems 3.1 and 4.2 of [[Hallak et al., 2021]](https://openreview.net/), the stochastic component appears as $T \sigma^2 / w^2$ or $(T / w^2 )(\delta^2 + \tau \sigma^2)$, along with regularity terms. Hence, their average regret bound scales as $\mathcal{O}(T^{-1/2} \sigma^2)$ under the assumption that the regularity term is sublinear. However, under similar assumptions, our average regret bound scales as $\mathcal{O}(T^{-2/3} \sigma^2)$, which shows significant improvement in dependence on $T$.
>
> Similarly, in the OBO setting (see Table 1), all existing methods incorporate regularity terms such as $V_T$, $H_{p,T}$, or $\Delta_T$ in their regret bounds. Specifically, all existing methods (e.g., OAGD, SOBOW, SOBBO) yield regret bounds like $T/w + V_T + H_{2,T}$ or $(T/w) \sigma^2 + V_T + H_{2,T}$ with $w = o(T)$; please see [[Ataee Tarzanagh et al., 2024]](https://openreview.net/forum?id=dzmWT2UWMD), [[Lin et al., 2024]](https://openreview.net/forum?id=Xx1_bVkiuZ), and [[Bohne et al., 2024]](https://openreview.net/forum?id=VYU605FWmS).
>
> However, our average regret bound features a $\sigma^2$ term that is multiplied by a factor strictly better in $T$ than previous methods, which often incur an $\mathcal{O}(T^{-1/2})$ rate due to window averaging (please refer to Table 1).
>
> Overall, our method is able to achieve regret bounds that improve over both single-level and bilevel settings.
>
> ---
>
> **Question 3: Is the dimensional dependence in the zeroth order setting optimal? In Theorem 3.2, Line 200, the dimensional dependence seems to be $\mathcal{O}((d_1 + d_2)^{3/2})$, but in Remark 3.3, this is reported as $\mathcal{O}(d_1 + d_2)$. Which statement is correct?**
>
> **Response:** Thank you for catching this important discrepancy. In Remark 3.3, we wrote $O(d_1 + d_2)$ to highlight the dependence on both the upper-level and lower-level dimensions. However, we simplified the expression and omitted the exponent for brevity. The correct dimensional dependence appears in Theorem 3.2, where the regret bound consists of two dominant terms: the first term scales as $O((d_1 + d_2)^{3/4})$, and the second as $O((d_1 + d_2)^{3/2})$.
>
> This higher-order dependence is expected in the zeroth-order bilevel setting, where gradients, Jacobians, and Hessians must be estimated using only function evaluations.
>
> We will revise Remark 3.3 to clarify that although $O(d_1 + d_2)$ reflects the qualitative dimensional dependence, the exact bound includes powers stated in Theorem 3.2.
>
> ---
>
> **Weakness 2: Paper lacks clear articulation of limitations and future work**
>
> **Limitations: The discussion of limitations is rather superficial. For example, there are many more natural questions that are left open after reading this paper. Are the regret bounds optimal? If not, what are the lower bounds for regret in each problem setting?**
>
> **Response:** Thank you for raising this important point. We agree that the current discussion of limitations can be expanded. While our work establishes the first sublinear regret guarantees for stochastic OBO without window smoothing, we do not claim these bounds are optimal. Indeed, deriving lower bounds for bilevel regret remains an open and challenging problem, in both first-order and zeroth-order settings. To date, all existing works have focused exclusively on upper bounds (please refer to Table 1).
>
> We will revise the limitations section to explicitly state that the optimality of our regret bounds is not known, and that lower-bound analysis is an important direction for future work.
>
> Beyond this, another limitation is that the work focuses on nonconvex online bilevel learning, which is more challenging than the convex case. Extending our results to cover convex and strongly convex bilevel problems, particularly in the zeroth-order setting, would be a valuable direction for future research. While first-order convex and strongly convex guarantees exist for some prior OBO methods [[Tarzanagh et al., 2022]](https://proceedings.mlr.press/v238/ataee-tarzanagh24a/ataee-tarzanagh24a.pdf), no such guarantees currently exist for zeroth-order bilevel optimization. This represents an open and important area for further investigation.
>
> We will add a discussion of these potential improvements and outline lower bound questions for future work.

---

> > ### Comment · Reviewer_yzQD · 2025-08-05
> >
> > I want to thank the authors for addressing my concerns. I read the rebuttal and I have no further questions.

---

### Official Review · Reviewer_LBrC · 2025-07-02

**Clarity:** 2
**Significance:** 2
**Originality:** 3
**Rating:** 3
**Confidence:** 4

**Summary:**

This paper studies the online bilevel optimization problem with stochastic and zero-order feedback. It proposes a new performance measure for OBO that avoids the need for window-smoothed objectives, as used in prior work. The paper then presents algorithms for both the stochastic and zero-order feedback settings, providing theoretical guarantees under the proposed performance metric. Experiments on black-box attacks and loss tunning are conducted to demonstrate the effectiveness of the proposed methods.

**Questions:**

**Questions:**

1. Could you provide additional justification for the proposed performance measure?
2. Could you offer a more detailed discussion on the tightness of the proposed regret bounds?
3. In the experiments, what is the benefit of applying an algorithm designed for time-varying objectives instead of treating the data as stochastic samples from a fixed distribution?

**Ethical Concerns:**

["NO or VERY MINOR ethics concerns only"]

**Final Justification:**

One of my concerns about the paper lies in the theoretical guarantee, specifically the $O(T\^{1/3}\Delta\_T + T\^{2/3}\Psi\_T)$ dynamic regret bound (Theorem 2.6). This bound implies that the dynamic regret is sublinear only when $\Delta\_T = o(T^{2/3})$ and $\Psi_T = o(T\^{1/3})$. In the worst case, the bound becomes superlinear at a rate of $O(T^{5/3})$. By contrast, the algorithms in Table 1 achieve at worst $O(T)$ dynamic regret bound (though the performance measure is slightly different). Since no lower bound is provided, it is unclear whether this gap is due to the inherent hardness of the problem or the suboptimality of the proposed algorithm.

That said, I acknowledge that the paper introduces an interesting approach by avoiding window smoothing, and my concerns regarding the experimental results have been adequately addressed. Therefore, I am willing to raise my score to 4.

In the revision, I encourage the authors to provide a more comprehensive discussion of their results. For example, in Remark 2.7, it would be helpful to clearly specify the conditions under which the regret is sublinear, identify the scenarios in which the proposed bound is tighter than existing approaches, and discuss its limitations in the worst-case setting.

**Quality:**

2

**Strengths And Weaknesses:**

**Strengths:**

* The proposed performance measure eliminates the need for smoothing used in previous works, making it more suitable for scenarios where the objective function changes over time.
* Experiments are conducted to empirically validate the effectiveness of the proposed algorithms.


**Weaknesses:**

* **Clarity of the performance measure:**
The notation for $f_t$ and $g_t$ is somewhat confusing to me, as it appears these notations are used to denote both expected and individual functions in (1), leading to ambiguity in understanding the performance measure. If $f_t$ represents the expected function, it is unclear to me over what randomness the expectation in equation (2a) is taken. On the other hand, if $f_t$ denotes the individual realization, it seems that BL-Regret cannot reach zero even with optimal predictions at each iteration. Additionally, the BL-Regret depends on $\alpha_t$, which is a parameter of the algorithm itself. It is not clear what is an appropriate choice of $\alpha_t$ for evaluating different methods, or how to fairly compare regret across methods using different $\alpha_t$ schedules.

* **Theorem 2.6 and Theorem 3.2:**
The regret bounds presented appear superlinear in the worst case, as $\Delta_T$ can be as large as $\mathcal{O}(T)$. In such scenarios, the guarantees become worse than the trivial $\mathcal{O}(T)$ regret that one could achieve simply by choosing arbitrary predictions within a compact set $X$.

* **Claims about convergence rates:**
In line 161, the authors claim the proposed method surpasses the $\mathcal{O}(T^{-1/2}\sigma^2)$ rate for online stochastic optimization (OSO). However, this argument is unclear because the bound still depends on $\Delta_T$, which in the worst case can grow as $\mathcal{O}(T)$, resulting in a convergence rate worse than $\mathcal{O}(1)$. The same concern applies to the claim made at line 212.

* **Experiments:**
Although the paper focuses on OBO with time-varying $f_t$ and $g_t$, the two experimental cases studied appear essentially to be offline learning problems, where it is unclear why modeling time-varying objectives is necessary. Even if the validation datasets $\mathcal{D}_t^{val}$ and training datasets $\mathcal{D}_t^{tr}$ vary, in the offline setting it would be more natural to assume they are drawn from the same fixed distribution rather than a time-varying distribution.

---

> ### Author Rebuttal · Authors · 2025-07-31
>
> We thank Reviewer LBrC for the detailed feedback.
>
> **Q1: Additional justification for proposed performance measure?**
> **Weakness 1- Clarity of the performance measure: - The notation for $f_t$ and $g_t$ ... iteration.**
>
> **Response:**  Thank you for your comment. We use $f_t(x,y;\xi_t)$ and $g_t(x,y;\zeta_t)$ to denote individual stochastic realizations, while the expected functions are defined as: $f_t(x,y) := E_{\xi_t  \sim D_f} \quad [f_t(x,y;\xi_t)]$ and $g_t(x,y) := E_{\zeta_t \sim D_g}\quad [g_t(x,y;\zeta_t)]$,as introduced in Equation (1).
>
> The key insight is that $x_t$ represents the output of a stochastic bilevel optimization algorithm at iteration $t$ (detailed in Section 2). As such, $x_t$ is a random variable that depends on all stochastic information observed by the algorithm up to time $t-1$, including the random samples used in the estimation of (implicit) gradients and Hessians in earlier iterations.
>
> Our bilevel regret is defined in Equation (2a) as $BL-Reg_T := \sum_{t=1}^T E [||P_{X,\alpha_t}(x_t; \nabla f_t(x_t, y^*_t(x_t)))||^2]$, where the gradient of the expected function $f_t$ is evaluated at the random point $x_t$. Although $f_t$ is an expected function (not a random sample), its gradient becomes a random quantity due to its dependence on $x_t$, which is itself random.
>
> The outer expectation $E[\cdot]$ in our regret is over all randomness in the algorithm up to time $t$, including sampled gradients and Hessians (see Algorithms 1 and 2). Regret thus measures the expected stationarity gap, with bounds given in Theorems 2.6 and 3.2.
>
> ---
>
> **Q1: Additional justification for proposed performance measure?**
> **Weakness 1- Clarity of the performance ... BL-Regret depends on $\alpha_t$, ...  schedules.**
>
> **Response:** Thank you for your comment. This dependency is standard in the analysis of online algorithms involving projected gradient descent. We refer the reviewer to Definitions 2.2 and 2.5 in the seminal work of [[Hazan, 2017](https://proceedings.mlr.press/v70/hazan17a/hazan17a.pdf)], where the regret for projected gradient descent is defined as:
> $$
> Regret_T = \sum_{t=1}^T ||\frac{1}{\eta_t}(x_t-x_{t+1})||^2,
> $$
> with updates
> $$
> x_{t+1} = \Pi_{\mathcal{K}}[x_t - \eta_t \nabla F_{t,w}(x_t)],
> $$
> where
> $$ F_{t,w}(x_t)=\frac{1}{w}\sum_{i=0}^{w-1}f_{t-i}(x_t).$$
>
> Their Proposition 2.3 shows projected gradient is a continuous mapping, so Brouwer’s fixed point theorem guarantees at least one fixed point where projected gradient (and thus average regret) vanishes. Definition 2.5 also shows single-level regret depends on stepsize, as does our bilevel regret on $\alpha_t$. Our analysis extends these results to the bilevel case.
> It is important to contextualize our regret relative to prior OBO work. Previous OBO methods in Table 1 (e.g. OAGD and SOBOW) address the unconstrained case ($\mathcal{X} = \mathbb{R}^{d_1}$), where the projection operator becomes the identity. Thus, our bilevel regret definition (Eq. 2b) simplifies to
> $$
> BL-Reg_T = \sum^T_{t=1} E [||\nabla f_t(x_t, y^*_t(x_t))||^2],
> $$
> which is independent of $\alpha_t$ and matches the regret definitions used by OAGD and SOBOW. Our work extends OBO to the **constrained** setting and establishes an improved $O(T^{1/3})$ bound. The stepsize $\alpha_t = 1/(c+t)^{1/3}$ in Theorem 2.6 (Eq. 15) ensures convergence and is standard in non-convex online optimization.
>
> For fair experimental comparison (Section 4):
> (1) In unconstrained settings, all methods use plain gradient descent, so regret is independent of $\alpha_t$;
> (2) In constrained settings, all use the same projection and tune $\alpha \in {0.0001, 0.0005, 0.001, 0.005, 0.01}$.
>
> ---
>
> **Q2: Detailed discussion on tightness of regret bounds?**
> **Weakness 2- Theorem 2.6 and Theorem 3.2: The regret bounds presented appear superlinear  ...  a compact set $X$.**
>
> **Response:**  Thank you for your comment.   It is widely recognized that it is impossible to achieve dynamic regret in general. For example,  [[Zhang et al. (2018)](https://proceedings.neurips.cc/paper/2018/file/10a5ab2db37feedfdeaab192ead4ac0e-Paper.pdf)], on Page 3, Section 2.2 states: *"While **it is impossible to achieve a sublinear dynamic regret in general**, we can bound the dynamic regret in terms of certain regularity of the comparator sequence or the function sequence."*
>
> The condition $\Delta_T = o(T)$ is standard in OBO literature [[Tarzanagh et al., (2022)](https://proceedings.mlr.press/v238/ataee-tarzanagh24a/ataee-tarzanagh24a.pdf)] and online  single-level optimization [[Besbes et al., (2015)](https://arxiv.org/pdf/1307.5449)].
>
>
> Even with $\Delta_T = O(T)$, our bound scales as $O(T^{4/3})$ compared to window-smoothed methods requiring $w = \sqrt{T}$, which give $O(T^{3/2})$ regret; albeit neither regret is usable as they are not sublinear.
>
> When $V_T = c T$ for some $c > 0$, the number of objective functions $f_t$ differing from previous ones can scale linearly with $T$, making good performance impossible—information from $f_1$ to $f_{t-1}$ is uninformative for $f_t$. $\Delta_T$ consists of $E_1$ and $V_T$. We assume $\|y_1 - y_1^*\| \leq D$ for $E_1$, and provide examples where $V_T = o(T)$ in many online settings.
>
> **Example 1:**
> Consider $f_t(x) = \|A_t x - b_t\|^2$, where
> $A_t = \begin{bmatrix} 1 & 0 \\ 0 & 1 + 1/t \end{bmatrix}$, $x = b_t = (1,1)$.
> Then,
> $$
> V_T := \sum_{t=2}^T \max_x |f_t(x) - f_{t-1}(x)| = \sum_{t=2}^T |(1/t)^2 - (1/(t-1))^2| .
> $$
> By $a^2 - b^2 = (a-b)(a+b)$,
> $$
> V_T = \sum_{t=2}^T |(1/t - 1/(t-1))(1/t + 1/(t-1))| = \sum_{t=2}^T |2/(t(t-1)^2)| .
> $$
> Then,
> $V_T \leq \sum_{t=2}^T 2/t^3 \approx \int_{2}^T 2/t^3 dt = 1/4 - 1/T^2.$
> As $T \to \infty$, $V_T$ approaches a constant, i.e., grows slower than $T$.
>
> Moreover, if $A_t = \begin{bmatrix} 1 & 0 \\ 0 & 1 + 1/\log t \end{bmatrix}$, then
> $$
> V_T = \sum_{t=3}^T |(1/\log t)^2 - (1/\log (t-1))^2| = \sum_{t=3}^T |1/\log t - 1/\log (t-1)| (1/\log t + 1/\log (t-1)) .
> $$
> Since $1/\log t - 1/\log (t-1) \approx -1/(t (\log t)^2)$, we have $V_T = \sum_{t=3}^T 1/(t (\log t)^3) = O(\log T)$.
>
>
> **Example 2:**
> Let $x \in [-1,1]$, $y \in \mathbb{R}$, and define
> $$
> f_t(x, y) = \frac{1}{2} (x + 2 a_t^{(1)})^2 + \frac{1}{2} (y - a_t^{(2)})^2,
> $$
> $$
> g_t(x, y) = \frac{1}{2} y^2 - (x - a_t^{(2)}) y,
> $$
> where $a_t^{(1)} = 1/t$ and $a_t^{(2)} = 1/\sqrt{t}$.
> Then $y_t^{\star}(x) = x - a_t^{(2)}$ and by bounding term by term one finds
> $$
> V_T = O(1).
> $$
>
> In summary, when the variations decay faster than $1/t$, we obtain $V_T = o(T)$. If the decay is like $1/\sqrt{t}$, then $V_T = O(\sqrt{T})$.
>
> ---
>
> **Weakness 3- Claims about convergence rates:.... same concern applies to the claim made at line 212.**
>
> **Response:** We appreciate the reviewer’s comment.
>
> As discussed in response to **Weakness 2** and Remark 2.7, and as shown in Theorems 3.1 and 4.2 of  [[Hallak et al., 2021](https://openreview.net/forum?id=XkXb99Eeg6G)], dynamic regret bounds in single-level online optimization include a regularity term (e.g., $V_T$ or $C_T$) that quantifies the temporal variation of the cost functions. The rate $\mathcal{O}(T^{-1/2} \sigma^2)$  is meaningful only when this regularity term is sublinear; please also see [[Aydore et al., 2019](https://openreview.net/forum?id=HJxWTNrg8B)] and [[Zhang et al., 2020](https://openreview.net/forum?id=Z_8hoytvdi)]
>
> Specifically, our method's regularity term $\Delta_T$ is analogous to those terms in prior work, but the key difference is that our dependence on $T$ and $\sigma$ yields strictly better dependence than that of previous methods when $\Delta_T$ grows slowly. For example, in Theorems 3.1 and 4.2 of [[Hallak et al., 2021](https://openreview.net/forum?id=XkXb99Eeg6G)], the stochastic component appears as $O(T \sigma^2/w^2)$ alongside regularity terms. Hence, their bound also depends on a regularity term, and it is commonly assumed that this term is sublinear in order to achieve a sublinear regret.
>
>
> Similarly, in the bilevel setting ( please refer to Table 1), all existing methods incorporate regularity terms such as $V_T$, $H_{p,T}$, or $\Delta_T$ in the regret bounds. These methods (e.g., OAGD, SOBOW, SOBBO) yield regret bounds like $T/w + V_T + H_{2,T}$ or
> $(T/w) \sigma^2 + V_T + H_{2,T}$, with $w = o(T)$; please see [[Ataee Tarzanagh et al., 2024](https://openreview.net/forum?id=dzmWT2UWMD)] and [[Lin et al., 2024](https://openreview.net/forum?id=Xx1_bVkiuZ)]. In contrast, our approach features a $\sigma^2$ term that is multiplied by a factor strictly better in $T$ than previous methods, which often incur a  $\mathcal{O}(T^{-1/2})$  due to window averaging; please refer to Table 1.
>
> ---
>
> **Q3: Benefit of time-varying ... from fixed distribution?**
> **Weakness 4- Experiments: Although the paper .. .a time-varying distribution.**
>
> **Response:** We respectfully disagree with the characterization of our experiments as "essentially offline learning problems." Both experimental settings are genuinely online with time-varying objectives. For example, in the adversarial attack setting (Section 4), the optimal attack strategy changes abruptly between stages (see Eq. (28)), reflecting dynamic defenses in real-world scenarios. These are not i.i.d. samples, but represent structural shifts requiring online adaptation of hyperparameters $x^*_s$. This setting is well-established in the online learning literature, where distribution shifts are a central motivation for online algorithms [[Hall and Willett, 2015]](https://arxiv.org/pdf/1307.5944).
>
> Our method is also well-suited for online meta-learning (OML), where agents must quickly adapt to sequential tasks; please see Section 4.3 in [[Ataee Tarzanagh et al., 2024](https://openreview.net/forum?id=dzmWT2UWMD)] for discussion on OBO for OML. For additional applications, including traffic flow prediction and wireless network control, please refer to Appendix A in [[Lin et al., 2024](https://openreview.net/forum?id=Xx1_bVkiuZ)].

---

> > ### Comment · Reviewer_LBrC · 2025-08-04
> >
> > Thank you for the detailed explanation of the notations and the clarifications regarding the experiments. However, I remain unclear about the arguments concerning the regret guarantees. While I agree with the authors that sublinear dynamic regret is generally unachievable, the results presented in the paper appear to be *superlinear* in the worst case, specifically of the order \$O(T^{\frac{4}{3}})\$. It seems that the proposed method only achieves sublinear regret under the condition \$\Delta\_T = o(T^{\frac{2}{3}})\$. In the case of single-level optimization (Besbes et al. 2015), the worst-case regret is \$O(T)\$, which implies that it does not exceed the trivial bound derived from the boundedness of the loss function. Regarding the comparison with Tarzanagh et al. (2022), the definition of regret differs from that used in this paper. Nevertheless, their bound of \$O(T/W + H\_{1,T} + H\_{2,T})\$ seems still result in \$O(T)\$ regret in the worst case when \$W = \sqrt{T}\$.

---

> ### Author Response · Authors · 2025-08-05
> **Response to Reviewer LBrC- Part I**
>
> **Response:** We appreciate the reviewer for providing additional feedback. Below, we provide further discussion regarding the worst-case superlinear regret bound.
>
> We agree with the reviewer that under worst-case regularity, such as $\Delta_T = O(T)$, the dynamic regret bound of our method is at most $O(T^{4/3})$. As previously discussed with Reviewer **yzQD**, we do not claim any lower bounds, and the regret may be loose in specific worst-case scenarios involving the regularity $\Delta_T$. Superlinear regret is often unavoidable when the environment exhibits linear variation, and many existing online algorithms can similarly incur superlinear dynamic regret in such non-stationary settings.
>
> We first review seminal works on online gradient descent [[Zinkevich (2003)](https://www.martin.zinkevich.org/publications/ICML03.pdf)] and online mirror descent [[Hall and Willett (2015)](https://arxiv.org/pdf/1307.5944)] that can lead to superlinear regret bounds under worst-case regularity shifts. We then highlight recent results showing that such bounds are not uncommon in online learning algorithms operating in non-stationary environments—particularly when assumptions similar to ours on the rate of change are not imposed. Next, we compare these results to the regret bounds in [[Tarzanagh et al. (2022)](https://proceedings.mlr.press/v238/ataee-tarzanagh24a/ataee-tarzanagh24a.pdf)] and [[Besbes et al. (2015)](https://arxiv.org/pdf/1307.5449)]. Finally, we discuss adaptive algorithms in the single-level setting that have the potential to be extended to the bilevel setting to improve regret dependence on $T$ and regularity measures such as $\Delta_T$.
>
> **Superlinear regret in worst-case regularity scenarios under non-stationary environments.**  In the seminal work of [[Zinkevich (2003)](https://www.martin.zinkevich.org/publications/ICML03.pdf)] for online gradient descent and its extension to online mirror descent by Theorem 2 of [[Hall and Willett (2015)](https://arxiv.org/pdf/1307.5944)], the regret is $\text{Reg} = O(\sqrt{T}(1 + P_T))$, where
> $$P_T = \sum_{t=1}^{T-1} ||x^{\star}_{t+1} - x^{\star}_t||$$
> is the path-length of the comparator sequence $x^{\star}_t$. This bound becomes superlinear when $P_T = O(T)$, resulting in $O(T^{3/2})$ regret. For example, consider a sequence of loss functions $f_t(x) = \frac{1}{2}||x - x^{\star}_t||^2$ where the optimal solutions drift linearly: $x^{\star}_t = t \cdot v$ for some unit vector $v$ with $||v|| = 1$. In this case
>
> $$P_T = \sum_{t=1}^{T-1} ||(t+1)v - tv|| =  T-1 = O(T).$$
> This scenario naturally occurs when tracking a moving target with constant velocity or in trending non-stationary environments.
>
> We now discuss further examples showing that worst-case superlinear regret bounds are not uncommon in online learning algorithms operating in non-stationary environments, particularly when  assumptions similar to ours on the rate of change are not imposed.
>
> - [Yi et al. (2021)](https://proceedings.mlr.press/v139/yi21b/yi21b.pdf) in Table 1 show that their Algorithm 1 achieves regret bound $O(\sqrt{T}(1 + P_T))$ in the convex setting.  When $P_T = O(T)$, this leads to a regret of $O(T^{3/2})$.
> - [Guo et al. (2022)](https://proceedings.neurips.cc/paper_files/paper/2022/file/ec360cb73d322e80a877b7ec7e13c79a-Paper-Conference.pdf) in Theorem 3 and Remark 3 show that their online learning algorithm achieves $O(\sqrt{T} P_T)$ regret in the convex setting.  When $P_T = O(T)$, this gives a regret of $O(T^{3/2})$.
>
> - Theorem 3.2 in [[Zhang et al. (2020)](https://arxiv.org/pdf/2010.07378)] shows that the regret includes a term $V_f^2 T^{3/4}$ in the convex setting, where $V_f$ denotes the maximum squared variation of the objective between consecutive time steps. When $V_f = O(T^{1/4})$, this leads to a regret of $O(T^{5/4})$, which is superlinear. Theorem 4.2 shows that the accumulated gradient of the smoothed functions has regret bounded by $W_T T^{1/2}$, where $W_T$ is the sum of expected function variations over time. When $W_T = O(T^{2/3})$, the regret becomes $O(T^{7/6})$, again resulting in a superlinear bound.
>
> - Theorem 1 in [[Kalhan et al. (2021)](https://koppel.netlify.app/assets/papers/2019_kalhan_etal.pdf)] shows the regret bound $\mathrm{Reg}_T^D \leq \mathcal{O}\left(\sqrt{T}\left(1 + V_T + \sqrt{D_T}\right)\right)$, where $V_T$ is the function variation and $D_T$ is the gradient variation.  When $V_T = O(T)$, this leads to a regret of $O(T^{3/2})$.
>
> - Theorem 1 in [[Chen et al. (2017)](https://arxiv.org/pdf/1701.03974)] shows that the regret bound is $P_T T^{1/3}$ with step size $T^{-1/3}$ where $P_T$ is the path-length of the comparator sequence.  When $P_T = O(T)$, this leads to a regret of $O(T^{4/3})$. In Corollary 1, they set $P_T=o(T^{2/3})$ to get $\text{Reg} = o(T)$.

---

> ### Author Response · Authors · 2025-08-05
> **Response to Reviewer LBrC- Part II**
>
> **Comparison with  OAGD [[Tarzanagh et al. (2022)](https://proceedings.mlr.press/v238/ataee-tarzanagh24a/ataee-tarzanagh24a.pdf)] and SOBBO [[Bohne et al. (2024)](https://arxiv.org/pdf/2409.10470?)].** As the reviewer pointed out, our regret definition differs from that of Tarzanagh et al. (2022), as ours is projection-based and window-free in the stochastic setting, which requires more technical effort to achieve a sublinear regret bound. Besides many algorithmic novelties of our algorithm such as single step, recursive momentum update, window-free, and derivative-free algorithm, we compare their regret bound below.
>
> - In deterministic bilevel optimization, Tarzanagh et al. (2022) show that with $W = \sqrt{T}$ and the path-length  regularity $H_{1,T} = \sqrt{T}$, the dynamic regret is $O(T/W + H_{1,T} + H_{2,T}) = O(\sqrt{T})$. Importantly, regardless of how benign the environment is (i.e., how small $H_{1,T} + H_{2,T}$ are), their regret bound remains $O(\sqrt{T})$. In contrast, our method adapts to the complexity of the environment. While our worst-case bound is $O(T^{4/3})$, in benign settings—such as when the regularity $\Delta_T = o(T^{2/3})$—we can achieve sublinear regret. Moreover, as the regularities decrease further, our regret bound continues to improve.
>
> - In stochastic bilevel optimization, our regret remains sublinear under suitable conditions on regularities. Importantly, in the stochastic bilevel optimization setting, our method achieves an (average regret) $T^{-2/3}\sigma^2$ dependence on the noise variance, which improves dependency on $T$ over the $T^{-1/2}\sigma^2$ bound for stochastic OBO [[Bohne et al. (2024)](https://arxiv.org/pdf/2409.10470?)] and surpasses the $T^{-1/2}\sigma^2$ rate in prior work;  please refer to Remark 2.7.
>
> **Comparison with [[Besbes et al. (2015)](https://arxiv.org/pdf/1307.5449)].** We agree with the reviewer that Besbes's work has better dependency on function variation $V_T^{1/3}$. We note that our regret bound in the nonconvex case may not be directly comparable to the results of Besbes et al. (2015) in convex settings. Nevertheless, the key to achieving favorable dependency in the regret bound is to use adaptive function variation-based step size and batch size. For example, in their Theorem 3, the authors select an adaptive batch size and step size as follows:
>
> $$\Delta_T = \left\lceil (T / V_T)^{2/3} \right\rceil, \quad \eta_t = \frac{r}{G\sqrt{\Delta_T}}.$$
>
> where $\Delta_T$ is the batch size (as defined in their work) and $\eta_t$ is the step size at iteration $t$, both depending on the cumulative function variation $V_T$. This adaptive parameterization is essential for attaining the improved regret  bound $O(V_T^{1/3} T^{2/3})$.  However, the quantity $V_T$ is typically not accessible in practice. Therefore, we adopt a simple step size $1/(c + t)^{1/3}$ for some constant $c$, without incorporating any regularity-dependent terms.
>
> **Adaptive algorithms for improved regret.**   Finally, we note that a similar adaptive step size and batch size strategy in Eqn (117) could also improve our dependence on regularity measures such as function variation. However, this type of information—such as function variation or path length—is typically not available in practice. Therefore, it is crucial to develop an adaptive algorithm that can achieve the optimal rate without prior knowledge of $P_T$ or $V_T$, following the expert-tracking algorithm of [[Zhang et al. (2018)](https://proceedings.neurips.cc/paper/2018/file/10a5ab2db37feedfdeaab192ead4ac0e-Paper.pdf)], which improves $\text{Reg} = O(\sqrt{T}(1 + P_T))$ to $\text{Reg} = O(\sqrt{T(1 + P_T)})$. Exploring how to extend such adaptive techniques to improve regret bounds in bilevel learning remains an open and promising research direction.

---

### Official Review · Reviewer_ETap · 2025-07-02

**Clarity:** 3
**Significance:** 3
**Originality:** 3
**Rating:** 4
**Confidence:** 4

**Summary:**

The paper presents a unified first-order (SOGD) and zeroth-order (ZO-SOGD) framework for online bilevel optimization (OBO) that achieves sub-linear stochastic bilevel local regret without any window smoothing. Key ingredients are 1) momentum-type projected search direction that simultaneously updates leader, follower, and auxiliary variables in a single loop, 2) oracle-efficient hyper-gradient estimation that needs only one inner linear-system step per round, and 3) a fully function-value–only variant that estimates gradients, Hessians, and Jacobians via two-point finite differences.

Theoretical contributions include tight $O(T^{1/3})$ (first-order) and $O((d_1+d_2)^{3/4}T^{1/3})$ (zeroth-order) regret bounds under standard non-convex/strongly-convex assumptions. Empirical studies on online adversarial attacks and parametric loss tuning under distribution shift confirm the framework’s practical gains over state-of-the-art OBO and single-level baselines.

**Questions:**

1) Could you make a GitHub or submit your codes and experiments for the sake of reproducibility of your result?

2) Could you elaborate on concrete scenarios where the proposed bilevel local regret better captures performance than traditional window-smoothed metrics? Have you identified tasks where the two measures give conflicting evaluations?

3) The theoretical analysis uses order-$T^{-1/3}$ step sizes. In the experiments I suppose you tune them heuristically, did you observe a clear regime where the theory-suggested schedule outperforms alternative decays (e.g., $T^{-1/2}$)?

**Ethical Concerns:**

["NO or VERY MINOR ethics concerns only"]

**Final Justification:**

The author’s reply resolves my concerns; I’m keeping my current rating. However, the proofs read quite densely and could be clarified with revisions. Compared with other bilevel optimization papers I’ve read, I had more difficulty following these arguments.

**Limitations:**

Yes.

**Paper Formatting Concerns:**

None.

**Quality:**

3

**Strengths And Weaknesses:**

Strengths:
1) Removing the window-smoothed regret requirement is a clear conceptual advance.
2) Proofs carefully track stochastic variance, path-length, and gradient-drift terms and tighten the noise dependence to $T^{-2/3}$ which is sharper than the $T^{-1/2}$ factor in SOBBO/SOBOW.
3) Experiments show ZO-SOGD consistently outperforms strong black-box attack optimizers (ZO-O-Adam, SignSGD, etc.) in success rate and perturbation norm while maintaining reasonable run-time, and SOGD matches or exceeds OAGD / SOBOW on adaptive loss tuning with lower latency.
Weaknesses: The code for the experiments is not provided in the supplementary material. It must be required for reproducibility.

---

> ### Author Rebuttal · Authors · 2025-07-31
>
> We sincerely thank Reviewer ETap for the valuable and constructive feedback. We address each point below.
>
> **Question 1: Could you make a GitHub or submit your codes and experiments for the sake of reproducibility of your result?**
>
> **Response:** Thank you for raising this important point. We fully agree on the importance of reproducibility. To that end, we contacted the Area Chairs to inquire whether we could share an anonymous code link during the rebuttal phase and received the following guidance:
>
> > "Because of known concerns on identity leakage, we are prohibited using any links in the rebuttal, including, but not limited to anonymous or non-anonymous URL links, or updating your submitted GitHub repository."
>
> Accordingly, while we are currently unable to share the code during the rebuttal phase, we will release the complete codebase upon acceptance to facilitate full reproducibility of our results.
>
> ---
>
> **Question 2: Could you elaborate on concrete scenarios where the proposed bilevel local regret better captures performance than traditional window-smoothed metrics? Have you identified tasks where the two measures give conflicting evaluations?**
>
> **Response:**  We thank the reviewer for this insightful question. While the bilevel local regret values themselves may not always show dramatic differences between window-smoothed and non-smoothed approaches, the key distinction lies in the underlying algorithmic mechanisms and their real-world performance. Our contribution is not just the regret metric, but the first and zeroth order novel recursive momentum-type search directions (Eq. 6) that fundamentally differs from window-smoothed gradient methods used in prior work.
>
> Traditional OBO methods like OAGD, SOBOW, and SOBBO all rely on window-smoothed search directions of the form:
>
> $$
> \nabla F_{t,w}(x,y) = \frac{1}{w}\sum_{i=0}^{w-1} \nabla f_{t-i}(x,y),
> $$
>
> which inherently averages past gradients. This averaging introduces lag and can lead to conflicting gradient directions when objectives change rapidly. In contrast, our recursive momentum approach
>
> $$
> d^x_t = \eta\nabla f_t(x_t, y_t) + (1-\eta)d^x_{t-1} + (1-\eta)(\nabla f_t(x_t, y_t) - \nabla f_t(x_{t-1}, y_{t-1}))
> $$
>
> maintains a single adaptive direction that immediately responds to changes while preserving useful historical information through the momentum term.
>
> *Experiment: Periodic Objective Oscillations:* Following your valuable suggestion, we evaluate a synthetic scenario where the leader's objective exhibits oscillations with frequency parameter $c$:
> $$
> f_t(x,y) = (1 + \sin(ct)) \cdot \|x - y^*_t(x)\|^2 + \cos(2ct) \cdot \|x\|^2
> $$
> $$
> g_t(x,y) = \|y - (\cos(x) + 0.5\sin(ct))\|^2
> $$
>
> **Table 1: Gradient-based Regret Under Different Change Frequencies**
>
> | Method             | Time (s) | Our Regret |  Smoothed Regret |
> |--------------------|----------|-------------------------------|-----------------------------------|
> | *For $c = 0.1$ (Slow changes)* |          |                               |                                   |
> | OAGD ($w = 20$)     | 45.2     | **498.0**                         | **485.3**                             |
> | SOBOW ($w = 10$)    | 68.4     | 542.6                         | 521.8                             |
> | SOGD ($w = 1$)      | **22.6** | 596.9                     | 596.9                         |
> | *For $c = 5$ (Rapid changes)* |          |                               |                                   |
> | OAGD ($w = 20$)     | 52.3     | 1842.7                        | **127.5**                          |
> | SOBOW ($w = 10$)    | 75.8     | 1205.3                        | 186.2                             |
> | SOGD ($w = 1$)      | **24.9** | **758.4**                     | 758.4                         |
>
>
> In rapidly changing environments ($c = 5$), OBO   [[Tarzanagh et al., (2022)](https://proceedings.mlr.press/v238/ataee-tarzanagh24a/ataee-tarzanagh24a.pdf)]  with smoothing regret ($\sum_{t=1}^T ||\nabla F_{t,w}||$) shows extremely low window-smoothed regret (127.5) because opposing gradients cancel out when averaged, suggesting near-optimal performance. However, its local regret (1842.7) reveals it's actually performing poorly—the algorithm is stuck between conflicting directions.
>
> This demonstrates why window-smoothed metrics can be fundamentally misleading: they evaluate performance against averaged gradients that may not represent any actual objective the algorithm needs to track. Our local regret correctly captures that SOGD (758.4) significantly outperforms window-based methods in dynamic environments.
>
>
> ---
>
> **Question 3: The theoretical analysis uses $O(T^{-2/3})$ step sizes. In the experiments I suppose you tune them heuristically, did you observe a clear regime where the theory-suggested schedule outperforms alternative decays (e.g., $T^{-1/2}$)?**
>
> **Response:** We appreciate the reviewer's careful observation. From Theorem 2.6, we use $\alpha_t = 1/((c_1 + t)^{1/3})$ and $\beta_t = c_\beta \alpha_t$. For the zeroth-order case in Theorem 3.2, we have $\alpha_t = 1/((d_1 + d_2)^{3/4}(c_1+ t)^{1/3})$, with similar scaling for the other parameters. These $O(t^{-1/3})$ schedules are derived to balance the bias-variance trade-off in our stochastic bilevel setting and to achieve the $O(T^{1/3})$ regret bound.
>
> While it is difficult to validate this theory in our real-world  highly nonconvex experiments--and we heuristically tuned the parameters--we can demonstrate its advantages in synthetic settings. To validate our theoretical step size schedule, we conducted experiments using the periodic objective oscillation scenario from our response to Question 2.
>
> **Table 2: Step Size Schedule Comparison on Periodic Objectives**
>
> | Step Size Schedule                 | Our Regret ($c = 0.1$) | Our Regret ($c = 5$) |
> |-----------------------------------|---------------------|------------------|
> | $\alpha_t = 0.001$ (Fixed)         | 396.6               | 892.4            |
> | $\alpha_t \propto t^{-1/2}$       | **325.5**               | 850.6            |
> | $\alpha_t \propto t^{-1/3}$ (Theory) | 596.9         | **758.4**        |
>
> The faster $O(t^{-1/2})$ decay proves too aggressive for our bilevel setting, especially in rapidly changing environments. This schedule reduces the step size too quickly, preventing the algorithm from adapting to sudden changes in the bilevel landscape.
>
> The advantage of our theoretical schedule becomes particularly pronounced in environments with rapid changes. When the oscillation frequency is high ($c = 5$), the performance gap widens dramatically: the $O(t^{-1/3})$ schedule achieves a regret of 758.4, while the $O(t^{-1/2})$ schedule yields 850.6. Conversely, in smoother settings with $c = 0.1$, the performance difference narrows (596.9 vs. 325.6), as rapid adaptation is less critical.
>
> We emphasize that our approach significantly reduces computation time and memory usage, as we do not use window smoothing. Please refer to **Table 1**.

---

> > ### Comment · Reviewer_ETap · 2025-08-08
> >
> > My concerns have been taken care of by them!

---

### Decision · Program_Chairs · 2025-09-17

**Decision:**

Accept (poster)

**Comment:**

This paper proposes a new online bilevel optimization method. It introduces a novel search direction that  does not rely on deterministic window-smoothed regret minimization as existing approaches. It then shows that both first- and zeroth-order (ZO) stochastic OBO algorithms leveraging this direction achieve sublinear stochastic bilevel regret without window smoothing.

All reviewers agree that eliminating the dependence on window smoothing is a significant contribution to online bilevel optimization. In addition, the experiments confirm the effectiveness of the proposed method.  A minor concern after the rebuttal phase is that the paper lacks a more comprehensive discussion of its regret bound, particularly with regard to its limitations in worst-case scenarios. It is highly recommended that this issue be addressed in the camera-ready version. Overall, this paper makes important contributions to online bilevel optimization and is recommended for acceptance.